# Understanding Optimization in Deep Learning with Central Flows

**Jeremy M. Cohen[13]\* & Alex Damian[2]\* & Ameet Talwalkar[1] & J. Zico Kolter[1] & Jason D. Lee[2]**

[1] Carnegie Mellon University, [2] Princeton University, [3] Flatiron Institute

\*Alex and Jeremy contributed equally; author order determined by coin flip

Corespondence to: {`ad27@princeton.edu, jcohen@flatironinstitute.org`}

## Abstract

Optimization in deep learning remains poorly understood. A key difficulty is that optimizers exhibit complex oscillatory dynamics, referred to as "edge of stability," which cannot be captured by traditional optimization theory. In this paper, we show that the path taken by an oscillatory optimizer can often be captured by a *central flow*: a differential equation which directly models the time-averaged (i.e. smoothed) optimization trajectory. We empirically show that these central flows can predict long-term optimization trajectories for generic neural networks with a high degree of numerical accuracy. By interpreting these flows, we are able to understand how gradient descent makes progress even as the loss sometimes goes up; how adaptive optimizers "adapt" to the local loss landscape; and how adaptive optimizers implicitly seek out regions of weight space where they can take larger steps. These insights (and others) are not apparent from the optimizers' update rules, but are revealed by the central flows. Therefore, we believe that central flows constitute a promising tool for reasoning about optimization in deep learning.

## 1 Introduction

While there is a rich body of work on the theory of optimization, few works attempt to analyze optimization in "real" deep learning settings. Instead, even works motivated by deep learning often rely on unrealistic assumptions such as convexity, or restrict their analyses to simplified architectures. Practitioners cannot use such theories to reason directly about their optimization problems. Our goal in this paper is to develop optimization theory that applies *directly* to deep learning problems. This is a difficult task: prior research has shown that, even in the seemingly simple setting of deterministic (i.e. full-batch) training, optimization typically operates in a complex, oscillatory regime called the *edge of stability* (Xing et al., 2018; Wu et al., 2018; Jastrzębski et al., 2019; 2020; Cohen et al., 2021; 2022). The dynamics in this regime cannot be captured by traditional optimization theory.

In this paper, we devise a methodology for analyzing these complex deep learning dynamics. We argue that an oscillatory optimizer should be conceived of as "oscillating around" a particular path through weight space, which we call a *central flow*. The central flow can be interpreted as the "time-averaged" (i.e. smoothed) optimization trajectory (Figure 2). We show that this central flow

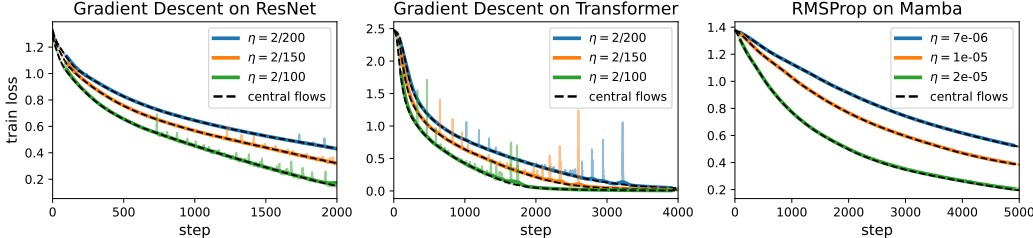

Figure 1: **Our theory can deliver accurate numerical predictions on actual neural networks**. For example, our central flows can accurately predict (dashed black) the time-averaged (i.e. smoothed) loss curves of gradient descent and RMSProp on three architectures at different learning rates.

Figure 2: **The central flow models the time-averaged (smoothed) trajectory of the oscillatory optimizer**. In this representative cartoon, gradient descent takes an oscillatory path through weight space. The central flow is a smooth curve that characterizes this trajectory, whereas gradient flow takes a different path.

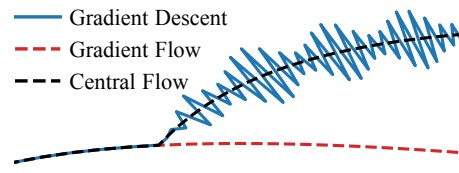

can be captured by a differential equation which explicitly characterizes how the optimizer's path through weight space depends on the local loss landscape and on the optimizer's hyperparameters. As a smooth curve, the central flow is easier to understand than the original oscillatory trajectory. Hence, by reasoning about the central flow, we can reason more easily about the original optimizer.

We start in Section 3 by analyzing gradient descent, the simplest optimizer. The dynamics of gradient descent in deep learning are complex and potentially intimidating; for example, the training loss curve behaves non-monotonically. Yet we show that gradient descent is merely "oscillating around" a well-behaved central flow that *does* decrease the loss monotonically. We then examine a simple adaptive optimizer in Section 4, before turning to RMSProp (i.e. Adam without momentum) in Section 5. We show that much of the behavior of these optimizers is actually *implicit* in their oscillatory dynamics, and we render such behaviors *explicit* via our central flow analysis. Our analysis reveals precisely how these adaptive optimizers implicitly: (1) adapt their step size(s) to the local curvature, and (2) steer towards lower-curvature regions where they can take larger steps.

While we derive each central flow using informal mathematical reasoning, we show that these flows accurately predict long-term optimization trajectories in a variety of deep learning settings — a high standard of empirical proof. Thus, we believe that central flows hold promise as a framework for analyzing, reasoning about, and perhaps even inventing, deep learning optimizers.

Our code can be found at: `http://github.com/locuslab/central_flows`.

## 2 RELATED WORK

**Edge of Stability** The dynamics of optimization in deep learning remain poorly understood, even in the seemingly simple setting of deterministic (i.e. full-batch) training. Indeed, recent research showed that gradient descent on neural networks typically operates in a regime termed the "edge of stability" (EOS) in which (1) the largest Hessian eigenvalue equillibrates around the *critical threshold* $2/\eta$, and (2) the algorithm oscillates along high-curvature directions without diverging (Xing et al., 2018; Wu et al., 2018; Jastrzębski et al., 2019; 2020; Cohen et al., 2021). These dynamics could not be explained by existing optimization theory, which led Cohen et al. (2021) to observe that there was no explanation for how or why gradient descent can function properly in deep learning.

Subsequently, several studies sought to theoretically explain EOS dynamics. Some works analyzed EOS dynamics on specific objectives (Agarwala et al., 2023; Ahn et al., 2024; Chen & Bruna, 2023; Even et al., 2024; Kreisler et al., 2023; Song & Yun, 2023; Li et al., 2022; Wu et al., 2024; Zhu et al., 2023), while other works (Arora et al., 2022; Lyu et al., 2022; Damian et al., 2023), gave generic analyses based on a local *third-order* Taylor expansion of the loss, which is one order higher than is normally used to analyze gradient descent. Similar arguments were first used by Blanc et al. (2019) to study implicit regularization in SGD. Our work is most directly inspired by Damian et al. (2023), as their analysis applies to generic objective functions, and holds throughout training, not just near convergence. However, whereas they analyze the *fine-grained* oscillatory dynamics, we argue that analyzing the *time-averaged* dynamics is simpler, and is sufficient for most purposes.

**Continuous-time models** Prior works have proposed continuous-time processes to approximate the trajectory of gradient descent (Barrett & Dherin, 2021; Rosca et al., 2023), SGD (Li et al., 2017; Smith et al., 2021; Li et al., 2021), and adaptive optimizers (Malladi et al., 2022; Cattaneo et al., 2024; Compagnoni et al., 2024). However, in the deterministic setting, these processes cannot model the optimization trajectory over long timescales in the EOS regime, which is our goal.

**Understanding adaptive optimizers** Ma et al. (2022) observed that RMSProp and Adam oscillate, and Cohen et al. (2022) showed that such dynamics can be viewed as an adaptive version of the edge of stability — a finding which we will leverage. Khaled et al. (2023) and Mishkin et al. (2024) observed that on quadratic functions, certain adaptive optimizers adapt their effective step size to the maximum stable step size; we show this holds more generally, beyond quadratics. Experiments in Roulet

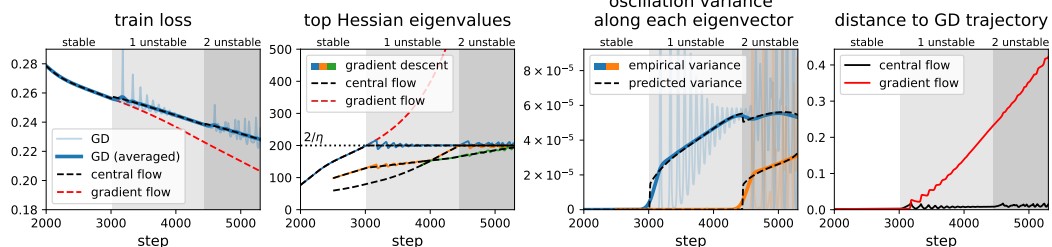

Figure 3: **Central flow for gradient descent.** A ViT is trained on CIFAR-10 using gradient descent with $\eta = 2/200$ (blue). Gradient descent enters EOS at step 3000 and after step 4500, multiple eigenvalues are unstable (dark gray). The central flow (black) accurately models gradient descent even at EOS, whereas gradient flow (red) follows a different path. Our analysis also accurately predicts the covariance $\Sigma$ with which gradient descent oscillates around the central flow (third panel). Note that the first panel plots the central flow's *prediction* for the time-averaged train loss (eq. (6))

et al. (2024) and Wang et al. (2024d) can be explained by the phenomenon we call "acceleration via regularization." Many works have also conducted rigorous convergence analyses of adaptive optimizers, under various regularity assumptions on the objective (Duchi et al., 2011; Reddi et al., 2018; Chen et al., 2019a;b; Zaheer et al., 2018; Zou et al., 2019; Défossez et al., 2022; Li & Lin, 2024; Chen et al., 2022; Wang et al., 2024a; Yang et al., 2024; Guo et al., 2021; Shi et al., 2021; Zhang et al., 2022; Crawshaw et al., 2022; Li et al., 2024; Wang et al., 2024b; Hong & Lin, 2024; Zhang et al., 2024; Wang et al., 2024c; Hübler et al., 2024).

## 3    GRADIENT DESCENT

The simplest optimizer is deterministic gradient descent with a fixed learning rate $\eta$:

$$w_{t+1} = w_t - \eta \nabla L(w_t). \tag{GD}$$

Cohen et al. (2021) showed that traditional optimization analyses cannot capture the typical dynamics of gradient descent in deep learning. We now present an analysis that *does* capture these dynamics.

### 3.1    THE DYNAMICS OF GRADIENT DESCENT

To understand the dynamics of gradient descent in deep learning, it is instructive to first consider the case of quadratic objectives. On quadratics, gradient descent oscillates if the *curvature* (i.e. Hessian) is too large relative to the learning rate. For example, consider a one-dimensional quadratic $L(x) = \frac{1}{2}Sx^2$, with global curvature $S$. The gradient descent iterates $\{x_t\}$ evolve via $x_{t+1} = (1 - \eta S)x_t$. If $S$ exceeds the *critical threshold* $2/\eta$, then $(1 - \eta S) < -1$, so the iterate $x_t$ flips signs and grows in magnitude at each step, i.e. gradient descent oscillates with exponentially growing magnitude. More generally, on a quadratic in multiple dimensions, gradient descent oscillates with exponentially growing magnitude along any Hessian eigenvector with an eigenvalue exceeding $2/\eta$.

While deep learning objectives are not globally quadratic, a local quadratic Taylor approximation suggests that if the Hessian $H(w)$ has any eigenvalues exceeding $2/\eta$, gradient descent will oscillate along the corresponding eigenvectors. This suggests that gradient descent cannot function properly if the *sharpness* $S(w) := \lambda_1(H(w))$, defined as the largest Hessian eigenvalue, exceeds $2/\eta$.

Why, then, does gradient descent converge in deep learning? The natural explanation is that the sharpness $S(w)$ remains below $2/\eta$ throughout training. Yet, Cohen et al. (2021) empirically observed a very different reality. When training neural networks, the sharpness tends to rise (a still-unexplained phenomenon termed *progressive sharpening*), and usually reaches the critical threshold $2/\eta$ during training. When this occurs, gradient descent oscillates along the top Hessian eigenvector(s), as expected from a quadratic Taylor approximation. Yet, perhaps surprisingly, these oscillations do not cause divergence. Instead, gradient descent enters a regime termed the *edge of stability* (EOS) in which the sharpness equilibrates around the critical threshold $2/\eta$, the algorithm oscillates without diverging along the top Hessian eigenvector(s), and the loss behaves non-monotonically. Cohen et al. (2021) noted that these dynamics could not be explained by traditional optimization theory.

Damian et al. (2023) showed that the key for understanding these EOS dynamics is to Taylor-expand the objective to *third* order, which is one order higher than traditionally used in analyses of gradient

descent. Let us informally sketch this argument. Suppose that gradient descent is oscillating around a reference point $\overline{w}$, along the top Hessian eigenvector $u$, with current magnitude $x$, so that $w = \overline{w} + xu$. Due to the oscillation, the optimizer follows the gradient at $w$ rather than the gradient at $\overline{w}$. How do the two relate? A Taylor expansion of $\nabla L$ around $\overline{w}$ yields: (see Lemma 1):

$$\nabla L(w) = \underbrace{\boxed{\nabla L(\overline{w})}}_{\text{(1) gradient at reference point}} + \underbrace{\boxed{xS(\overline{w})u}}_{\text{(2) oscillation}} + \underbrace{\boxed{\tfrac{1}{2}x^2\nabla S(\overline{w})}}_{\text{(3) sharpness reduction}} + o(x^2) \qquad (1)$$

The third term, which arises from the cubic term in the Taylor expansion of the loss, reveals that a gradient step on the loss with step size $\eta$ automatically includes a gradient step on the *sharpness* of the loss with step size $\tfrac{1}{2}\eta x^2$. Thus, **oscillations automatically trigger reduction of sharpness.** This is the crucial ingredient that is missing from traditional optimization theory.

In the special case where only the largest Hessian eigenvalue crosses $2/\eta$ (e.g. steps 3000-4500 in Figure 3), which was analyzed in Damian et al. (2023), the EOS dynamics consist of repeated cycles in which: (a) the sharpness rises above $2/\eta$; (b) this triggers growing oscillations along the top Hessian eigenvector; (c) such oscillations reduce sharpness via eq. (1), pushing it below $2/\eta$; (d) the oscillations consequently shrink in magnitude. In the more typical case when *multiple* Hessian eigenvalues have reached $2/\eta$ (e.g. steps 4500-5500 in Figure 3), gradient descent oscillates along all the corresponding eigenvectors, causing all such eigenvalues to stay regulated around $2/\eta$.

While the *fine-grained* EOS dynamics are difficult to analyze, we will now show that a simple time-averaging argument allows us to tractably characterize the *macroscopic* path taken by gradient descent. This analysis will recover the main result of Damian et al. (2023) (albeit non-rigorously), while also generalizing to the more realistic and challenging setting of multiple oscillating directions.

### 3.2 DERIVING THE GRADIENT DESCENT CENTRAL FLOW

So long as the sharpness $S(w_t)$ remains below the critical threshold $2/\eta$, gradient descent is said to be *stable*. While gradient descent is stable, it does not exhibit sustained oscillations, and its trajectory is empirically well-approximated[1] by that of gradient flow:[2] $\frac{dw}{dt} = -\eta\nabla L(w)$; however, once gradient descent enters the EOS regime, its trajectory rapidly departs from that of gradient flow (Cohen et al., 2021). We will now derive a more general ODE, which we call a *central flow*, which models the *time-averaged* trajectory of gradient descent in both the stable and EOS regimes. We will derive the central flow using a heuristic time-averaging argument, and we will empirically show that it can accurately predict gradient descent trajectories in a variety of neural network settings.

> We abuse notation and use $\mathbb{E}$ to denote "local time-averages" of deterministic quantities — see Appendix A.2. The central flow is intended to model the time-averaged trajectory $\mathbb{E}[w_t]$. To simplify notation, we will also use $\overline{w}_t := \mathbb{E}[w_t]$ to denote this time-averaged trajectory.

#### 3.2.1 THE SPECIAL CASE OF ONE UNSTABLE EIGENVALUE

We will introduce our time-averaging methodology by analyzing the special case when only the largest Hessian eigenvalue has crossed the critical threshold $2/\eta$. In this setting, gradient descent oscillates along a single direction — the top Hessian eigenvector. We will therefore model the GD trajectory by $w_t = \overline{w}_t + x_t u_t$ where $w_t$ is the gradient descent iterate, $\overline{w}_t$ is the time-averaged iterate, $u_t$ is the top Hessian eigenvector at $\overline{w}_t$, and $x_t$ denotes the displacement between $w_t$ and $\overline{w}_t$ along the $u_t$ direction. Note that by definition, $\mathbb{E}[x_t] = 0$, i.e. the time-averaged displacement is zero. To track the evolution of $\overline{w}_t$, we time-average both sides of the gradient descent update and Taylor expand the time-averaged gradient[3] using eq. (1):

$$\overline{w}_{t+1} = \overline{w}_t - \eta\,\mathbb{E}[\nabla L(w_t)] \approx \overline{w}_t - \eta\Big[\nabla L(\overline{w}_t) + \underbrace{S(\overline{w}_t)\,\mathbb{E}[x_t]\,u_t}_{} + \tfrac{1}{2}\mathbb{E}[x_t^2]\nabla S(\overline{w}_t)\Big].$$

Note that the second term in the time-averaged gradient is 0 because $\mathbb{E}[x_t] = 0$. Therefore, we model the time-averaged iterates $\overline{w}_t$ by the sharpness-penalized gradient flow $w(t)$ defined by:

$$\frac{dw}{dt} = -\eta\Big[\nabla L(w) + \underbrace{\tfrac{1}{2}\sigma^2(t)\nabla S(w)}_{\text{implicit sharpness penalty}}\Big]. \qquad (2)$$

Here, $\sigma^2(t)$ is a still-unknown quantity intended to model $\mathbb{E}[x_t^2]$, the instantaneous variance of the oscillations at time $t$. This quantity also controls the strength of the implicit sharpness penalty. To

determine $\sigma^2(t)$, we argue that only one value is consistent with empirical observation. Empirically, the sharpness stays dynamically regulated around $2/\eta$. Therefore, we will enforce that the central flow keeps the sharpness *fixed* at $S(w(t)) = 2/\eta$.

The time derivative of the sharpness under eq. (2) can be easily computed using the chain rule:

$$\frac{dS(w)}{dt} = \left\langle \nabla S(w), \frac{dw}{dt} \right\rangle = \underbrace{\eta \left\langle \nabla S(w), -\nabla L(w) \right\rangle}_{\text{progressive sharpening}} - \underbrace{\tfrac{1}{2}\eta\sigma^2(t)\|\nabla S(w)\|^2}_{\text{sharpness reduction from oscillations}} \tag{3}$$

Solving for $\frac{dS(w)}{dt} = 0$ yields the unique $\sigma^2(t)$ that keeps the sharpness fixed in place:

$$\sigma^2(t) = \frac{2 \left\langle \nabla S(w), -\nabla L(w) \right\rangle}{\|\nabla S(w)\|^2}. \tag{4}$$

Intuitively, this is the unique $\sigma^2(t)$ for which the downward force of oscillation-induced sharpness reduction "cancels out" the upwards force of progressive sharpening so the sharpness is locked at $2/\eta$. The central flow for a single unstable eigenvalue is given by substituting this $\sigma^2(t)$ into eq. (2).

### 3.2.2 THE GENERAL CASE (MULTIPLE UNSTABLE EIGENVALUES)

We now generalize this analysis to the setting where multiple eigenvalues have reached the critical threshold $2/\eta$, and gradient descent oscillates in the span of the corresponding eigenvectors. We assume the displacement $\delta_t := w_t - \overline{w}_t$ between the true process and the time-averaged process lies in the span of these eigenvectors. Note that by definition, $\mathbb{E}[\delta_t] = 0$ because $\overline{w}_t := \mathbb{E}[w_t]$. By taking a cubic Taylor expansion similar to Lemma 1, we obtain a similar ansatz for the central flow:

$$\frac{dw}{dt} = -\eta\Big[ \nabla L(w) + \underbrace{\tfrac{1}{2}\nabla_w \left\langle H(w), \Sigma(t) \right\rangle}_{\text{implicit curvature penalty}} \Big]. \tag{5}$$

Here, $\Sigma(t)$ is a still-unknown quantity intended to model $\mathbb{E}[\delta_t \delta_t^T]$, the instantaneous covariance of the oscillations at time $t$. This matrix also controls the strength and direction of the implicit curvature penalty. As above, we will argue that only one value of $\Sigma(t)$ is consistent with empirical observation. First, since Hessian eigenvalues that have reached $2/\eta$ empirically do not rise further, we impose that the flow eq. (5) should not increase any Hessian eigenvalues beyond $2/\eta$. Second, since gradient descent oscillates within the span of the unstable eigenvectors, we impose that $\Sigma(t)$, which models the covariance of these oscillations, should be supported within the span of the Hessian eigenvectors with eigenvalue $2/\eta$. Third, since $\Sigma(t)$ models a covariance matrix, we impose that $\Sigma(t)$ should be PSD. These three conditions turn out to imply a unique value of $\Sigma(t)$. In particular, we detail in Appendix A.3 that $\Sigma(t)$ must be the unique solution to a type of convex program known as a *cone complementarity problem* (CCP). The central flow for gradient descent is defined as eq. (5) with this value of $\Sigma(t)$ (Definition 2). This central flow can also be interpreted as a *projected* gradient flow on the constrained problem: $\min_w L(w)$ such that $S(w) \le 2/\eta$, as we detail in Appendix A.3.3. Note that the central flow automatically reduces to the gradient flow when $S(w) < 2/\eta$.

### 3.2.3 EXPERIMENTAL VERIFICATION

Although this derivation employs informal mathematical reasoning, our experiments demonstrate that this central flow can successfully predict long-term optimization trajectories on a variety of neural networks with a high degree of numerical accuracy. For example, in Figure 3, we run gradient descent side by side with the central flow, and monitor the distance between the two trajectories in weight space. Observe that the distance between these two trajectories stays small, whereas the distance between the gradient descent and gradient *flow* trajectories grows large. Moreover, observe that the central flow's $\Sigma(t)$ accurately predicts the *covariance* with which gradient descent oscillates around the central flow; in particular, each eigenvalue of $\Sigma(t)$ accurately predict the instantaneous variance of oscillations along the corresponding eigenvector of $\Sigma(t)$ (see Figure 11 for more explanation).

Appendices B and C.1 show our full set of experiments, and Appendix B.4 discusses the factors that affect the quality of the central flow approximation. A promising direction for future research is to rigorously identify the conditions under which gradient descent follows the central flow.

### 3.3 INTERPRETING GRADIENT DESCENT VIA ITS CENTRAL FLOW

As a smooth flow, the central flow is an simpler object to reason about than the oscillatory gradient descent trajectory. For example, the training loss $L(w_t)$ along the gradient descent trajectory behaves non-monotonically, as shown in Figure 4.

This led Cohen et al. (2021) to observe that there was no explanation for why the loss still decreases over longer timescales. The central flow provides a simple way to reason about this phenomenon.[4] Because the central flow is a smooth curve, the chain rule quantifies the rate of loss decrease: $\frac{dL(w)}{dt} = \langle \nabla L(w), \frac{dw}{dt} \rangle$. Using this, we prove in Proposition 1 that $\frac{dL(w)}{dt} \le 0$, i.e. that the central flow loss $L(w(t))$ is monotonically decreasing. Thus, the central flow loss is a **hidden progress metric** for the optimization process.

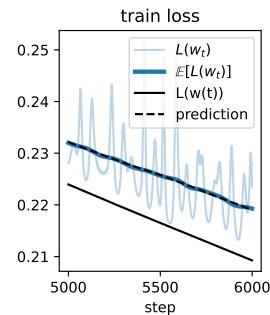

Figure 4: Train loss curve

As visible in Figure 4, the gradient descent train loss $L(w_t)$ is generally higher than the central flow train loss $L(w(t))$, due to the oscillations. However, since the central flow also models the oscillation covariance $\Sigma(t)$, it can render predictions for the *time-averaged* train loss (Figure 4):

$$\mathbb{E}[L(w_t)] \approx L(w(t)) + \tfrac{1}{2} \langle H(w(t)), \Sigma(t) \rangle = \underbrace{L(w(t))}_{\text{central flow loss}} + \underbrace{\tfrac{1}{2} S(w(t)) \operatorname{tr}(\Sigma(t))}_{\text{effect of oscillations}}. \tag{6}$$

Thus, the central flow allows us to decompose the time-averaged training loss curve into the loss along the central flow, which decreases monotonically, plus a contribution from the oscillations.

## 4  SCALAR RMSPROP

As a stepping stone to RMSProp, we now study "Scalar RMSProp," a simple adaptive optimizer which uses one global adaptive step size, rather than separate adaptive step sizes for each coordinate:

$$\nu_t = \beta_2 \nu_{t-1} + (1 - \beta_2) \|\nabla L(w_t)\|^2, \quad w_{t+1} = w_t - \tfrac{\eta}{\sqrt{\nu_t}} \nabla L(w_t). \qquad \text{(Scalar RMSProp)}$$

The algorithm maintains an exponential moving average (EMA), $\nu$, of the squared gradient norm, and takes gradient steps of size $\eta / \sqrt{\nu}$, which we call the *effective step size*.[56] The hyperparameter $\beta_2$ interpolates the algorithm from GD when $\beta_2 = 1$ to normalized GD (NGD) when $\beta_2 = 0$.[7]

Our analysis in this section will make precise how Scalar RMSProp adapts its effective step size to the local curvature, and will reveal that Scalar RMSProp's efficacy as an optimizer relies on a hidden ability to implicitly steer towards low-curvature regions in which it can take larger steps.

### 4.1  THE DYNAMICS OF SCALAR RMSPROP

The dynamics of Scalar RMSProp revolve around the *effective sharpness*, defined as $S^{\text{eff}} := \eta S(w) / \sqrt{\nu}$. First, the effective sharpness controls the oscillations: when $S^{\text{eff}} > 2$, Scalar RM-SProp oscillates with growing magnitude along high curvature direction(s). Second, such oscillations in turn trigger a reduction of effective sharpness. This occurs via a combination of two distinct mechanisms. One mechanism, shared with gradient descent, is that oscillations implicitly reduce sharpness due to Equation (1), thereby decreasing the effective sharpness via its *numerator*. The other mechanism, new to Scalar RMSProp, is that oscillations increase the gradient norm and hence $\nu$, thereby decreasing effective sharpness via its *denominator*. These dynamics give rise to a negative feedback loop that keeps the effective sharpness automatically regulated around 2, as depicted in Figure 5. The fine-grained dynamics are challenging to analyze, even in the case of a single oscillatory direction. Fortunately, we will see the *time-averaged* dynamics are more tractable.

### 4.2  DERIVING THE CENTRAL FLOW

Recall that while gradient descent trains stably, it is well-approximated by gradient flow. One can derive an analogous "stable flow" for Scalar RMSProp (Ma et al., 2022, cf.):[8]

$$\tfrac{d}{dt} w(t) = -\tfrac{\eta}{\sqrt{\nu(t)}} [\nabla L(w(t))], \quad \tfrac{d}{dt} \nu(t) = \tfrac{1 - \beta_2}{\beta_2} \left[ \|\nabla L(w(t))\|^2 - \nu(t) \right]. \tag{7}$$

However, at the edge of stability, the trajectory of Scalar RMSProp deviates from eq. (7). We will now derive a more general central flow that characterizes the time-averaged trajectory even at EOS. In the main text, we will focus on the case where one eigenvalue is unstable.

In section 3.2.1, we derived an approximation for the time-averaged gradient, $\mathbb{E}[\nabla L(w)]$. Using the first two terms of eq. (1), we can also derive a time-averaged approximation for $\mathbb{E}[\|\nabla L(w)\|^2]$:

$$\mathbb{E}[\|\nabla L(w)\|^2] \approx \|\nabla L(\overline{w})\|^2 + 2 \underbrace{\langle \nabla L(\overline{w}), u \rangle}_{} S(\overline{w}) \mathbb{E}[x] + S(\overline{w})^2 \mathbb{E}[x^2]$$

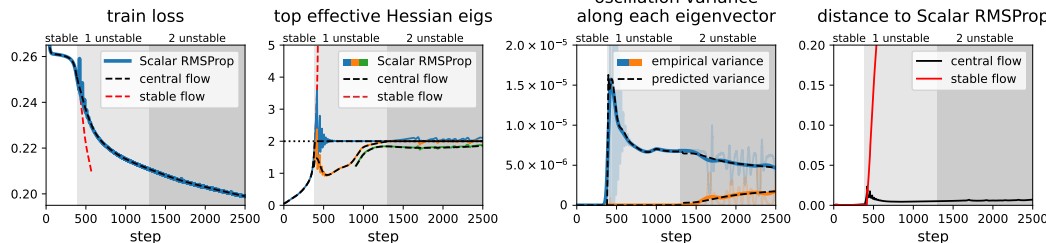

Figure 5: **Central flow for Scalar RMSProp.** A Mamba is trained on a synthetic sequence prediction task using Scalar RMSProp with $\eta = 2/400$ and $\beta_2 = 0.99$ (blue). The central flow (black) accurately models the time-averaged trajectory of Scalar RMSProp even at the edge of stability, whereas the naive stable flow (red) follows a different path. As with gradient descent, our analysis can predict the exact covariance $\Sigma$ with which Scalar RMSProp oscillates around the central flow (third panel).

where we again used $\mathbb{E}[x] = 0$ to ignore the middle term. Based on these time averages, we make the ansatz that the joint dynamics of $(w_t, \nu_t)$ follow a central flow $(w(t), \nu(t))$ of the form:

$$\frac{dw}{dt} = -\frac{\eta}{\sqrt{\nu}}\Big[\underbrace{\nabla L(w(t)) + \tfrac{1}{2}\sigma^2(t)\nabla S(w)}_{\mathbb{E}[\nabla L(w_t)]}\Big], \quad \frac{d\nu}{dt} = \frac{1-\beta_2}{\beta_2}\Big[\underbrace{\|\nabla L(w)\|^2 + S(w)^2\sigma^2(t)}_{\mathbb{E}[\|\nabla L(w_t)\|^2]} - \nu\Big] \quad (8)$$

where $\sigma^2(t)$ is a still-unknown quantity intended to model $\mathbb{E}[x_t^2]$, the instantaneous variance of the oscillations. As in our analysis of gradient descent, there is a unique value of $\sigma^2(t)$ that maintains $S^{\text{eff}}(w, \nu) = 2$. To compute it, we expand $\frac{dS^{\text{eff}}}{dt}$ using the chain rule: $\frac{dS^{\text{eff}}}{dt} = \langle \frac{\partial S^{\text{eff}}}{\partial w}, \frac{dw}{dt}\rangle + \frac{\partial S^{\text{eff}}}{\partial \nu} \cdot \frac{d\nu}{dt}$. Plugging in $\frac{dw}{dt}, \frac{d\nu}{dt}$ from eq. (8) shows that $\frac{dS^{\text{eff}}}{dt}$ is linear in $\sigma^2$. Thus, there is a unique value of $\sigma^2$ that will ensure $\frac{dS^{\text{eff}}}{dt} = 0$, which is given by:

$$\sigma^2(w; \eta, \beta_2) = \frac{\beta_2 \overbrace{\langle -\nabla L(w), \nabla S(w)\rangle}^{\text{progressive sharpening}} + (1-\beta_2)\overbrace{\big[S(w)^2/4 - \|\nabla L(w)\|^2/\eta^2\big]}^{\text{effect of mean reversion on }\nu}}{\beta_2 \underbrace{\tfrac{1}{2}\|\nabla S(w)\|^2}_{\text{sharpness reduction}} + (1-\beta_2)\underbrace{S(w)^2/\eta^2}_{\text{effect of oscillation on }\nu}}. \quad (9)$$

The central flow for Scalar RMSProp with a single unstable eigenvalue is given by eq. (8) with this value of $\sigma^2$. In Appendix A.4 we extend this flow to the case of multiple oscillating directions. In Figure 5 and Appendix C.2 we validate this flow empirically.

### 4.3 Interpreting Scalar RMSProp via its Central Flow

We now interpret the Scalar RMSProp central flow to shed light on the behavior of the algorithm and the function of its hyperparameters $\eta$ and $\beta_2$. Because the dynamics usually transition from stable to EOS quite early in training, we focus on interpreting the central flow in the EOS regime.[9]

#### 4.3.1 Implicit step size selection

The central flow renders *explicit* the step size strategy that is *implicit* in the oscillatory dynamics of Scalar RMSProp. Recall that while the central flow is at EOS, the effective sharpness $S^{\text{eff}} := \eta S(w)/\sqrt{\nu}$ is fixed at 2. This condition can be rearranged into a statement about the effective step size: $\eta/\sqrt{\nu} = 2/S(w)$. Notably, the value $2/S(w)$ is the *maximal locally stable step size* for the current location $w$ in weight space. In other words, while the algorithm is at EOS, the oscillatory dynamics continually adapt the effective step size to the current maximal stable step size (figure on right). This is the precise sense in which Scalar RMSProp "adapts" to the local loss landscape.

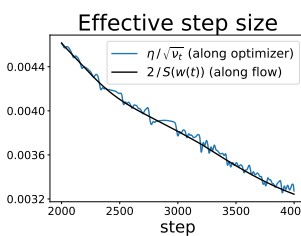

Notice that while Scalar RMSProp only accesses the loss function via first-order *gradients*, its oscillatory dynamics implicitly adapt the effective step size to the *curvature*.

Figure 6: **"Acceleration via regularization" for Scalar RMSProp**. Starting from the same initialization, we run the Scalar RMSProp central flow at two different learning rates (in blue and orange), as well as an ablated flow $\frac{dw}{dt} = -\frac{2}{S(w)}\nabla L(w)$ (in black) with curvature regularization removed. These three flows all use the same step size strategy but differ in the strength of implicit curvature regularization. Initially (see inset), the flows with higher curvature regularization optimize slower; however, over the longer run, they take larger steps and optimize faster (Mamba / MSE).

### 4.3.2 Implicit curvature reduction

Understanding the implicit step size strategy employed by Scalar RMSProp is not sufficient to fully characterize the behavior of the algorithm. To do so, we need to return to the central flow, which additionally accounts for the curvature regularization induced by oscillations. In general, the Scalar RMSProp central flow is a joint flow over $(w, \nu)$. However, at EOS, because $\eta/\sqrt{\nu} = 2/S(w)$, we can eliminate $\nu$ from the expression for $\frac{dw}{dt}$, and write the central flow in terms of $w$ alone:[10]

$$\frac{dw}{dt} = -\underbrace{\frac{2}{S(w)}}_{\text{effective step size}}\big[\nabla L(w) + \underbrace{\tfrac{1}{2}\sigma^2(w; \eta, \beta_2)\nabla S(w)}_{\text{implicit sharpness penalty}}\big] \tag{10}$$

where $\sigma^2(w; \eta, \beta_2)$ is given by eq. (9). In other words, the time-averaged trajectory of Scalar RMSProp at EOS is essentially equivalent to that of the following simpler-to-understand algorithm: at each iteration, compute the sharpness $S(w)$, and take a gradient step of size $2/S(w)$ on a sharpness-regularized objective, where the strength of the sharpness regularizer is given by eq. (9).

Notably, the hyperparameters $\eta, \beta_2$ are not used to determine the effective step size. Instead, their only role is to modulate $\sigma^2$, which controls the strength of the implicit sharpness penalty. The effect of the learning rate hyperparameter $\eta$ is to *monotonically increase* $\sigma^2$ — indeed, the numerator of eq. (9) is increasing in $\eta$ while the denominator is decreasing in $\eta$, which implies the overall expression for $\sigma^2$ is increasing in $\eta$.[11] Meanwhile, the effect of the hyperparameter $\beta_2$ is to monotonically interpolate $\sigma^2$ between that of NGD when $\beta_2 = 0$ and that of gradient descent when $\beta_2 = 1$.[12] The interpretations of $\eta, \beta_2$ generalize to the setting of multiple oscillating directions, as detailed in Lemma 3.

### 4.3.3 Acceleration via regularization

To fully grasp the *modus operandi* of Scalar RMSProp, it is necessary to consider the link between step size adaptation and curvature regularization. By regularizing sharpness $S(w)$, Scalar RMSProp is able to steer itself towards regions where the maximal locally stable step size of $2/S(w)$ is larger. In such regions, Scalar RMSProp can and does take larger steps. Thus, by regularizing sharpness, Scalar RMSProp enables larger steps later in training. We call this mechanism *acceleration via regularization*. Our experiments suggest that this mechanism is a critical component of the algorithm's effectiveness. In Figures 6 and 15, we compare the Scalar RMSProp central flow to an ablated version which adapts the step size to $2/S(w)$ but does not regularize sharpness. Over the long term, this ablated flow optimizes slower than the Scalar RMSProp central flow, because it traverses very sharp regions of weight space in which it is forced to take small steps.

The mechanism of "acceleration via regularization" is also key for understanding the function of the learning rate hyperparameter $\eta$. We have seen that at EOS, the only direct effect of $\eta$ on the central flow is to modulate the strength of sharpness regularization, with higher $\eta$ inducing stronger sharpness regularization. Thus, counterintuitively, the *instantaneous* effect of a higher $\eta$ is often to *slow down* optimization. However, as we illustrate in Figures 6 and 15, over longer timescales, higher $\eta$ steers the trajectory into lower-sharpness regions, in which Scalar RMSProp's effective step size will be larger, thereby tending to *speed up* optimization.

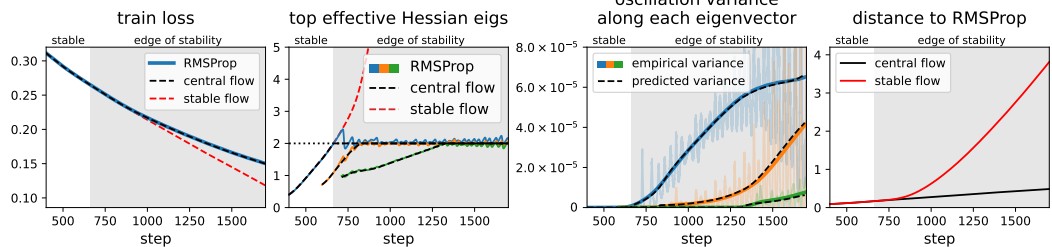

Figure 7: **Central Flow for RMSProp.** A ResNet is trained on a subset of CIFAR-10 using RMSProp with $\eta = 2 \times 10^{-5}$ and $\beta_2 = 0.99$ (blue). The central flow (black) accurately models the time-averaged trajectory of RMSProp even at EOS, whereas the naive stable flow (red) follows a different path. As for gradient descent and Scalar RMSProp, we are able to predict the exact covariance $\Sigma$ with which RMSProp oscillates around the central flow (third panel).

## 5   RMSPROP

We now study RMSProp (Tieleman & Hinton, 2012), which maintains an EMA $\nu$ of the elementwise squared gradients $\nabla L(w)^{\odot 2}$, and uses *per-coordinate* effective step sizes of $\eta / \sqrt{\nu}$ :[13]

$$\nu_t = \beta_2 \nu_{t-1} + (1 - \beta_2) \nabla L(w_t)^{\odot 2}, \quad w_{t+1} = w_t - \frac{\eta}{\sqrt{\nu_t}} \odot \nabla L(w_t), \qquad \text{(RMSProp)}$$

where $\odot$ represents the entrywise product. It is useful to view RMSProp as preconditioned gradient descent $w_{t+1} = w_t - P_t^{-1} \nabla L(w_t)$ with the dynamic preconditioner $P_t := \text{diag}(\sqrt{\nu_t}/\eta)$.

In this section, we will show that RMSProp's preconditioner is *implicitly* determined by the algorithm's oscillatory dynamics, and we will make this preconditioner *explicit* for the first time. Interpreting this preconditioner sheds light on its efficacy, while also revealing a potential direction for future improvement. Finally, we will show that adaptive preconditioning is not the full story: RMSProp also implicitly regularizes curvature, and this behavior is important for its success.

### 5.1   THE DYNAMICS OF RMSPROP; DERIVING A CENTRAL FLOW

The dynamics of RMSProp revolve around the *effective sharpness* $S^{\text{eff}}(w; \nu)$, defined as the largest eigenvalue of the preconditioned Hessian $P^{-1} H(w)$ where $P = \text{diag}(\sqrt{\nu}/\eta)$.[14] When $S^{\text{eff}} > 2$, the iterates oscillate along the top right eigenvector of the preconditioned Hessian. In turn, such oscillations reduce $S^{\text{eff}}$, via a combination of two mechanisms: (1) they implicitly reduce curvature, and (2) they increase the gradient, growing $\nu$ and therefore $P$. The net effect is that the effective sharpness $S^{\text{eff}}$ stays regulated around 2 throughout training (Cohen et al., 2022).[15]

In Appendix A.5 we derive a central flow $(w(t), \nu(t))$, in the same way as above, which models the time-averaged trajectory of RMSProp. We verify its accuracy in Figure 7 and Appendix C.3.

### 5.2   INTERPRETING RMSPROP VIA ITS CENTRAL FLOW

We now interpret the RMSProp central flow to understand the behavior of RMSProp. Because the dynamics usually transition from stable to EOS early in training, we focus on the EOS regime.

#### 5.2.1   THE STATIONARY PRECONDITIONER

The central flow for RMSProp is harder to interpret than that for Scalar RMSProp, because even at EOS, $\nu$ cannot be expressed as a closed-form function of $w$, and instead remains an independent variable.[16] Nevertheless, it turns out that, in some circumstances, $\nu$ implicitly converges under the dynamics of RMSProp to a value that depends on the current $w$ alone. Namely, imagine holding the weights fixed at some value $w$, and letting $\nu$ "catch up." What value would $\nu$ converge to? We show in Proposition 3 that for any $w$, there is a unique $\nu$ that satisfies the stationarity condition $\frac{d\nu}{dt} = 0$. We call this the *stationary $\nu$* for the weights $w$, denoted as $\bar{\nu}(w)$. Empirically, we observe that $\nu(t)$ usually converges to its stationary value $\bar{\nu}(w(t))$ during training (Figure 8; Appendix B.3).

In Proposition 2, we show that the corresponding *stationary preconditioner* $\overline{P}(w) = \text{diag}(\sqrt{\nu(w)}/\eta)$ is, remarkably, the optimal solution to a convex optimization problem over preconditioners:

$$\overline{P}(w) \quad := \quad \underset{P \text{ diagonal}, \, P \succeq 0}{\arg\min} \quad \text{tr}(P) + \frac{1}{\eta^2} \underbrace{\|\nabla L(w)\|_{P^{-1}}^2}_{\text{optimization speed}} \quad \text{such that} \quad \underbrace{H(w) \preceq 2P}_{\text{local stability}}. \tag{11}$$

That is, **RMSProp implicitly solves the convex program eq. (11) to compute its preconditioner**. This is the precise sense in which RMSProp "adapts" to the local loss landscape.

We can now understand RMSProp's preconditioning strategy by interpreting the optimization problem eq. (11). The constraint $H(w) \preceq 2P$ is equivalent to $S^{\text{eff}} \leq 2$ and hence stipulates that the preconditioner $P$ should keep RMSProp locally stable. The first term of the objective, $\text{tr}(P)$, is the sum of the inverse effective step sizes. If this were the only term in the objective, RMSProp's preconditioning strategy could be summarized as maximizing the *harmonic mean* of the effective step sizes while maintaining local stability — a sensible preconditioning strategy. In fact eq. (11) would reduce to the dual of the max-cut SDP (Goemans & Williamson, 1995).

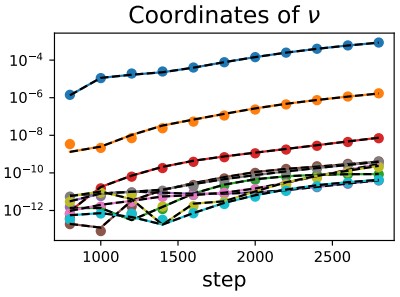

Coordinates of $\nu$

Figure 8: Across 10 coordinates $i$ (colors), the EMA $\nu(t)_i$ (dots) matches its stationary value $\hat{\nu}(w(t))_i$ (dashed).

However, matters are complicated by the presence of the second term in the eq. (11) objective. The quantity $\|\nabla L(w)\|_{P^{-1}}^2$ is the instantaneous rate of loss decrease under preconditioned gradient flow with preconditioner $P$. Minimizing this term actually *slows down* optimization (Figures 24 and 25). Therefore, the presence of this term implies that RMSProp's preconditioning strategy is not optimal.

### 5.2.2 IMPLICIT CURVATURE REDUCTION AND ACCELERATION VIA REGULARIZATION

Understanding the preconditioning strategy implicitly employed by RMSProp is not sufficient to fully characterize the behavior of the algorithm. To do so, we need to return to the central flow, which additionally accounts for the implicit curvature reduction induced by the oscillations. Substituting $\overline{P}$ into the RMSProp central flow, we can obtain a *stationary flow* in terms of $w$ alone, which assumes that the preconditioner $P$ is always fixed at its stationary value $\overline{P}$ (eq. 11):

$$\frac{dw}{dt} \quad = \quad - \underbrace{\overline{P}(w)^{-1}}_{\text{stationary preconditioner}} \Big[ \nabla L(w) + \underbrace{\tfrac{1}{2} \nabla_w \langle \Sigma, H(w) \rangle}_{\text{implicit curvature penalty}} \Big]. \tag{12}$$

where $\Sigma = \Sigma(w; \eta; \beta_2)$ is defined as the solution to a cone complementarity problem (eq. 24). Figure 26 and Figure 27 show that this stationary flow can accurately predict the instantaneous optimization speed of the central flow, given only access to $w(t)$ and not $\nu(t)$.

Empirically, we find that the implicit curvature reduction effect is beneficial for the efficacy of the optimizer. In Figure 19, we compare the central flow against an ablated flow that leaves out the implicit curvature penalty. We find that in the long run, this ablated flow navigates into sharper regions in which it takes smaller steps, and optimizes slower.

## 6 CONCLUSION

In this paper, we have developed a methodology for analyzing deep learning optimizers. To analyze an optimization algorithm, we derive a *central flow* which models the optimizer's time-averaged trajectory, rendering explicit what was implicit in the oscillatory dynamics. We have empirically shown that these central flows can accurately predict long-term optimization trajectories of neural networks, and by interpreting these flows we have obtained new insights about optimizers' behavior.

These advances are made possible by the fact that we adopt different goals from most works in optimization. Rather than try to characterize global convergence rates, we set ourselves the more modest goal of characterizing the *local* optimization dynamics throughout training. The local dynamics are important, they are more interesting than may have been assumed (even vanilla gradient descent gives rise to rich, complex dynamics), and they are empirically consistent across different deep learning settings, which suggests that a general theory is feasible. We believe that similar analyses can be fruitfully conducted for other optimizers, and we hope to inspire work in that direction.

## 7 ACKNOWLEDGEMENTS

AD acknowledges support from an NSF Graduate Research Fellowship and a Jane Street Graduate Research Fellowship. JDL acknowledges support of Open Philanthropy, NSF IIS 2107304, NSF CCF 2212262, NSF CAREER Award 2144994, and NSF CCF 2019844. AT acknowledges support from National Science Foundation grants IIS1705121, IIS1838017, IIS2046613, IIS2112471, and funding from Meta, Morgan Stanley, Amazon, Google, and Scribe. JC would like to thank Jim and Marilyn Simons for their support of basic research via the Flatiron Institute. Any opinions, findings and conclusions or recommendations expressed in this material are those of the author(s) and do not necessarily reflect the views of any of these funding agencies.

The authors are grateful for extensive feedback from Zhili Feng, Nikhil Ghosh, Yiding Jiang, and Sam Sokota.

**Reproducibility statement** All code is available at `http://github.com/locuslab/central_flows`.

**Ethics statement** This paper presents work whose goal is to advance the field of machine learning. As such, there are many potential societal consequences of our work.

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

TABLE OF CONTENTS

# A   CENTRAL FLOW DERIVATIONS

We will now derive the general central flows for gradient descent, Scalar RMSProp, and RMSProp. We begin by defining the necessary tensor notation.

## A.1   TENSOR NOTATION

For a $k$-tensor $T$ and a $s$-tensor $A$ with $s < k$ we define $T[A]$ to be the contraction of $T$ with $A$ along its last $s$ indices, i.e.

$$(T[A])_{i_1,\dots,i_{k-s}} = \sum_{j_1,\dots,j_s} T_{i_1,\dots,i_{k-s},j_1,\dots,j_s} A_{j_1,\dots,j_s}. \tag{13}$$

We also define $\langle T, T' \rangle$ for two $k$-tensors $T, T'$ of the same shape by:

$$\langle T, T' \rangle = \sum_{i_1,\dots,i_k} T_{i_1,\dots,i_k} T'_{i_1,\dots,i_k}. \tag{14}$$

We will also use $T[u_1, \dots, u_j]$ to denote $T[u_1 \otimes \cdots \otimes u_j]$.

## A.2   ON LOCAL TIME AVERAGING

We intentionally do not specialize to a specific notion of "local time-average". The only properties of the local time-averaging operator $\mathbb{E}$ that we use are:

1. linearity, i.e. $\mathbb{E}[f + g] = \mathbb{E}[f] + \mathbb{E}[g]$ and $\mathbb{E}[cf] = c\,\mathbb{E}[f]$ for any constant $c$

2. the local time average of a constant $c$ is itself: $\mathbb{E}[c] = c$

3. in the EOS regime when the sharpness oscillates around $2/\eta$, the time-average is coarse enough to smooth out these oscillations so that $S(\mathbb{E}[w_t]) = 2/\eta$

When we empirically verify the accuracy of our central flows, we use Gaussian smoothing with a standard deviation of $\tau$ steps, i.e. we define

$$\mathbb{E}[f_t] := \frac{\sum_s f_s c_{t-s}}{\sum_s c_{t-s}} \quad \text{where} \quad c_j := \exp\left(\frac{-j^2}{2 \times \tau^2}\right).$$

## A.3   GRADIENT DESCENT

We begin by proving the Taylor expansion used in Section 3.1:

**Lemma 1.** *Assume that $\overline{w}$ is such that the top eigenvalue of $H(\overline{w})$ has multiplicity $1$ and let $w = \overline{w} + xu$ where $u$ is the top eigenvector of $H(\overline{w})$. Furthermore, assume that $\sup_{w' \in B} \left\| \nabla^4 L(w') \right\| < \infty$ for some open neighborhood $B$ of $\overline{w}$. Then,*

$$\nabla L(w) = \nabla L(\overline{w}) + S(\overline{w})xu + \tfrac{x^2}{2} \nabla S(\overline{w}) + \mathcal{O}(x^3).$$

*Proof.* By Taylor's theorem,

$$\nabla L(w) = \nabla L(\overline{w}) + H(\overline{w})xu + \tfrac{x^2}{2} \nabla^3 L(\overline{w})[u, u] + \mathcal{O}(x^3).$$

Because $u$ is an eigenvector of $H(\overline{w})$ with eigenvalue $S(\overline{w})$, the second term can be simplified to $S(\overline{w})xu$. Finally, by Danskin's theorem (or equivalently the standard formula for the derivative of an eigenvalue):

$$\nabla_{\overline{w}} S(\overline{w}) = \nabla_{\overline{w}}\left[ \max_{\|v\|=1} v^T H(\overline{w}) v \right] = \nabla_{\overline{w}}\left[ u^T H(\overline{w}) u \right] = \nabla^3 L(\overline{w})[u, u]$$

where $u$ is the argmax of the second expression, i.e. the top eigenvector of the Hessian at $\overline{w}$.   $\square$

We now define some notation that will be necessary for the remainder of the section. First, let $U(w) := \ker[\eta H(w) - 2I]$ denote the *critical subspace*, i.e. the directions in weight space that are at the edge of stability. Let $\mathrm{Sym}(U(w)) \subseteq \mathbb{R}^{d \times d}$ denote the subspace of symmetric matrices whose span is contained in $U(w)$.

### A.3.1 THE DIFFERENTIAL VARIATIONAL INEQUALITY FORMULATION

As in the main text, we will assume that GD oscillates around the central flow $w(t)$ with covariance $\Sigma(t) \in \mathrm{Sym}(U(w))$ so that it follows:

$$\frac{dw}{dt} = -\eta\big[\nabla L(w) + \tfrac{1}{2}\nabla^3 L(w)[\Sigma(t)]\big]. \tag{15}$$

To determine $\Sigma$, we impose three conditions for all times $t$:

- **non-negativity:** As a covariance matrix, $\Sigma(t)$ is positive semidefinite, i.e. $\Sigma(t) \succeq 0$

- **stability:** The sharpness remains bounded by $2/\eta$, i.e. $H(w(t)) \preceq (2/\eta)I$

- **complementarity:** The span of $\Sigma(t)$ (i.e. the span of the oscillations) is contained within the critical subspace $U(w(t))$. Equivalently, $\Sigma(t) \in \mathrm{Sym}(U(w(t)))$.

We say that $(w(t), \Sigma(t))$ follow the GD central flow if they follow eq. (15) along with these conditions:

**Definition 1** (GD Central Flow, DVI Formulation). We say that $\{w(t), \Sigma(t)\}_{t \geq 0}$ follow the GD central flow if for almost all $t$ they satisfy eq. (15) along with the conditions: $\Sigma(t) \succeq 0$, $\eta H(w(t)) \preceq 2I$, and $\Sigma(t) \in \mathrm{Sym}(U(w))$.

Definition 1 is an example of a *differential variational inequality* (DVI). It provides conditions that the central flow must satisfy but it is not a-priori clear that such a flow exists or is unique. To prove this, and to turn Definition 1 into a form that we can simulate numerically, we will show that for almost all times $t$, $\Sigma(t)$ solves a low dimensional convex *cone complementarity program* (CCP) This CCP will be guaranteed to have a unique solution, which can be easily computed numerically.

### A.3.2 THE CONE COMPLEMENTARITY PROBLEM FORMULATION

We will start with the DVI formulation, Definition 1, and prove two additional conditions on $\Sigma(t)$ which will allow us to solve for $\Sigma(t)$. For simplicity, we will look for a choice of $\Sigma(t)$ is right-continuous so that we can reason about the immediate future using the chain rule. First, we consider the change in the Hessian along the central flow ansatz eq. (15), restricted to the critical subspace. We will denote this quantity by $\dot{H}(t) \in \mathrm{Sym}(U(w(t)))$. This generalizes the computation of $\frac{dS}{dt}$ in Section 3. To compute $\dot{H}$, we define the linear operator $\mathcal{T}_w : \mathrm{Sym}(U(w)) \to \mathbb{R}^d$ by

$$\mathcal{T}_w[\Sigma] := \nabla_w \langle H(w), \Sigma \rangle \quad \forall\, \Sigma \in \mathrm{Sym}(U(w)).$$

Intuitively, $\mathcal{T}_w$ will play a role analogous to that of $\nabla S(w)$ in Section 3. The operator $\mathcal{T}_w : \mathrm{Sym}(U(w)) \to \mathbb{R}^d$ takes as input a $d \times d$ matrix $\Sigma \in \mathrm{Sym}(U(w))$ and returns the gradient of $\langle H(w), \Sigma \rangle$, the $\Sigma$-weighted Hessian. Meanwhile, its transpose $\mathcal{T}_w^\top : \mathbb{R}^d \to \mathrm{Sym}(U(w))$ takes as input a direction, and returns the directional derivative of the Hessian, restricted to the critical subspace. Therefore, $\frac{dw}{dt} = -\eta\big[\nabla L(w) + \tfrac{1}{2}\mathcal{T}_w[\Sigma]\big]$, and $\dot{H}$ is given by:

$$\dot{H}(t) := \left.\frac{dH}{dt}\right|_{U(w)} = \mathcal{T}_w^T\left[\frac{dw}{dt}\right] = \underbrace{\mathcal{T}_w^T[-\nabla L(w)]}_{=:\alpha(w)} - \underbrace{\tfrac{1}{2}\mathcal{T}_w^T \mathcal{T}_w}_{=:\beta(w)}[\Sigma].$$

To simplify notation, we define $\alpha(w), \beta(w)$ by:

$$\alpha(w) := \mathcal{T}_w^T[-\nabla L(w)] \in \mathrm{Sym}(U(w)), \quad \beta(w) := \tfrac{1}{2}\mathcal{T}_w^T \mathcal{T}_w \in \mathrm{Sym}(U(w)) \otimes \mathrm{Sym}(U(w)),$$

so that $\dot{H} = \alpha - \beta[\Sigma]$. In order to maintain stability, i.e. $H \preceq (2/\eta)I$, we need to ensure $\dot{H} \preceq 0$, i.e. $\alpha \preceq \beta[\Sigma]$. However, this condition only lower bounds $\Sigma$ and does not yet uniquely determine it. To do so, we will need to differentiate the complementarity condition. Recall that $\dot{H}$ measures the change in the Hessian, restricted to the critical subspace. The directions in the kernel of $\dot{H}$ will remain at EOS. However, the curvature in the directions where $\dot{H} \prec 0$ will drop below $2/\eta$, so these directions will leave the critical subspace. In other words, the critical subspace at time $t + \epsilon$ will be contained in $\ker[\dot{H}]$, which is a subspace of the critical subspace. Because we assumed $\Sigma$ is right-continuous, this implies that $\Sigma(t)$ must lie in this subspace, i.e. $\mathrm{span}[\Sigma] \in \ker[\dot{H}]$. Equivalently, because $\Sigma \succeq 0$ and $\dot{H} \preceq 0$, we can rewrite this condition as $\langle \Sigma, \dot{H} \rangle = 0$. Together, these constraints:

$$\Sigma \succeq 0, \quad \dot{H} \preceq 0, \quad \left\langle \Sigma, \dot{H} \right\rangle = 0, \quad \dot{H} = \alpha - \beta[\Sigma]$$

define a *cone complementarity problem* for $\Sigma$ (Definition 8). We show in Lemma 5 that if $\beta$ is symmetric and has full rank, the solution to this CCP is unique, and we denote it by $\mathrm{CCP}(\alpha, \beta)$. Otherwise it denotes the set of all solutions. We can now define the central flow in terms of this CCP:

**Definition 2** (GD Central Flow, CCP Formulation). We say $\{w(t)\}_{t \geq 0}$ follows the GD central flow if for almost all $t \geq 0$, $w$ satisfies eq. (15) with $\Sigma \in \mathrm{CCP}(\alpha(w), \beta(w))$.

Definition 2 is the simplest version of the central flow to simulate numerically. By picking a basis for $\mathrm{Sym}(U(w))$, which has dimension $\frac{k(k+1)}{2}$, we can materialize the linear operator $\mathcal{T}_w$ as a $\frac{k(k+1)}{2} \times d$ dimensional matrix, compute $\alpha(w), \beta(w)$, solve the low-dimensional CCP to compute $\Sigma$, and take a small Euler step on the flow using this $\Sigma$.

In the next section, we will see that this CCP formulation can be reinterpreted as a projected gradient flow, which will help to interpret the behavior of the gradient descent central flow.

### A.3.3 THE PROJECTION FORMULATION

Let $\mathbb{S}_\eta := \{w : S(w) \leq 2/\eta\}$ denote the *stable set*. In this section we will show that the gradient descent central flow can be reinterpreted as projected gradient flow constrained to this set.

**Definition 3** (GD Central Flow, Projection Formulation). We say that $\{w(t)\}_{t \geq 0}$ follows the gradient descent central flow if for almost all $t$,

$$\frac{dw}{dt} = \mathrm{proj}_{T_{\mathbb{S}_\eta}(w)}[-\eta \nabla L(w)] \quad \text{where} \quad \mathbb{S}_\eta := \{w : S(w) \leq 2/\eta\}, \tag{16}$$

where $\mathrm{proj}_{T_{\mathbb{S}_\eta}(w)}$ denotes the orthogonal projection onto the *tangent cone* of $\mathbb{S}_\eta$ at iterate $w$, i.e. the set of directions that, if followed, would not immediately increase the curvature in the critical subspace: $T_{\mathbb{S}_\eta(w)} = \{v \in \mathbb{R}^d : \mathcal{T}_w^T[v] \preceq 0\}$.

The equivalence between Definition 3 and the CCP formulation Definition 2 ia a consequence of this simple lemma:

**Lemma 2.** *Let* $w \in \mathbb{S}_\eta$. *Then*

$$\mathrm{proj}_{T_{\mathbb{S}_\eta}(w)}[v] = v - \tfrac{1}{2}\mathcal{T}_w[\Sigma] \quad \text{where} \quad \Sigma \in CCP(\mathcal{T}_w^T[v], \tfrac{1}{2}\mathcal{T}_w^T \mathcal{T}_w).$$

*Proof.* Note that the tangent space of $\mathbb{S}_\eta$ is given by the set: $\{v : \mathcal{T}_w^T[v] \preceq 0\}$. Therefore the projection is given by solving the quadratic program:

$$\min_\delta \|\delta\|^2 \quad \text{such that} \quad \mathcal{T}_w^T[v + \delta] \preceq 0.$$

The KKT conditions imply that there exists a $\Sigma$ such that:

$$\delta = -\tfrac{1}{2}\mathcal{T}_w[\Sigma], \quad \langle \Sigma, \mathcal{T}_w^T[v + \delta] \rangle = 0, \quad \Sigma \succeq 0,$$

which implies that $\Sigma \in CCP(\mathcal{T}_w^T[v], \tfrac{1}{2}\mathcal{T}_w^T \mathcal{T}_w)$. $\qquad \square$

We now intuitively reconcile eq. (16) with the CCP definition:

- When $S(w) < 2/\eta$, $w$ is in the interior of $\mathbb{S}_\eta$ so $U(w) = \emptyset$, the tangent cone is the entire space, and the projection is the identity map. Therefore eq. (16) reduces to gradient flow.

- When there is a single eigenvalue at $2/\eta$, $w$ is on the boundary of $\mathbb{S}_\eta$ and the tangent cone is given by the halfspace: $T_{\mathbb{S}_\eta}(w) = \{v : \langle \nabla S(w), v \rangle \leq 0\}$. If the negative gradient lies outside this halfspace (i.e. if gradient flow threatens to increase the sharpness above $2/\eta$), then the projection onto the halfspace is given by the projection onto the hyperplane: $-\eta \Pi_{\nabla S(w)}^\perp \nabla L(w)$. Otherwise, if the negative gradient already lies in the halfspace, the projection is the identity map, so the central flow follows gradient flow and leaves EOS.

- In general, computing the orthogonal projection onto $T_{\mathbb{S}_\eta(w)} = \{v : \mathcal{T}_w^\top[v] \preceq 0\}$ requires solving a semidefinite quadratic program for which $\Sigma$ is the Lagrangian dual variable. The KKT conditions of this quadratic program are equivalent to the CCP that defines $\Sigma$ above.

### A.3.4 THE RATE OF LOSS DECREASE

**Proposition 1.** *Under the GD central flow (definition 3), for almost all $t$ we have*

$$\frac{dL(w)}{dt} = -\eta \left\| \text{proj}_{T_{\mathbb{S}_\eta}(w)}[-\nabla L(w)] \right\|^2.$$

*This implies $\frac{dL}{dt} \leq 0$ (i.e. the loss monotonically decreases) and $\frac{dL}{dt} \geq -\eta \|\nabla L(w)\|^2$ (i.e. the loss decreases at a slower rate than under the gradient flow).*

*Proof.* By the chain rule we have

$$\begin{aligned}
\frac{dL(w)}{dt} &= \left\langle \nabla L(w), \frac{dw}{dt} \right\rangle \\
&= \left\langle \nabla L(w), \text{proj}_{T_{\mathbb{S}_\eta}(w)}[-\eta \nabla L(w)] \right\rangle \\
&= -\eta \left\langle -\nabla L(w), \text{proj}_{T_{\mathbb{S}_\eta}(w)}[-\nabla L(w)] \right\rangle \\
&= -\eta \left\| \text{proj}_{T_{\mathbb{S}_\eta}(w)}[-\nabla L(w)] \right\|^2
\end{aligned}$$

where we used that for any orthogonal projection $\text{proj}[cv] = \text{proj}[c]$ when $c \geq 0$ to pull out the $\eta$, and that $\langle v, \text{proj}[v] \rangle = \|\text{proj}[v]\|^2$ to get the last equality. Finally, the comparison with gradient flow follows from the inequality $\|\text{proj}[v]\| \leq \|v\|$ for any orthogonal projection. □

### A.4 SCALAR RMSPROP

We follow a similar derivation to gradient descent. We define $U(w, \nu) := \ker[\eta \nu^{-1/2} H(w) - 2I]$, and let $\text{Sym}(U(w, \nu))$ denote the subspace of symmetric matrices with span in $U(w, \nu)$. We now proceed with the time-averaging derivation.

If $w = \overline{w} + \delta$ with $\mathbb{E}[\delta] = 0$, we have that

$$\mathbb{E} \|\nabla L(w)\|^2 \approx \mathbb{E} \|\nabla L(\overline{w})\|^2 + \mathbb{E} \|\nabla^2 L(\overline{w})\delta\|^2 = \|\nabla L(\overline{w})\|^2 + \langle \nabla^2 L(\overline{w})^2, \Sigma \rangle \qquad (17)$$

where $\Sigma = \mathbb{E}[\delta \delta^T]$. Because we assume that $\Sigma \in \text{Sym}(U(\overline{w}, \nu))$, $\nabla^2 L(\overline{w})\Sigma = S(\overline{w})\Sigma$, so this expression is equal to

$$\mathbb{E} \|\nabla L(w)\|^2 \approx \|\nabla L(\overline{w})\|^2 + S(\overline{w})^2 \text{tr}(\Sigma).$$

This suggests the central flow ansatz:

$$\begin{aligned}
\frac{dw}{dt} &= -\frac{\eta}{\sqrt{\nu}} \left[ \nabla L(w) + \tfrac{1}{2} \nabla^3 L(w)[\Sigma(t)] \right] \\
\frac{d\nu}{dt} &= \tfrac{1-\beta_2}{\beta_2} \left[ \|\nabla L(w)\|^2 + S(w)^2 \text{tr}(\Sigma(t)) - \nu \right].
\end{aligned} \qquad (18)$$

We can now give the differential variational inequality definition for the Scalar RMSProp central flow:

**Definition 4** (Scalar RMSProp Central Flow, Differential Variational Inequality Formulation)**.** We say that $\{(w(t), \nu(t), \Sigma(t))\}_{t \geq 0}$ satisfy the Scalar RMSProp central flow if they satisfy eq. (18) along with the conditions: $\Sigma(t) \succeq 0$, $\eta \nu(t)^{-1/2} H(w(t)) \preceq 2$, and $\Sigma(t) \in \text{Sym}(U(w(t), \nu(t)))$ for almost all $t$.

As for gradient descent, this definition is fairly opaque and does not give a way of actually computing $\Sigma$. To do so, we will again derive a cone-complementarity problem formulation of the Scalar RMSProp central flow. The first step in this derivation is to differentiate the stability constraint: $\eta \nu^{-1/2} H(w) \preceq 2$. We will fix a time $t$ and use $U$ to denote $U(w, \nu)$ to simplify notation. We will also define the linear operator $\mathcal{T} : \text{Sym}(U(w, \nu)) \to \mathbb{R}^d$ by

$$\mathcal{T}[\Sigma] = \nabla_w \langle H(w), \Sigma \rangle \quad \text{for} \quad \Sigma \in \text{Sym}(U(w, nu)).$$

Let $\dot{H} := \frac{d}{dt}\eta\nu^{-1/2}H(w)\big|_U$ be the time-derivative of the preconditioned Hessian restricted to the critical subspace. In order to avoid violating the stability condition, we need to enforce $\dot{H} \preceq 0$. We can compute $\dot{H}$ using the chain rule.

$$
\begin{aligned}
\dot{H} &:= \eta\frac{d}{dt}\nu^{-1/2}H(w)\bigg|_U \\
&= \eta\nu^{-1/2}\left(\frac{d}{dt}H(w)\right)\bigg|_U + \left(\frac{d}{dt}\nu^{-1/2}\right)\eta H(w)\bigg|_U \\
&= \eta\nu^{-1/2}\mathcal{T}^T\left[\frac{dw}{dt}\right] - \frac{1}{2\nu^{3/2}}\frac{d\nu}{dt}2\sqrt{\nu}I \\
&= \frac{\eta^2}{\nu}\mathcal{T}^T\left[-\nabla L(w) - \tfrac{1}{2}\mathcal{T}[\Sigma]\right] + \frac{1-\beta_2}{\beta_2}\frac{1}{\nu}\left[\nu - \|\nabla L(w)\|^2 - S(w)^2\operatorname{tr}(\Sigma)\right]I.
\end{aligned}
$$

We can now assume that $(w, \nu)$ are at EOS so that $\nu = \frac{\eta^2 S(w)^2}{4}$. Otherwise, complementarity forces $\Sigma = 0$. Substituting this gives:

$$
\begin{aligned}
\dot{H} = \;&\frac{4}{S(w)^2}\mathcal{T}^T\left[-\nabla L(w) - \tfrac{1}{2}\mathcal{T}[\Sigma]\right] \\
&+ \frac{1-\beta_2}{\beta_2}\frac{4}{\eta^2 S(w)^2}\left[\frac{\eta^2 S(w)^2}{4} - \|\nabla L(w)\|^2 - S(w)^2\operatorname{tr}(\Sigma)\right]I.
\end{aligned}
$$

We will now group the constant terms and the terms linear in $\Sigma$. Define:

$$
\begin{aligned}
\alpha(w) &= \frac{4}{S(w)^2}\mathcal{T}^T[-\nabla L(w)] + \frac{1-\beta_2}{\beta_2}\frac{4}{\eta^2 S(w)^2}\left[\frac{\eta^2 S(w)^2}{4} - \|\nabla L(w)\|^2\right]I \\
\beta(w) &= \frac{2}{S(w)^2}\mathcal{T}^T\mathcal{T} + \frac{1-\beta_2}{\beta_2}\frac{4}{\eta^2}I \otimes I.
\end{aligned}
$$

Then $\dot{H} = \alpha - \beta[\Sigma]$. As in the derivation for GD, if $\dot{H}$ is strictly negative definite in some direction, these directions will drop from the critical subspace so right continuity forces $\Sigma \in \ker[\dot{H}]$. This gives us the cone complementarity definition of the Scalar RMSProp central flow:

**Definition 5** (Scalar RMSProp Central Flow, CCP Formulation). We say that $\{(w(t), \nu(t)\}_{t\geq 0}$ follow the Scalar RMSProp central flow if they satisfy eq. (18) where $\Sigma \in CCP(\alpha(w), \beta(w))$.

We can now use this formulation to prove eq. (9) when $U(w, \nu)$ is one dimensional. In the one dimensional case, $\alpha, \beta$ are both scalars and assuming that $\alpha(w) > 0$ the solution to the CCP is simply $\Sigma = \alpha(w)/\beta(w)$. In addition, in this one dimensional case we simply have $\mathcal{T} = \nabla S(w)$. Therefore,

$$
\sigma^2 = \frac{\frac{4}{S(w)^2}\langle -\nabla L(w), \nabla S(w)\rangle + \frac{1-\beta_2}{\beta_2}\frac{4}{\eta^2 S(w)^2}\left[\frac{\eta^2 S(w)^2}{4} - \|\nabla L(w)\|^2\right]}{\frac{2}{S(w)^2}\|\nabla S(w)\|^2 + \frac{1-\beta_2}{\beta_2}\frac{4}{\eta^2}} \tag{19}
$$

$$
= \frac{\beta_2\langle -\nabla L(w), \nabla S(w)\rangle + (1-\beta_2)\left[\frac{S(w)^2}{4} - \frac{\|\nabla L(w)\|^2}{\eta^2}\right]}{\beta_2\frac{1}{2}\|\nabla S(w)\|^2 + (1-\beta_2)S(w)^2/\eta^2}. \tag{20}
$$

### A.4.1 THE EFFECT OF THE HYPERPARAMETERS $\eta, \beta_2$

**Lemma 3.** *Fix an iterate $w = w(t)$ and define $\nu = \nu(w; \eta) := \frac{\eta^2 S(w)^2}{4}$ so that $S^{eff}(w, \nu) = 2$. Let $U$ be the top eigenspace of $H(w)$. Then if $\Sigma(t) \succ 0$ under the Scalar RMSProp central flow (Definition 5), we have that*

$$
\frac{\partial}{\partial\eta}\frac{dH}{dt}\bigg|_U = -cI_U \quad \text{where} \quad C \geq 0
$$

*and where $I_U$ denotes the identity matrix on the subspace $U$. In other words, larger learning rates $\eta$ more aggressively decrease the curvature and they do so uniformly across all eigenvalues in the critical subspace $U$. In addition,*

$$
\frac{\partial}{\partial\beta_2}\frac{dH}{dt}\bigg|_U = -CI_U
$$

*where $C \geq 0$ when $\operatorname{tr}\Sigma\big|_{\beta_2=1} \leq \operatorname{tr}\Sigma\big|_{\beta_2=0}$ and $C \leq 0$ otherwise. Therefore $\beta_2$ can either monotonically increase or decrease the curvature regularization depending on whether gradient descent ($\beta_2 = 1$) or normalized gradient descent ($\beta_2 = 0$) would have a larger oscillation variance.*

*Proof.* The condition $\Sigma(t) \succ 0$ implies that $\dot{H}(t) = 0$ and $\Sigma = \beta^{-1}\alpha$. We will rescale $\alpha, \beta$ by $S(w)^2/4$:

$$\hat{\alpha} = \mathcal{T}^T[-\nabla L(w)] + \frac{1-\beta_2}{\beta_2}\left[\frac{S(w)^2}{4} - \frac{\|\nabla L(w)\|^2}{\eta^2}\right]I$$

$$\hat{\beta} = \tfrac{1}{2}\mathcal{T}^T\mathcal{T} + \frac{1-\beta_2}{\beta_2}\frac{S(w)^2}{\eta^2}I \otimes I.$$

We will also use $\hat{\alpha}_w, \hat{\beta}_w$ to refer to just the first terms in $\hat{\alpha}, \hat{\beta}$:

$$\hat{\alpha}_w = \mathcal{T}^T[-\nabla L(w)] \quad \text{and} \quad \hat{\beta}_w = \tfrac{1}{2}\mathcal{T}^T\mathcal{T}.$$

Then $\Sigma = \hat{\beta}^{-1}\hat{\alpha}$. Differentiating this with respect to $\eta, \beta_2$ gives:

$$\frac{\partial}{\partial\eta}\Sigma = \frac{1-\beta_2}{2\eta^3\beta_2}\hat{\beta}^{-1}[I]\left[\|\nabla L(w)\|^2 + S(w)^2\operatorname{tr}\Sigma\right]$$

$$\frac{\partial}{\partial\beta_2}\Sigma = -\frac{1}{\beta_2^2}\hat{\beta}^{-1}[I]\left[\frac{S(w)^2}{4} - \frac{\|\nabla L(w)\|^2}{\eta^2} - \frac{S(w)^2}{\eta^2}\operatorname{tr}\Sigma\right]$$

Then:

$$\frac{\partial}{\partial\eta}\frac{dH}{dt}\bigg|_U = -\hat{\beta}_w\left[\frac{\partial}{\partial\eta}\Sigma\right], \quad \frac{\partial}{\partial\beta_2}\frac{dH}{dt}\bigg|_U = -\hat{\beta}_w\left[\frac{\partial}{\partial\beta_2}\Sigma\right]$$

Therefore it suffices to compute $\hat{\beta}_w\hat{\beta}^{-1}[I]$. To compute this, we use the Sherman–Morrison formula. Let $c = \frac{1-\beta_2}{\beta_2}\frac{S(w)^2}{\eta^2}$. Then,

$$\hat{\beta}_w\hat{\beta}^{-1}[I] = I\left[1 - \frac{c\beta_w^{-1}[I,I]}{1 + c\beta_w^{-1}[I,I]}\right] = I\left[\frac{1}{1 + c\beta_w^{-1}[I,I]}\right].$$

Finally, note that $\left[\frac{1}{1+c\beta_w^{-1}[I,I]}\right] \geq 0$, which immediately implies the result for $\eta$. For $\beta_2$, we have shown that

$$\frac{\partial}{\partial\beta_2}\frac{dH}{dt}\bigg|_U = -CI_U$$

where $C \geq 0$ if and only if

$$\operatorname{tr}\Sigma \leq \frac{\eta^2}{4} - \frac{\|\nabla L(w)\|^2}{S(w)^2}.$$

To rewrite this in a form that is independent of $\Sigma$, note that

$$\operatorname{tr}\Sigma = \operatorname{tr}\left[\hat{\beta}^{-1}\hat{\alpha}\right]$$

$$= \left\langle \hat{\beta}^{-1}[I], \alpha\right\rangle$$

$$= \left\langle \alpha, \hat{\beta}_w^{-1}[I]\right\rangle\left[\frac{1}{1 + c\beta_w^{-1}[I,I]}\right]$$

$$= \frac{\operatorname{tr}\left[\hat{\beta}_w^{-1}\hat{\alpha}_w\right] + c\beta_w^{-1}[I,I]\left[\frac{\eta^2}{4} - \frac{\nabla L(w)^2}{S(w)^2}\right]}{1 + c\beta_w^{-1}[I,I]}.$$

Therefore $\operatorname{tr}\Sigma$ is a weighted average between $\operatorname{tr}\left[\hat{\beta}_w^{-1}\hat{\alpha}_w\right]$ and $\frac{\eta^2}{4} - \frac{\nabla L(w)^2}{S(w)^2}$, so $\operatorname{tr}[\Sigma] \leq \frac{\eta^2}{4} - \frac{\nabla L(w)^2}{S(w)^2}$ if and only if $\operatorname{tr}\left[\hat{\beta}_w^{-1}\hat{\alpha}_w\right] \leq \frac{\eta^2}{4} - \frac{\nabla L(w)^2}{S(w)^2}$. This first expression is just $\operatorname{tr}\Sigma\big|_{\beta_2=1}$ and the second is just $\operatorname{tr}\Sigma\big|_{\beta_2=0}$. $\qquad\square$

The condition $\Sigma(t) \succ 0$ is equivalent to requiring that we are not currently at a "breakpoint," i.e. a time $t$ at which an eigenvalue drops from EOS. The set of such $t$ constitute a measure zero set, so the above lemma holds for almost all $t$.

A.4.2   THE SMALL $\beta_2$ CORRECTION

The following lemma shows that the $\beta_2 \to \frac{1-\beta_2}{\beta_2}$ correction for the Scalar RMSProp and RMSProp central flows allows the continuous EMA to match the discrete EMA for linear targets $f(t)$:

**Lemma 4.** *Let* $f : \mathbb{R} \to \mathbb{R}$ *be a continuous time process with* $|f^{(2)}(t)| \le \Delta$ *for all t. Then if*

$$\nu_t = \beta_2 \nu_{t-1} + (1 - \beta_2)f(t)$$
$$\nu'(t) = \frac{1 - \beta_2}{\beta_2}[f(t) - \nu(t)]$$

*we have that for all integers* $t \ge \tilde{O}((1 - \beta_2)^{-1})$,

$$|\nu(t) - \nu_t| \le O\left(\frac{\beta_2 \Delta}{(1 - \beta_2)^2}\right).$$

*Proof.* First define

$$\delta_t := \nu_t - f(t) - \frac{\beta_2}{1 - \beta_2}f'(t)$$
$$\delta(t) := \nu(t) - f(t) - \frac{\beta_2}{1 - \beta_2}f'(t).$$

Then we have that:

$$\delta_t = \beta_2 \nu_{t-1} + (1 - \beta_2)f(t) - f(t) - \frac{1 - \beta_2}{\beta_2}f'(t)$$

$$= \beta_2 \nu_{t-1} - \beta_2 f(t) - \frac{\beta_2}{1 - \beta_2}f'(t)$$

$$= \beta_2 \delta_{t-1} - \beta_2[f(t) - f(t-1)] - \frac{\beta_2}{1 - \beta_2}f'(t) + \frac{\beta_2^2}{1 - \beta_2}f'(t-1)$$

$$= \beta_2 \delta_{t-1} - \beta_2[f(t) - f'(t) - f(t-1)] + O\left(\frac{\beta_2^2 \Delta}{1 - \beta_2}\right)$$

$$= \beta_2 \delta_{t-1} + O\left(\frac{\beta_2 \Delta}{1 - \beta_2}\right).$$

Therefore,

$$\delta_t = \beta_2^t \delta_0 + O\left(\frac{\beta_2 \Delta}{(1 - \beta_2)^2}\right).$$

Similarly,

$$\delta'(t) = \frac{1 - \beta_2}{\beta_2}[f(t) - \nu(t)] - f'(t) - \frac{\beta_2}{1 - \beta_2}f''(t)$$

$$= -\frac{1 - \beta_2}{\beta_2}\delta(t) + O\left(\frac{\beta_2 \Delta}{1 - \beta_2}\right)$$

so

$$\delta(t) = e^{-\frac{1-\beta_2}{\beta_2}t}\delta(0) + O\left(\frac{\beta_2^2 \Delta}{(1 - \beta_2)^2}\right).$$

Therefore subtracting the bounds on $\delta_t$ and $\delta(t)$ gives:

$$|\nu_t - \nu(t)| \lesssim \frac{\beta_2 \Delta}{(1 - \beta_2)^2}$$

as desired.                                                                                    □

A.5 RMSPROP

We first motivate the stability condition $\lambda_{max}(P^{-1}H) \leq 2$ on a quadratic. First, note that $P^{-1}H$ is similar to $P^{-1/2}HP^{-1/2}$ which is symmetric so $P^{-1}H$ has real eigenvalues and real eigenvectors. Next, consider gradient descent on the quadratic $\frac{1}{2}w^T H w$ with preconditioner $P$. The update is:

$$w \leftarrow w - P^{-1}Hw = (I - P^{-1}H)w.$$

Therefore, $w_t = (I - P^{-1}H)^t w_0$. If the eigenvalues of $P^{-1}H$ are in the range $(0, 2)$, then $w \to 0$. Otherwise, it diverges along the corresponding *right* eigenvectors of $P^{-1}H$. We can reinterpret this stability condition using the equivalent condition $H \preceq 2P$. Note that the top right eigenvectors of $P^{-1}H$ correspond to the top eigenvectors for the generalized eigenvalue problem $Hv = \lambda Pv$.

We again follow a similar derivation to gradient descent. We define the critical subspace by $U(w, \nu) := \ker[H - 2P]$ where $P = \mathrm{diag}(\sqrt{\nu}/\eta)$, and use $\mathrm{Sym}(U(w, \nu))$ to denote the set of symmetric matrices on this subspace. We can now proceed with the time-averaging argument.

As before, we assume $w = \overline{w} + \delta$ with $\mathbb{E}[\delta] = 0$. Rather than expanding the squared gradient norm, we expand the element wise squared gradient:

$$\mathbb{E}[\nabla L(w)^{\odot 2}] \approx \nabla L(\overline{w})^{\odot 2} + \mathbb{E}[(H(\overline{w})\delta)^{\odot 2}] \tag{21}$$

$$= \nabla L(\overline{w})^{\odot 2} + \mathrm{diag}\left[H(\overline{w})\Sigma H(\overline{w})\right] \tag{22}$$

where $\Sigma := \mathbb{E}[\delta\delta^T]$. This can be further simplified using the fact that $\Sigma$ is in the top eigenspace, so $H(\overline{w})\Sigma = 2P\Sigma$. Therefore,

$$\mathrm{diag}\left[H(\overline{w})\Sigma H(\overline{w})\right] = 4P \,\mathrm{diag}[\Sigma] P = \frac{4\nu}{\eta^2} \odot \mathrm{diag}[\Sigma].$$

This suggests the central flow ansatz:

$$\begin{aligned}
\frac{dw}{dt} &= -\frac{\eta}{\sqrt{\nu}} \odot \left[\nabla L(w) + \tfrac{1}{2}\nabla^3 L(w)[\Sigma(t)]\right] \\
\frac{d\nu}{dt} &= \frac{1 - \beta_2}{\beta_2}\left[\nabla L(w)^{\odot 2} + \frac{4\nu}{\eta^2} \odot \mathrm{diag}[\Sigma(t)] - \nu\right].
\end{aligned} \tag{23}$$

As for gradient descent and Scalar RMSProp, this immediately implies a differential variational inequality definition:

**Definition 6** (RMSProp Central Flow, Differential Variational Inequality Formulation). We say that $\{(w(t), \nu(t)\}_{t \geq 0}$ follow the RMSProp central flow if for almost all $t \geq 0$, they satisfy eq. (23) along with the conditions: $\Sigma(t) \succeq 0$, $H(w(t)) \preceq 2P(t)$, and $\langle H(w(t)) - 2P(t), \Sigma(t)\rangle = 0$ where $P(t) = \sqrt{\nu(t)}/\eta$.

We will now show that $\Sigma$ can be computed as the solution to a cone complementarity problem. Fix some time $t$, let $U = U(w, \nu)$, and define the linear operator $\mathcal{T} : \mathrm{Sym}(U) \to \mathbb{R}^d$ by $\mathcal{T}[\Sigma] = \nabla_w \langle H(w), \Sigma\rangle$. We can differentiate the stability condition in the critical subspace as:

$$\begin{aligned}
\dot{H} &:= \frac{d}{dt}(H(w) - 2P)\Big|_U \\
&= \mathcal{T}^T\left[\frac{dw}{dt}\right] - 2\frac{\partial P}{\partial \nu}\frac{d\nu}{dt}\Big|_U \\
&= \mathcal{T}^T P^{-1}\left[-\nabla L(w) - \tfrac{1}{2}\mathcal{T}[\Sigma]\right] \\
&\quad + \frac{1 - \beta_2}{\beta_2}\mathrm{diag}\left[\frac{1}{\eta\nu^{1/2}} \odot \left[\nu - \nabla L(w)^{\odot 2} - \frac{4\nu}{\eta^2} \odot \mathrm{diag}[\Sigma]\right]\right]\Big|_U.
\end{aligned}$$

We again group the constant terms and the terms linear in $\Sigma$. Define:

$$\alpha(w, P) := \mathcal{T}^T[-P^{-1}\nabla L(w)] + \frac{1 - \beta_2}{\beta_2}P^{-1}\left[P^2 - \frac{\mathrm{diag}[\nabla L(w)^{\odot 2}]}{\eta^2}\right]\Big|_U$$

$$\beta(w, P)[\Sigma] := \tfrac{1}{2}\mathcal{T}^T P^{-1}\mathcal{T}[\Sigma] + \frac{1 - \beta_2}{\beta_2}\frac{4}{\eta^2}P \,\mathrm{diag}[\Sigma]\Big|_U.$$

Note that we chose to define $\beta$ through its action on $\Sigma$ to avoid unnecessarily complicated tensor notation. Then $\dot{H} = \alpha - \beta[\Sigma]$. As in the derivations of gradient descent and Scalar RMSProp, we require that $\Sigma$ solves the CCP:

$$\Sigma \succeq 0, \quad \dot{H} \preceq 0, \quad \left\langle \Sigma, \dot{H} \right\rangle = 0. \tag{24}$$

Together these define the CCP formulation, which can be efficiently simulated:

**Definition 7** (RMSProp Central Flow, CCP Formulation). We say that $\{(w(t), \nu(t)\}_{t \geq 0}$ follow the RMSProp central flow if for almost all $t \geq 0$, they satisfy eq. (23) with $\Sigma(t) \in CCP(\alpha(w(t), P(t)), \beta(w(t), P(t)))$ where $P(t) = \text{diag}[\sqrt{\nu(t)}/\eta]$.

### A.5.1 THE RMSPROP STATIONARY $\nu$

Setting $\frac{d\nu}{dt} = 0$ in eq. (23) gives:

$$\nu = \nabla L(w)^{\odot 2} + (4/\eta^2)\nu \odot \text{diag}[\Sigma]. \tag{25}$$

In addition, by complementarity we know that

$$\Sigma\big(H(w) - 2\,\text{diag}[\sqrt{\nu}/\eta]\big) = 0. \tag{26}$$

**Proposition 2.** If $w, \nu$ satisfy eqs. (25) and (26), then $P := \text{diag}(\sqrt{\nu}/\eta)$ minimizes the convex program eq. (11).

*Proof.* Let $p = \sqrt{\nu}/\eta$. The convex program eq. (11) can be written as:

$$\min_p \sum_i p_i + \frac{1}{\eta^2} \frac{\nabla L(w)_i^2}{p_i} \quad \text{such that} \quad H(w) \preceq 2\,\text{diag}(p). \tag{27}$$

If $\hat{\Sigma}$ is the dual variable for the semidefinite constraint, the KKT conditions for this program are:

$$1 - \frac{1}{\eta^2} \frac{\nabla L(w)_i^2}{p_i^2} - 2\hat{\Sigma}_{ii} = 0 \quad \forall i, \quad \hat{\Sigma}(H(w) - 2\,\text{diag}(p)) = 0. \tag{28}$$

We will prove that $(p, \hat{\Sigma}) = (\frac{\sqrt{\nu}}{\eta}, \frac{2}{\eta^2}\Sigma)$ solve the KKT conditions eq. (28). Note that the complementary slackness KKT condition is equivalent to the complementarity condition on $\Sigma$. In addition, dividing the stationarity condition for $\nu$ (eq. (25)) by $\nu$ gives

$$1 - \frac{\nabla L(w)^{\odot 2}}{\nu_i} - (4/\eta^2)\Sigma_{ii} = 0 \quad \forall i \tag{29}$$

which is equivalent to the first condition in eq. (28) after substituting $\hat{\Sigma} = \frac{2}{\eta^2}\Sigma(t)$ and $\nu = \eta^2 p^2$ $\square$

**Proposition 3.** For any $\nabla L(w), H(w)$, the solution to eq. (11) is unique.

*Proof.* For notational simplicity, let $g = \nabla L(w)$ and let $H = H(w)$.

Assume there are two minimizers $P, P'$ and let $p := \text{diag}(P), \delta := \text{diag}(P' - P)$. Then by convexity, $\text{diag}[p + \epsilon\delta]$ also minimizes eq. (11) for any $\epsilon \leq 1$. Therefore, differentiating the objective function in this direction gives:

$$\sum_i \delta_i \left[ 1 - \frac{1}{\eta^2} \frac{g_i^2}{p_i^2} \right] = 0.$$

Taking another derivative implies that:

$$\sum_i \frac{g_i^2}{p_i^3} \delta_i^2 = 0.$$

This implies that $\delta_i = 0$ in any direction where $g_i \neq 0$. Let $I$ be the set of indices for which $g_i \neq 0$, and for any vector $p$, let $p$ denote the vector $p$ restricted to the indices in $I$. Define the linear map $g$ by

$$g[v_I]_i := \begin{cases} v_i & i \in I \\ p_i & i \notin I \end{cases}.$$

In other words, $g$ takes a reduced vector $v_I$ and fills in the missing entries with $p$. Next, define the operator $\mathcal{A}$ by

$$\mathcal{A}^T[v_I] = \mathrm{diag}[g[v_I]] \oplus \mathrm{diag}[v_I].$$

Then both $p_I, p_I'$ minimize the following reduced SDP:

$$\min_{p_I} \sum_{i \in I} p_i \quad \text{such that} \quad \tfrac{1}{2} H(w) \oplus 0_{|I| \times |I|} \preceq \mathcal{A}^T(p).$$

Now we apply de Carli Silva & Tunçel (2018, Proposition 1) with $(\mathcal{A}, \mathbf{1}_{|I|})$. First, note that $\mathcal{A}[I_{d+|I|}] = 2\mathbf{1}_{|I|}$ which satisfies the first condition. Next, for any $y \neq 0$, we can take $z = |y|$ to satisfy the second condition, as in the proof of (de Carli Silva & Tunçel, 2018, Corollary 2) Therefore $p_I = p_I'$, and as we have already shown equality on $I^c$, we must have $p = p'$. □

**Numerically computing the RMSProp stationary $\nu$ and preconditioner** To numerically compute $\overline{\nu}(w)$ and/or $\overline{P}(w)$ at some weights $w$, we use a fixed point iteration to solve eq. (11) for $\overline{P}(w)$. Then, one can easily recover $\overline{\nu}(w)$ from $\overline{P}(w)$.

To solve eq. (11), we apply a fixed point iteration based on the fixed point equations for $\Sigma, P$. To derive our fixed point iteration, we parameterize $\Sigma$ in the factorized form $\Sigma = DD^T$ where $D \in \mathbb{R}^{d \times r}$ (as in the Burer-Monteiro factorization (Burer & Monteiro, 2005)) and $r \ll d$ is intended to upper bound the number of unstable eigenvalues. Then the formula for $\frac{d\nu}{dt}$ reduces to:

$$\nu = \nabla L(w)^{\odot 2} + \mathrm{diag}[H\Sigma H] = \nabla L(w)^{\odot 2} + (HD)^{\odot 2}\mathbf{1},$$

where define $H = H(w)$ for notational simplicity.

In addition, we will use that the span of $\Sigma$ (and of $D$) lies in the critical subspace, so

$$\mathrm{diag}[\nu^{-1/2}]HD = \frac{2}{\eta}D.$$

We begin with a random initial guess for $D$ and iteratively update $D, \nu$ by:

$$\nu \leftarrow \nabla L(w)^{\odot 2} + (HD)^{\odot 2}\mathbf{1}$$
$$D \leftarrow \frac{\eta}{2}\mathrm{diag}[\nu^{-1/2}]HD.$$

If this fixed point iteration converges, we have

$$\nu = \nabla L(w)^{\odot 2} + (HD)^{\odot 2}\mathbf{1} \quad \text{and} \quad \mathrm{diag}[\nu^{-1/2}]HD = \frac{2}{\eta}D$$

so that $\frac{d\nu}{dt} = 0$ and the span of $\Sigma = DD^T$ is in the critical subspace.

To verify the results of this fixed point iteration, we compare the objective values of the primal and dual programs. The dual program to eq. (11) is:

$$\max_{\Sigma \succeq 0} \langle \Sigma, H(w) \rangle + 2|\nabla L(w)| \cdot \sqrt{1 - 2\,\mathrm{diag}\,\Sigma} \quad \text{such that} \quad \mathrm{diag}\,\Sigma \preceq 1/2.$$

### A.6 Arbitrary Preconditioned Methods

In this section, we derive a central flow for an abstract preconditioned method. This general central flow will reduce to our central flows for gradient descent, Scalar RMSProp, and RMSProp. However, we emphasize that this paper does not claim that the central flow derived in this section will be correct for any preconditioned method. We include this section both because it allows us to easily generalize our central flows to minor variants of the same algorithms (e.g. gradient descent with a learning rate schedule, RMSProp with bias correction), and because we hope it can be a starting point for others to derive central flows for other optimizers.

We will let $\nu$ be the "state" of the preconditioner, and let $P$ be a function that maps the state $\nu$ to a symmetric, positive definite matrix. Specifically, we consider the discrete update:

$$\nu_t = \nu_{t-1} + f(\nu_{t-1}, w_t), \quad w_{t+1} = w_t - P(\nu_t)^{-1}\nabla L(w_t).$$

This formulation is very general and includes a wide variety of updates including:

- *GD with a learning rate schedule $\eta(t)$:* Set $f(\nu, w) = 1$ so that $\nu(t) = t$, and $P(t) = \eta(t)^{-1}I$

- *Vanilla RMSProp:* Set $f(\nu, w) = (1 - \beta_2)[\nabla L(w)^{\odot 2} - \nu]$ and $P(\nu) = \mathrm{diag}[\sqrt{\nu}/\eta]$

- *RMSProp with $\epsilon$, bias correction, and learning rate schedule $\eta(t)$:* Set $\nu = [v, t]$, $f([v, t], w) = [(1 - \beta_2)[\nabla L(w)^{\odot 2} - v], 1]$ and define

$$P([v, t]) = \mathrm{diag}\left[\frac{1}{\eta(t)} \cdot \sqrt{\frac{v}{1 - \beta_2^t} + \epsilon}\right].$$

Note that this trick of embedding $t$ into the state variable $\nu$ allows us to automatically derive central flows for any smooth hyperparameter schedules (e.g. $\eta(t), \beta_2(t), \epsilon(t)$) as a simple corollary.

As for RMSProp, the stability of this algorithm requires $\lambda_{max}(P^{-1}H) \leq 2$ or equivalently $H \preceq 2P$. To derive the central flow, we assume that $w = \overline{w} + \delta$ with $\mathbb{E}[\delta] = 0$ and $\mathbb{E}[\delta\delta^T] = \Sigma$. Taylor expanding and time-averaging the update for $\nu$ gives:

$$\mathbb{E}[f(\nu, w)] = \mathbb{E}[f(\nu, \nabla L(\overline{w} + \delta))] \approx f(\nu, \nabla L(\overline{w})) + \tfrac{1}{2}\nabla_w^2 f(\nu, \overline{w})[\Sigma]$$

These motivate the central flow ansatz:

$$
\begin{aligned}
\frac{dw}{dt} &= -P(\nu)^{-1}\left[\nabla L(w) + \tfrac{1}{2}\nabla^3 L(w)[\Sigma]\right] \\
\frac{d\nu}{dt} &= f(\nu, w) + \tfrac{1}{2}\nabla_w^2 f(\nu, w)[\Sigma].
\end{aligned}
\tag{30}
$$

We say that $\{w(t), \nu(t), \Sigma(t)\}_{t \geq 0}$ satisfy the DVI formulation of the central flow if for almost all $t \geq 0$ they satisfy eq. (30), $\Sigma(t) \succeq 0$, $H(w(t)) \preceq 2P(\nu(t))$, and $\langle\Sigma(t), H(w(t)) - 2P(\nu(t))\rangle$.

To derive the CCP formulation which is efficiently computable, we differentiate the stability condition. Fix an iterate $t$, let $U := \ker[H - 2P(\nu(t))]$ be the critical subspace at time $t$, and define $\dot{H} := \frac{dH}{dt}\big|_U$ under eq. (30). Then for the stability condition to remain true, we need $\dot{H} \preceq 0$. To compute it, define $\mathcal{T} : \mathrm{Sym}(U) \to \mathbb{R}^d$ by $\mathcal{T}[\Sigma] := \nabla_w \langle H(w), \Sigma\rangle$. Then,

$$
\begin{aligned}
\dot{H} := \frac{dH}{dt}\bigg|_U \\
= \mathcal{T}^T\left[\frac{dw}{dt}\right] - 2\nabla_\nu P(\nu)\left[\frac{d\nu}{dt}\right]\bigg|_U \\
= \mathcal{T}^T P(\nu)^{-1}\left[-\nabla L(w) - \tfrac{1}{2}\mathcal{T}[\Sigma]\right] - 2\nabla_\nu P(\nu)\left[f(\nu, w) + \tfrac{1}{2}\nabla_w^2 f(\nu, w)[\Sigma]\right]\bigg|_U.
\end{aligned}
$$

Splitting up the constant terms and the terms linear in $\Sigma$, we define:

$$
\begin{aligned}
\alpha(w, \nu) &:= \mathcal{T}^T[-P(\nu)^{-1}\nabla L(w)] - 2\nabla_\nu P(\nu)|_U f(\nu, w) \in \mathrm{Sym}(U) \\
\beta(w, \nu) &:= \tfrac{1}{2}\mathcal{T}^T P(\nu)^{-1}\mathcal{T} + \nabla_\nu P(\nu)|_U \nabla_w^2 f(\nu, w) \in \mathrm{Sym}(U) \otimes \mathrm{Sym}(U).
\end{aligned}
$$

Then $\dot{H} = \alpha - \beta[\Sigma]$, so by the same arguments as for gradient descent, Scalar RMSProp, RMSProp, $\Sigma$ needs to solve $CCP(\alpha(w, \nu), \beta(w, \nu))$, and plugging this $\Sigma$ into eq. (30) gives the CCP formulation for the central flow for this preconditioned method.

## A.7 CONE COMPLEMENTARITY PROBLEMS

In this section we will let $k \geq 1$ be a positive integer, and we will use $\mathrm{Sym}_k(\mathbb{R})$ to denote the set of $k \times k$ symmetric matrices.

**Definition 8** (Cone Complementarity Problem). Let $\alpha \in \mathrm{Sym}_k(\mathbb{R})$, and let $\beta \in \mathrm{Sym}_k(\mathbb{R}) \otimes \mathrm{Sym}_k(\mathbb{R})$ be a linear operator on symmetric matrices. We say that the matrix $X \in \mathrm{Sym}_k(\mathbb{R})$ solves the cone complementarity problem $\mathrm{CCP}(\alpha, \beta)$ if:

$$X \succeq 0, \quad \alpha - \beta[X] \preceq 0, \quad \langle X, \alpha - \beta[X]\rangle = 0.$$

**Lemma 5.** *Let $\beta \in \mathrm{Sym}_k(\mathbb{R}) \otimes \mathrm{Sym}_k(\mathbb{R})$ be a symmetric linear operator on symmetric matrices.*

*1. If $\beta \succeq 0$ as an operator on $\mathrm{Sym}_k(\mathbb{R})$, then $CCP(\alpha, \beta)$ has a solution for all $\alpha$.*

2. *If $\beta \succ 0$, this solution is unique. Otherwise, all solutions $X, X'$ differ by a matrix in the kernel of $\beta$.*

*Proof.* Consider the quadratic program

$$\min_X \tfrac{1}{2}\beta[X, X] - \langle X, \alpha \rangle \quad \text{such that} \quad X \succeq 0. \tag{31}$$

Because $\beta \succeq 0$, this program is convex. The KKT conditions for this program are $\beta[X] - \alpha \succeq 0$ and the complementary slackness condition $X(\beta[X] - \alpha) = 0$. Therefore any KKT point to the original quadratic program, including its global optimum, solve the conic complementarity problem which proves (1). To prove (2), let $X, X'$ be solutions to the conic complementarity program. Then taking the trace of the complementarity relations we have:

$$\langle \alpha, X \rangle - \beta[X, X] = 0, \quad \langle \alpha, X' \rangle - \beta[X', X'] = 0. \tag{32}$$

In addition, because both $\beta[X] - \alpha$ and $X'$ are PSD and vice-versa, we must have

$$\langle X', \beta[X] - \alpha \rangle \geq 0, \quad \langle X, \beta[X'] - \alpha \rangle \geq 0. \tag{33}$$

Adding these four conditions gives:

$$\beta[X, X'] + \beta[X', X] - \beta[X, X] - \beta[X', X'] \geq 0 \iff \beta[X - X', X - X'] \leq 0. \tag{34}$$

However, since $\beta \succeq 0$ this implies that $\beta[X - X', X - X'] = 0$. When $\beta \succ 0$ this implies that $X = X'$, and when $\beta \succeq 0$ this implies that $X - X'$ is in the kernel of $\beta$. $\square$

### A.8 DISCRETIZING THE CENTRAL FLOWS

It is not immediately trivial to discretize the central flows (Definitions 2, 5 and 7). These are defined in terms of the *critical subpace*, i.e. the subspace of (effective) Hessian eigenvectors with eigenvalues at the critical threshold. However, when simulating these flows numerically, there are never any eigenvalues "exactly" equal to the critical threshold, so a naive discretization is not possible.

Instead, we will describe an alternate Euler discretization scheme that recovers these central flows as the size of each discretization time step goes to 0. We will first discuss the case of gradient descent as the intuition is clearest (Appendix A.8.1), and then we we will discuss the preconditioned methods (Appendix A.8.2).

### A.8.1 DISCRETIZING THE GRADIENT DESCENT CENTRAL FLOW

Recall that the gradient descent central flow can be written as the unique trajectory $\{w(t), \Sigma(t)\}$ that satisfies: (Definition 1)

1. $\frac{dw}{dt} = -\eta\left[\nabla L(w) + \frac{1}{2}\nabla_w \langle H(w), \Sigma \rangle\right]$

2. $H(w) \preceq 2/\eta I$

3. $\Sigma \succeq 0$

4. $\Sigma \in \ker[H(w) - 2/\eta I]$

where we write $w$ and $\Sigma$ rather than $w(t)$ and $\Sigma(t)$, for notational simplicity.

As discussed above, the last constraint creates challenges when simulating these flows in discrete time, as $H(w)$ will never have eigenvalues "exactly" equal to $2/\eta$ on a computer. As a result, we will slightly relax these constraints to allow them to be practically simulated. We will fix a discretization step size $\epsilon$ and we will require:

1. $w_{t+\epsilon} \leftarrow w_t - \epsilon\eta\left[\nabla L(w_t) + \frac{1}{2}\nabla_{w_t} \langle H(w_t), \Sigma_t \rangle\right]$

2. $H(w_{t+\epsilon}) \preceq 2/\eta I$

3. $\Sigma_t \succeq 0$

4. $\Sigma_t \in \ker[H(w_{t+\epsilon}) - 2/\eta I]$.

This is again not possible to directly discretize, since it is challenging to reason analytically about $H(w_{t+\epsilon})$ where $H$ is the Hessian of the deep learning objective. Therefore, we *linearize* $H(w_{t+\epsilon})$ *around* $w_t$, and approximate $H(w_{t+\epsilon})$ as:

$$H(w_{t+\epsilon}) \approx \underbrace{H(w_t)}_{\text{current Hessian}} - \underbrace{\epsilon\eta\nabla H(w_t)[\nabla L(w_t) + \tfrac{1}{2}\nabla_{w_t}\langle H(w_t), \Sigma_t\rangle]}_{\text{change in Hessian after the Euler step } t \to t+\epsilon} =: H'[\Sigma_t].$$

We will refer to this approximation as $H'[\Sigma_t]$, and crucially, this approximation is *linear in* $\Sigma$. We can therefore ask for a trajectory $\{w_t, \Sigma_t\}$ that satisfies:

1. $w_{t+\epsilon} \leftarrow w_t - \epsilon\eta\big[\nabla L(w_t) + \tfrac{1}{2}\nabla_{w_t}\langle H(w_t), \Sigma_t\rangle\big]$

2. $H'[\Sigma_t] \preceq 2/\eta I$

3. $\Sigma_t \succeq 0$

4. $\Sigma_t \in \ker[H'[\Sigma_t] - 2/\eta I]$.

We can now use these constraints to derive a CCP for $\Sigma$ that can be solved numerically on a computer. First, note that $\Sigma_t$ is supported on the span of the eigenvectors *that will be at the edge of stability in the next Euler step*. To reduce the search space, we will track all eigenvectors of $H(w_t)$ above some fixed threshold (e.g. $1.95/\eta$), as these have the potential to become unstable after the next Euler step. We will store these eigenvectors in a matrix $V \in \mathbb{R}^{d\times k}$ where $k$ denotes the number of such eigenvectors. We will then parameterize $\Sigma = VXV^T$ where $X \in \mathbb{R}^{k\times k}$ is PSD. The above four conditions now imply that $X$ solves a $k \times k$ dimensional CCP:

$$X \succeq 0, \quad \alpha - \beta[X] \preceq 0, \quad \langle X, \alpha - \beta[X]\rangle = 0$$

where

$$\alpha := V^T \underbrace{[H(w_t) - \epsilon\eta\nabla H(w_t)[\nabla L(w_t)]]}_{\text{Hessian after 1 sub-step if no regularization}} V, \quad \beta[X] := V^T \underbrace{\epsilon\eta\tfrac{1}{2}\nabla H(w_t)[\nabla H(w_t)[VXV^T]]}_{\text{effect of curvature regularization}} V.$$

This gives a $k \times k$ CCP for $X$ where $k$ is the number of eigenvectors with eigenvalues near the stability threshold (i.e. above $1.95/\eta$). When simulating these flows numerically, we first compute $\alpha \in \mathbb{R}^{k\times k}$ and $\beta \in \mathbb{R}^{k\times k\times k\times k}$, we then solve this CCP to get $X \in \mathbb{R}^{k\times k}$, and we then update using condition (1) above. As $\epsilon \to 0$, this process will converge to the continuous time central flow Definition 2.

**Interpretation as discretization of projected gradient flow**    We note that while this discretization scheme may seem a bit strange, it actually matches the natural discretization scheme for the projection interpretation for the gradient descent central flow (Definition 3). Recall that the gradient descent central flow can be written as gradient flow on the constrained optimization problem: $\min_w L(w)$ such that $w \in \mathbb{S}_\eta := \{w : H(w) \preceq 2/\eta I\}$. The natural idea is to discretize this using a projected Euler method with discretization step-size $\epsilon$:

$$w_{t+\epsilon} \leftarrow \text{proj}_{\mathbb{S}_\eta}[w_t - \epsilon\eta\nabla L(w_t)]$$

However, this requires computing a projection onto $\mathbb{S}_\eta$, which is intractable since $H(w)$ is the Hessian of a neural network. Therefore, we approximate this projection by its *linearization around* $w_t$. Specifically, rather than project onto the set where $H(w) \preceq 2/\eta I$, we will project onto the linearization of this constraint around $w_t$:

$$H(w_t) + \nabla H(w_t)[w - w_t] \preceq 2/\eta I.$$

Computing the projection onto this set requires solving the *exact same CCP* as the one described above, and the proof is exactly the same as the proof for Lemma 2. This is therefore the "natural" discretization for these types of differential variational inequalities and their related forms.

### A.8.2    DISCRETIZING PRECONDITIONED OPTIMIZERS

In our codebase, we view GD, Scalar RMSProp, and RMSProp as special cases of the more general framework described in Appendix A.6. In this section we assumed that $\nu, w$ follow the discrete time updates:

$$\nu_t = \nu_{t-1} + f(\nu_{t-1}, w_t), \quad w_{t+1} = w_t - P(\nu_t)^{-1}\nabla L(w_t).$$

Recall that GD, Scalar RMSProp, and RMSProp can all be written as special cases of this framework. To derive a central flow for updates of this form, we imposed the conditions:

1. $\frac{dw}{dt} = -P(\nu)^{-1}\big[\nabla L(w) + \frac{1}{2}\nabla^3 L(w)[\Sigma]\big]$

2. $\frac{d\nu}{dt} = f(\nu, w) + \frac{1}{2}\nabla_w^2 f(\nu, w)[\Sigma].$

3. $H(w) \preceq 2P(\nu)$

4. $\Sigma \succeq 0$

5. $\Sigma \in \ker[H(w) - 2P(\nu)]$

As for GD, the key to numerically simulating such a process is to linearize both $H$ and $P$. Given an oscillation covariance $\Sigma_t$ that we will describe how to compute, we will take a sub-step of size $\epsilon$ and update $w, \nu$ by:

$$w_{t+\epsilon} \leftarrow w_t - \epsilon P(\nu_t)^{-1}\big[\nabla L(w_t) + \frac{1}{2}\nabla^3 L(w_t)[\Sigma_t]\big]$$
$$\nu_{t+\epsilon} \leftarrow \nu_t + \epsilon\big[f(\nu_t, w_t) + \frac{1}{2}\nabla_{w_t}^2 f(\nu_t, w_t)[\Sigma_t]\big].$$

We will also define $H', P'$ to be the linearizations of $H, P$ around $w_t$:

$$H' := H(w_t) + \nabla H(w_t)[w_{t+\epsilon} - w_t], \quad P' := P(\nu_t) + \nabla P(\nu_t)[\nu_{t+\epsilon} - \nu_t].$$

Note that because $w_{t+\epsilon}, \nu_{t+\epsilon}$ are linear in $\Sigma_t$, so are $H', P'$. We will therefore enforce that:

- $H' \preceq 2P'$

- $\Sigma_t \succeq 0$

- $\Sigma_t \in \ker[H' - 2P'].$

This will again give us a CCP for $\Sigma$. As for GD, note that the last condition implies that $\Sigma$ is supported on the generalized eigenvectors at the edge of stability *at the next step*. We therefore track all generalized eigenvectors which solve $H(w_t)u = \lambda P(w_t)u$ with eigenvalue $\lambda$ above some fixed threshold (e.g. 1.95), as these have the potential to become unstable at the next substep. Let $V \in \mathbb{R}^{d \times k}$ store these vectors as in the GD section. Then we decompose $\Sigma = VXV^T$ where $X \in \mathbb{R}^{k \times k}$. Plugging this into the above conditions gives a CCP for $X$:

$$X \succeq 0, \quad \alpha - \beta[X] \preceq 0, \quad \langle X, \alpha - \beta[X]\rangle = 0$$

where

$$\alpha := V^T\Big[H(w_t) + \epsilon\nabla H(w_t)\big[\widetilde{\Delta w_t}\big] - 2\big[P(\nu_t) + \epsilon\nabla P(\nu_t)\big[\widetilde{\Delta \nu_t}\big]\big]\Big]V$$

$$\beta[X] := \frac{\epsilon}{2}V^T\big[\nabla H(w_t)[P(\nu_t)^{-1}\nabla H(w_t)[\Sigma]] + 2\nabla P(w_t)[\nabla_{w_t}^2 f(\nu_t, w_t)[\Sigma]]\big]V$$

$$\Sigma := VXV^T, \quad \widetilde{\Delta w_t} := -P(\nu_t)^{-1}\nabla L(w_t), \quad \widetilde{\Delta \nu_t} := f(\nu_t, w_t).$$

Again as for GD, computing $\alpha, \beta$ only require matrix-vector product access to the Hessian and tensor-vector-vector product access to the third derivatives. When simulating the central flows, we first construct the basis $V$ by solving the generalized eigenvalue problem on $H(w_t), P(\nu_t)$ and keeping the eigenvalues/eigenvectors near 2, and we then compute these values of $\alpha \in \mathbb{R}^{k \times k}$ and $\beta \in \mathbb{R}^{k \times k \times k \times k}$. We then solve the CCP to get $X \in \mathbb{R}^{k \times k}$ and then use this to take an Euler step of size $\epsilon$ on the central flow. As $\epsilon \to 0$, this process reduces to the continuous-time central flow described in Appendix A.6, and in the special case of GD where $\nu = \emptyset$ and $P(\nu) := 1/\eta$, this reduces to the discrete time update for gradient descent described above.

## B    EXPERIMENTS

This section contains an overview of our experimental results. More experimental data can be found in Appendix C, which contains "bulk" data from the full set of experiments.

Our code can be found at: `https://github.com/locuslab/central_flows`.

**Architectures**    We experiment on six architectures:

1. a convolutional neural network (CNN);

2. a ResNet (He et al., 2016);

3. a Vision Transformer (ViT) (Dosovitskiy et al., 2021);

4. an LSTM (Hochreiter & Schmidhuber, 1997);

5. a (sequence) Transformer (Vaswani et al., 2017);

6. and a Mamba sequence model (Gu & Dao, 2024).

Due to the computational expense of discretizing the central flows (see Appendix B.5), we experiment on small-scale instances of these architectures. Architectural details can be found in Appendix B.6.

**Datasets**    We test the vision architectures (CNN, ResNet, ViT) on a subset of CIFAR-10 (Krizhevsky, 2009). We test the sequence architectures (LSTM, Transformer, Mamba) on a synthetic sorting task (Karpathy, 2020). Further details on these datasets can be found in Appendix B.7. For each architecture and each dataset, we test both cross-entropy loss and MSE loss. As discussed in Appendix B.4, the central flow tends to be somewhat more accurate with MSE loss.

**Implementation**    We discretize the central flows using the variant of Euler's method outlined in Appendix A.8. Implementation details can be found in Appendix B.5.

The remainder of this section is structured as follows:

1. In Appendix B.1, we present experiments for **gradient descent**.

2. In Appendix B.2, we present experiments for **Scalar RMSProp**, and we experimentally demonstrate the "acceleration via regularization" effect that was described in Section 4.3.

3. In Appendix B.3, we present experiments for **RMSProp**, and we experimentally support the stationarity analysis that was described in Section 5.2, as well as the "acceleration via regularization" phenomenon that was described in Section 5.2.

4. In Appendix B.4, we discuss known **failure modes** for the central flows, i.e. situations where the central flow poorly approximates the optimizer trajectory.

### B.1 GRADIENT DESCENT EXPERIMENTS

We tested the gradient descent central flow in the six deep learning settings described above. We experimented with both mean squared error (MSE) loss and cross-entropy loss. Figure 9 and Figure 10 show that, in each of these six settings, the central flow can accurately predict the time-averaged (smoothed) loss curve of gradient descent across several different learning rates.

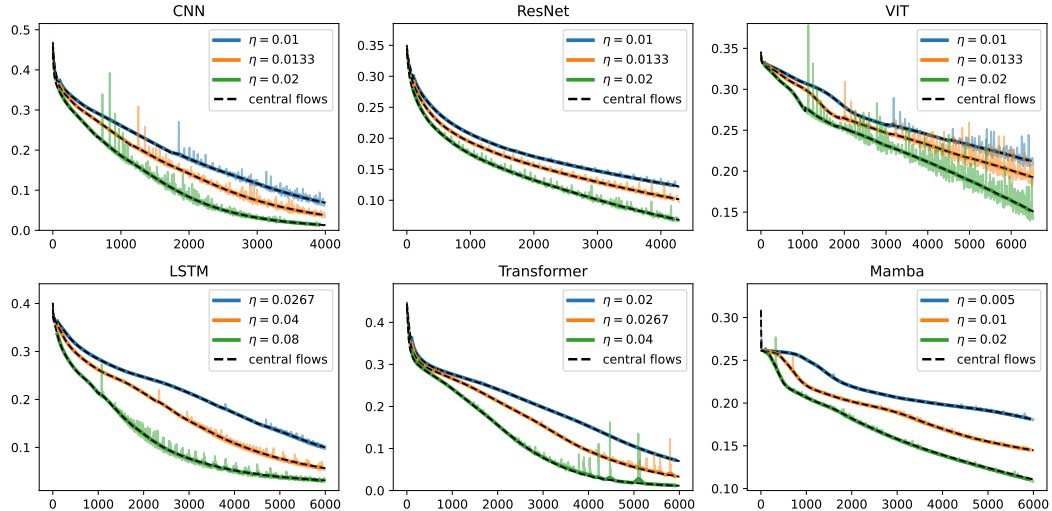

Figure 9: **Central flow predictions for gradient descent loss curves (MSE).** For six architectures, and three learning rates each, we compare the actual loss curve (colors) to the central flow's prediction for the time-averaged loss curve eq. (6) (black dashed). The faint colored lines are the raw loss curves; the thick colored lines are the time-averaged (smoothed) versions. Observe that the central flow accurately predicts the time-averaged loss curves.

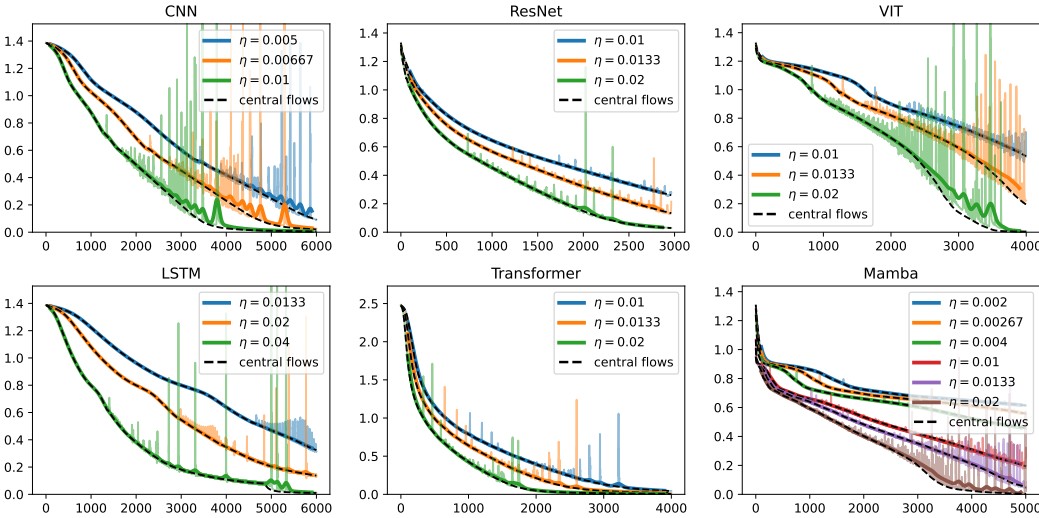

Figure 10: **Central flow predictions for gradient descent loss curves (CE).** Similar figure to Figure 9, but for cross-entropy loss. Observe that the prediction is sometimes a bit off, especially at the end, and especially with larger learning rates.

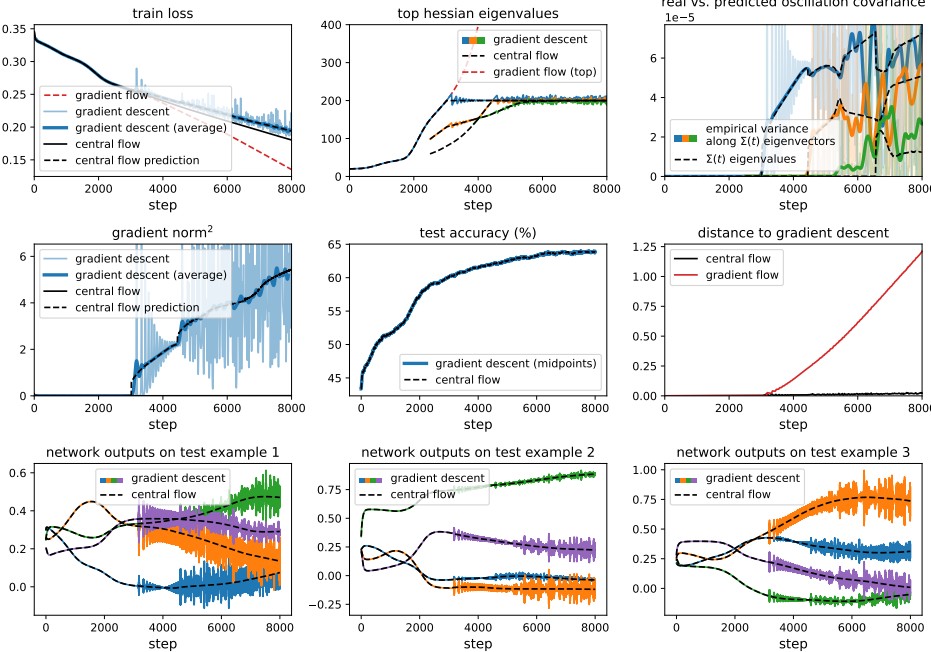

Figure 11: **A close look at the gradient descent central flow.** Using gradient descent with $\eta = 2/200$, we train a ViT on a subset of CIFAR-10. The central flow (black) accurately models the time-averaged trajectory of gradient descent (blue), whereas the gradient flow (red) takes a different path. As described in Appendix B.5, we stop running the gradient flow once the sharpness gets too high, which is why the red lines stop early.

**Top left**: The loss along the central flow (solid black) decreases monotonically, whereas the loss along the gradient descent trajectory (light blue) behaves non-monotonically once the dynamics enter EOS. While the loss is higher along the gradient descent trajectory than along the central flow, the central flow can accurately predict the *time-averaged* loss along the gradient descent trajectory, using eq. (6) (dashed black); this can be seen to accurately match the time average of the actual gradient descent loss curve (dark blue). Finally, the train loss along the gradient flow (in red) decreases faster, because it follows a different, unregularized path.

**Top center**: we plot the top three Hessian eigenvalues under gradient descent (colors) and under the central flow (black). Under GD, the top Hessian eigenvalues equilibrate around $2/\eta$; under the central flow they are fixed exactly at $2/\eta$. In red, we plot the top Hessian eigenvalue under the gradient flow, which rises beyond $2/\eta$. Note that for GD, we report the Hessian eigenvalues at the second-order midpoints (see Appendix B.5), rather than at the iterates themselves, as this makes for clearer plots.

**Top right**: we show that the central flow's $\Sigma(t)$ accurately predicts the covariance of the oscillations. In black, we plot the nonzero eigenvalues of $\Sigma(t)$; the number is always the same as the number of Hessian eigenvalues at $2/\eta$. In faint colors, we plot the squared magnitude of the displacement between gradient descent and the central flow along each eigenvector of $\Sigma(t)$. In thick colors, we plot the time-averages of these displacements, i.e. the empirical variance of the oscillations along each eigenvector of $\Sigma(t)$. Observe that the eigenvalues of $\Sigma(t)$ accurately predict the instantaneous variance of the oscillations along the corresponding eigenvector. More details in Appendix B.5.

**Middle left**: in solid black, we plot the squared gradient norm along the central flow; in dashed black, we plot the central flow's prediction for the time-averaged squared gradient norm along the trajectory (similar to eq. (6)). This accurately predicts the time average (dark blue) of the squared gradient norm at the iterates (light blue). Notice that most of the gradient norm comes from the oscillations.

**Middle center**: we plot the test accuracy under gradient descent (blue) and the central flow (black). For gradient descent, we report the test accuracy at second-order midpoint, as this removes much of the oscillations. Because the central flow matches the gradient descent trajectory, the test accuracy is nearly the same across both trajectories.

**Middle right**: The distance in weight space between gradient descent and the central flow (black) stays small over time, indicating that these two trajectories stay close. By contrast, the distance between gradient descent and the *gradient* flow (red) grows rapidly once the dynamics enter EOS.

**Bottom row**: we show the network's final-layer predictions on three arbitrary examples. Under gradient descent (colors) these predictions oscillate due to the oscillations in weight space. Under the central flow (black), the predictions evolve smoothly while following the same overall path.

For one of these runs, Figure 11 depicts more granular set of metrics. (Analogous figures for all of the other runs are given in Appendix C.) Note the following:

- The distance in weight space between the central flow and the real optimizer trajectory stays small over time. In comparison, the distance between the *gradient flow* and the real optimizer trajectory grows much faster over time.

- The central flow accurately predicts the evolution of individual network outputs over time. This can be viewed as an visualization of the central flow's accuracy in *function* space.

- The central flow's prediction $\Sigma(t)$ for the instantaneous covariance matrix of the oscillations accurately predicts the empirical variance of oscillations along each eigenvector of $\Sigma(t)$.

- The central flow can accurately predict the (smoothed) training loss curve, as well as the (smoothed) squared gradient norm curve.

Nevertheless, the central flow is just an approximation to the gradient descent trajectory, and this approximation can break down under several different circumstances which we detail in Appendix B.4.

## B.2 SCALAR RMSPROP EXPERIMENTS

We tested the Scalar RMSProp central flow in the six deep learning settings described above. We experimented with both mean squared error (MSE) loss and cross-entropy loss. Figure 12 and Figure 13 show that, in each of these six settings, the central flow can accurately predict the time-averaged (smoothed) loss curve of Scalar RMSProp across several different learning rates.

For one of these runs, Figure 14 depicts more granular set of metrics. (Analogous figures for all of the other runs are given in Appendix C.)

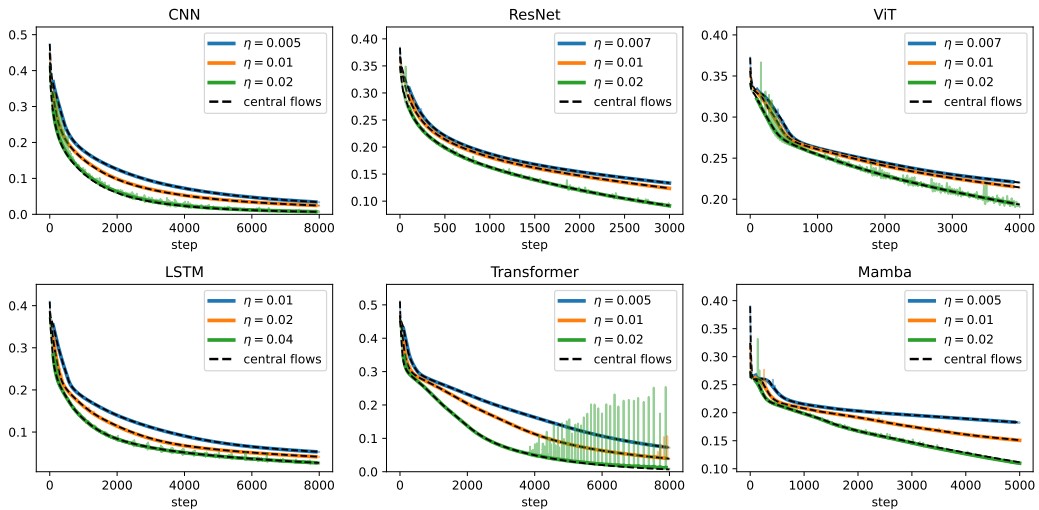

Figure 12: **Central flow predictions for Scalar RMSProp loss curves (MSE).** For six architectures, and three learning rates each, we compare the actual loss curve (colors) to the central flow's prediction for the time-averaged loss curve eq. (6) (black dashed). The faint colored lines are the raw loss curves; the thick colored lines are the time-averaged (smoothed) versions. Observe that the central flow accurately predicts the time-averaged loss curves.

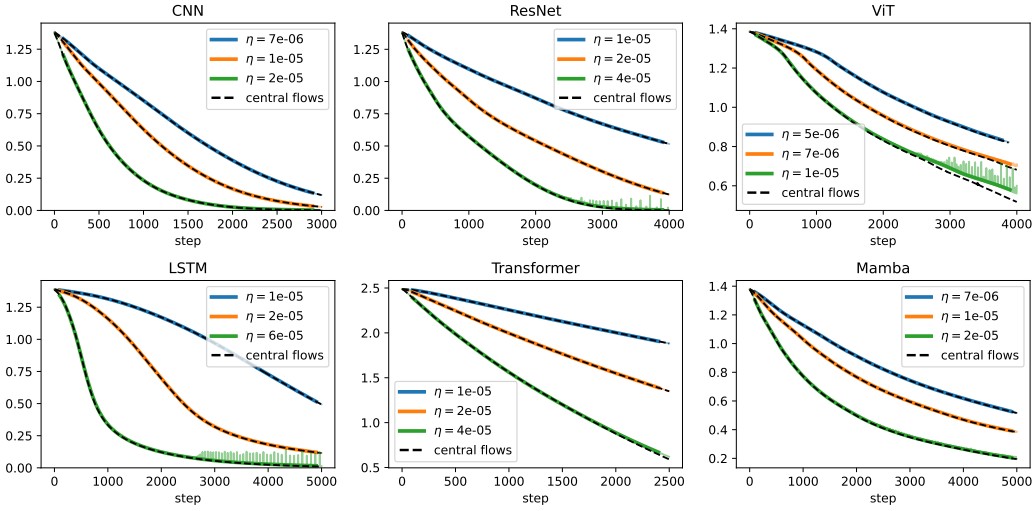

Figure 13: **Central flow predictions for Scalar RMSProp loss curves (CE).** Similar to Figure 12 but for cross entropy.

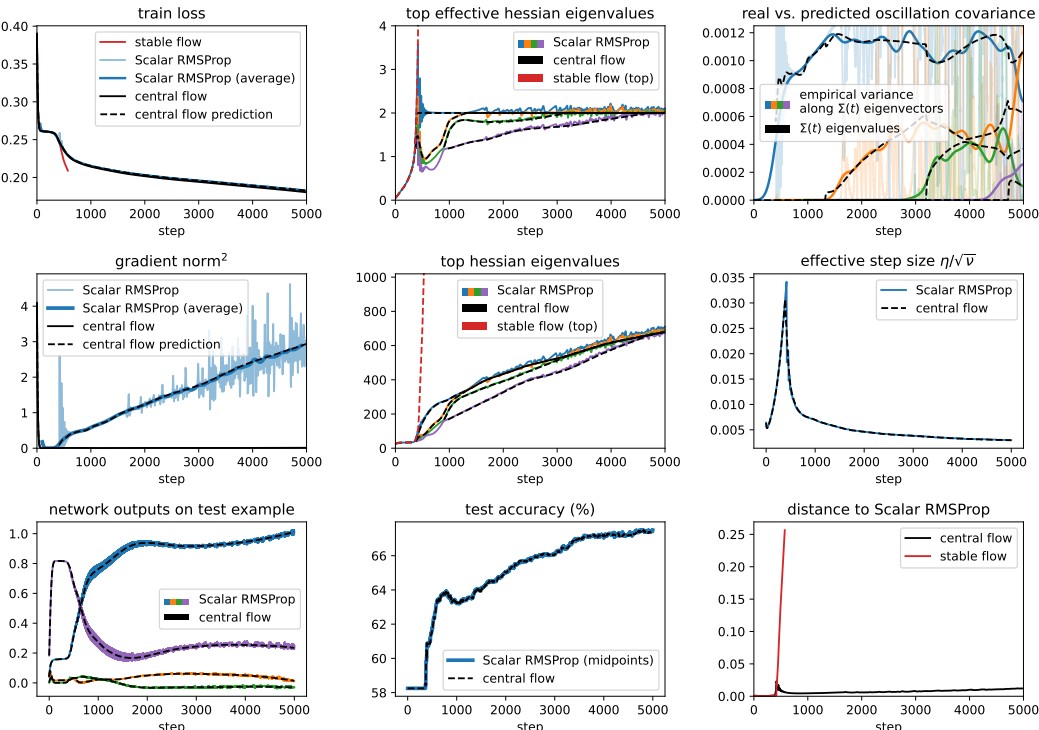

Figure 14: **A close look at the Scalar RMSProp central flow.** Using Scalar RMSProp with $\eta = 2/400$, $\beta_2 = 0.99$, and bias correction, we train a Mamba on a synthetic sequence prediction task with MSE loss. The central flow (black) accurately models the time-averaged trajectory of Scalar RMSProp (blue), whereas the stable flow (red) takes a different path. As described in Appendix B.5, we stop running the stable flow once the effective sharpness gets too high, which is why the red lines stop early.

**Top left**: see Figure 11 caption.
**Top center**: see Figure 11 caption.
**Top right**: see Figure 11 caption.
**Middle left**: see Figure 11 caption.
**Middle center**: while the top eigenvalues of the *effective* Hessian equilibrate at a fixed value, the top eigenvalues of the "raw" Hessian keep evolving. Once the dynamics enter EOS, both Scalar RMSProp (blue) and the central flow (black) start to regularize these eigenvalues relative to the stable flow (red).
**Middle right**: while the effective step size $\eta/\sqrt{\nu}$ along the Scalar RMSProp trajectory is oscillatory, the ESS along the central flow varies smoothly. At EOS, this quantity stays locked at $2/S(w)$.
**Bottom left**: we show the network's final-layer predictions on an arbitrary example. Under Scalar RMSProp (colors) these predictions oscillate due to the oscillations in weight space. Under the central flow (black), the predictions evolve smoothly while following the same overall path.
**Bottom center**: see Figure 11 caption.
**Bottom right**: see Figure 11 caption.

### B.2.1 ACCELERATION VIA REGULARIZATION

Here, we provide experimental support for the *acceleration via regularization* phenomenon that was described in the main text. We conduct the following experiment: starting from some initial point $w_0$ in weight space, we run the Scalar RMSProp central flow at multiple learning rates $\eta$. For each $\eta$, we set the initial $\nu_0$ so that the dynamics start at EOS, i.e $S(w_0)/\sqrt{\nu_0} = 2/\eta$ or equivalently $\eta/\sqrt{\nu_0} = 2/S(w_0)$. Thus, each flow has the form:

$$\frac{dw}{dt} = - \underbrace{\frac{2}{S(w)}}_{\text{adapt step size}} \left[ \nabla L(w) + \underbrace{\tfrac{1}{2}\nabla_w \langle H(w), \Sigma(t; \eta) \rangle}_{\text{regularize curvature}} \right] \tag{35}$$

These flows all adapt the current step size to $2/S(w)$, but differ in the curvature regularization term, with larger $\eta$ translating to stronger curvature regularization, per Lemma 3. We also run an ablated flow which adapts the step size to $2/S(w)$ but does not regularize curvature at all:

$$\frac{dw}{dt} = -\frac{2}{S(w)}\nabla L(w) \tag{36}$$

As shown in Figure 15, the stronger the curvature regularizer, the lower the sharpness $S(w)$ along the trajectory. Since the Scalar RMSProp central flow keeps the effective learning rate $\eta/\sqrt{\nu(t)}$ locked at $2/S(w)$, stronger curvature regularization thus leads to larger effective learning rates later in training. This, in turn, empirically causes the train loss to decrease faster over the long run (even though at the outset, stronger curvature regularization often causes the optimization to be slower).

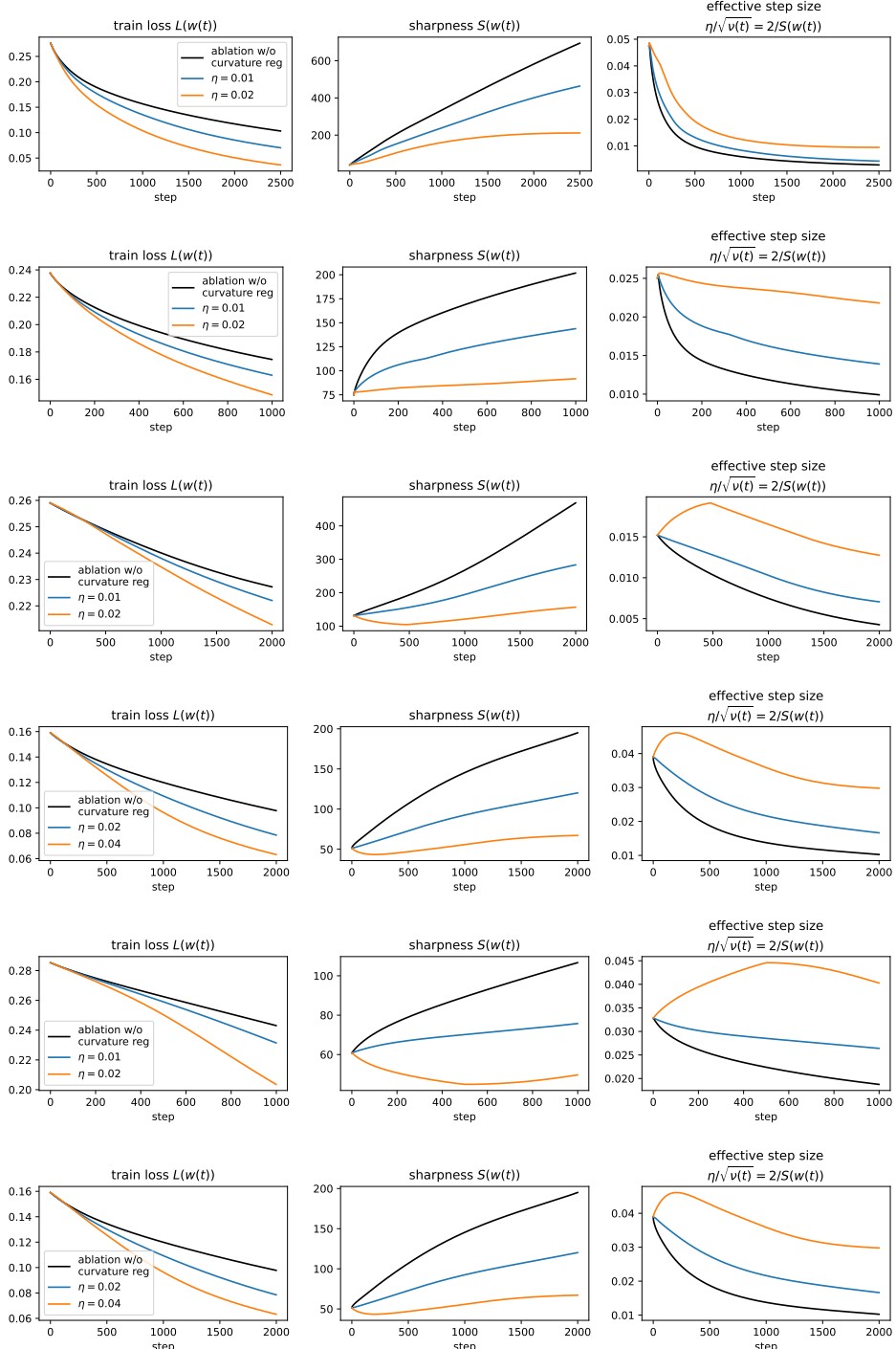

Figure 15: **"Acceleration via regularization" for Scalar RMSProp**. Starting from the same initialization, we run the Scalar RMSProp central flow at various learning rates, as well as an ablated flow $\frac{dw}{dt} = -\frac{2}{S(w)}\nabla L(w)$ with curvature regularization removed. These three flows all use the same step size strategy but differ in the strength of implicit curvature regularization. Initially (see inset), the flows with higher curvature regularization often optimize slower; however, over the longer run, they are able to take larger steps and optimize faster.

### B.3 RMSPROP EXPERIMENTS

We tested the RMSProp central flow in the six deep learning settings described above. We experimented with both mean squared error (MSE) loss and cross-entropy loss. Figure 16 and Figure 17 show that, in each of these six settings, the central flow can accurately predict the time-averaged (smoothed) loss curve of RMSProp across several different learning rates.

For one of these runs, Figure 18 depicts more granular set of metrics. (Analogous figures for all of the other runs are given in Appendix C.)

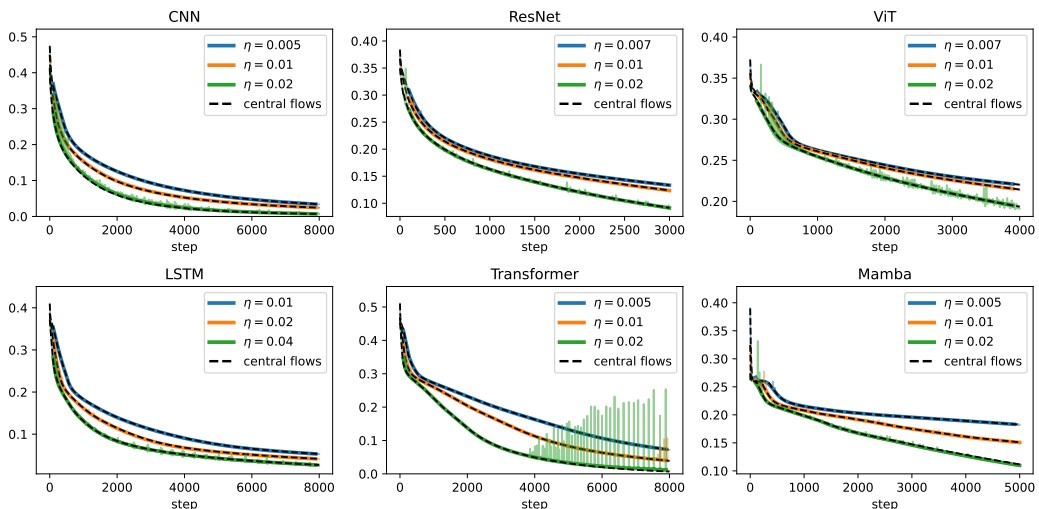

Figure 16: **Central flow predictions for RMSProp loss curves (MSE).** For six architectures, and three learning rates each, we compare the actual loss curve (colors) to the central flow's prediction for the time-averaged loss curve eq. (6) (black dashed). The faint colored lines are the raw loss curves; the thick colored lines are the time-averaged (smoothed) versions. Observe that the central flow accurately predicts the time-averaged loss curves.

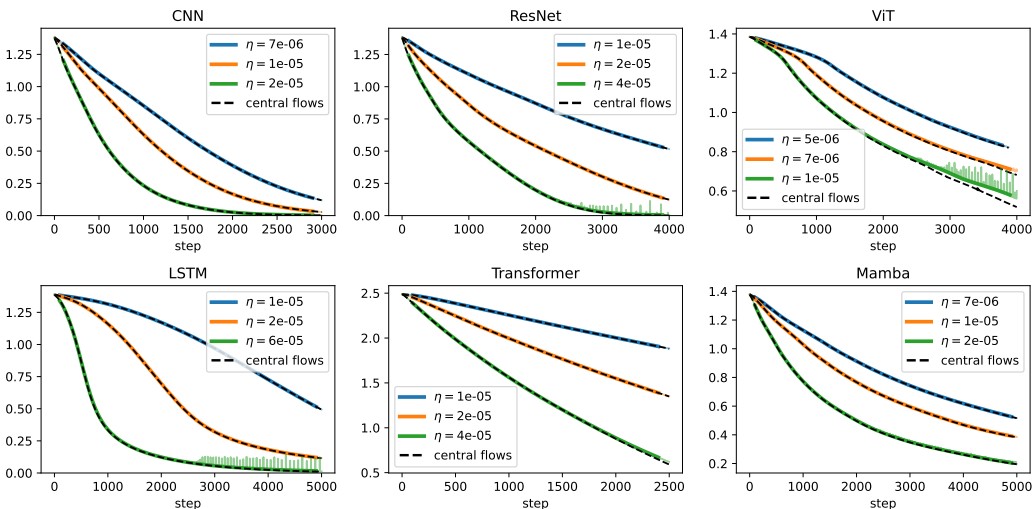

Figure 17: **Central flow predictions for RMSProp loss curves (CE).** Similar to Figure 16 but for cross-entropy loss.

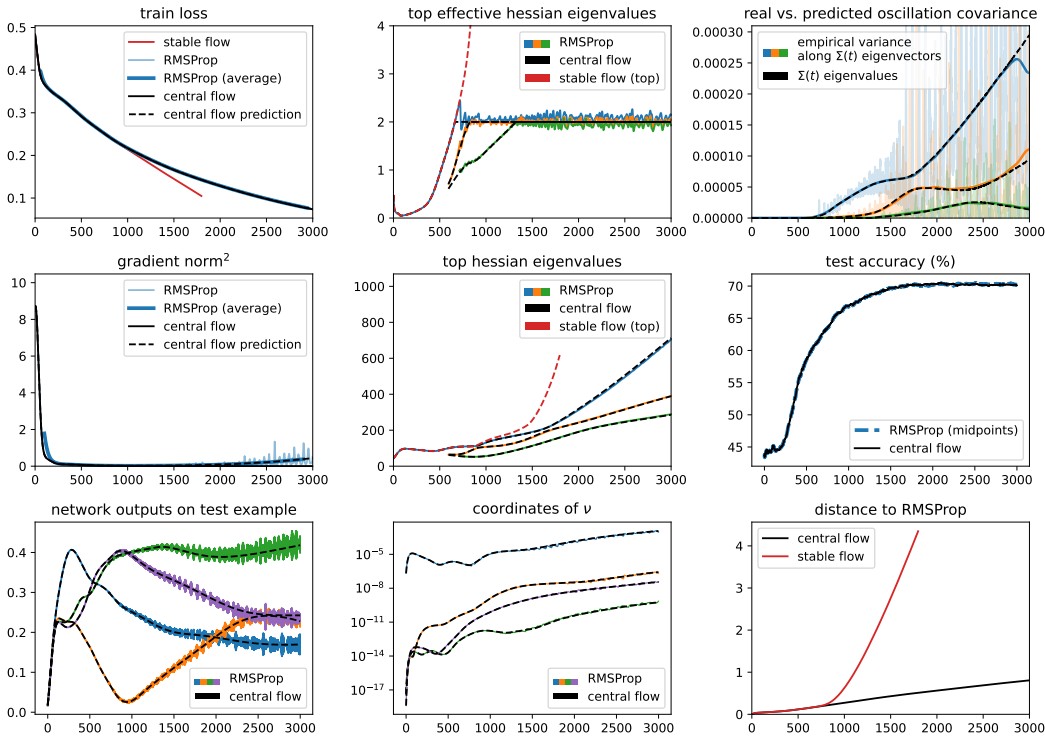

Figure 18: **A close look at the RMSProp central flow.** Using RMSProp with $\eta = 2 \times 10^{-5}$, $\beta_2 = 0.99$, $\epsilon = 10^{-8}$ and bias correction, we train a ResNet on a subset of CIFAR-10 with MSE loss. The central flow (black) accurately models the time-averaged trajectory of RMSProp (blue), whereas the stable flow (red) takes a different path. As described in Appendix B.5, we stop running the stable flow once the effective sharpness gets too high, which is why the red lines stop early.

**Top left**: see Figure 11 caption.
**Top center**: see Figure 11 caption.
**Top right**: we show that the central flow's $\Sigma(t)$ accurately predicts the covariance of the oscillations. In black, we plot the nonzero generalized eigenvalues of $\Sigma(t)$ with respect to the preconditioner $P(t)$; the number is always the same as the number of effective Hessian eigenvalues at 2. In faint colors, we plot the squared magnitude of the displacement between gradient descent and the central flow along each generalized eigenvector of $\Sigma(t)$ with respect to the preconditioner $P(t)$. In thick colors, we plot the time-averages of these displacements, i.e. the empirical variance of the oscillations along each generalized eigenvector of $\Sigma(t)$. Observe that the generalized eigenvalues of $\Sigma(t)$ accurately predict the instantaneous variance of the oscillations along the corresponding generalized eigenvectors. More details in Appendix B.5.
**Middle left**: see Figure 11 caption.
**Middle center**: while the top eigenvalues of the *effective* Hessian equilibrate at a fixed value, the top eigenvalues of the "raw" Hessian keep evolving.
**Middle right**: see Figure 11 caption.
**Bottom left**: we show the network's final-layer predictions on an arbitrary example. Under RMSProp (colors) these predictions oscillate due to the oscillations in weight space. Under the central flow (black), the predictions evolve smoothly while following the same overall path.
**Bottom center**: while the entries of $\nu$ are oscillatory along the RMSProp trajectory, they vary smoothly along the central flow.
**Bottom right**: see Figure 11 caption.

As with the other optimizers, we find that the central flow approximation becomes less accurate as $\eta$ becomes larger. We also expect the RMSProp central flow to break down when $\beta_2$ becomes very small, as then RMSProp would not resemble preconditioned gradient descent with a slowly varying preconditioner.

### B.3.1 ACCELERATION VIA REGULARIZATION

Here, we verify the *acceleration via regularization* phenomenon for RMSProp. Our main claim is that RMSProp's implicit curvature reduction behavior — which is made precise by its central flow — is beneficial for the RMSProp's efficacy as an optimizer.

Starting from some initial point $(w_0, \nu_0)$ from an RMSProp central flow trajectory, we run both the RMSProp central flow, as well as an ablated flow which has the implicit curvature regularization disabled. (That is, this flow sets $\Sigma(t)$ so as to stay at the edge of stability purely via the effect of oscillations on $\nu$.) Figure 19 shows that the original RMSProp central flow: (1) traverses a lower-sharpness trajectory than the ablated flow; (2) is able to use higher effective learning rates in the long run; (3) optimizes faster in the long run.

### B.3.2 STATIONARY PRECONDITIONER

Here, we empirically support our stationarity analysis from Section 5.2, which aims to understand the local dynamics of RMSProp based on the current $w$ alone, eliminating $\nu$ from the picture.

**Convergence to stationary $\nu$**   We first show empirically that under the RMSProp central flow $(w(t), \nu(t))$, the EMA $\nu(t)$ approximately converges to its stationary value $\overline{\nu}(w(t))$. To numerically estimate $\overline{\nu}(w)$, we use the fixed point scheme outlined in Appendix A.5.1 to obtain $\overline{P}(w)$, and then we obtain $\overline{\nu}(w)$ from $\overline{P}(w)$.

Figures 20 and 21 plot the cosine similarity between $\nu(t)$ and $\overline{\nu}(w(t))$ over time. We can see that this cosine similarity goes to nearly 1, indicating that the $\overline{\nu}(w(t))$ is indeed a good approximation to the RMSProp EMA $\nu(t)$.

**Suboptimality of stationary preconditioner**   We have shown that the stationary preconditioner $\overline{P}(w)$ at weights $w$ can be characterized as the optimal solution to eq. (11), a convex optimization problem over preconditioners:

$$\overline{P}(w) \quad := \quad \underset{P \text{ diagonal}, P \succeq 0}{\arg\min} \quad \operatorname{tr}(P) + \tfrac{1}{\eta^2} \underbrace{\|\nabla L(w)\|_{P^{-1}}^2}_{\text{optimization speed}} \quad \text{such that} \quad \underbrace{H(w) \preceq 2P}_{\text{local stability}}. \tag{11}$$

It is reasonable why minimizing the *first* term in eq. (11) would be desirable in a preconditioner: the diagonal elements of the preconditioner $P$ are the inverse effective learning rates ($P_{ii} = \sqrt{\nu_i}/\eta$), so the trace $\operatorname{tr}(P)$ is the sum of the inverse effective learning rates. Thus, minimizing $\operatorname{tr}(P)$ is equivalent to maximizing the harmonic mean of the effective learning rates. This is an intuitively reasonable criterion — we want the effective learning rates to be large while still maintaining local stability, and one natural-enough way to measure the "largeness" of a vector is via the harmonic mean.

By contrast, the *second* term in eq. (11) appears to be *undesirable*. Prepending a minus sign, the quantity $-\|\nabla L(w)\|_{P^{-1}}^2 = -\nabla L(w)^T P^{-1} \nabla L(w)$ is the instantaneous rate of change in the loss $L(w)$ under the preconditioned gradient flow $\frac{dw}{dt} = P^{-1}\nabla L(w)$ at weights $w$. This is not quite the rate of change in the loss under the central flow, as the central flow also posesses an implicit curvature regularization term, but it is a reasonable proxy. This suggests that for training to be fast, we want $-\|\nabla L(w)\|_{P^{-1}}^2$ to be large and negative, or equivalently, we want $\|\nabla L(w)\|_{P^{-1}}^2$ to be large and positive. Yet the optimization problem eq. (11) tries to *decrease* $\|\nabla L(w)\|_{P^{-1}}^2$; in other words, it tries to make training *slower*. Thus, its presence is undesirable.

One can consider an alternative optimization problem over preconditioners that omits the problematic second term:

$$\hat{P}(w) \quad := \quad \underset{P \text{ diagonal}, P \succeq 0}{\arg\min} \quad \operatorname{tr}(P) \quad \text{such that} \quad \underbrace{H(w) \preceq 2P}_{\text{local stability}}. \tag{37}$$

From the definitions of eq. (11) and eq. (37) as optimization problems, it is guaranteed that preconditioned gradient flow using this alternative preconditioner $\hat{P}(w)$ would decrease the loss at least as fast as preconditioned gradient flow using the RMSProp preconditioner $\overline{P}(w)$:

$$\|\nabla L(w)\|_{\hat{P}(w)}^2 \leq \|\nabla L(w)\|_{\overline{P}(w)}^2. \tag{38}$$

To illustrate this point empirically, and to see how large the gap is, in Figure 24 and Figure 25 we compute both $\|\nabla L(w)\|_{\hat{P}(w)}^2$ and $\|\nabla L(w)\|_{\overline{P}(w)}^2$ along the trajectory of the RMSProp central flow.

To compute $\hat{P}(w)$, we use a fixed point iteration similar to that described in Appendix A.5.1, but which leaves out the gradient term. We observe that $\|\nabla L(w)\|^2_{\overline{P}(w)}$ is often substantially bigger than $\|\nabla L(w)\|^2_{\hat{P}(w)}$, indicating that the alternative preconditioner would have been non-negligibly better. The degree to which the alternative preconditioner is better empirically depends on the RMSProp learning rate $\eta$. This makes sense, because as $\eta$ grows, the effect of the second term in Equation (11) diminishes.

Thus, our analysis suggests a natural direction for improvement for RMSProp's implicit preconditioning scheme. It is an interesting open question whether there exists an optimizer update rule that has eq. (37) as its implicit preconditioner.

**Stationary flow**     We have shown that RMSProp's preconditioner converges to the stationary value eq. (11). However, this statement does not fully capture the local dynamics of the optimizer, as the RMSProp central flow is not merely a preconditioned gradient flow — it also includes a curvature penalty which reflects the effect of the oscillations on the curvature.

To approximate the local dynamics of RMSProp as a closed system in $w$ alone, we propose the following *stationary flow*:

$$\frac{dw}{dt} = \underbrace{\overline{P}(w)^{-1}}_{\substack{\text{stationary} \\ \text{preconditioner}}} \Big[ \nabla L(w) + \underbrace{\tfrac{1}{2}\nabla_w \langle \Sigma, H(w)\rangle}_{\text{implicit curvature penalty}} \Big]. \tag{12}$$

where $\Sigma = \Sigma(w; \eta; \beta_2)$ is the unique value of $\Sigma$ which satisfies the RMSProp CCP eq. (24), but using the stationary preconditioner $\overline{P}(w)$ rather than the RMSProp preconditioner derived from $\nu$. Equation (12) models RMSProp as implicitly performing preconditioned gradient descent on a curvature-penalized objective, using the current stationary preconditioner $\overline{P}(w)$.

Figures 26 and 27 show that the stationary flow eq. (12) can indeed accurately predict the instantaneous rate of loss decrease along the central flow.

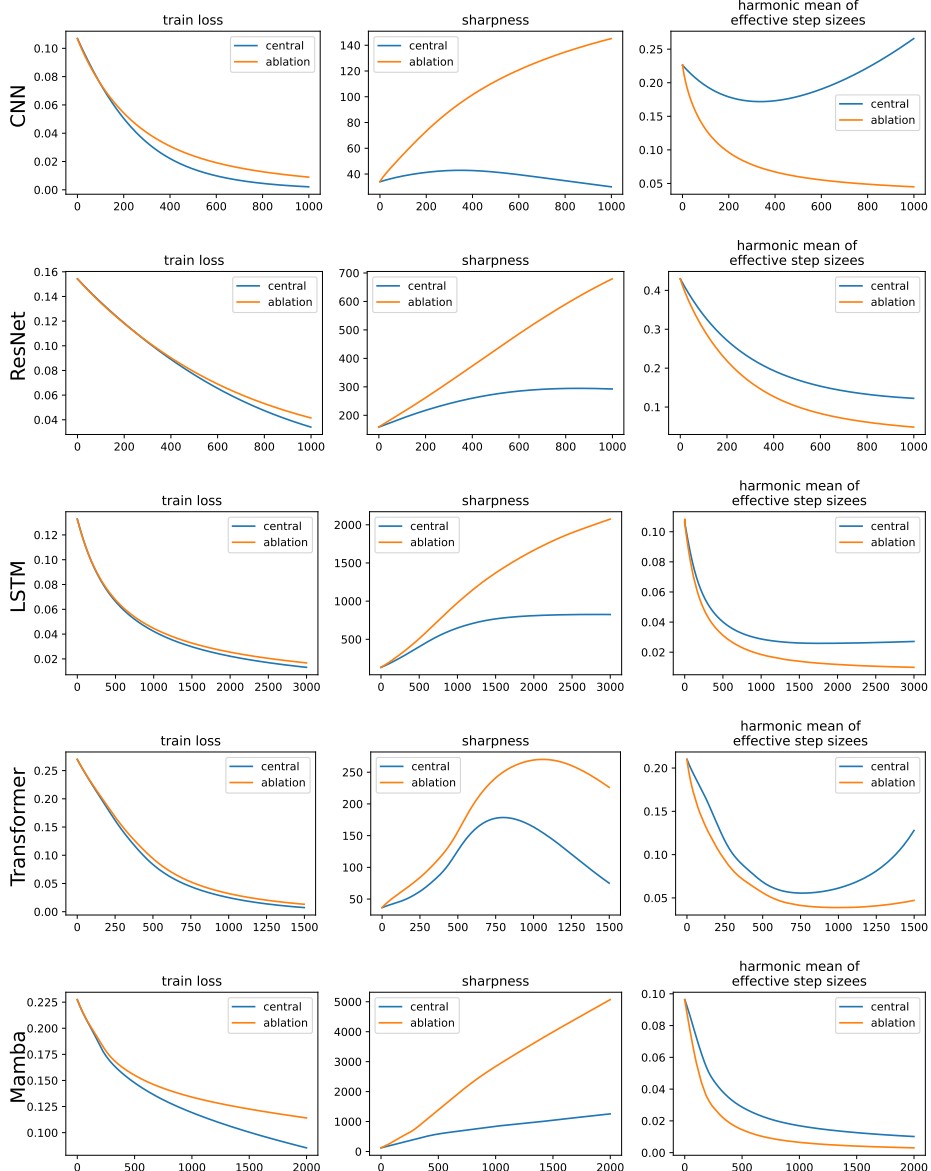

Figure 19: **"Acceleration via regularization" for RMSProp**. We compare the the RMSProp central flow (blue) to an ablated version (orange) which leaves out the implicit curvature regularization (maintaining stability purely by the effect of oscillations on $\nu$). At first, the central flow often optimizes slower than the ablated flow, due to the presence of implicit curvature regularization. But over time, it navigates to lower-curvature regions (right), where it takes larger steps (middle), and optimizes faster (left). Each row is a different DL setting. The left column plots the train loss, the middle column plots the harmonic mean of the effective learning rates, the right column plots the sharpness. These experiments all use MSE loss.

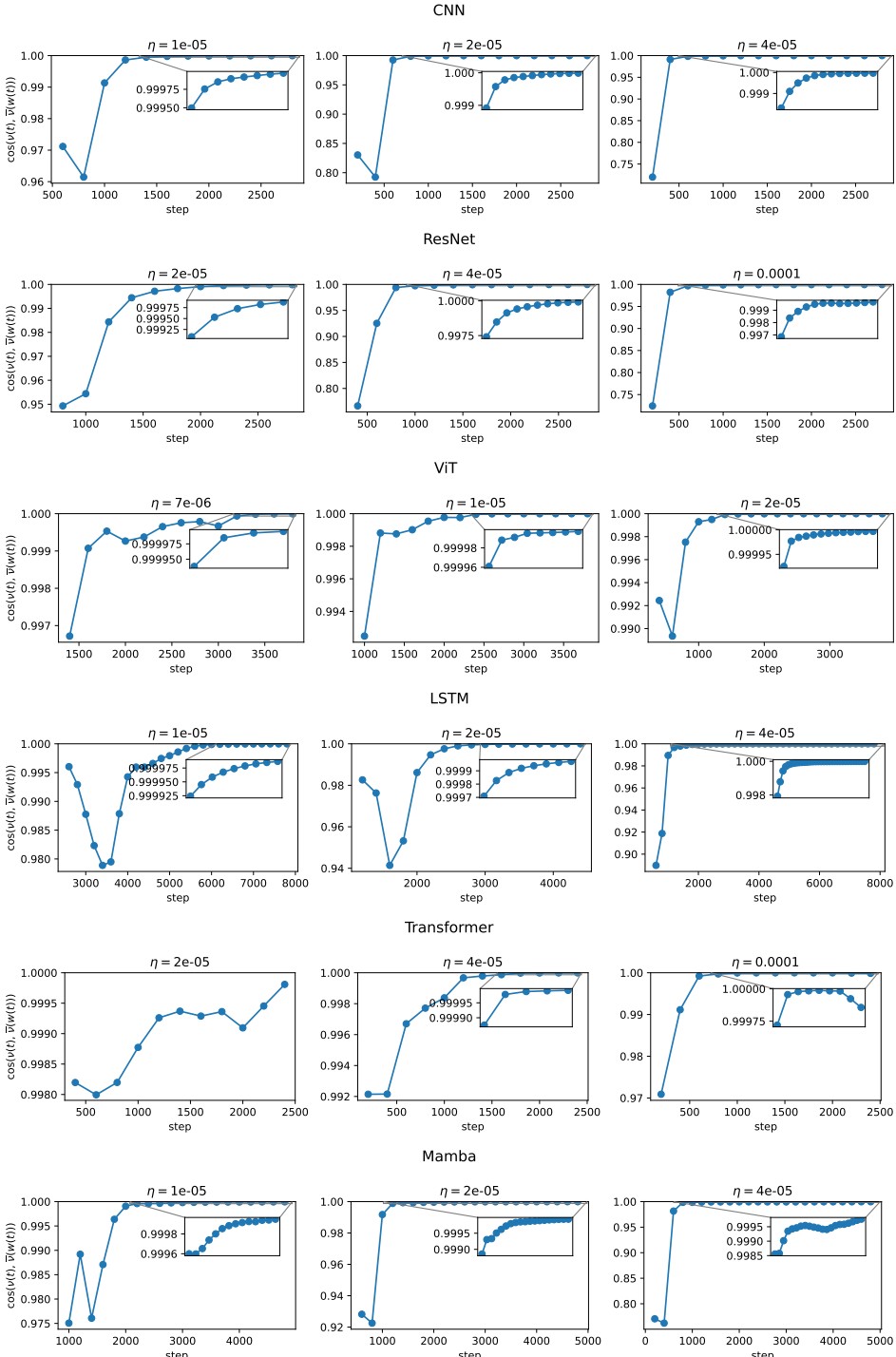

Figure 20: **The EMA $\nu$ reaches stationarity during training (MSE).** While running the RMSProp central flow, starting at the time when training enters EOS, we monitor the cosine similarity between the EMA $\nu(t)$ and the stationary EMA $\bar{\nu}(w(t))$. This cosine similarity rises to high values (nearly 1) during training, implying that $\nu(t)$ reaches stationarity.

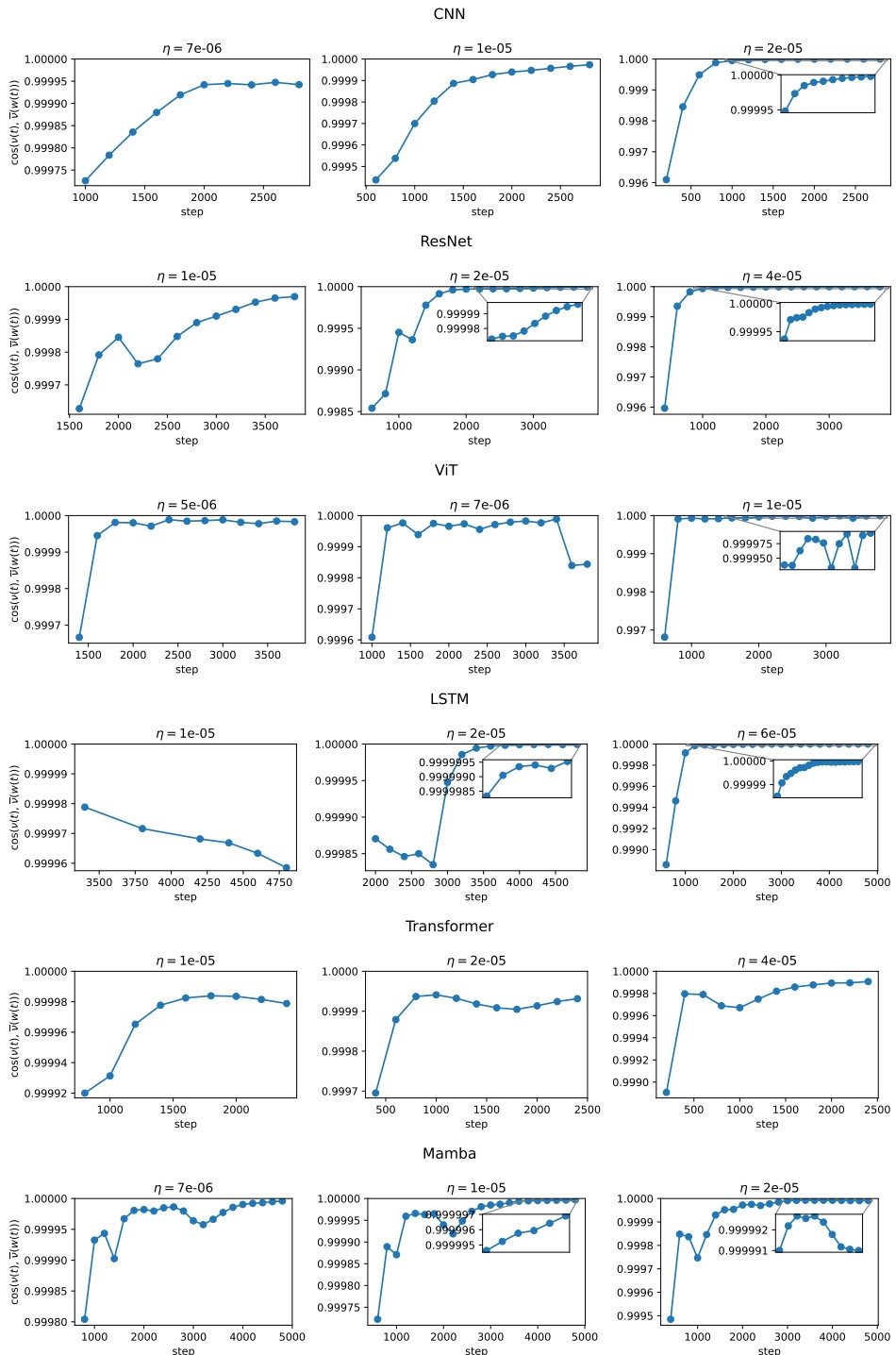

Figure 21: **The EMA $\nu$ reaches stationarity during training (cross-entropy).** This figure is analogous to Figure 20 but for cross-entropy loss.

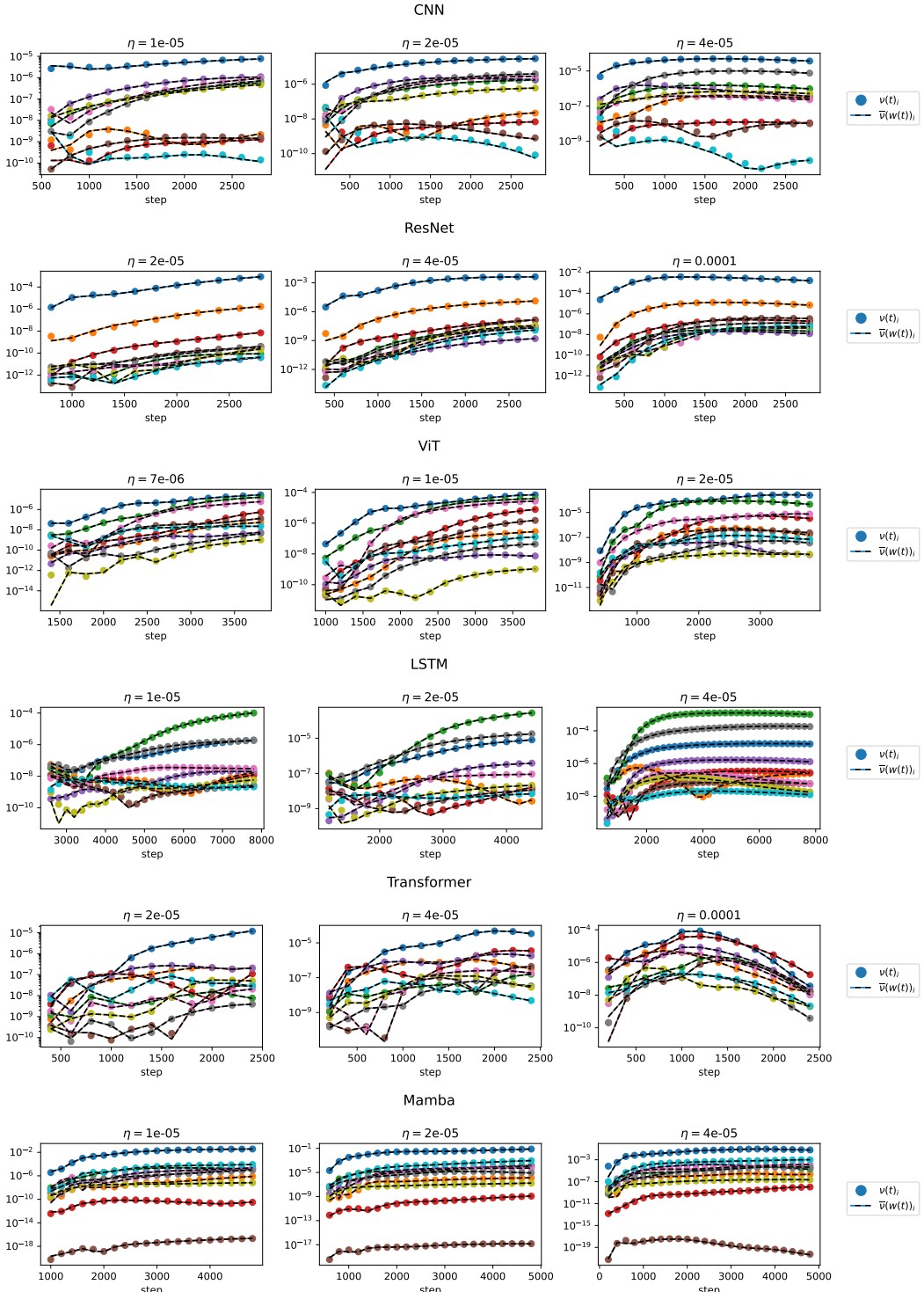

Figure 22: **Stationary EMA is accurate at a coordinate-wise level (MSE)**. While running the RMSProp central flow, starting at the time when training enters EOS, we plot the evolution of ten coordinates of the actual EMA $\nu(t)$ (dots) and the stationary EMA $\overline{\nu}(w(t))$ (half-black dashed lines). Each color is a different coordinate, and the ten coordinates are uniformly spaced throughout the network. We can see that, starting soon after training reaches EOS, the stationary EMA $\overline{\nu}(w(t))$ becomes an excellent approximation to the real EMA $\nu(t)$, on a coordinatewise level. We can also see that both the real EMA and the stationary EMA evolve significantly (in tandem) during this time.

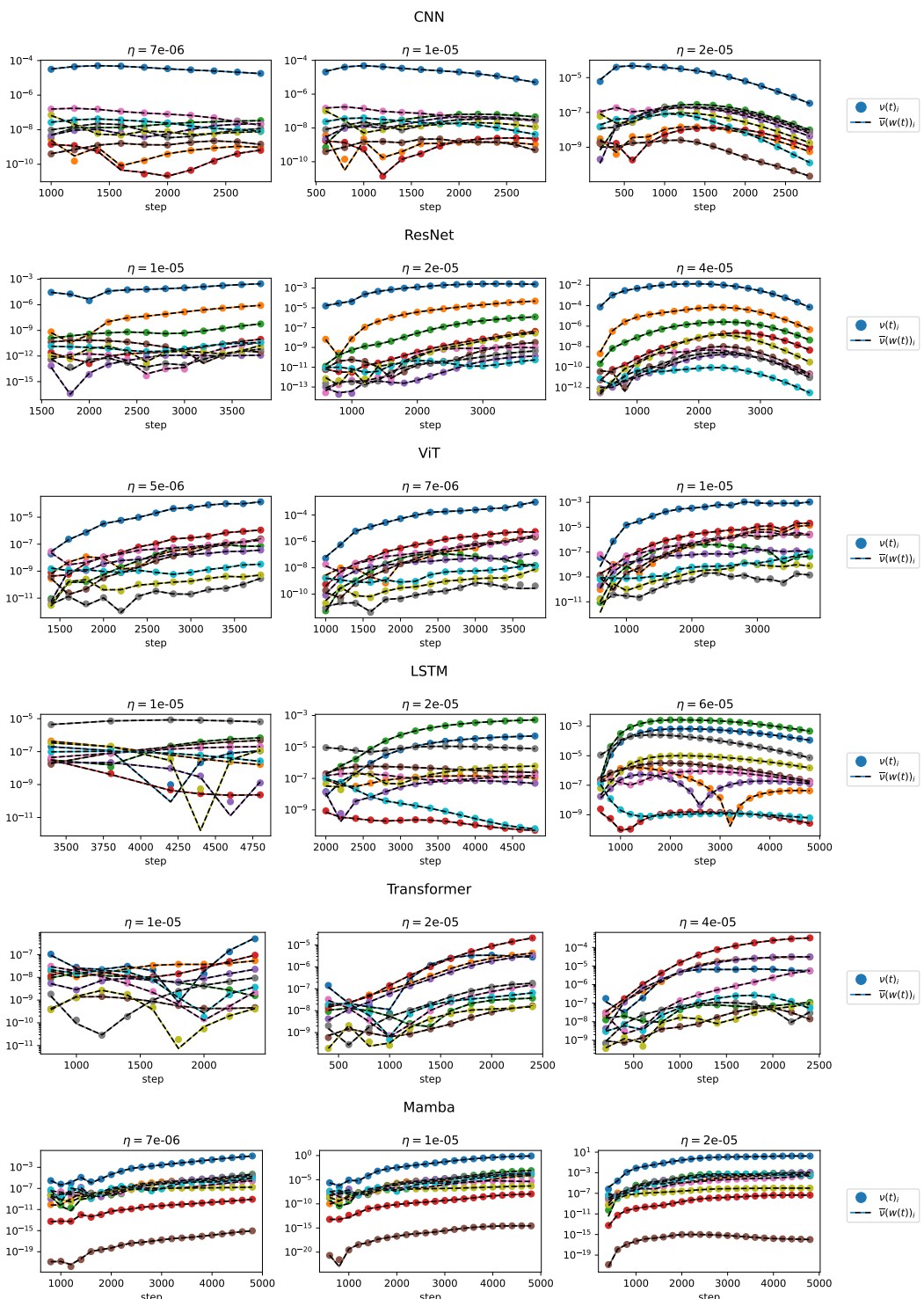

Figure 23: **Stationary EMA is accurate at a coordinate-wise level (CE)**. Analogous to Figure 22, but for cross-entropy loss.

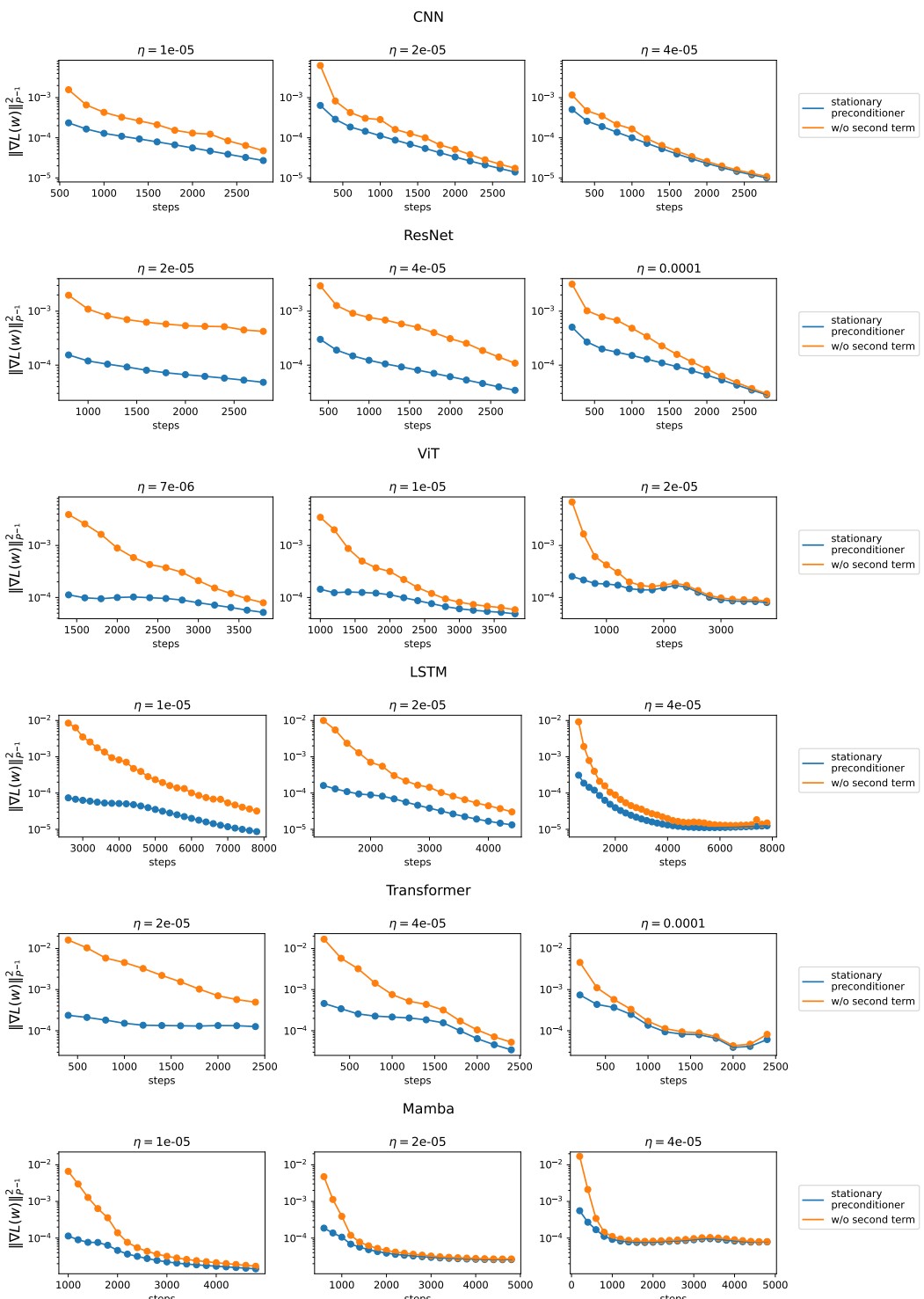

Figure 24: **RMSProp stationary preconditioner is suboptimal**. We compare the RMSProp stationary preconditioner, defined as the solution to the optimization problem eq. (11), to an alternative preconditioner defined as the solution to eq. (37), a similar optimization problem but without the second term in the objective. We assess each preconditioner $P$ by reporting $\|\nabla L(w)\|^2_{P^{-1}}$, the instantaneous rate of decrease in the loss under the preconditioned gradient flow with preconditioner $P$. Observe that this value is higher under the alternative preconditioner (orange) than under the RMSProp stationary preconditioner (blue), meaning that the alternative preconditioner would decrease the loss faster. The gap between the two preconditioners tends to be smaller when $\eta$ is larger, which makes sense, because the second term in eq. (11) is proportional to $\frac{1}{\eta^2}$.

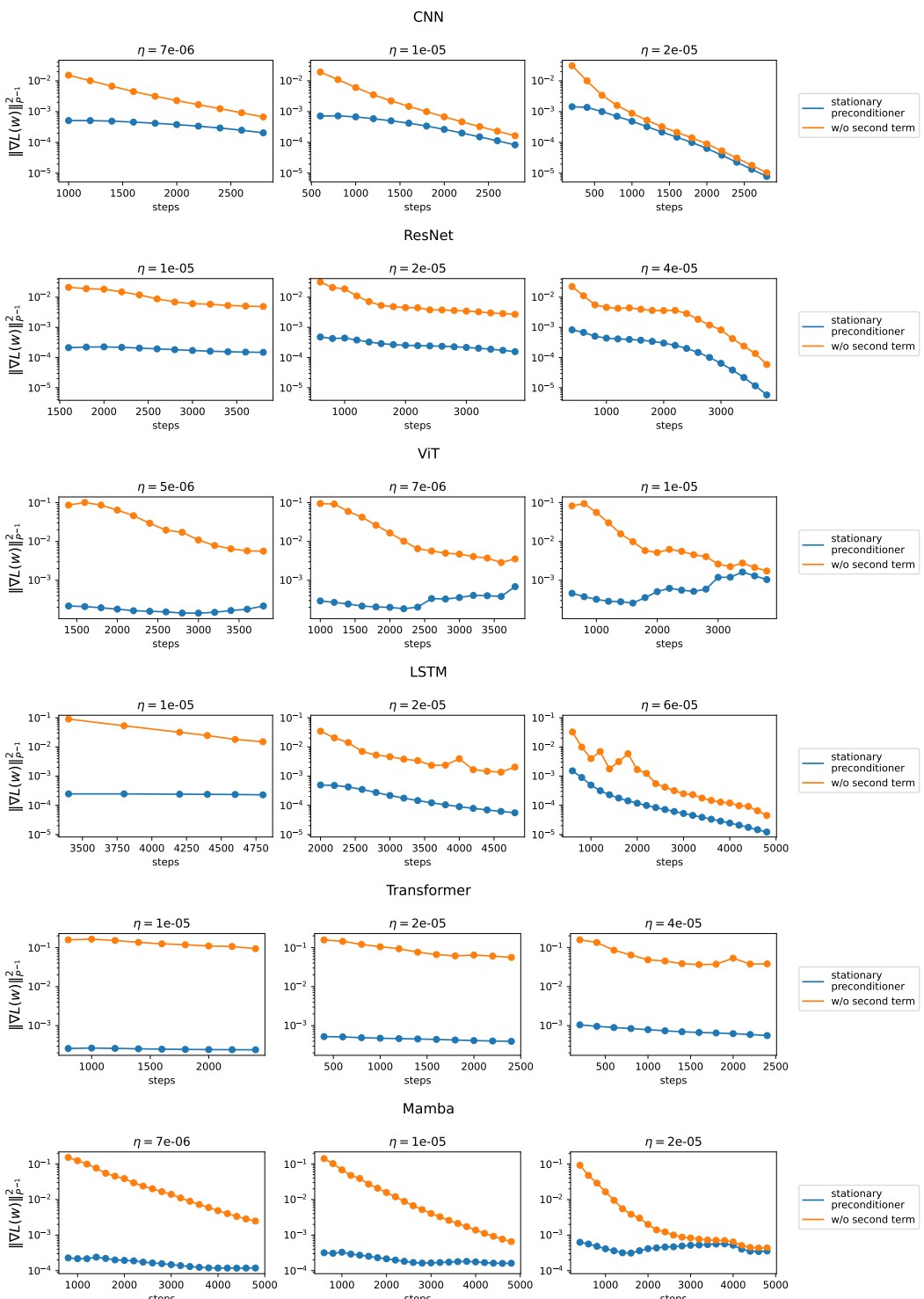

Figure 25: **RMSProp stationary preconditioner is suboptimal.** This figure is analogous to Figure 24, but for cross-entropy loss.

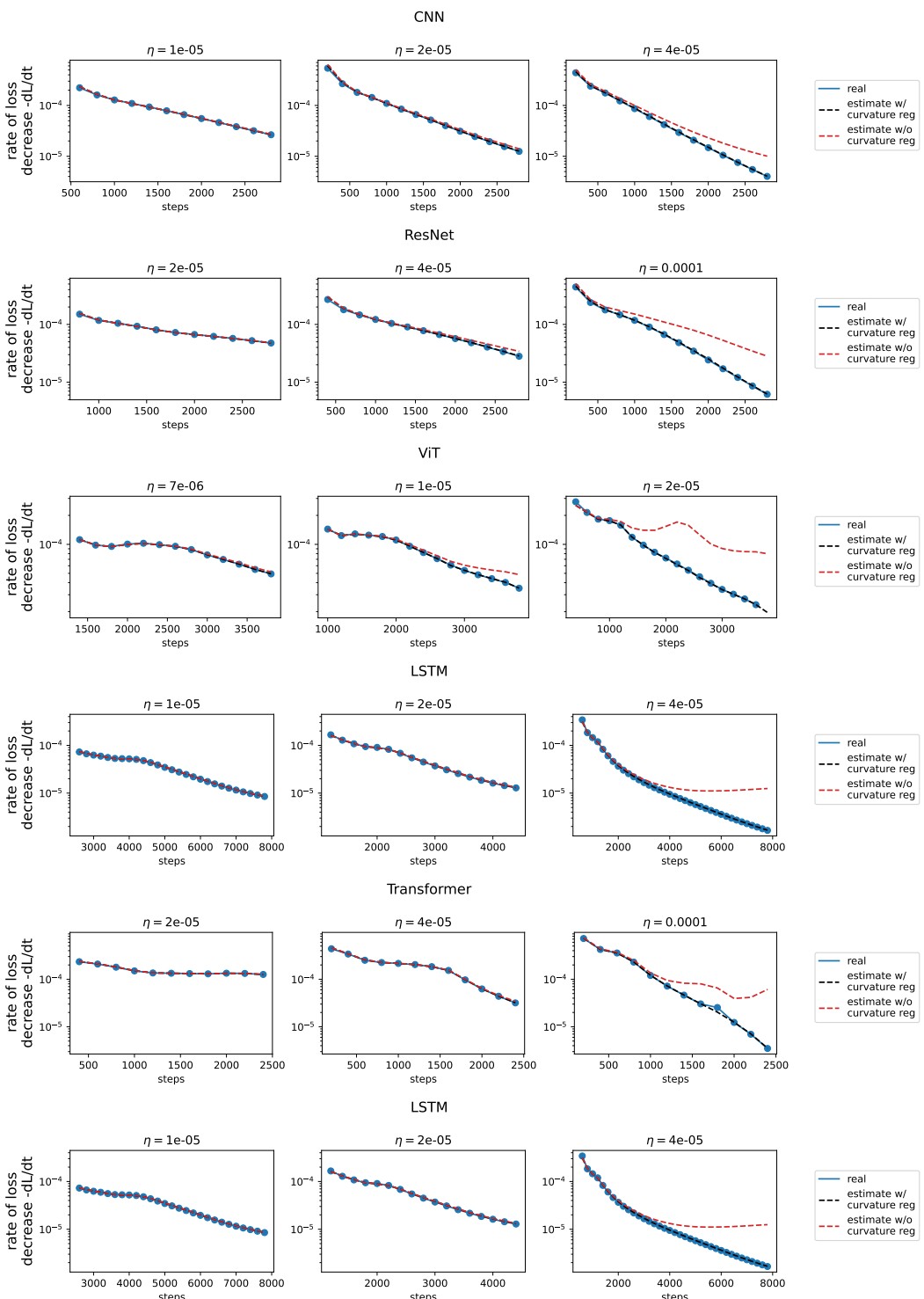

Figure 26: **Stationary flow accurately predicts the instantaneous speed of optimization (MSE)**. The stationary flow eq. (12), which incorporates an implicit curvature regularizer, predicts (black) the rate of loss decrease $-\frac{dL}{dt}$ (blue) more accurately than a naive estimate $\|\nabla L(w)\|^2_{\bar{P}^{-1}(w)}$ (in red) which uses the stationary preconditioner but does not incorporate curvature regularization. Observe that the gap between the two estimates is larger when $\eta$ is larger, suggesting that, like Scalar RMSProp, the implicit regularization of RMSProp increases in strength with $\eta$.

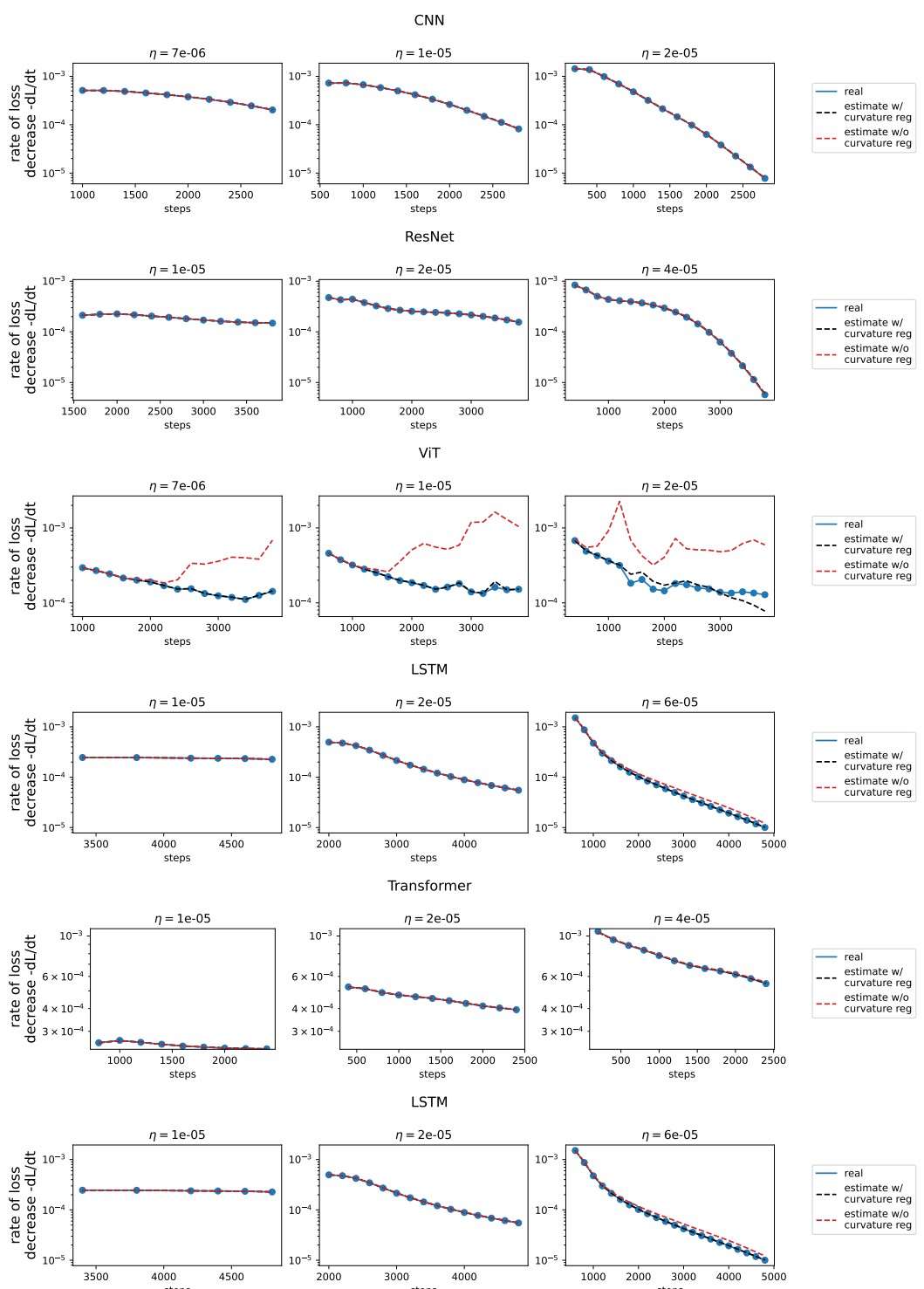

Figure 27: **Stationary flow accurately predicts the instantaneous speed of optimization (CE).** Same as Figure 26, but with cross-entropy loss.

## B.4 KNOWN FAILURE MODES FOR THE CENTRAL FLOWS

So long as an optimizer is initialized stably (i.e. with effective sharpness below the critical threshold), the central flow is intended to approximate the real optimizer trajectory over the long term. However,

this approximation can fail. Below, we describe some common failure modes which, in our experience, consistently cause the real optimization trajectory to deviate from the central flow. A promising direction for future work is to precisely identify conditions under which the central flow does or does not approximate the real optimizer trajectory.

**Sufficiently large learning rates**     For all three optimizers studied in this paper, we reliably observe that as the learning rate hyperparameter is made increasingly large, the real optimization trajectory tends to deviate more from the central flow, as illustrated in Figure 28. (As an extreme example, even if the optimizer is initialized stably, very large learning rates sometimes cause the real optimizer to explosively diverge in the middle of training, whereas this never happens under the central flow.) We do not know whether some corrected version of the central flow would be more successful at capturing the real optimization trajectory in these scenarios, or whether the real trajectory is simply too chaotic to be captured by any flow.

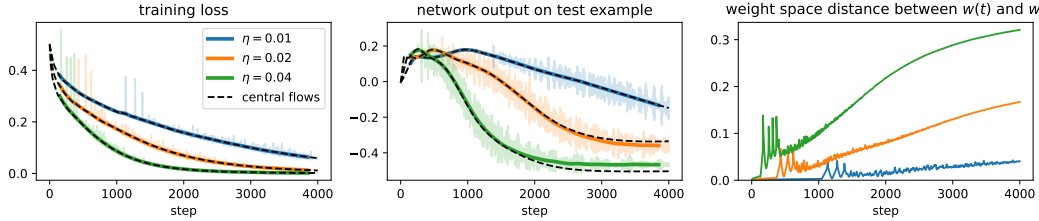

Figure 28: **Central flow approximation is less accurate at larger learning rates**. We run both gradient descent and its central flow at three learning rates (colors). The larger the learning rate, the faster the growth in the accumulated approximation error (right). Indeed, at larger learning rates, the network's output on an arbitrary test example can be visually seen to be slightly different between the central flow and gradient descent (middle). Nevertheless, the central flow approximation is still accurate enough here to accurately capture the train loss curves (left). (CNN / CIFAR-10 / MSE)

**Sub/super-quadraticity**     Sometimes, at the edge of stability, the sharpness does not equilibrate *exactly* at $2/\eta$; rather, it equilibrates slightly above or below $2/\eta$ (e.g. $2.1/\eta$ or $1.9/\eta$). Damian et al. (2023) showed that this is due to contributions from higher-order derivatives, beyond the third-order derivatives we study. For example, when the fourth derivative of the loss along the top eigenvector direction is positive, the local loss landscape is *super-quadratic*, and the sharpness at the iterates will be *above* $2/\eta$; when this fourth derivative is negative, the loss loss landscape is *sub-quadratic*, and the sharpness at the iterates will be *below* $2/\eta$. Since our derivation for the central flow assumes that the sharpness along the time-averaged trajectory is located *exactly* at $2/\eta$, this leads to a discrepancy between the central flow and the real optimizer which causes error to gradually accumulate over the long term. For the special case of one unstable eigenvalue, Damian et al. (2023) Appendix F derived a corrected version of their constrained trajectory (analogous to our central flow) which they empirically showed to match the real gradient descent trajectory even in the sub/super-quadratic setting. It would be interesting to try to re-derive this correction under the central flow framework, which would require extending this correction to the setting of multiple unstable directions.

**Smoothness of architecture**     The smoothness of the architecture (e.g. of the activation function) seems to affect the accuracy of the central flow approximation, although it is not clear how to precisely quantify "smoothness." In Figure 29, we show that as a network's activation function is interpolated from smooth to non-smooth, the accuracy the central flow approximation degrades, both in weight space and function space.

**Large spikes**     When the EOS dynamics lead to extremely large spikes (e.g. in the gradient norm), we have found such spikes can cause the real trajectory to deviate noticeably from the central flow. This may be related to the large learning rate issue described above.

**Interactions with loss criterion**     We have empirically found that sub/super-quadraticity issue and the large spike issue are more common with cross-entropy loss than with mean squared error loss (although we do not have a satisfactory explanation for these observations). Consequently, the central flow approximation is often more accurate under the MSE loss than the cross-entropy loss.

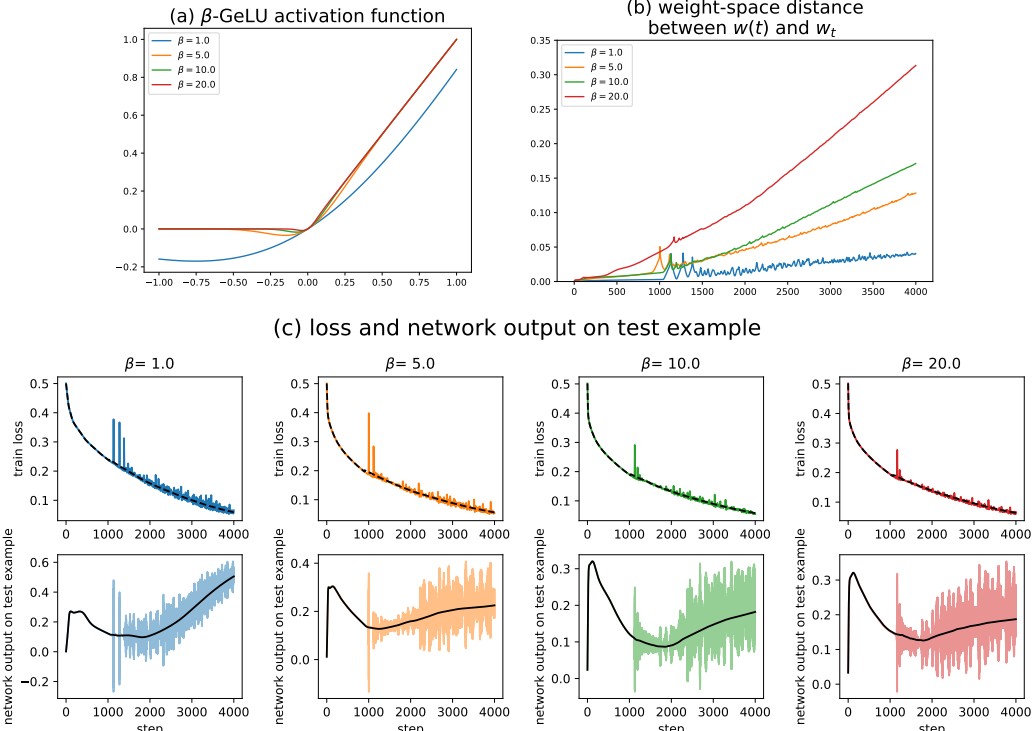

Figure 29: **Accuracy of central flow degrades as activation function becomes less smooth.** We consider networks with the $\beta$-GeLU activation function from Dauphin et al. (2024), defined as $x \mapsto x\Phi(\beta x)$ where $\Phi$ is the standard Gaussian CDF. This activation interpolates between (smooth) GeLU when $\beta = 1$ and (non-smooth) ReLU when $\beta = \infty$. Subfigure (a) plots this activation function with varying $\beta$. Subfigure (b) shows that when $\beta$ is larger (i.e. when the activation is less smooth), the approximation error between the central flow $w(t)$ and the optimizer trajectory $w_t$ grows faster. Subfigure (c) plots the loss curve, and the network's output on a test example, for both the optimizer trajectory and the central flow. Fortunately, even when $\beta = 20$, at which point $\beta$-GeLU is a very close approximation to ReLU, the central flow accurately predicts the overall training loss curve.

### B.5 IMPLEMENTATION DETAILS

Our code can be found at: `http://github.com/locuslab/central_flows`.

In order to reuse code between the central flows for gradient descent, Scalar RMSProp, and RMSProp, we view all three optimizers as instances of preconditioned gradient descent $w_{t+1} = w_t - P_t^{-1}\nabla L(w_t)$, as described in Appendix A.6. For example, for gradient descent, we have $P_t = \eta^{-1}I$. For all three optimizers, we define the effective Hessian as $P_t^{-1}H(w_t)$, and the EOS condition is that the largest eigenvalue of this matrix is 2.

**Eigenvalue computation** To regularly recompute the top eigenvalues and eigenvectors of the effective Hessian, we use the LOBPCG algorithm (Knyazev, 2001), which only requires access to the Hessian via Hessian-vector products, and which allows us to warm-start using the previously computed eigenvectors. We were originally inspired by the LOBPCG implementation in Jax's `jax.experimental.sparse.linalg` (Bradbury et al., 2018).

**Stable flows** To discretize the stable flows (e.g. gradient flow), we use Euler's method. We dynamically adapt the discretization step size based on the current effective sharpness. Discretizing the stable flow would take a prohibitively long time in regions of weight space where the effective sharpness is too high, so we automatically terminate the stable flow if the effective sharpness exceeds a certain threshold.

**Central flows**  To discretize the central flows, we use the scheme described in Appendix A.8. At each discretization time step, we re-estimate the top eigenvectors and eigenvalues of the effective Hessian, we compute the necessary third derivatives (quantities of the form $\nabla_w[u_i^T H(w)u_j]$, where $u_i, u_j$ are critical eigenvectors of the effective Hessian), we solve the CCP to compute $\Sigma(t)$ and to determine the update for $w$ (and, for adaptive optimizers, $\nu$), and we apply that update.

Discretizing the central flows is computationally expensive, due to the need to continually re-estimate the top eigenvectors/eigenvalues of the effective Hessian, and to compute the gradients of these eigenvalues (third derivatives). The time complexity of each discretization step grows quadratically with the number of eigenvalues that are at the edge of stability. In particular, if there are $k$ eigenvalues/eigenvectors at the edge of stability, then we need to compute a third derivative $\nabla_w[u_i^T H(w)u_j]$ for each distinct pair $u_i, u_j$ of these eigenvectors, including the case where $i = j$, which comes out to $k + \binom{k}{2} = \Theta(k^2)$ third derivative computations.

**Burn-in**  The quality of the central flow approximation is enhanced by starting it 10-15 steps into training. This is likely due to the size of the gradients during these first few steps, and the central flow could possibly be improved during this phase by incorporating the implicit gradient norm penalty from Barrett & Dherin (2021). Thus, most of our experiments first run the original optimizer for 10-15 steps and then use this as a starting point from which to run both the original optimizer and the central/stable flows.

**How many eigenvalues to track?**  For all processes (i.e. discrete optimizers, central flows, stable flows), we track all eigenvalues of the effective Hessian that are above the threshold $1.5$. We then track the same number of eigenvalues of the Hessian. Note that for gradient descent and Scalar RMSProp the Hessian eigenvalues are trivially related to the effective Hessian eigenvalues, whereas for RMSProp we need to do an extra eigenvalue solve to obtain the Hessian eigenvalues.

**Second-order midpoints**  When reporting metrics from the discrete optimizers (gradient descent, Scalar RMSProp, and RMSProp, as opposed to their central flows), we usually report the top Hessian eigenvalues measured not at the optimizer iterates $\{w_t\}$ themselves, but at the (second order) midpoints $\{\hat{w}_t\}$, defined as $\hat{w}_t := \frac{1}{4}[2w_t - w_{t-1} - w_{t+1}]$ (so named because it is the midpoint of the midpoints $\frac{1}{2}[w_t - w_{t-1}]$ and $\frac{1}{2}[w_{t+1} - w_t]$). This empirically makes the Hessian measurements crisper (less "noisy"), while not altering the fundamental patterns.

**Assessing the central flow's prediction for the oscillation covariance**  The central flow's $\Sigma(t)$ is intended to model the "instantaneous covariance matrix" of the oscillations at time $t$. Since $\Sigma(t)$ is continually evolving, it is not immediately obvious how to numerically verify this prediction. We now describe our solution, first for gradient descent. At time $t$, if the Hessian has $k$ eigenvalues at criticality, then $\Sigma(t)$ will have $k$ nonzero eigenvalues, which are intended to predict the instantaneous variance of the oscillations along the corresponding eigenvectors. Thus, as we simultaneously run both the optimizer and the central flow, at each time/step $t$ we measure the squared magnitude of the displacement between the discrete optimizer and the central flow along each eigenvector of $\Sigma(t)$, i.e. $(v_j^T(w_t - w(t)))^2$, where $v_j$ is the $j$-th eigenvector of $\Sigma(t)$. These quantities are plotted in faint lines colored lines in plots such as Figure 3 (third plot), and each eigenvalue/eignvector $j$ is given a different color. Note that the eigenvectors and eigenvalues of $\Sigma(t)$ are gradually evolving (except at breakpoints where a new Hessian eigenvalue enters or leaves EOS, causing $\Sigma(t)$ to be discontinuous). We then smooth these time series by convolving them with a Gaussian kernel, and plot the resulting smoothed time series as thick colored lines in plots such as Figure 3 (third plot). These are our empirical measurements of the variance along the different eignevector directions. We then compare these to the eigenvalues of $\Sigma(t)$, which are our predictions for these variances.

For the two adaptive optimizers, we assess whether the *generalized* eigenvalues of $\Sigma(t)$ with respect to the preconditioner $P(t)$ accurately predict the instantaneous variance of the oscillations along the corresponding *generalized* eigenvectors. That is, if $v_j$ is a generalized eigenvector of $\Sigma(t)$ so that $H(w(t))v = 2P(t)v$, then we measure $(v_j^T P(w_t - w(t)))^2$ and compare the Gaussian smoothing of that quantity to the $j$-th generalized eigenvalue of $\Sigma(t)$.

For the experiments in Appendix B and Appendix C, for each optimization algorithm (GD, Scalar RMSProp, RMSProp), we used a fixed Gaussian kernel bandwidth for all experiments and for all time within each experiment. However, we believe that it may be more appropriate to use different bandwidths at different times within a single run, and we used this in Figures 3, 5 and 7.

**Numerical issues in the central flow**    In some runs (e.g. Figure 34.17 ), the central flow's $\Sigma(t)$ predictions are erratic for stretches. This phenomenon seems to be a numerical issue that arises when discretizing the flow. We are optimistic that the issue is fixable.

### B.6    ARCHITECTURE DETAILS

Here we describe our architectures. Note that our code for all architectures can be found at:
`http://github.com/locuslab/central_flows`.

> Both our derivations and the analytic formulas for the central flows rely on higher-order information about the loss function (e.g. Hessians and third derivatives). As a result, we require that all of the architectures are smooth. This rules out the commonly used ReLU activation (Nair & Hinton, 2010). Note that it might be possible to run central flows on ReLU nets by replacing ReLU with a close approximation that is smooth, as in Figure 29.

**CNN**    Our CNN has four layers, an initial channel width of 32, and 3x3 convolutional kernels. It uses the GeLU activation function, average pooling, and a linear readout layer.

**ResNet**    We use a ResNet (He et al., 2016) with 20 layers and GeLU activations. We use GroupNorm (Wu & He, 2018) in place of BatchNorm (Ioffe & Szegedy, 2015), as we empirically find that BatchNorm often leads to sub/super-quadraticity (see the discussion in Appendix B.4).

**Vision Transformer**    We use the Vision Transformer (ViT) (Dosovitskiy et al., 2021) implementation from LucidRains (2024). Our ViT has depth 3, embedding dimension 64, number of heads 8, MLP dimension 256, and patch size 4. We use the ViT modifications proposed in Beyer et al. (2022). We initialize the weights and biases of the final linear layer to be zero, as this makes the curvature low at initialization. For unknown reasons, we found that the core PyTorch LayerNorm implementation (written in C++) leads to third derivatives being computed incorrectly; thus, we substituted in an alternative implementation written in vanilla PyTorch, which empirically fixed the issue.

**LSTM**    Our LSTM (Hochreiter & Schmidhuber, 1997) has 2 layers, an embedding dimension of 48, and a hidden dimension of 48.

**(Sequence) Transformer**    Our sequence transformer has 4 layers, an embedding dimension of 32, an MLP dimension of 128, and 4 attention heads. We disabled dropout, to make the network deterministic. As with the ViT (see above), we initialize the weights and biases of the final linear layer to be zero, and we substitute a vanilla PyTorch LayerNorm implementation in place of the default C++ LayerNorm implementation.

**Mamba**    We use the Mamba (Gu & Dao, 2024) implementation from Torres-Leguet (2024). Our Mamba has 2 layers and a model dimension of 64.

### B.7    DATASET DETAILS

Here we describe our datasets. The code can be found at:
`http://github.com/locuslab/central_flows`.

**CIFAR-10**    We test the vision architectures on a subset of CIFAR-10 that contains 1000 training examples, all from the first 4 CIFAR-10 classes. We use the standard preprocessing of subtracting the dataset-wide channel-wise mean, and dividing by the dataset-wide channel-wise standard deviation. When training using MSE loss, we encode the ground truth class as 1 and the others as 0.

**Sorting**    We test the sequence architectures on the synthetic sorting task described in Karpathy (2020). The network is fed a sequence of numbers and is then tasked (via a language modeling loss) with returning these numbers in sorted order. We used numbers 1 through 4, and sequences of length 8. The size of the training datraset was usually 1,000 (except for Mamba, where it was 250).

## C    BULK EXPERIMENTAL DATA

This section contains the "bulk" experimental data from all of our experiments. Appendix C.1 has GD experiments, Appendix C.2 has Scalar RMSProp experiments, and Appendix C.3 has RMSProp experiments. See Figure 11 for a fully annotated example of a gradient descent trajectory, Figure 14 for a fully annotated example of a Scalar RMSProp trajectory, and Figure 18 for a fully annotated example of a RMSProp trajectory.

## C.1 GRADIENT DESCENT

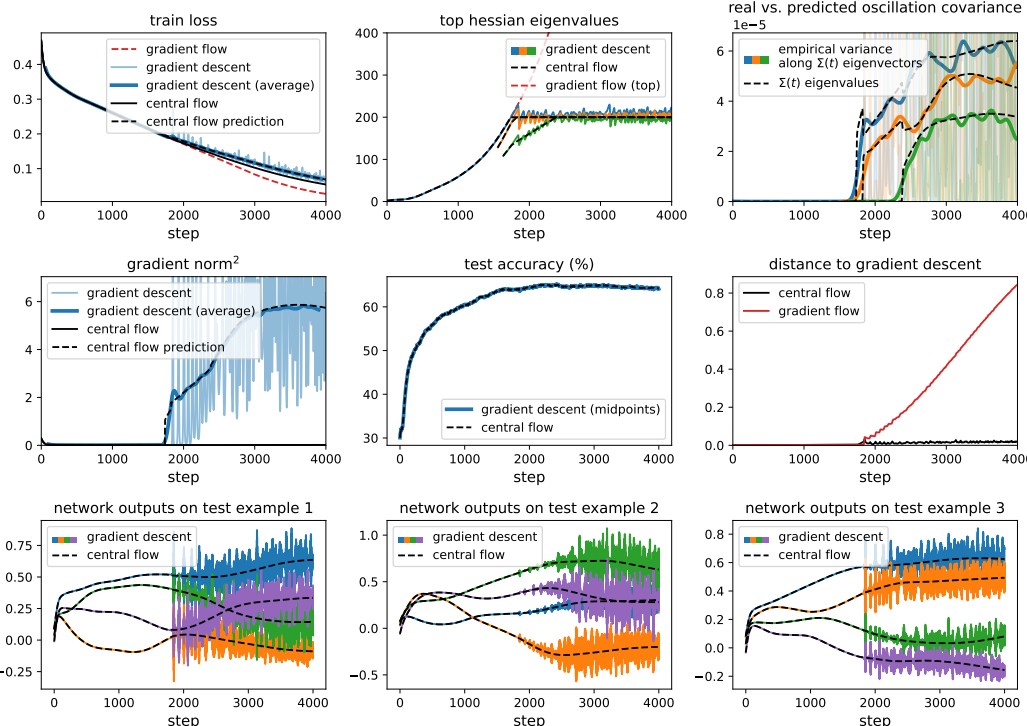

Figure 30.1: Gradient descent central flow for a CNN with MSE loss, $\eta = 0.01$.

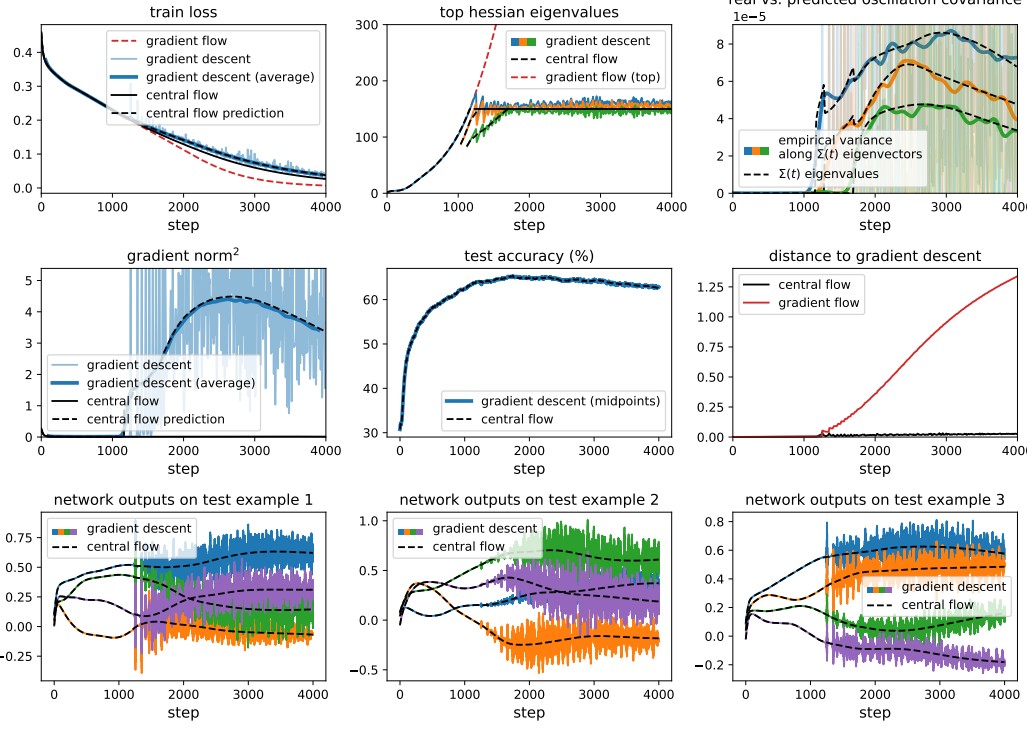

Figure 30.2: Gradient descent central flow for a CNN with MSE loss, $\eta = 0.013333$.

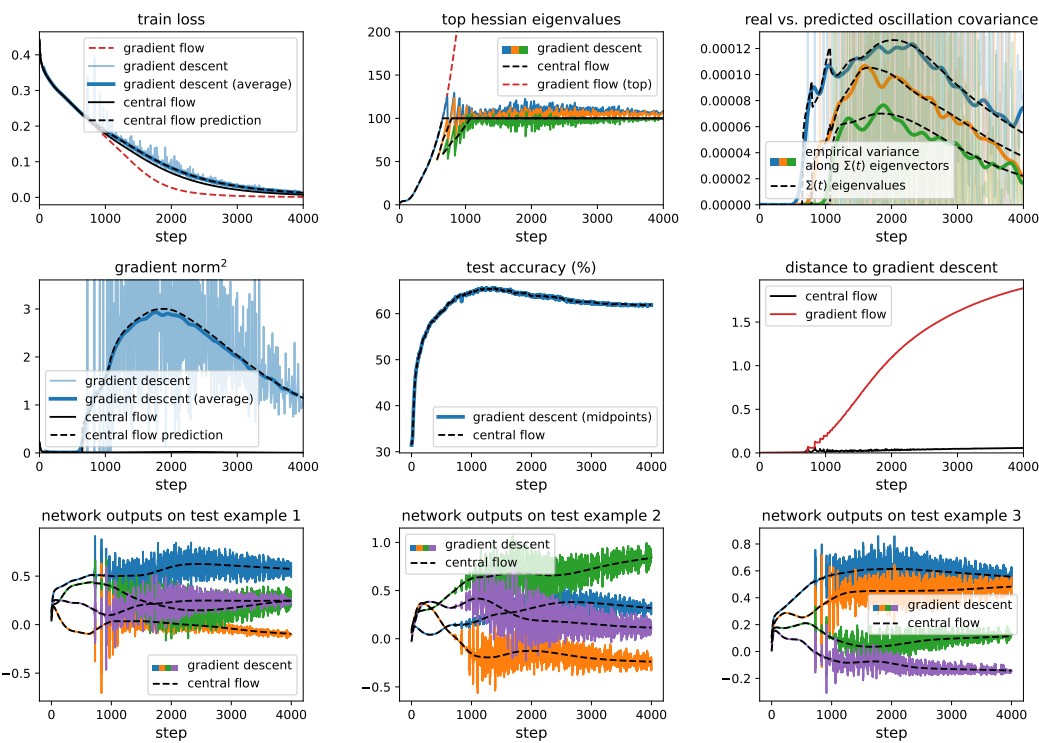

Figure 30.3: Gradient descent central flow for a CNN with MSE loss, $\eta = 0.02$.

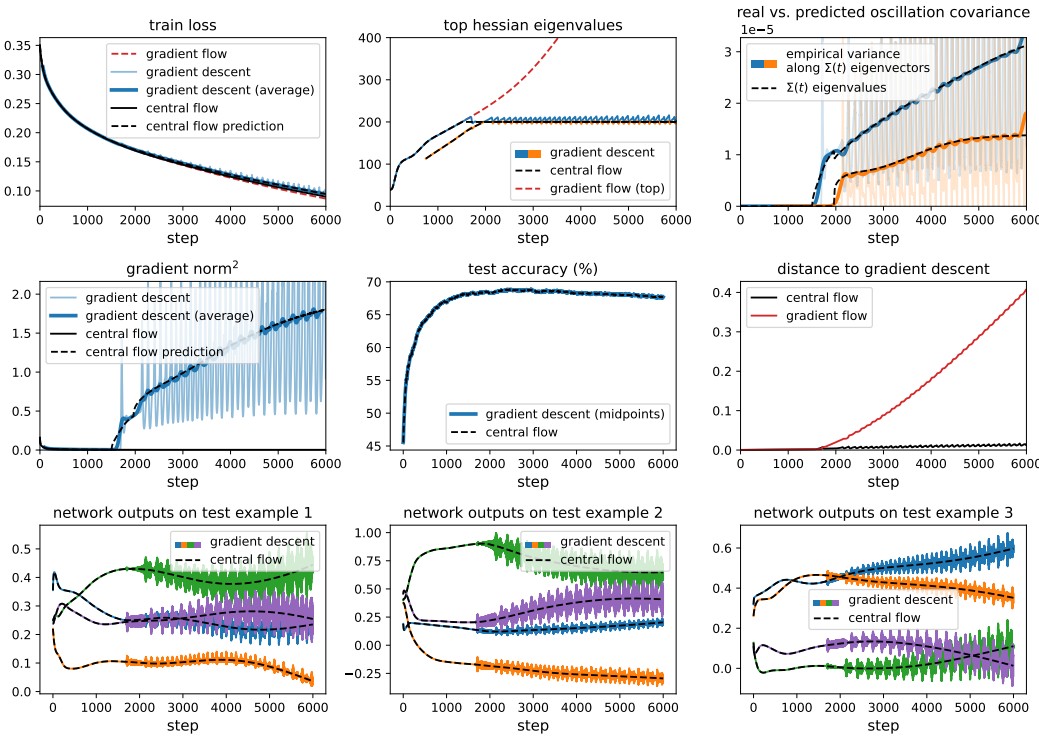

Figure 30.4: Gradient descent central flow for a ResNet with MSE loss, $\eta = 0.01$.

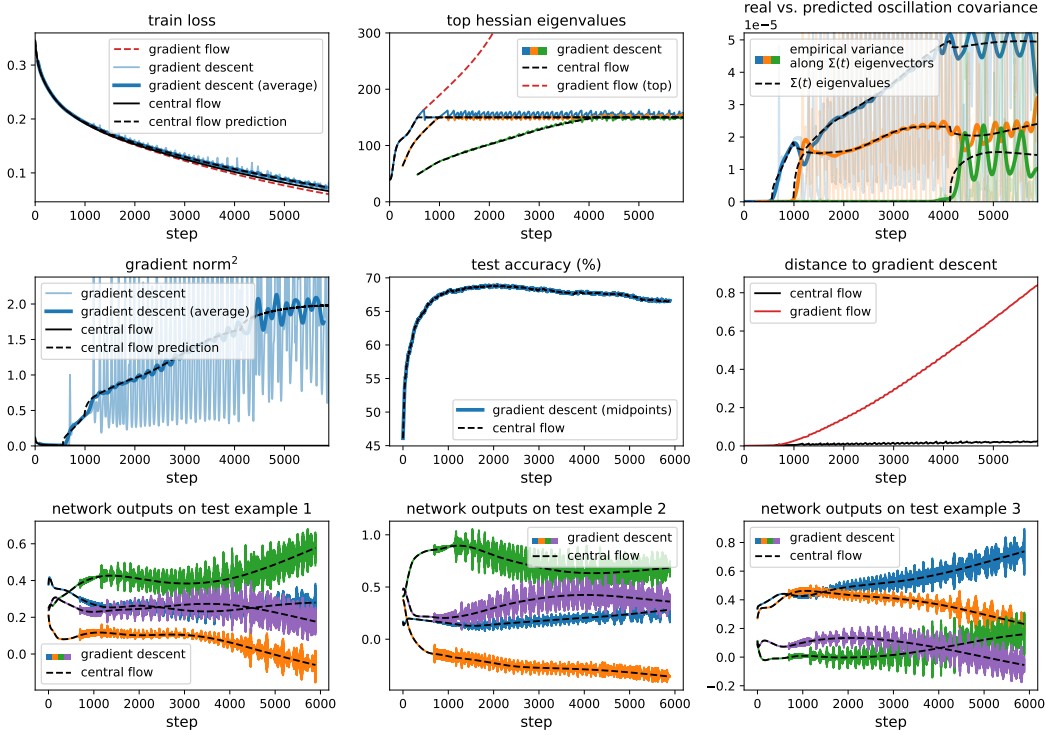

Figure 30.5: Gradient descent central flow for a ResNet with MSE loss, $\eta = 0.013333$.

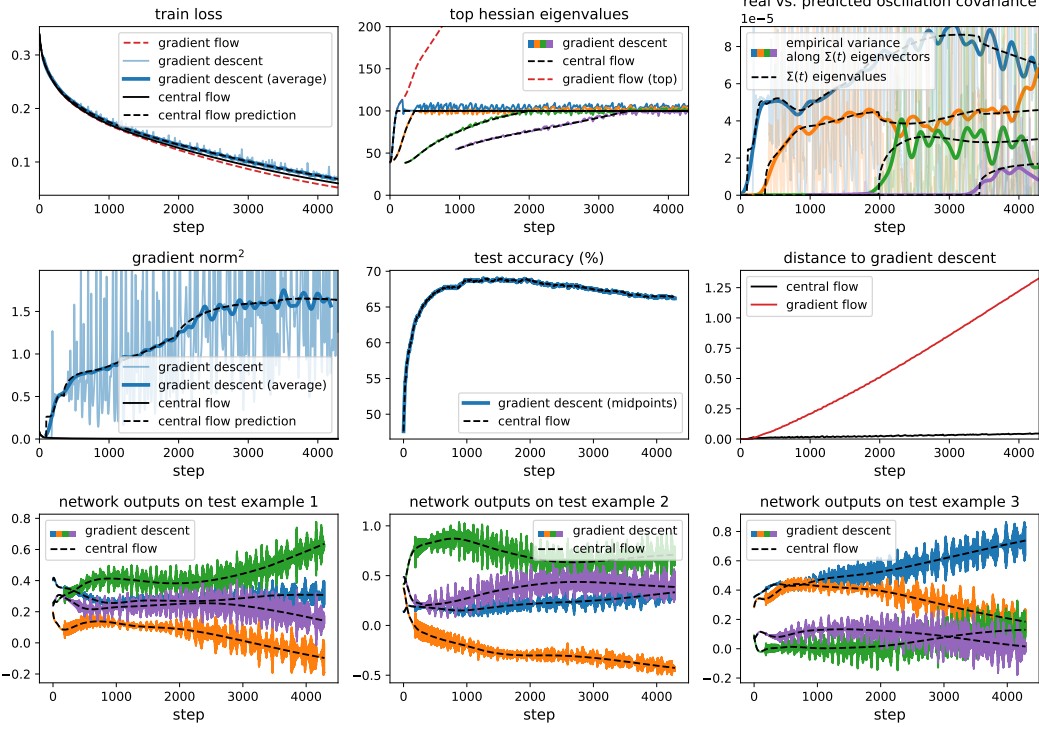

Figure 30.6: Gradient descent central flow for a ResNet with MSE loss, $\eta = 0.02$.

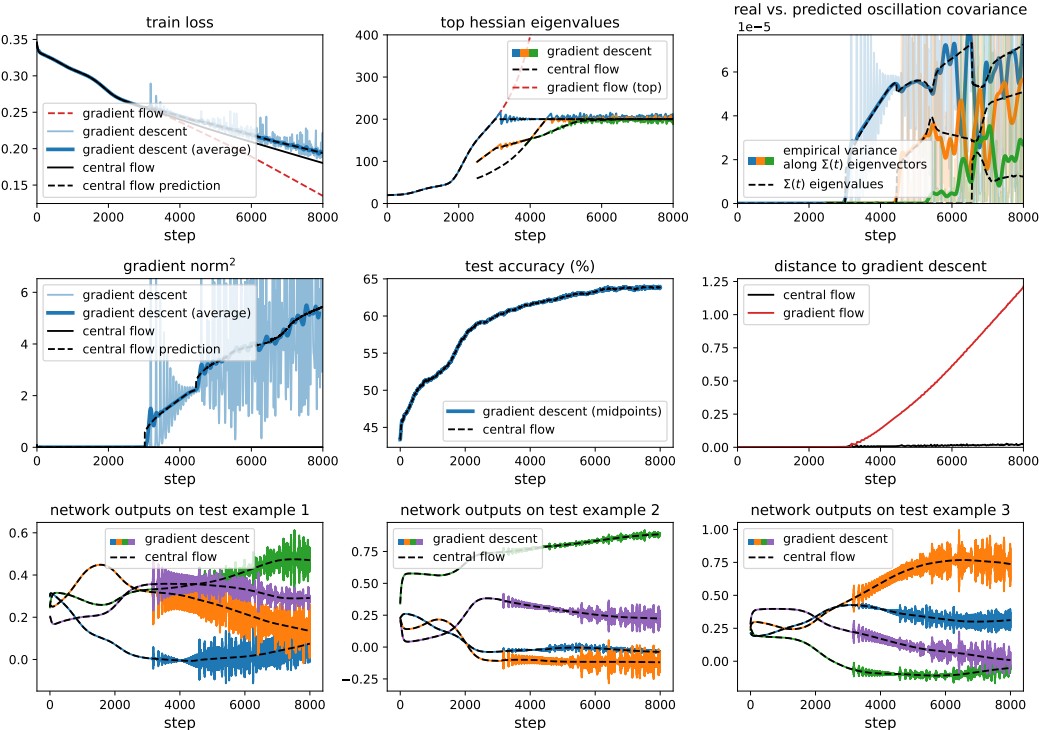

Figure 30.7: Gradient descent central flow for a VIT with MSE loss, $\eta = 0.01$.

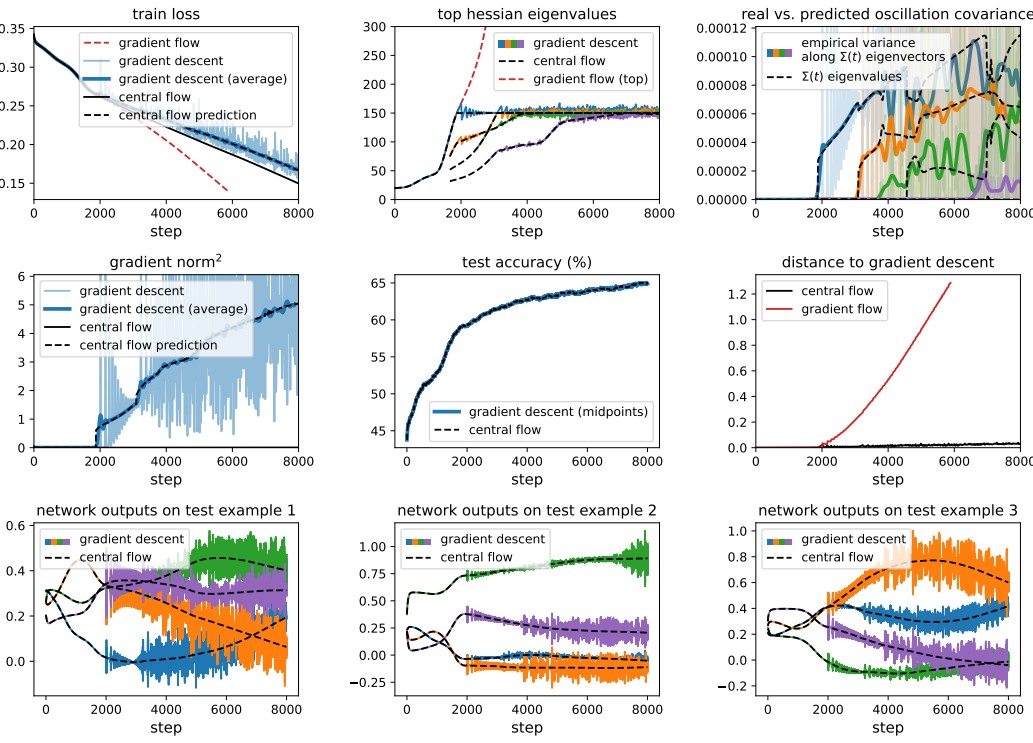

Figure 30.8: Gradient descent central flow for a VIT with MSE loss, $\eta = 0.0133333$.

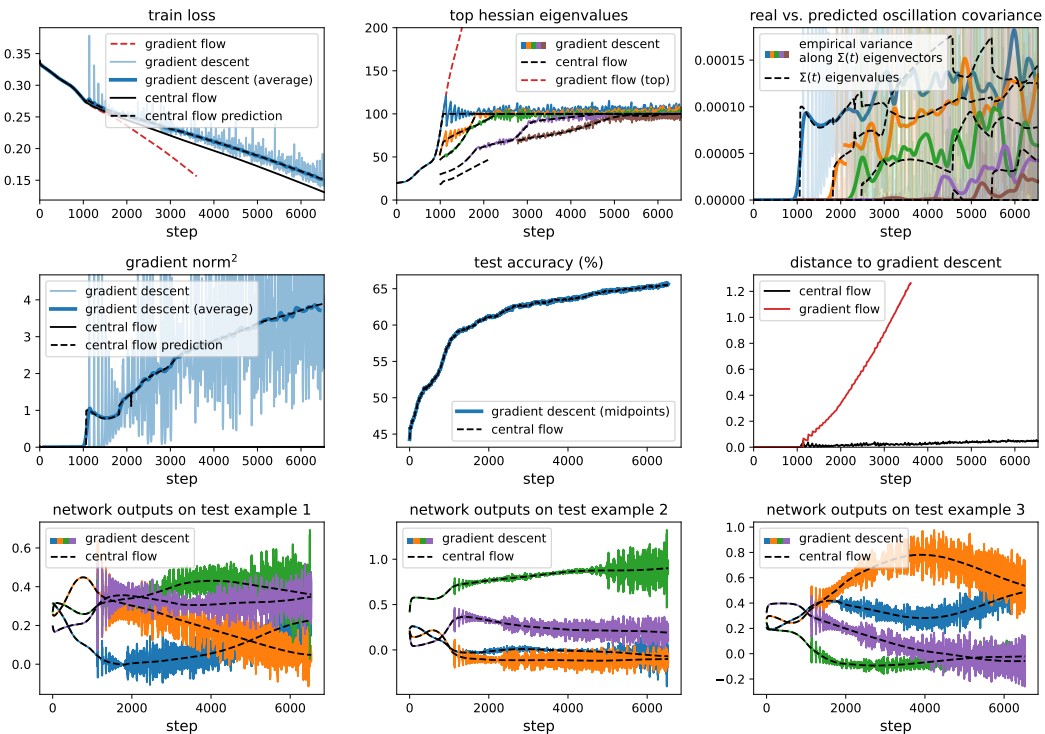

Figure 30.9: Gradient descent central flow for a VIT with MSE loss, $\eta = 0.02$.

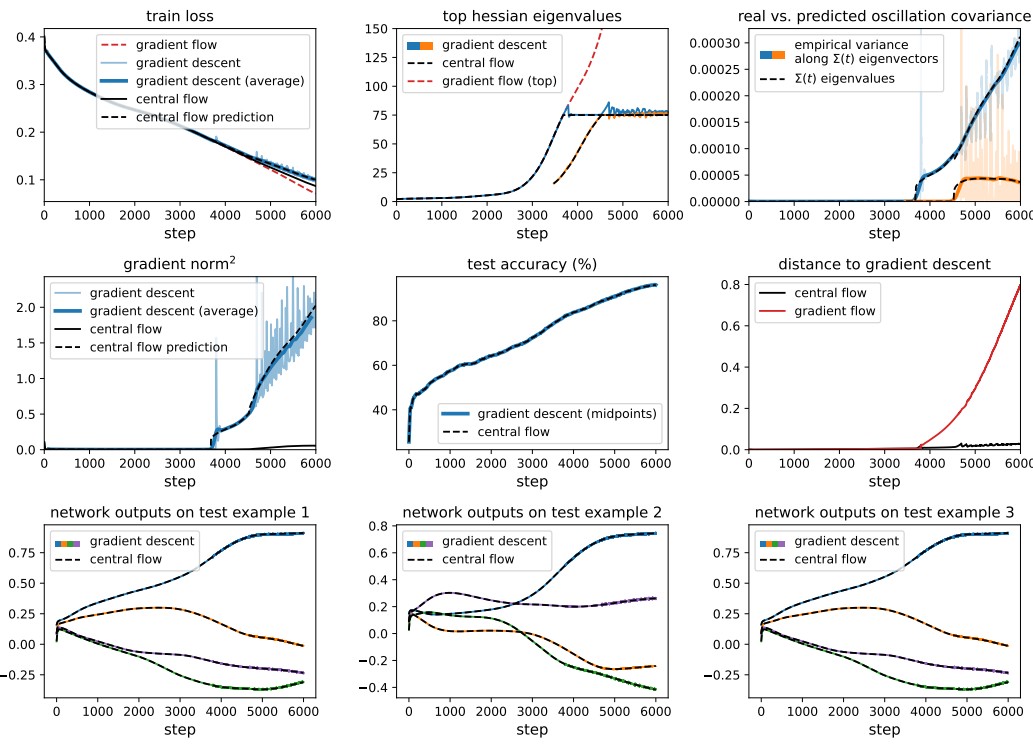

Figure 30.10: Gradient descent central flow for a LSTM with MSE loss, $\eta = 0.02666$.

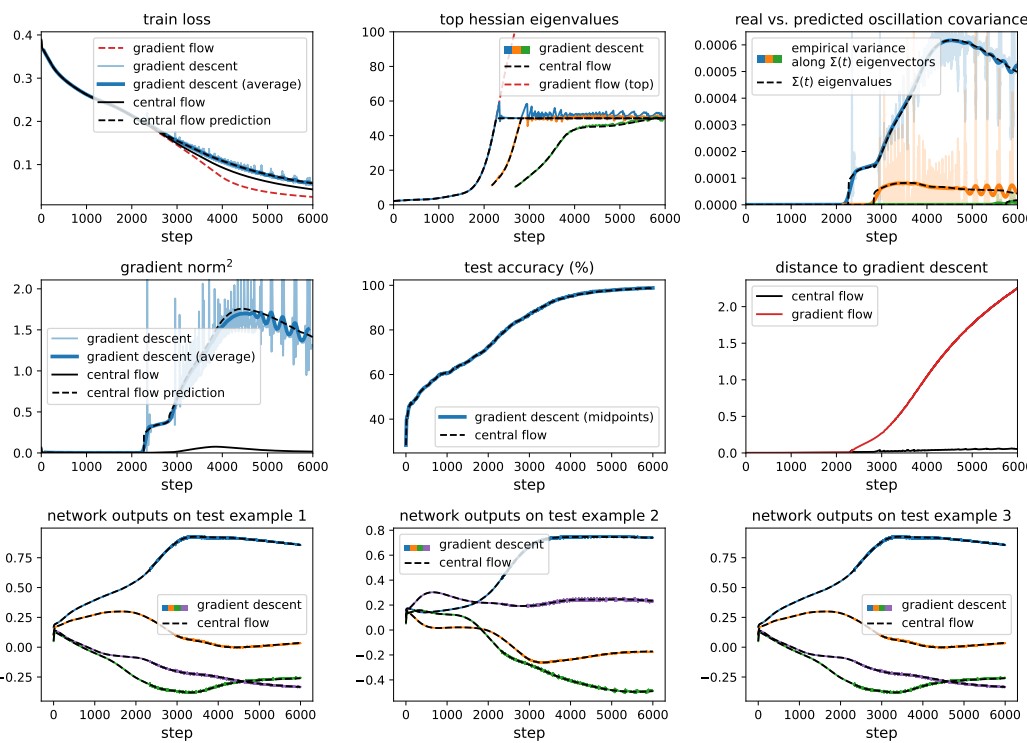

Figure 30.11: Gradient descent central flow for a LSTM with MSE loss, $\eta = 0.04$.

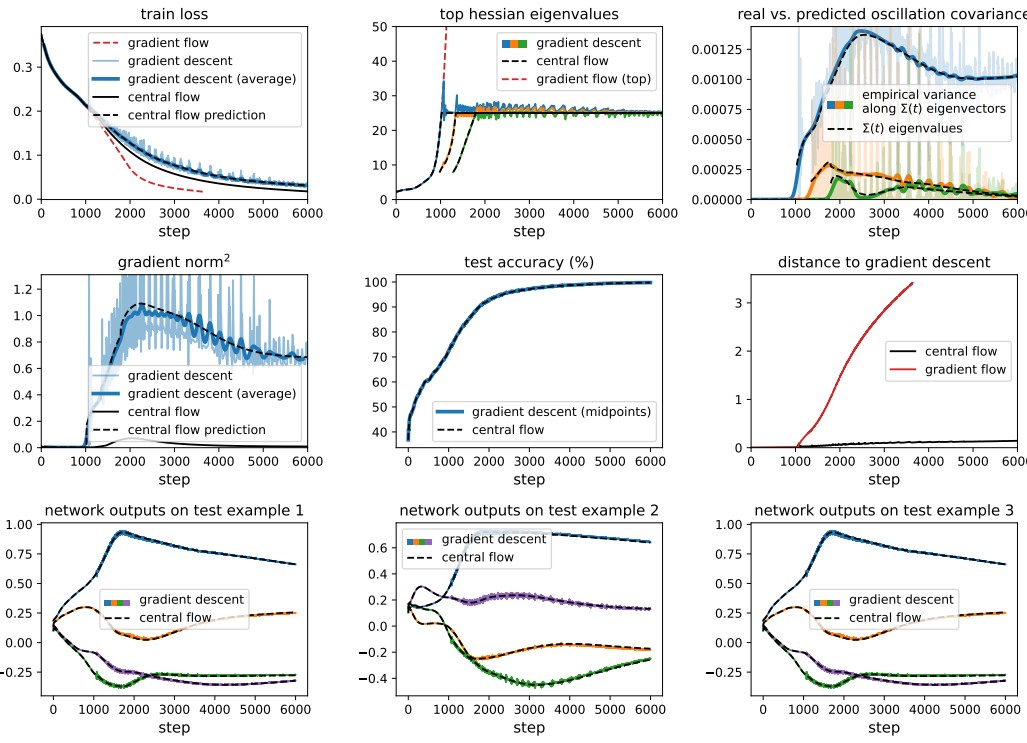

Figure 30.12: Gradient descent central flow for a LSTM with MSE loss, $\eta = 0.08$.

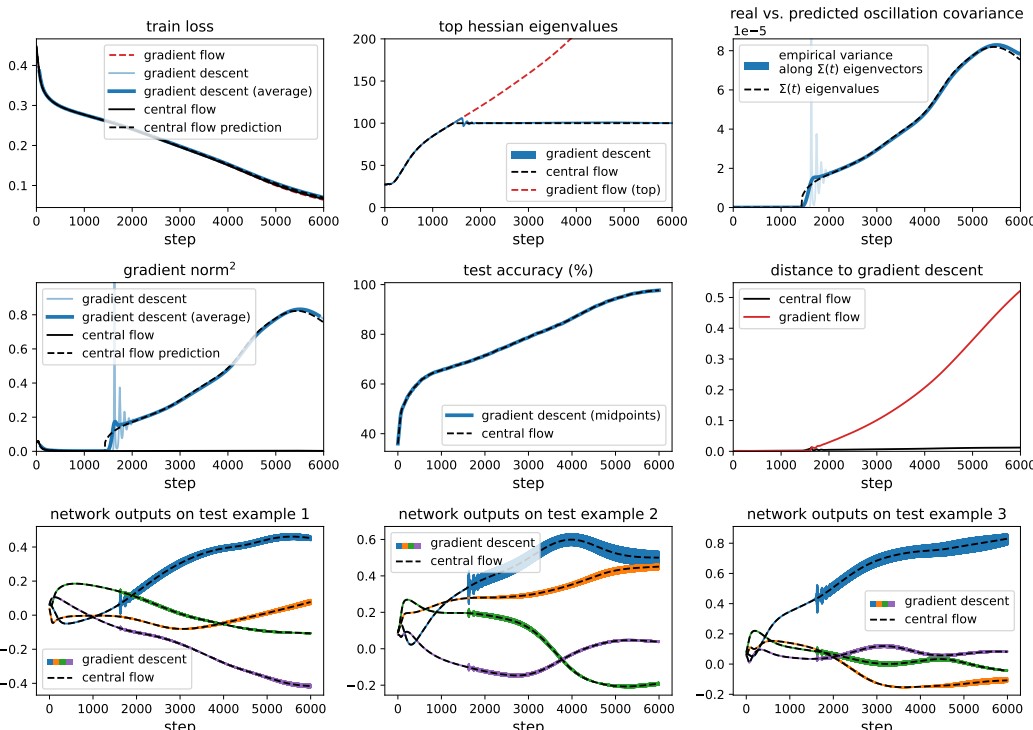

Figure 30.13: Gradient descent central flow for a Transformer with MSE loss, $\eta = 0.02$.

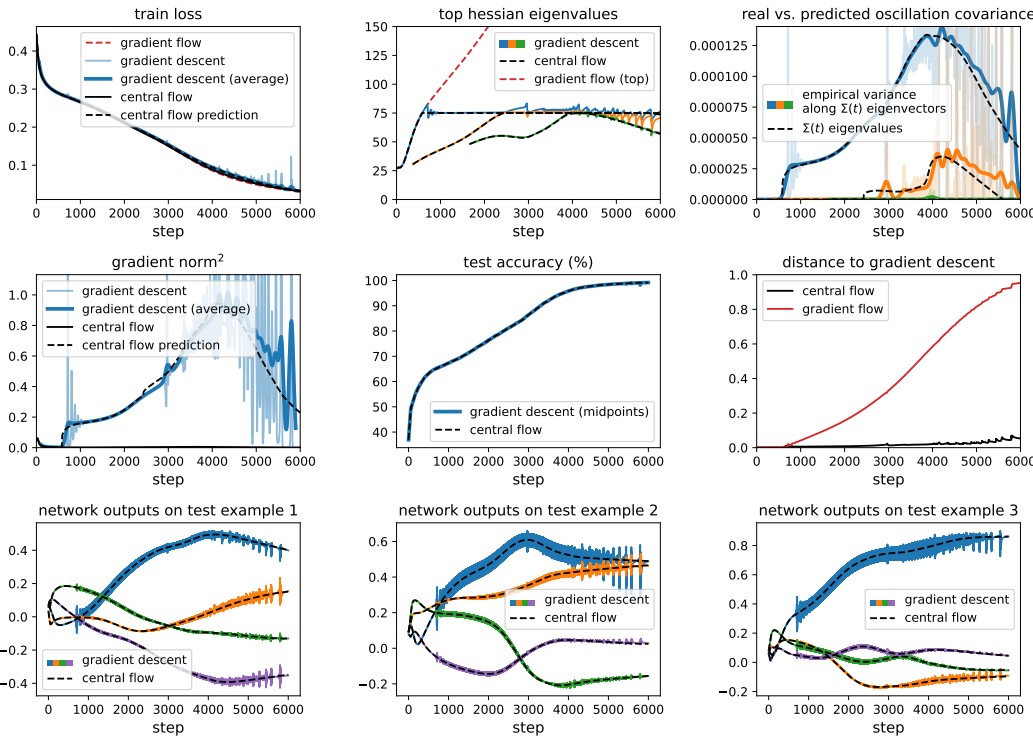

Figure 30.14: Gradient descent central flow for a Transformer with MSE loss, $\eta = 0.02666$.

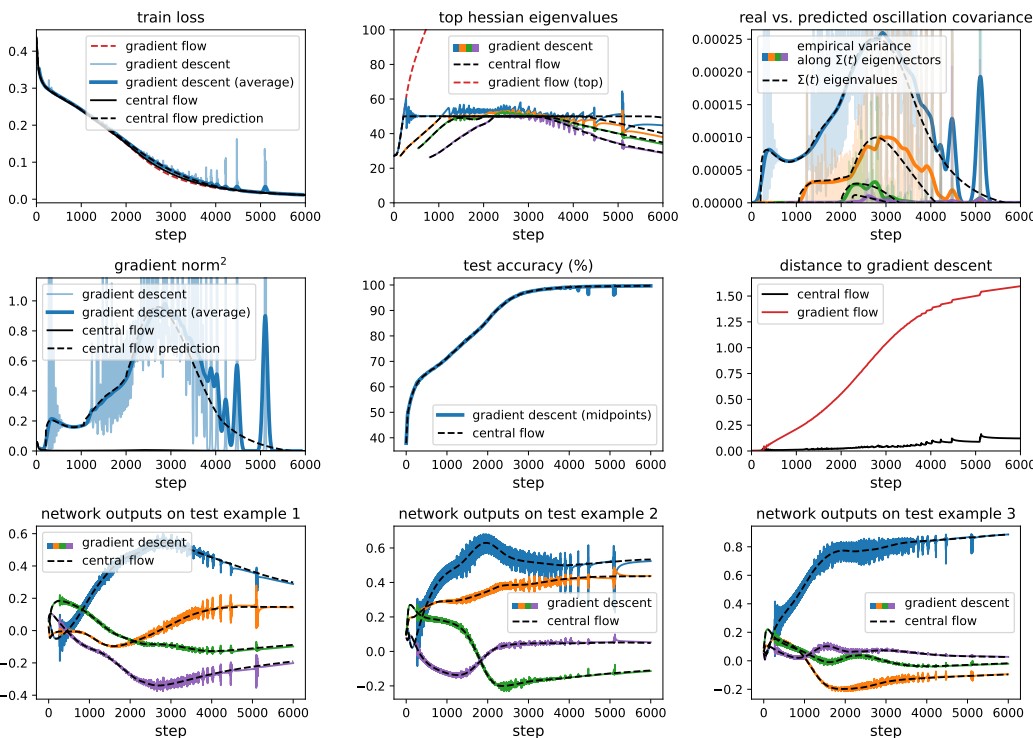

Figure 30.15: Gradient descent central flow for a Transformer with MSE loss, $\eta = 0.04$.

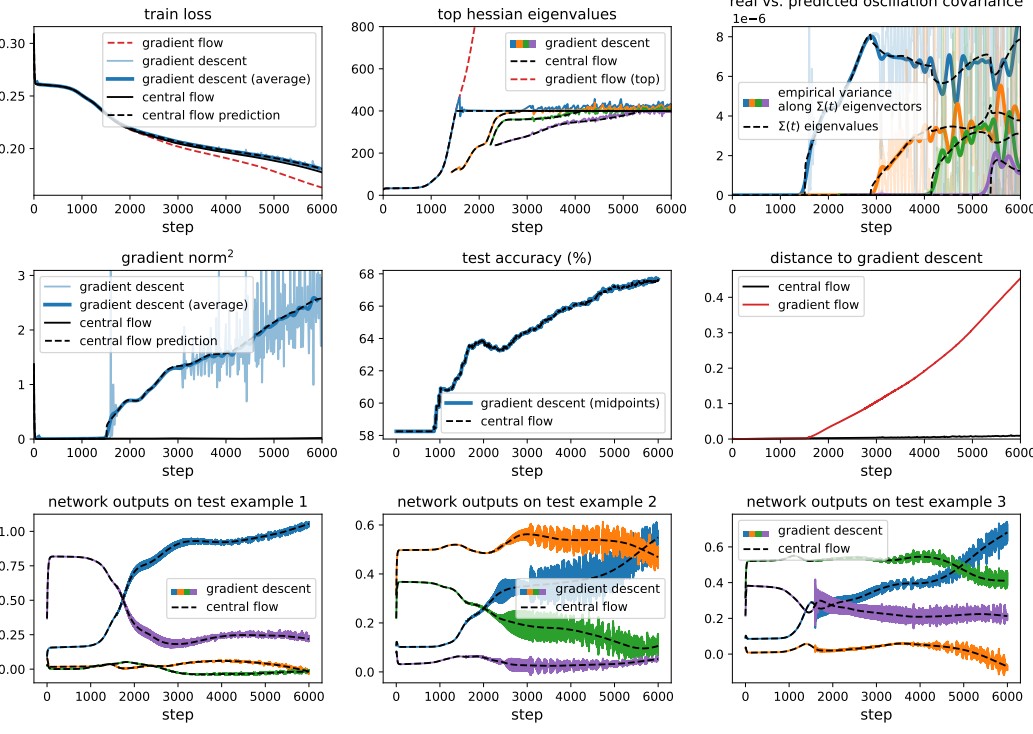

Figure 30.16: Gradient descent central flow for a Mamba with MSE loss, $\eta = 0.005$.

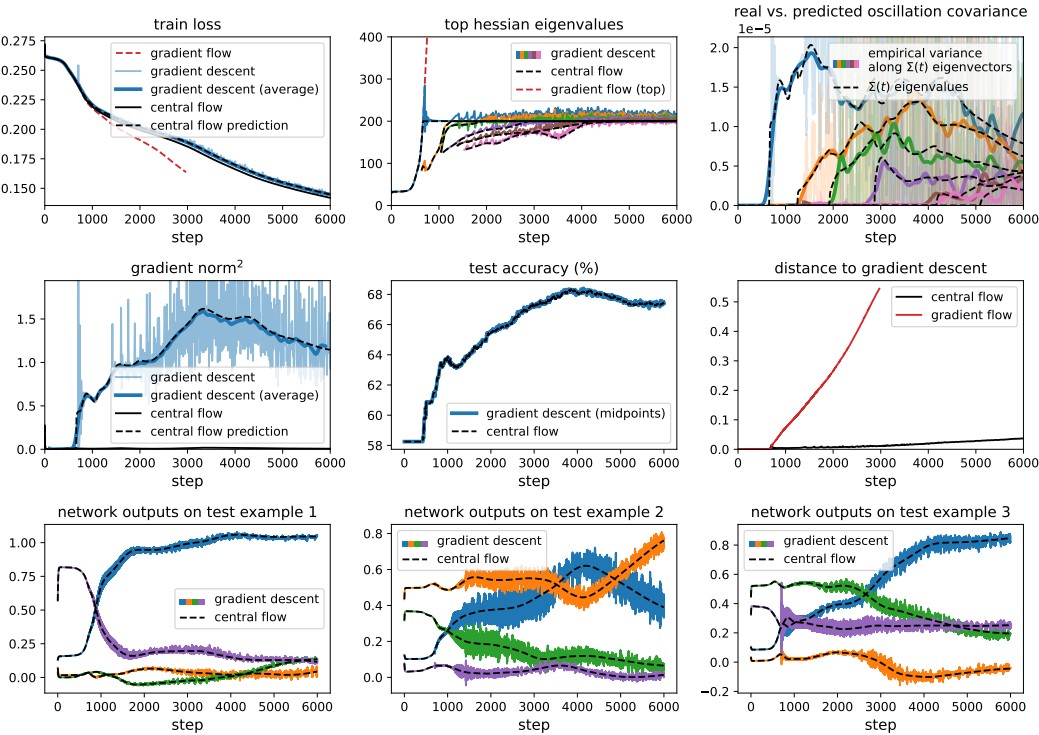

Figure 30.17: Gradient descent central flow for a Mamba with MSE loss, $\eta = 0.01$.

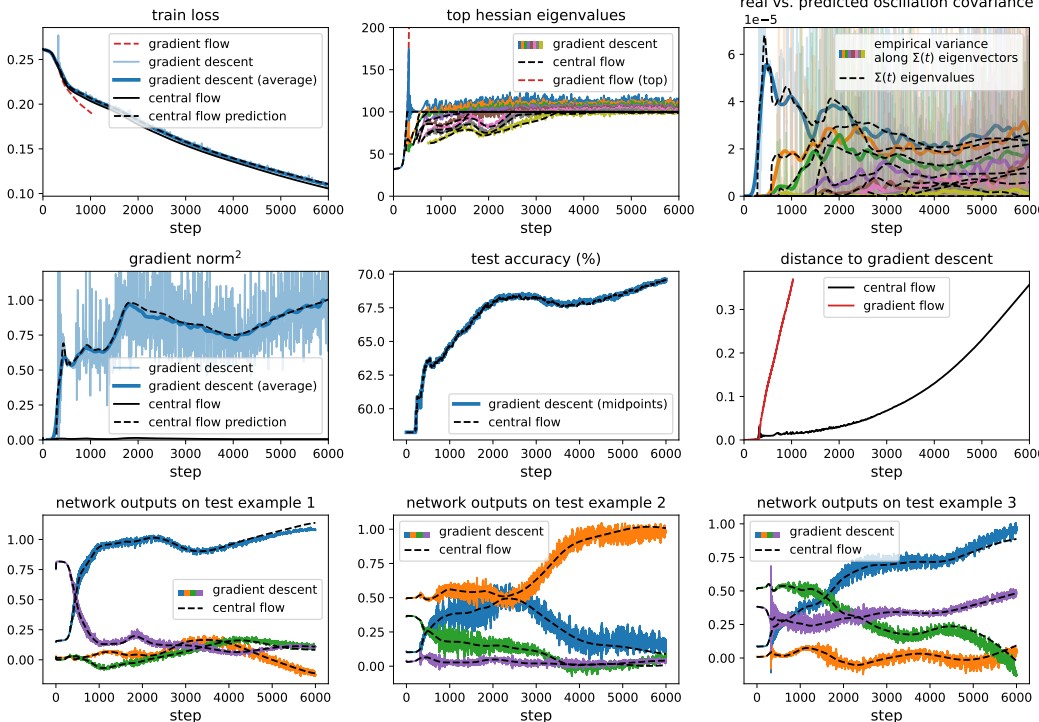

Figure 30.18: Gradient descent central flow for a Mamba with MSE loss, $\eta = 0.02$.

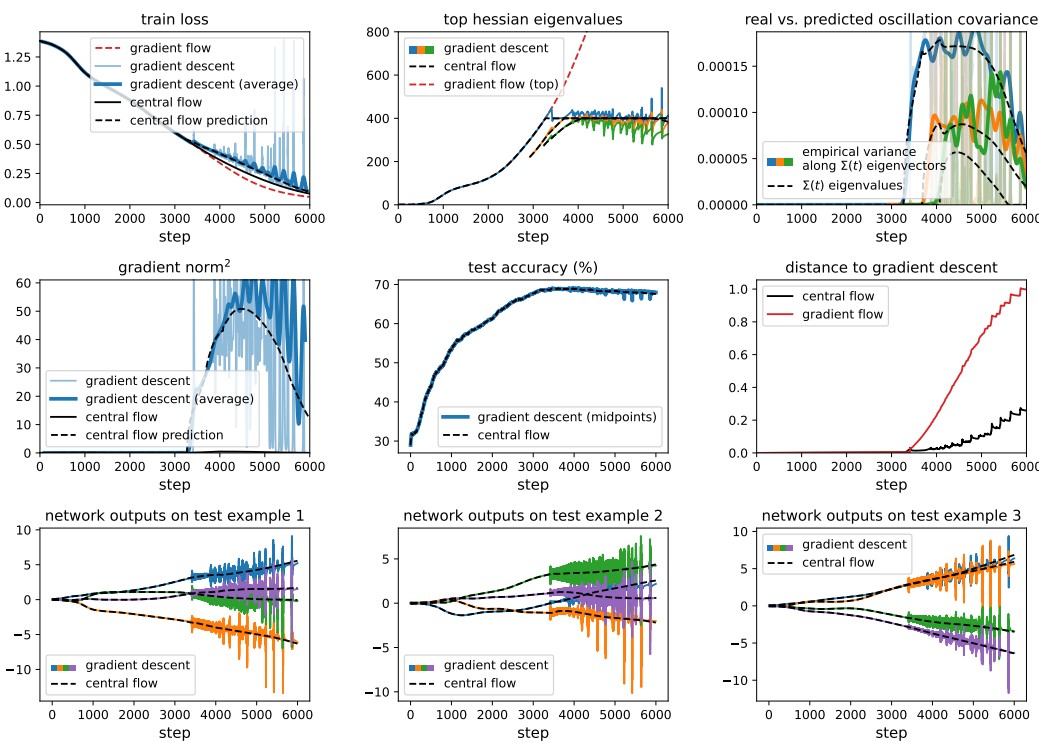

Figure 31.1: Gradient descent central flow for a CNN with CE loss, $\eta = 0.005$.

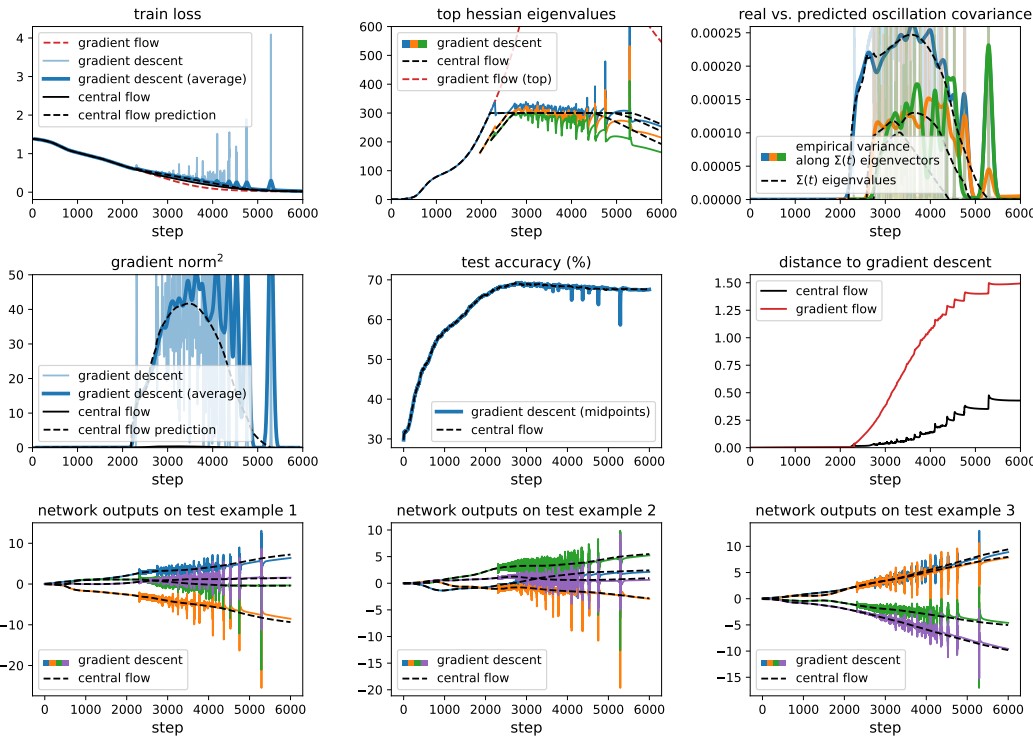

Figure 31.2: Gradient descent central flow for a CNN with CE loss, $\eta = 0.006666$.

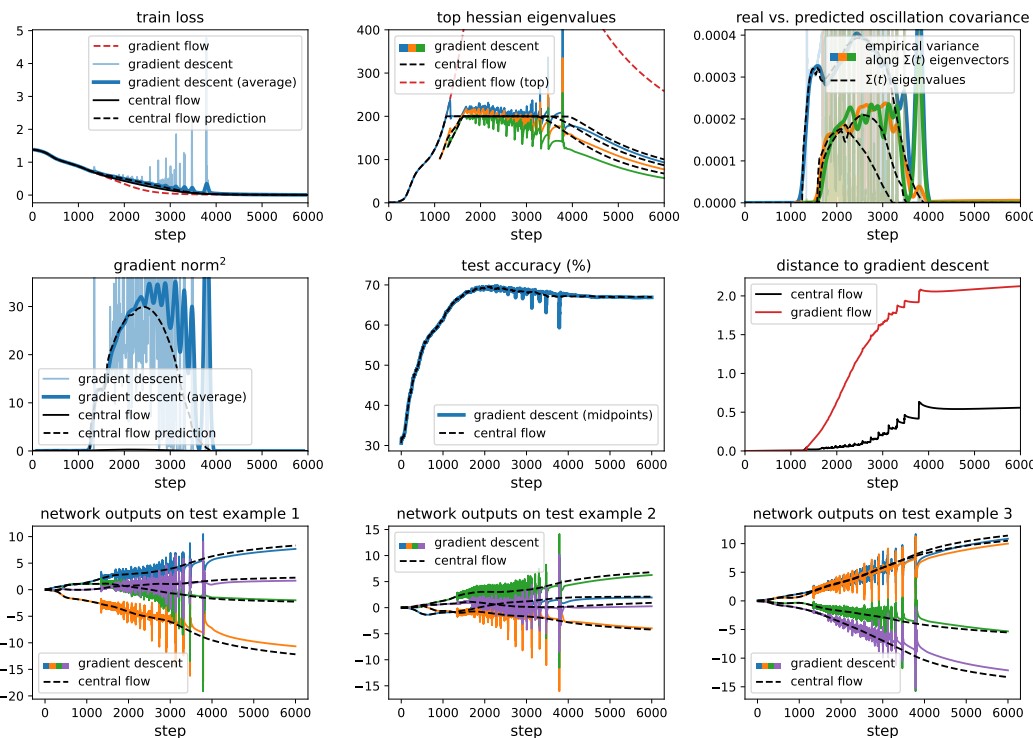

Figure 31.3: Gradient descent central flow for a CNN with CE loss, $\eta = 0.01$.

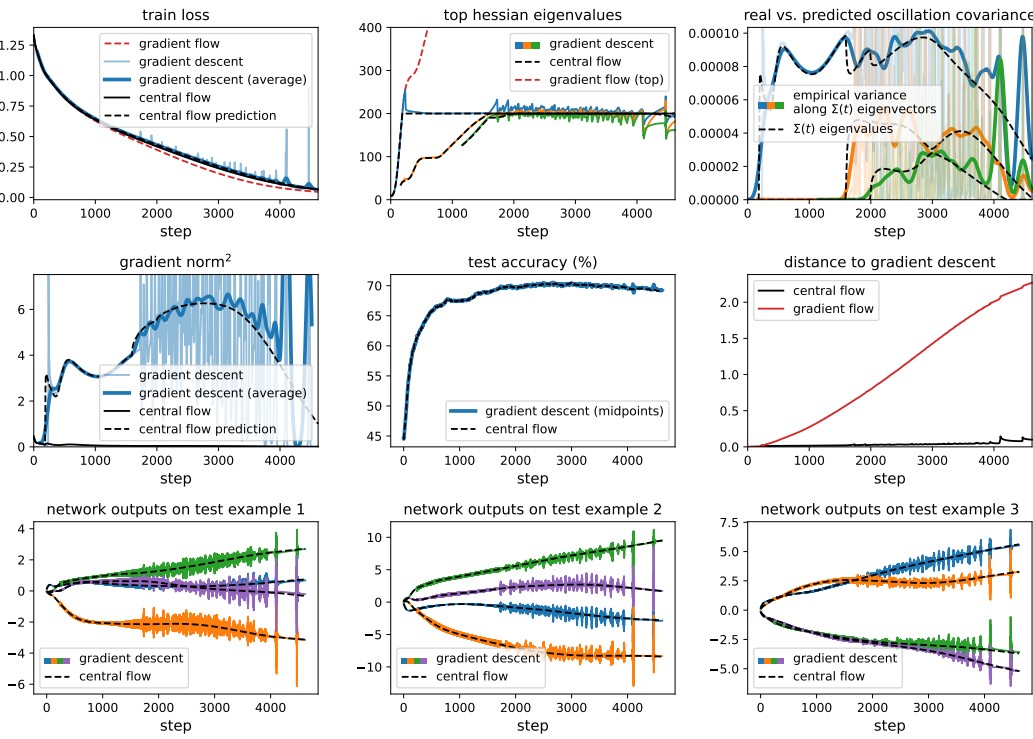

Figure 31.4: Gradient descent central flow for a ResNet with CE loss, $\eta = 0.01$.

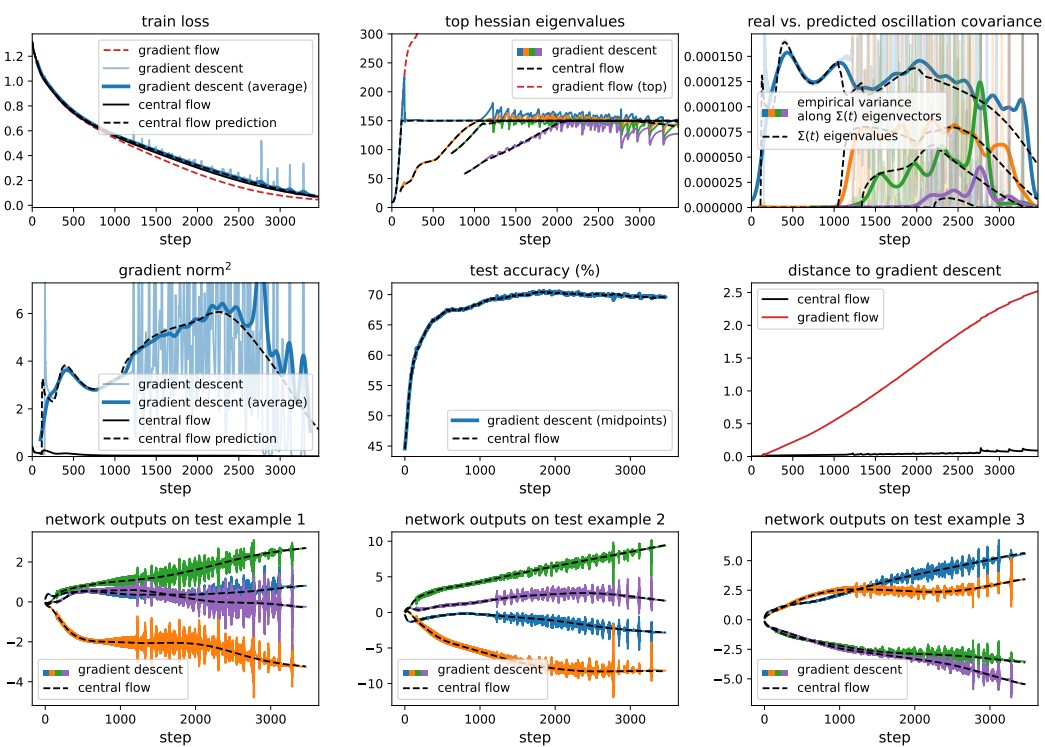

Figure 31.5: Gradient descent central flow for a ResNet with CE loss, $\eta = 0.013333$.

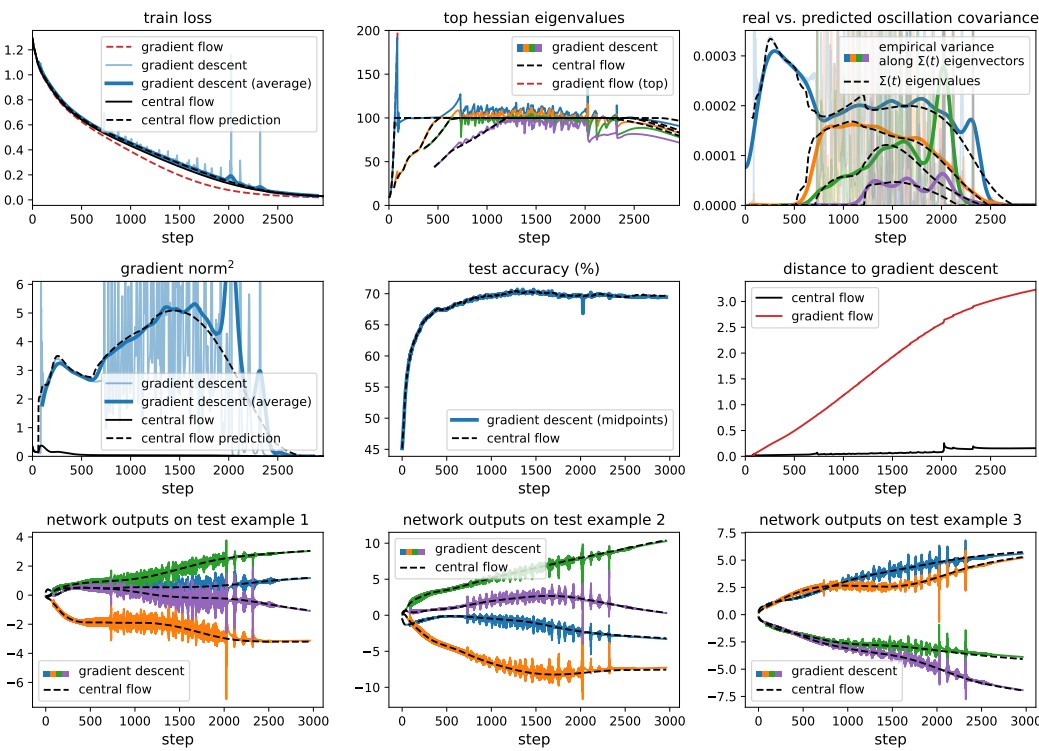

Figure 31.6: Gradient descent central flow for a ResNet with CE loss, $\eta = 0.02$.

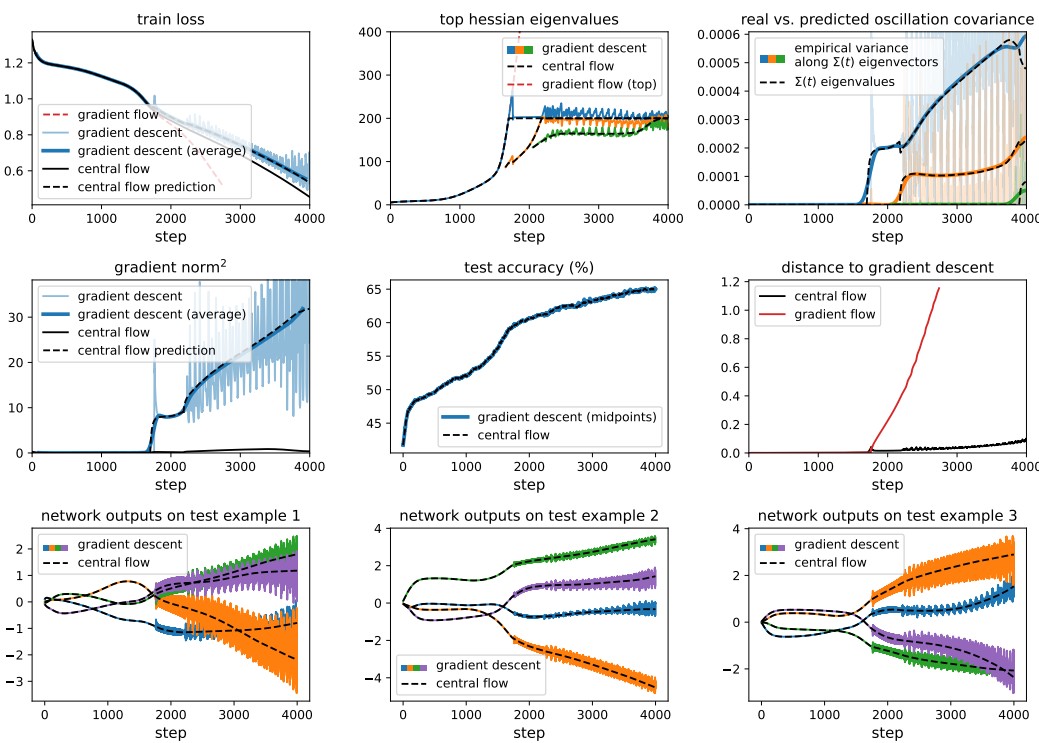

Figure 31.7: Gradient descent central flow for a VIT with CE loss, $\eta = 0.01$.

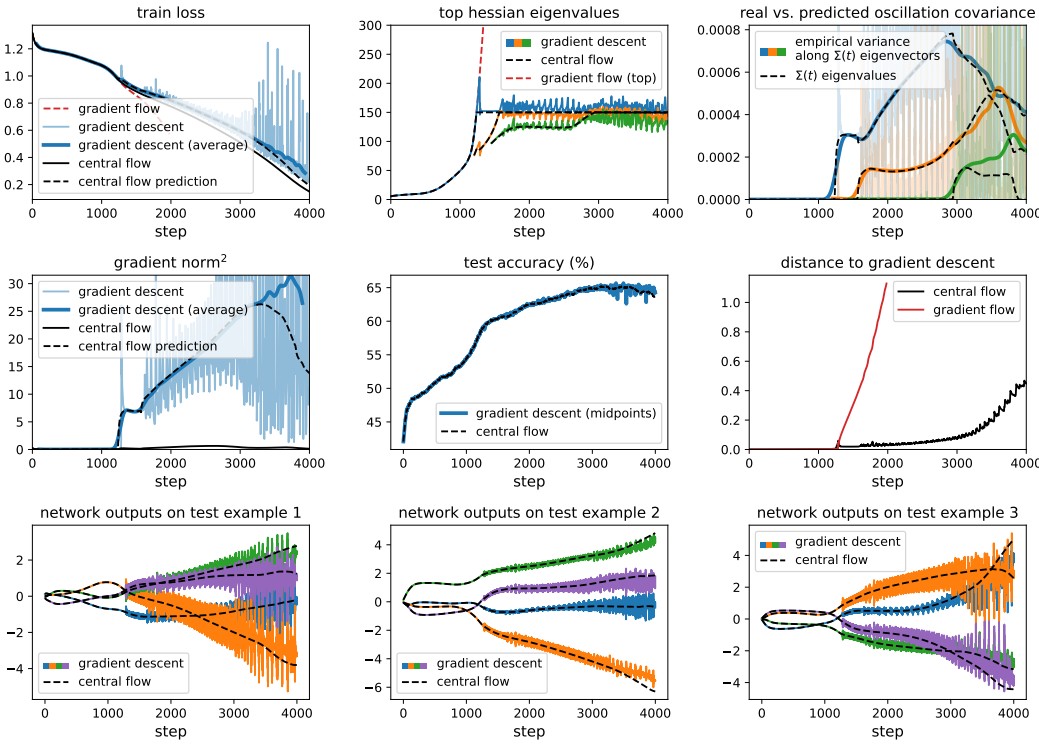

Figure 31.8: Gradient descent central flow for a VIT with CE loss, $\eta = 0.013333$.

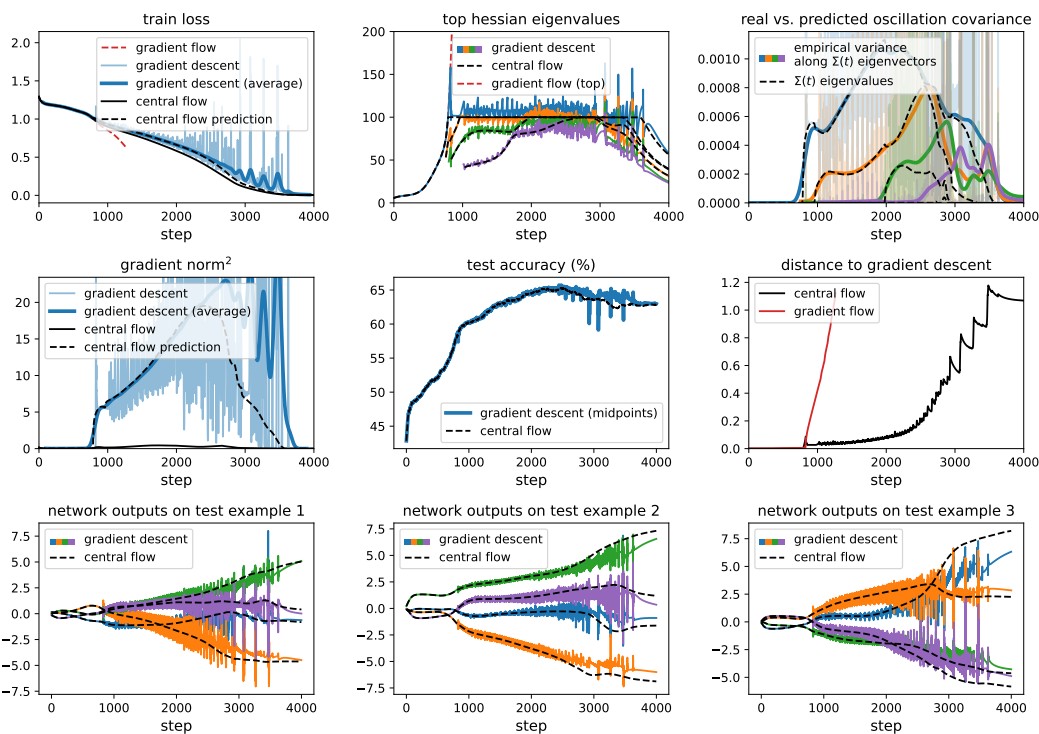

Figure 31.9: Gradient descent central flow for a VIT with CE loss, $\eta = 0.02$.

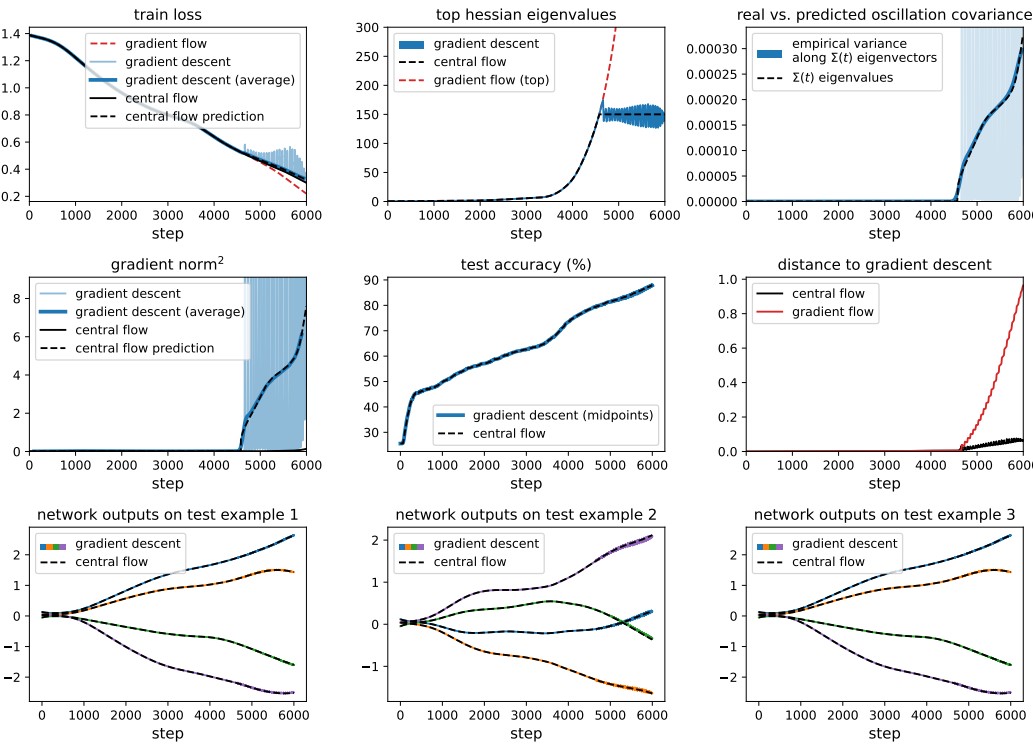

Figure 31.10: Gradient descent central flow for a LSTM with CE loss, $\eta = 0.01333$.

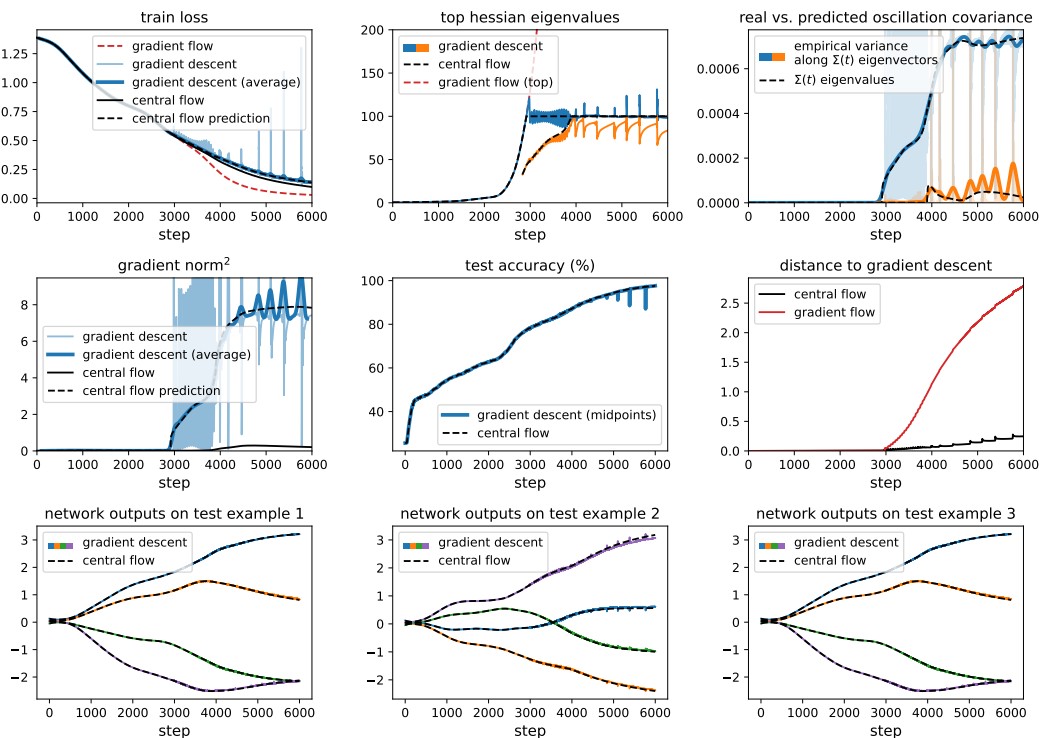

Figure 31.11: Gradient descent central flow for a LSTM with CE loss, $\eta = 0.02$.

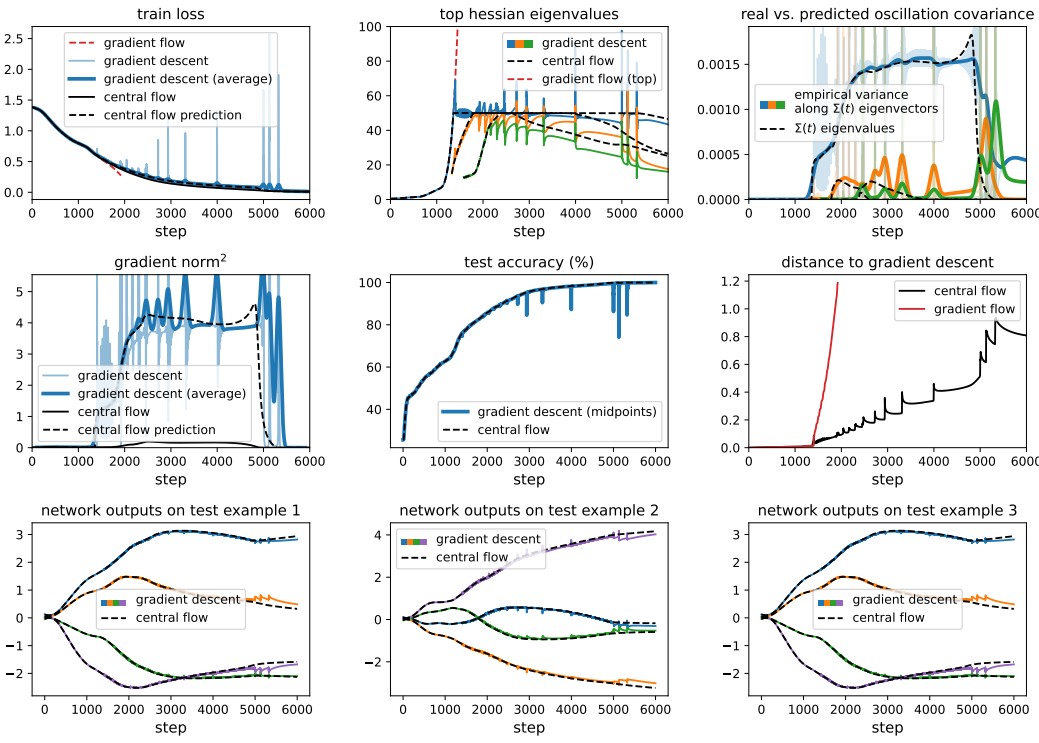

Figure 31.12: Gradient descent central flow for a LSTM with CE loss, $\eta = 0.04$.

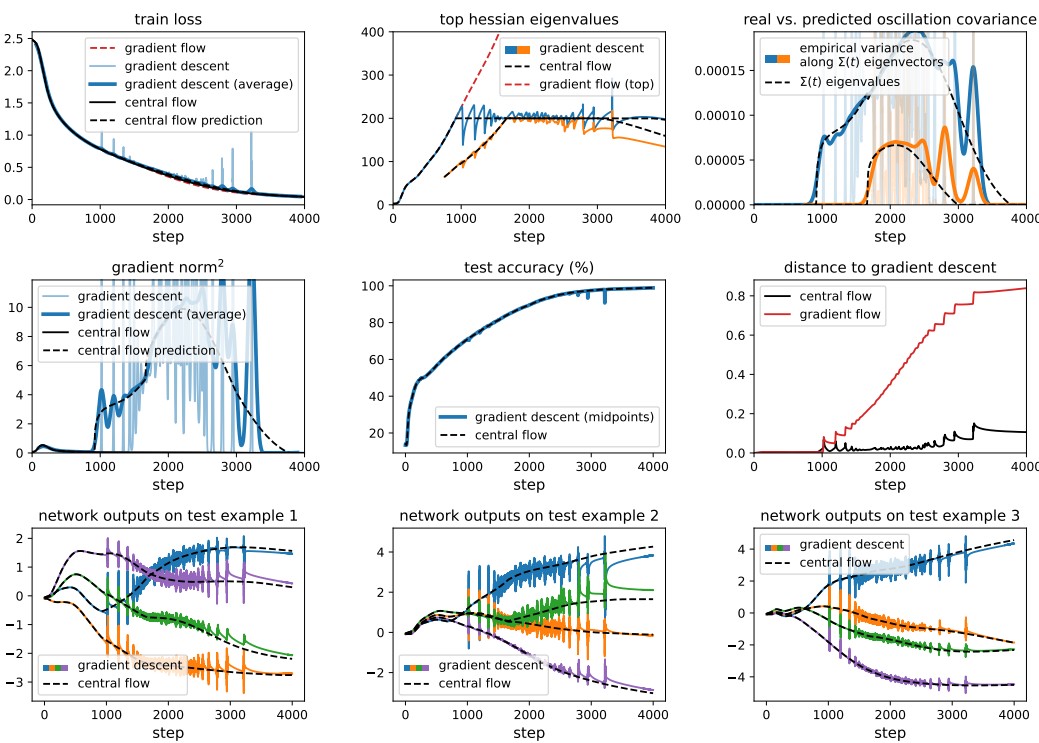

Figure 31.13: Gradient descent central flow for a Transformer with CE loss, $\eta = 0.01$.

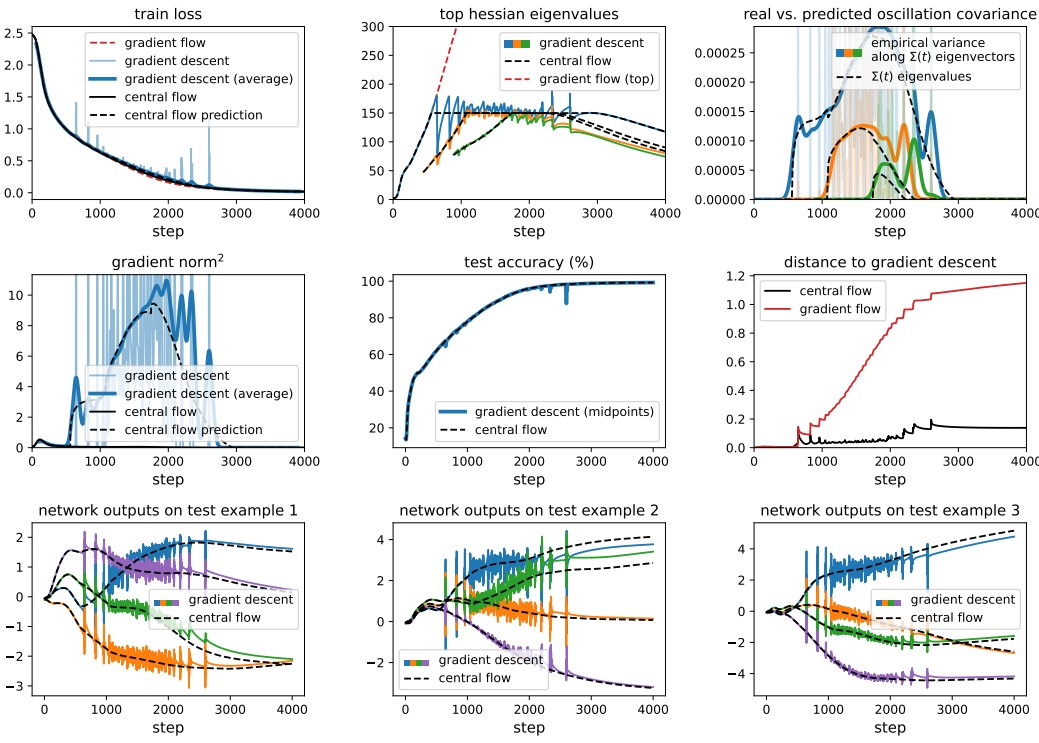

Figure 31.14: Gradient descent central flow for a Transformer with CE loss, $\eta = 0.013333$.

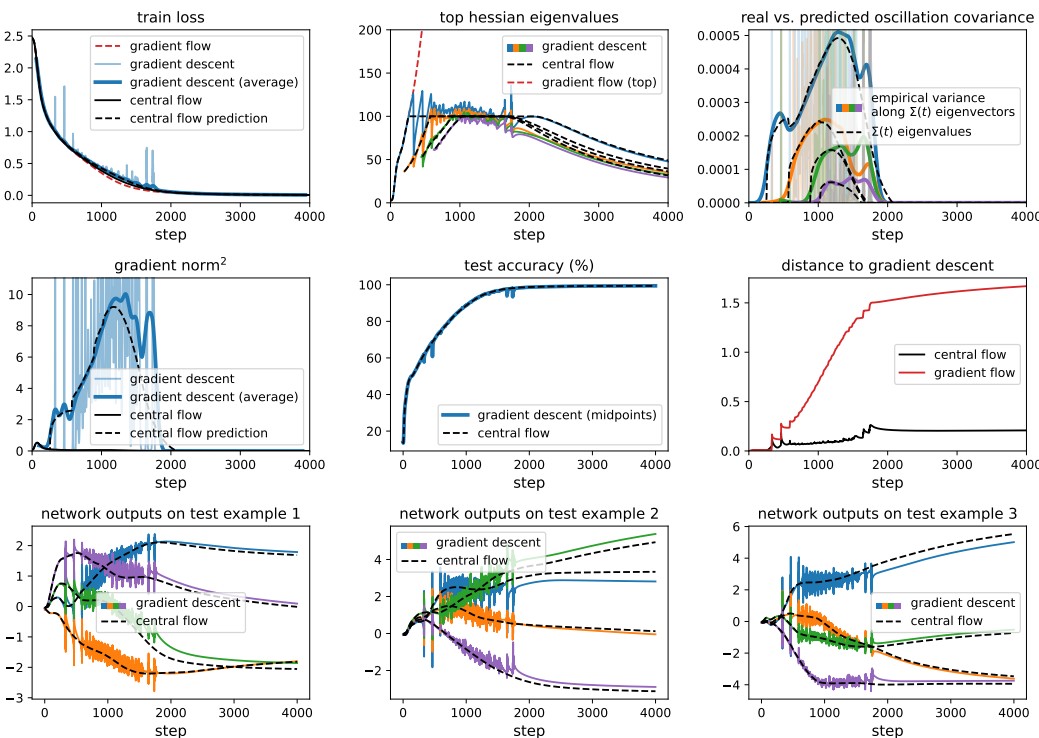

Figure 31.15: Gradient descent central flow for a Transformer with CE loss, $\eta = 0.02$.

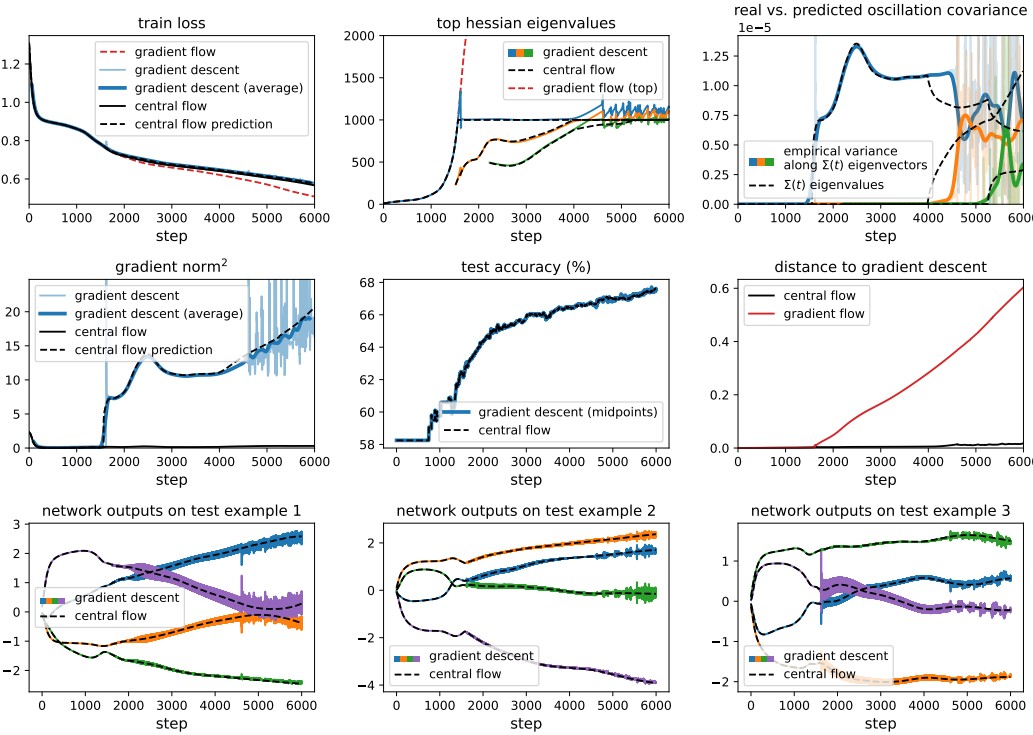

Figure 31.16: Gradient descent central flow for a Mamba with CE loss, $\eta = 0.002$.

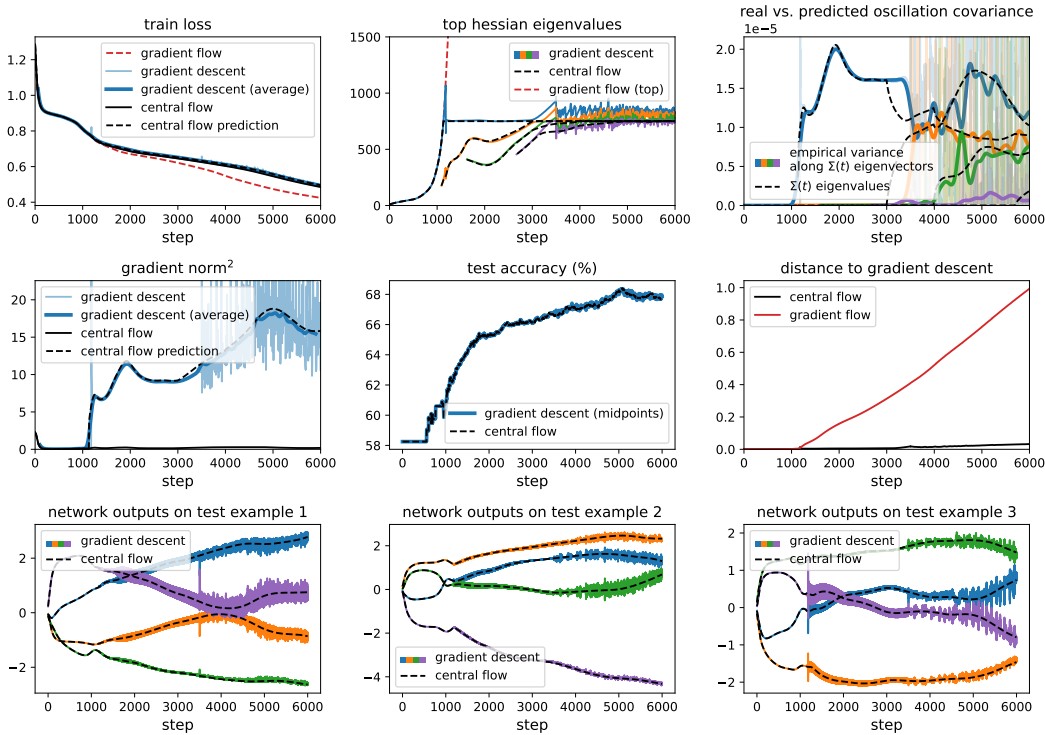

Figure 31.17: Gradient descent central flow for a Mamba with CE loss, $\eta = 0.002666$.

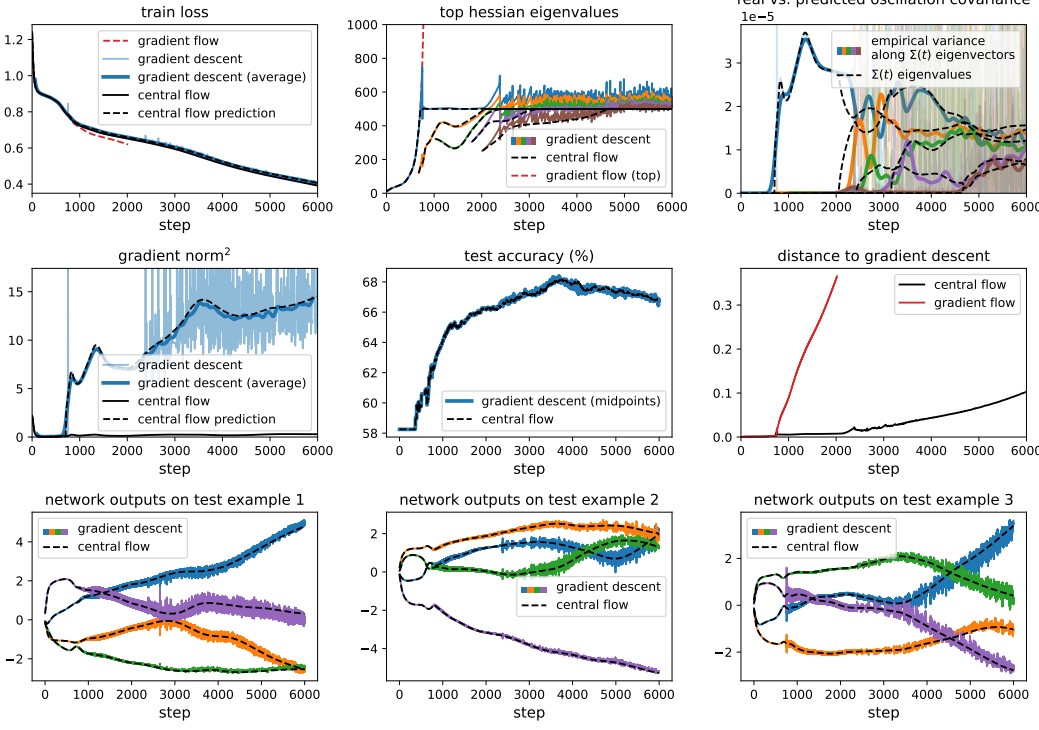

Figure 31.18: Gradient descent central flow for a Mamba with CE loss, $\eta = 0.004$.

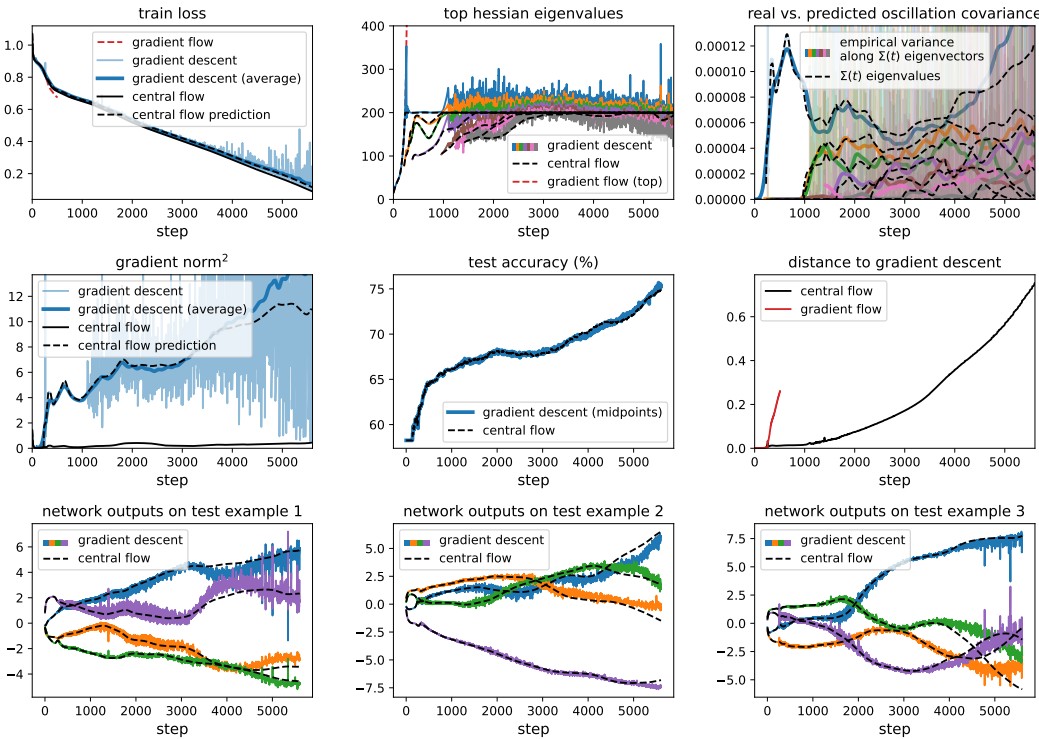

Figure 31.19: Gradient descent central flow for a Mamba with CE loss, $\eta = 0.01$.

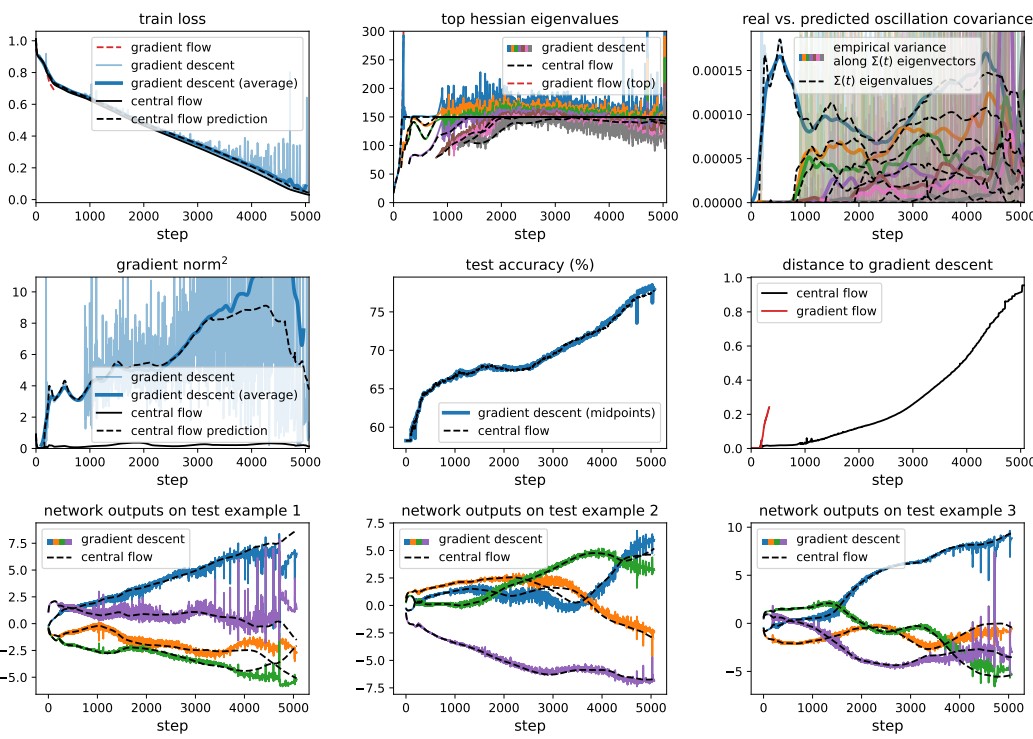

Figure 31.20: Gradient descent central flow for a Mamba with CE loss, $\eta = 0.013333$.

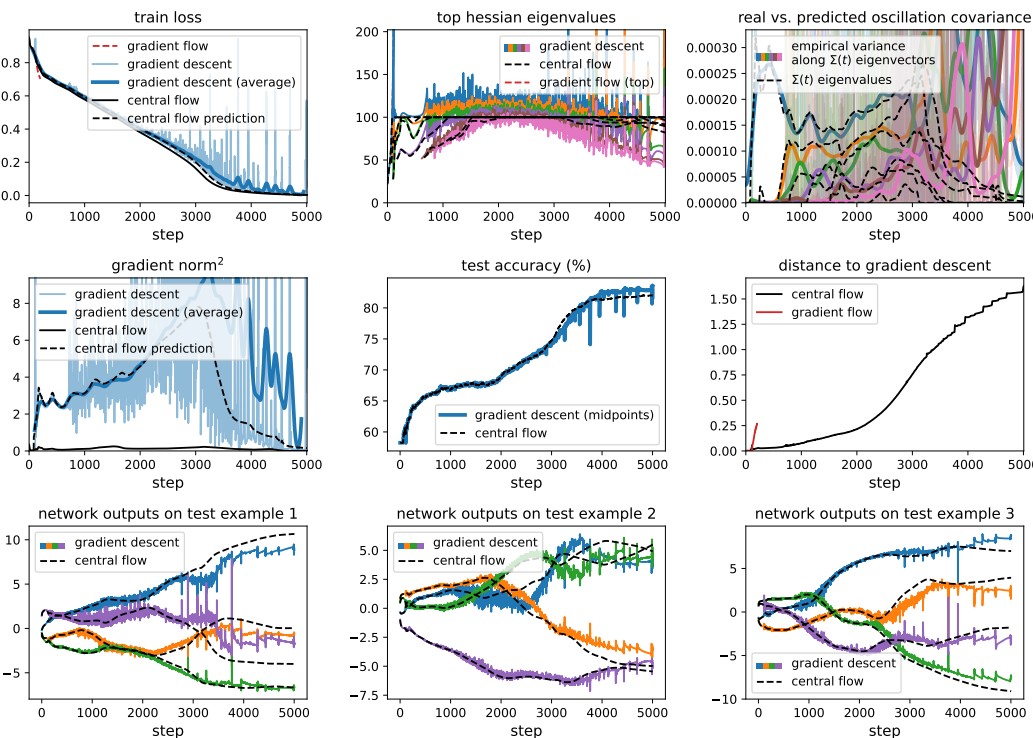

Figure 31.21: Gradient descent central flow for a Mamba with CE loss, $\eta = 0.02$.

## C.2    SCALAR RMSPROP

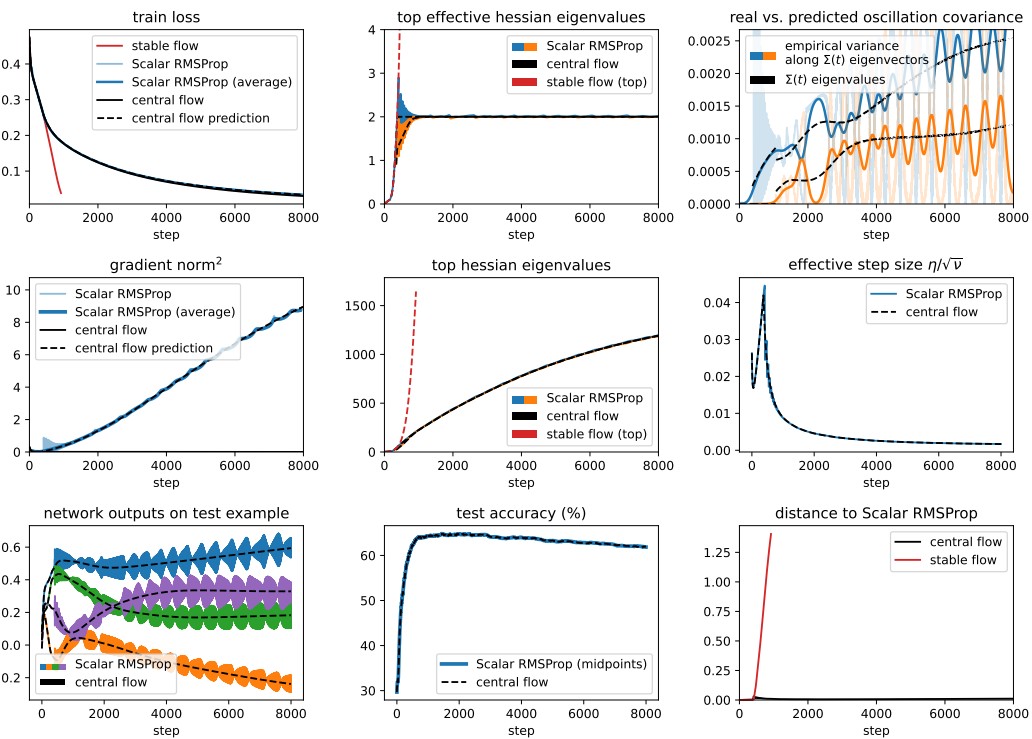

Figure 32.1: Scalar RMSProp central flow for a CNN with MSE loss, $\eta = 0.003$, $\beta_2 = 0.99$, and bias correction.

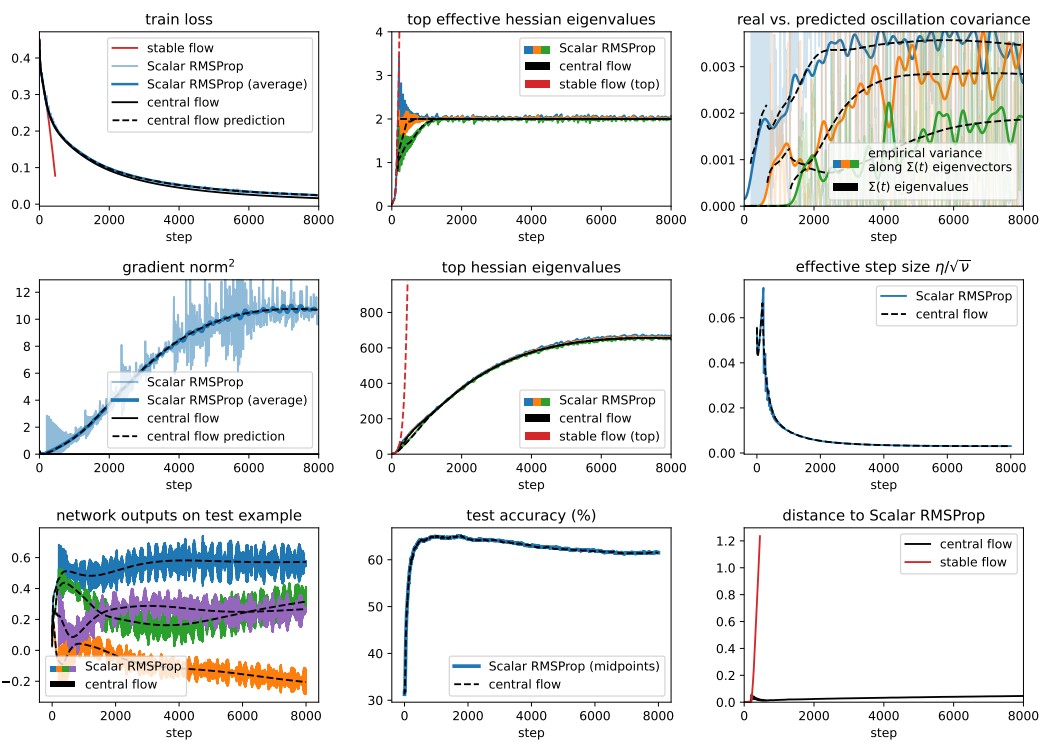

Figure 32.2: Scalar RMSProp central flow for a CNN with MSE loss, $\eta =0.006$, $\beta_2 =0.99$, and bias correction.

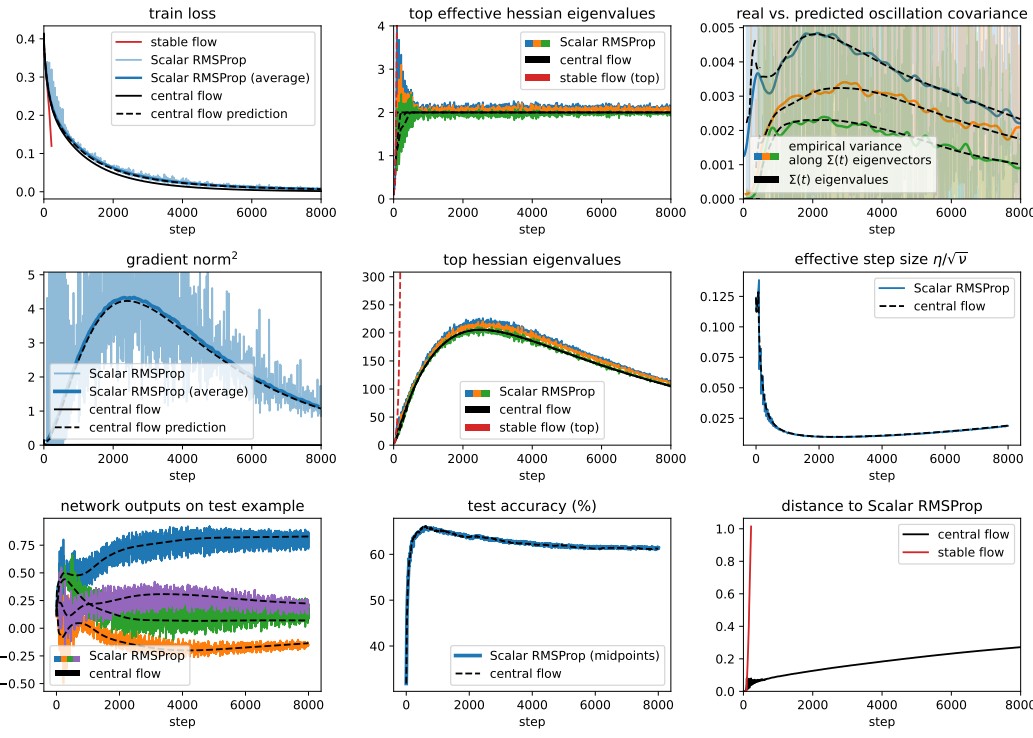

Figure 32.3: Scalar RMSProp central flow for a CNN with MSE loss, $\eta =0.01$, $\beta_2 =0.99$, and bias correction.

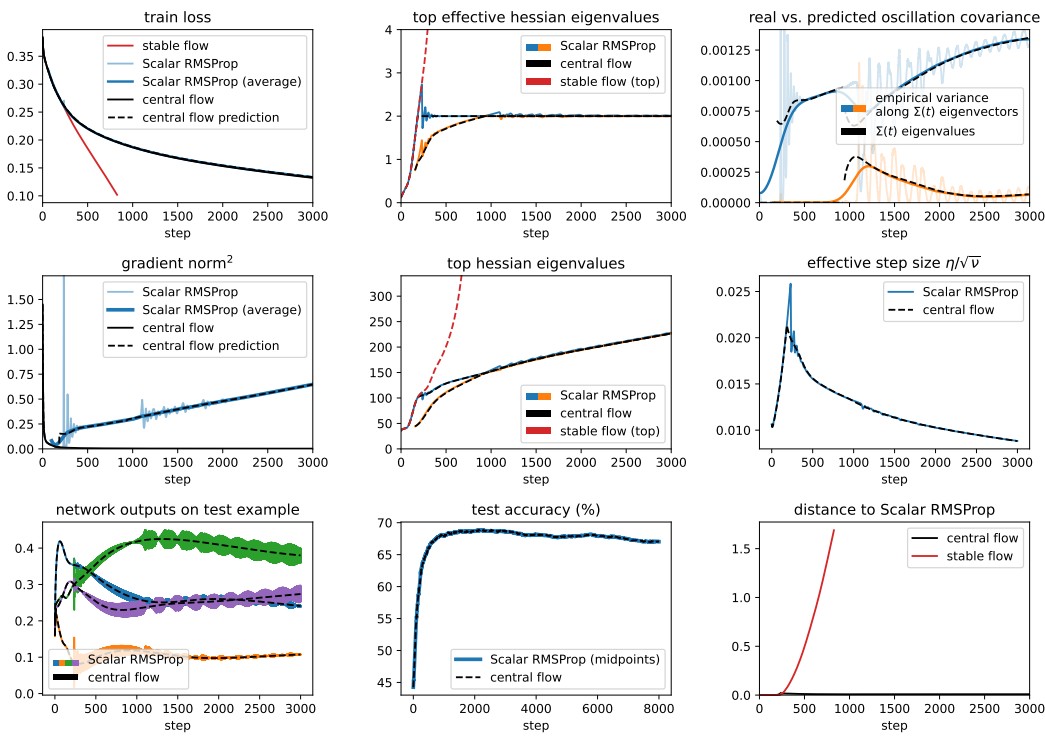

Figure 32.4: Scalar RMSProp central flow for a ResNet with MSE loss, $\eta = 0.01$, $\beta_2 = 0.99$, and bias correction.

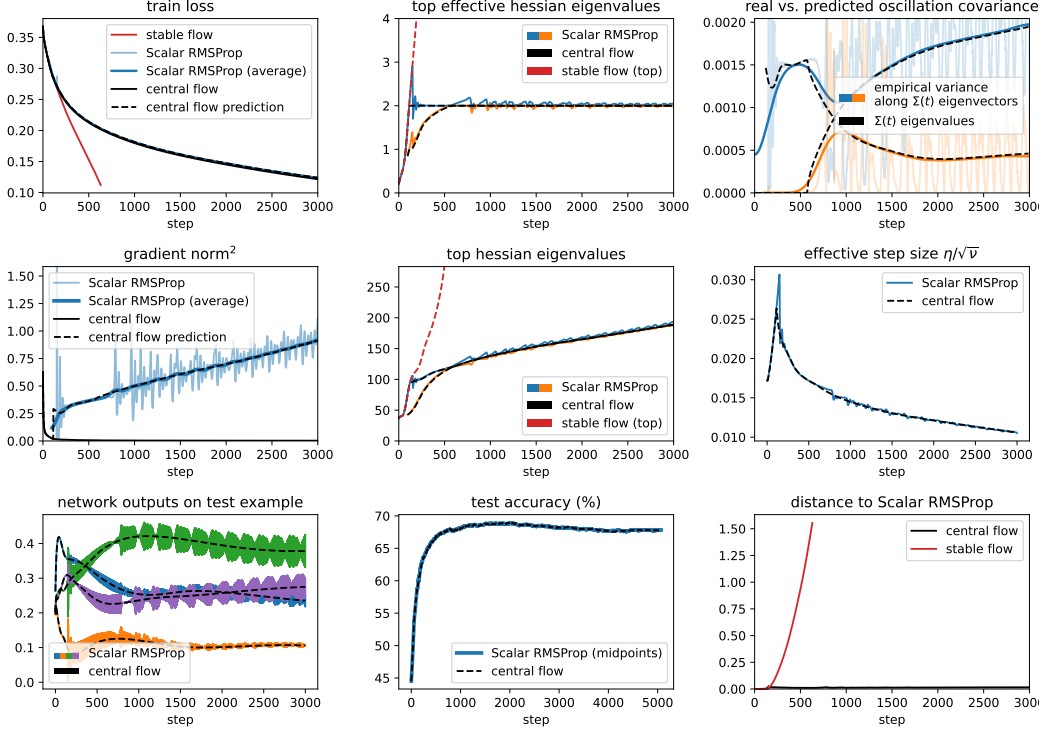

Figure 32.5: Scalar RMSProp central flow for a ResNet with MSE loss, $\eta = 0.02$, $\beta_2 = 0.99$, and bias correction.

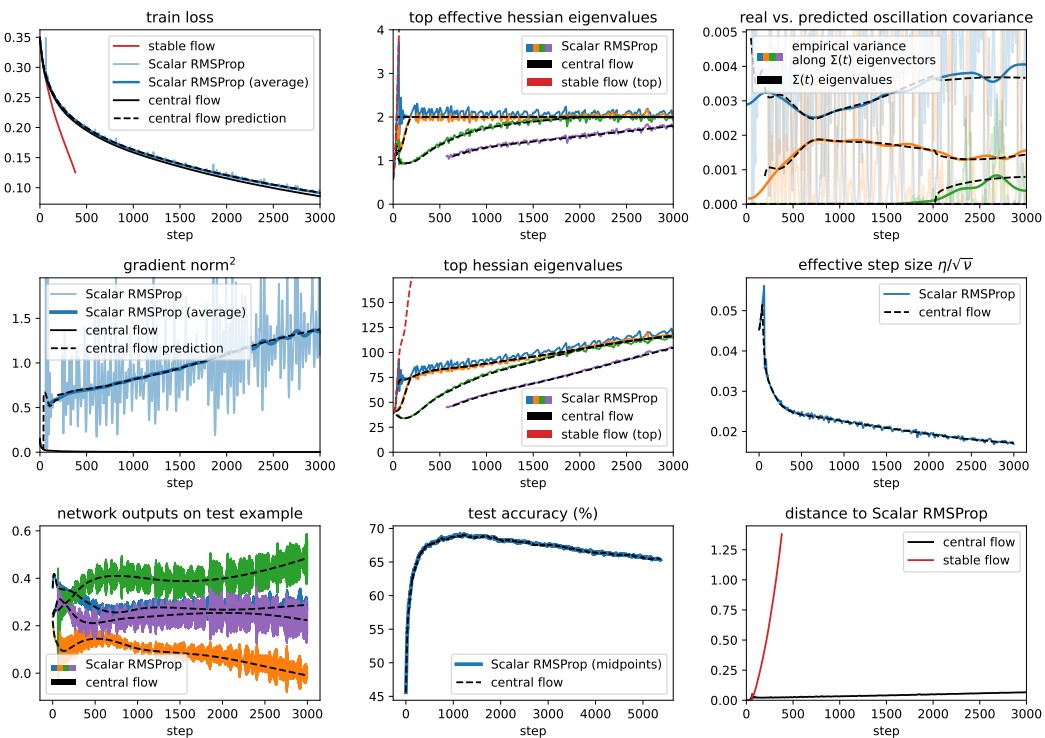

Figure 32.6: Scalar RMSProp central flow for a ResNet with MSE loss, $\eta = 0.03$, $\beta_2 = 0.99$, and bias correction.

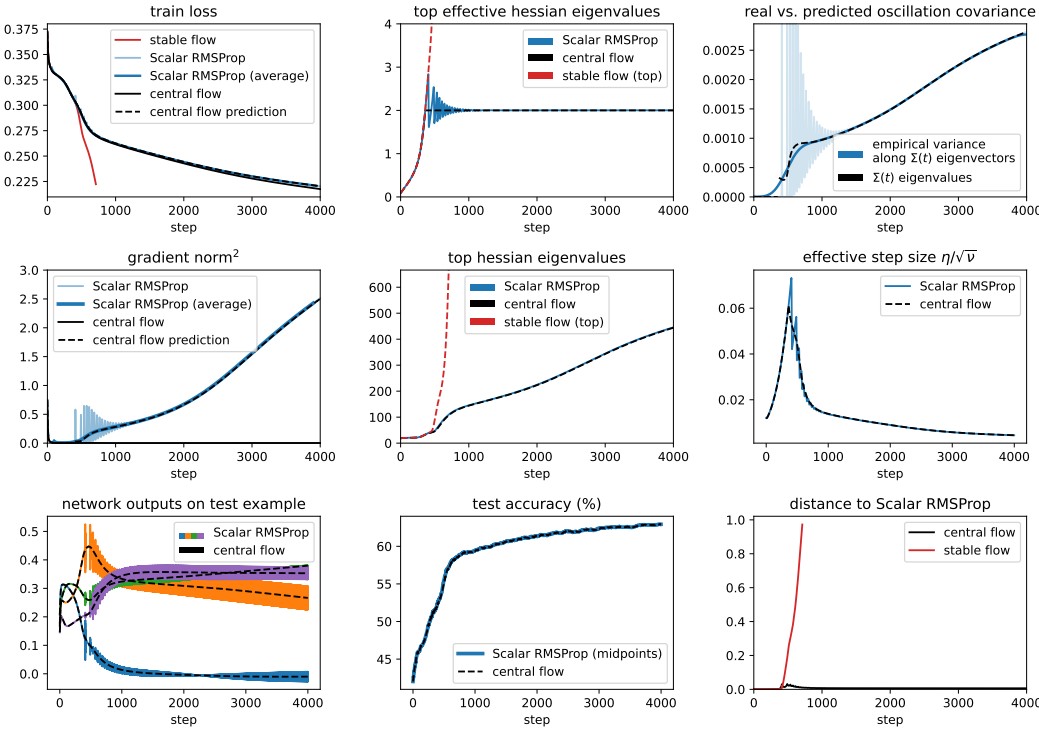

Figure 32.7: Scalar RMSProp central flow for a ViT with MSE loss, $\eta = 0.01$, $\beta_2 = 0.99$, and bias correction.

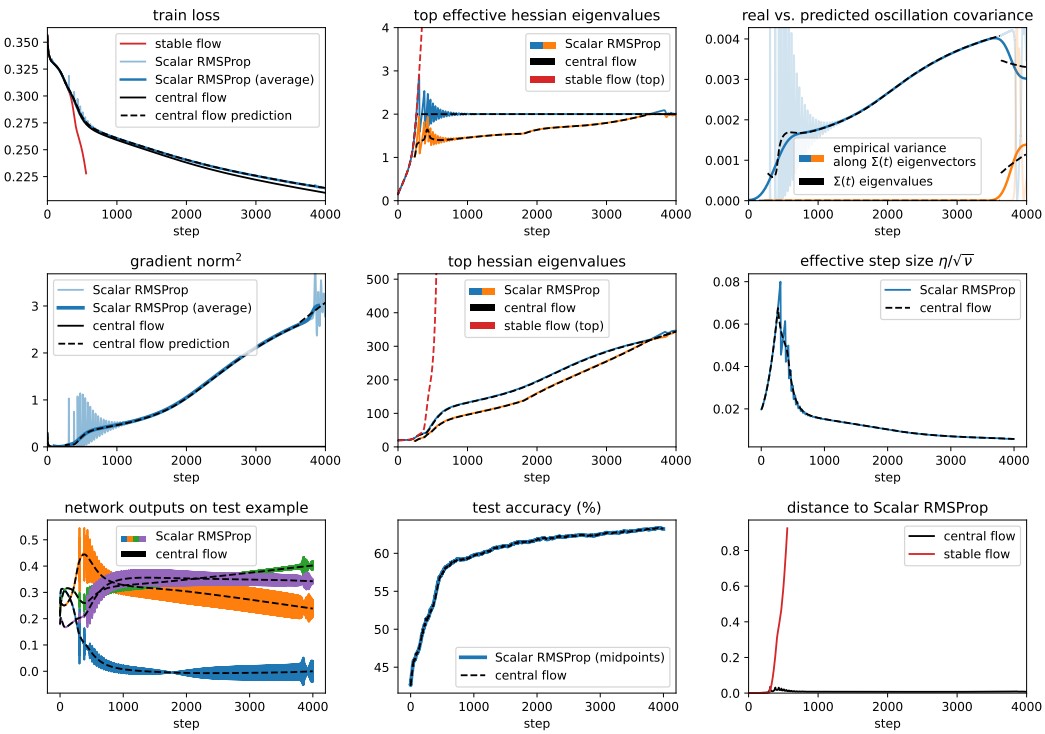

Figure 32.8: Scalar RMSProp central flow for a ViT with MSE loss, $\eta$ =0.02, $\beta_2$ =0.99, and bias correction.

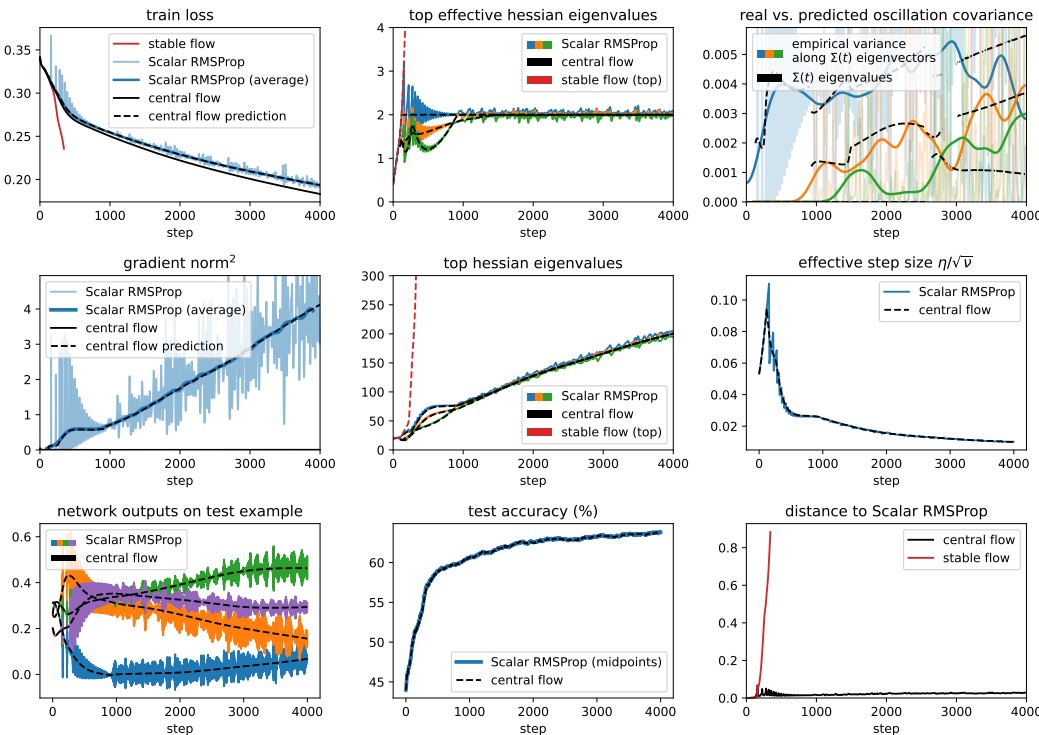

Figure 32.9: Scalar RMSProp central flow for a ViT with MSE loss, $\eta$ =0.03, $\beta_2$ =0.99, and bias correction.

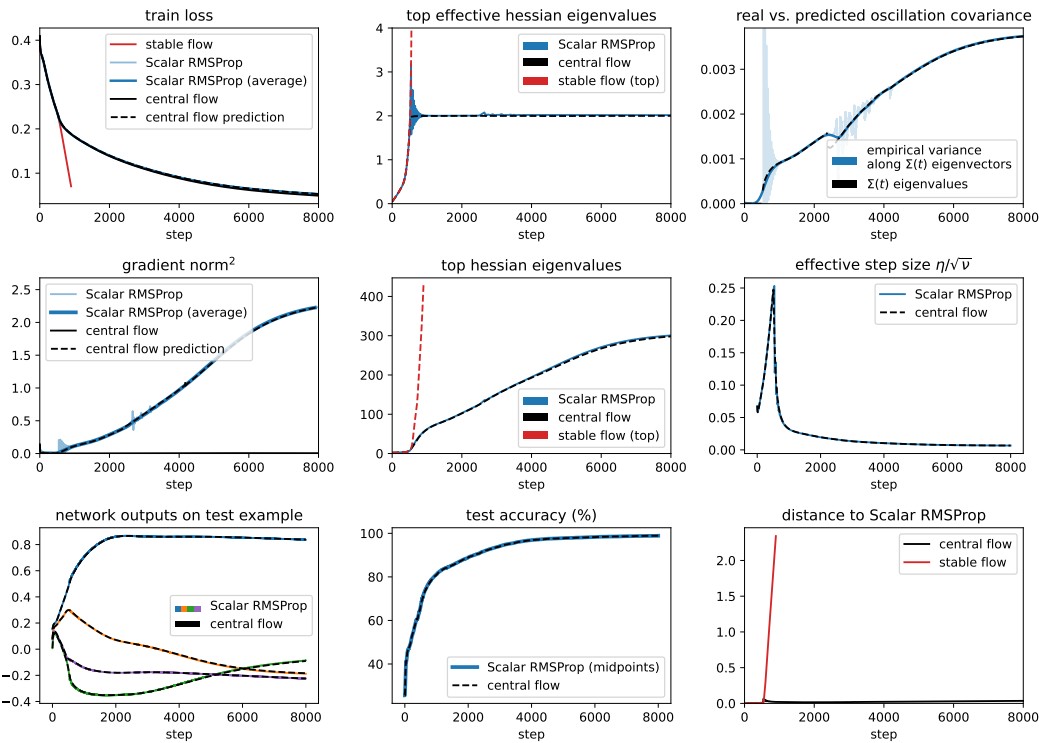

Figure 32.10: Scalar RMSProp central flow for a LSTM with MSE loss, $\eta$ =0.01, $\beta_2$ =0.99, and bias correction.

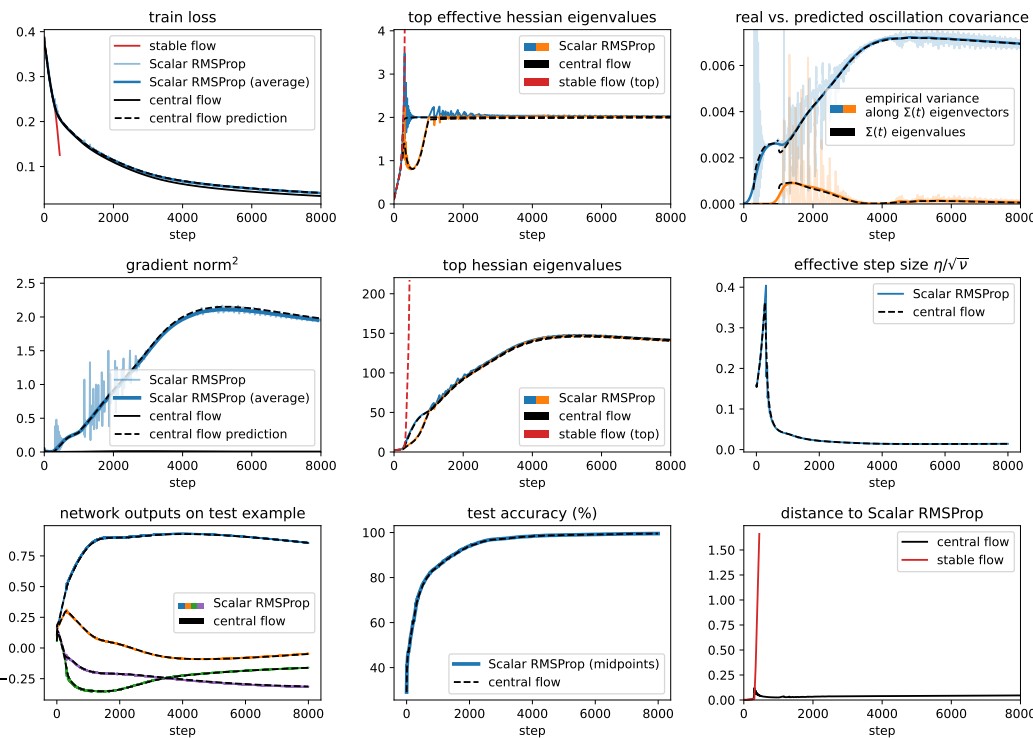

Figure 32.11: Scalar RMSProp central flow for a LSTM with MSE loss, $\eta$ =0.02, $\beta_2$ =0.99, and bias correction.

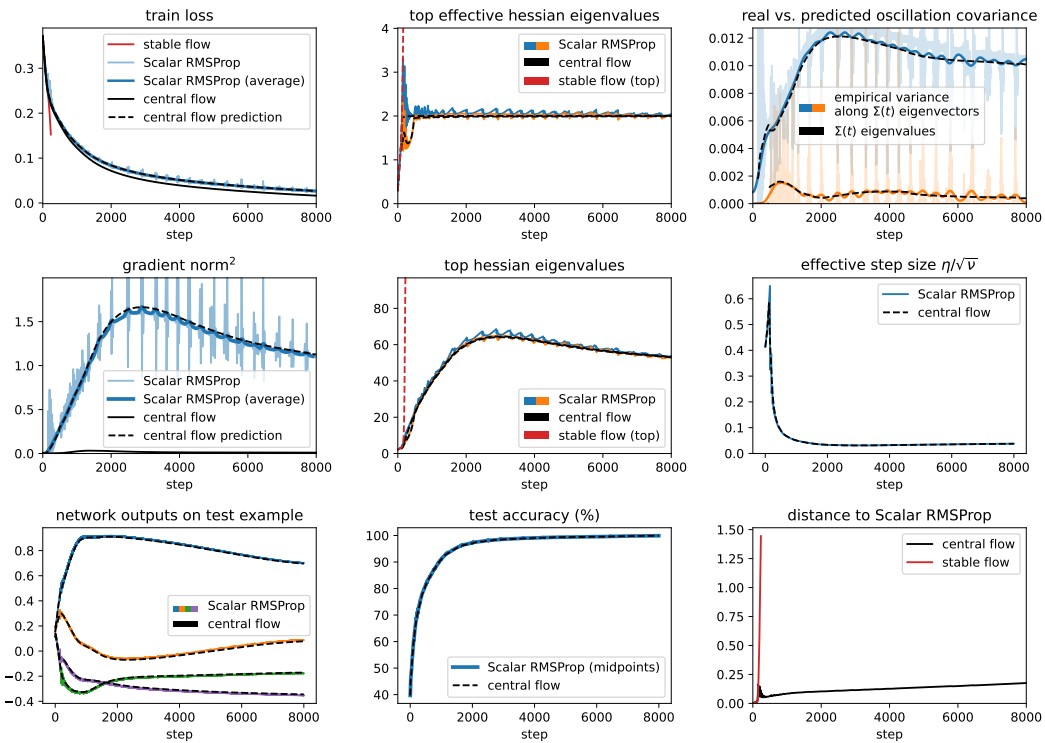

Figure 32.12: Scalar RMSProp central flow for a LSTM with MSE loss, $\eta =0.03$, $\beta_2 =0.99$, and bias correction.

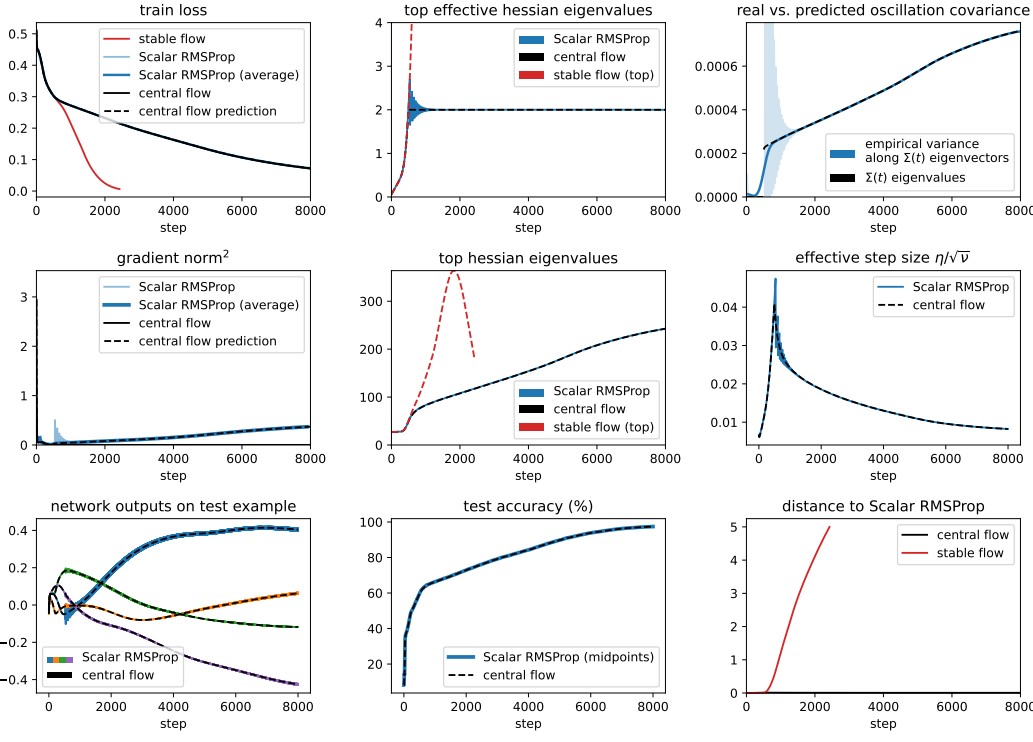

Figure 32.13: Scalar RMSProp central flow for a Transformer with MSE loss, $\eta =0.01$, $\beta_2 =0.99$, and bias correction.

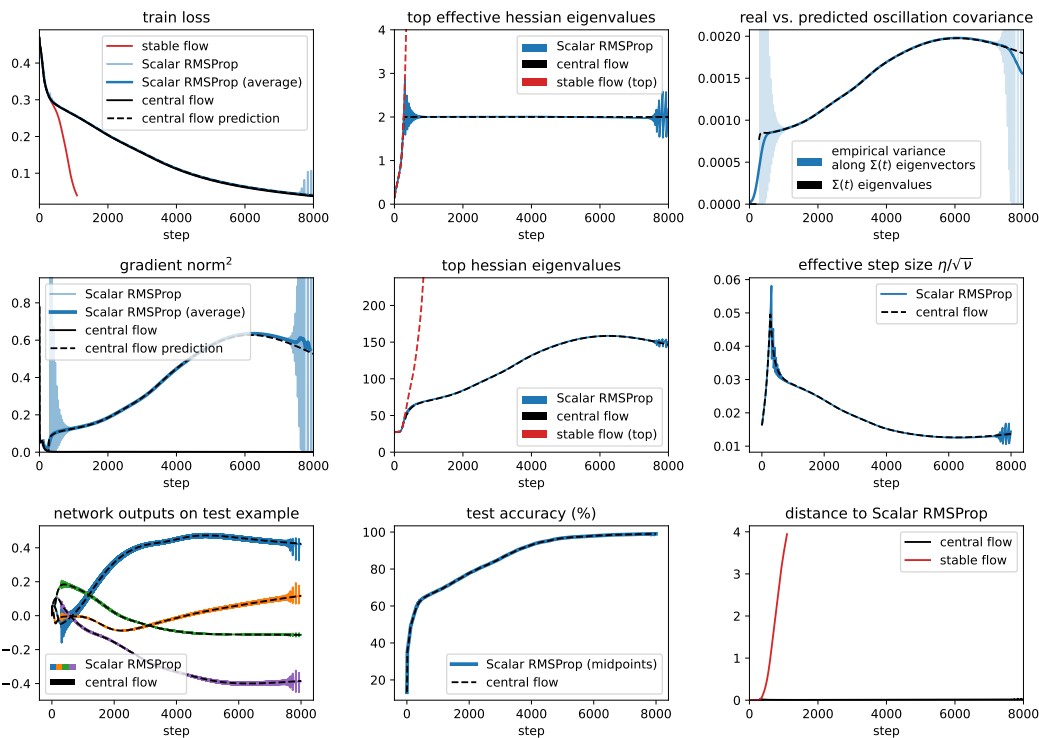

Figure 32.14: Scalar RMSProp central flow for a Transformer with MSE loss, $\eta = 0.02$, $\beta_2 = 0.99$, and bias correction.

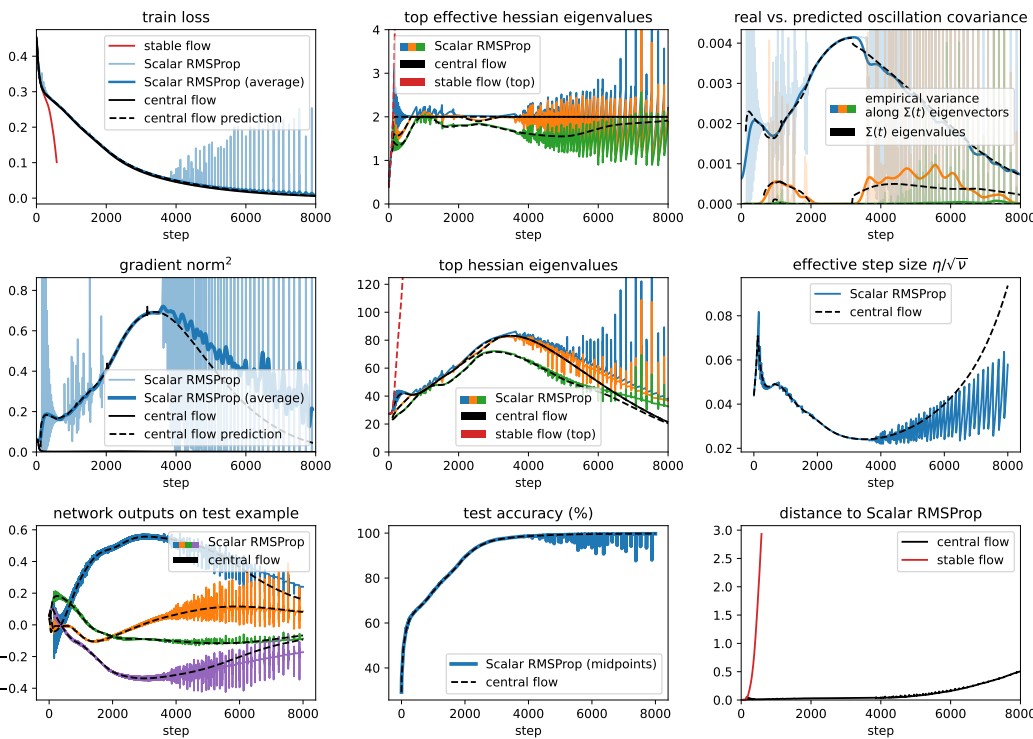

Figure 32.15: Scalar RMSProp central flow for a Transformer with MSE loss, $\eta = 0.03$, $\beta_2 = 0.99$, and bias correction.

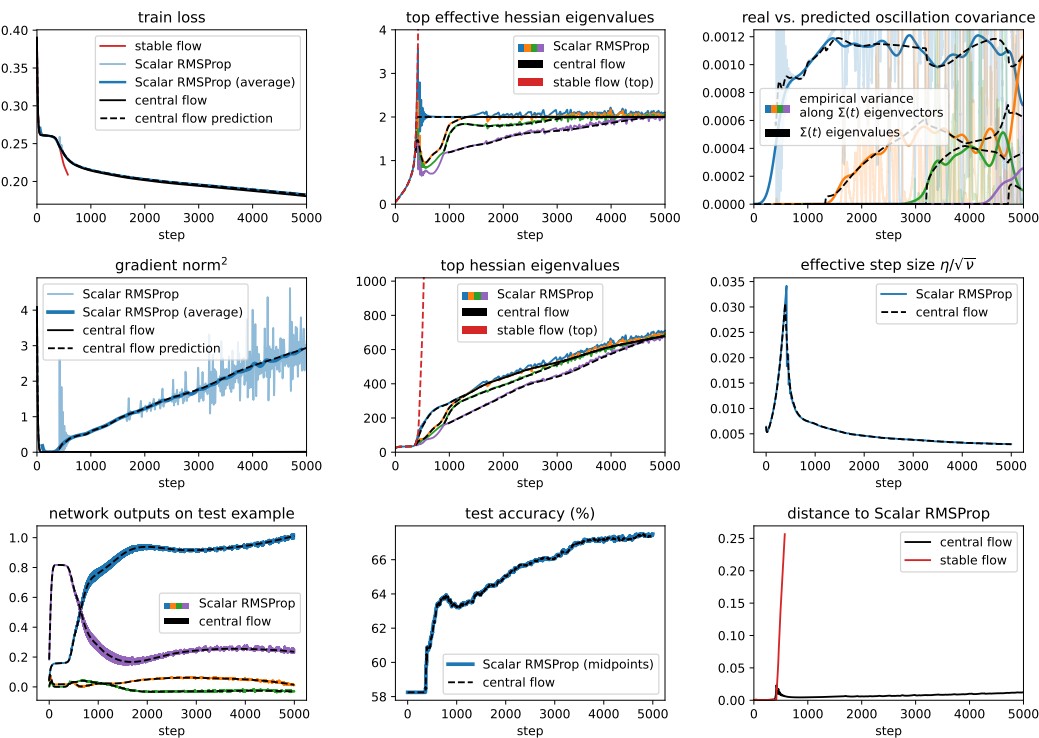

Figure 32.16: Scalar RMSProp central flow for a Mamba with MSE loss, $\eta$ =0.007, $\beta_2$ =0.99, and bias correction.

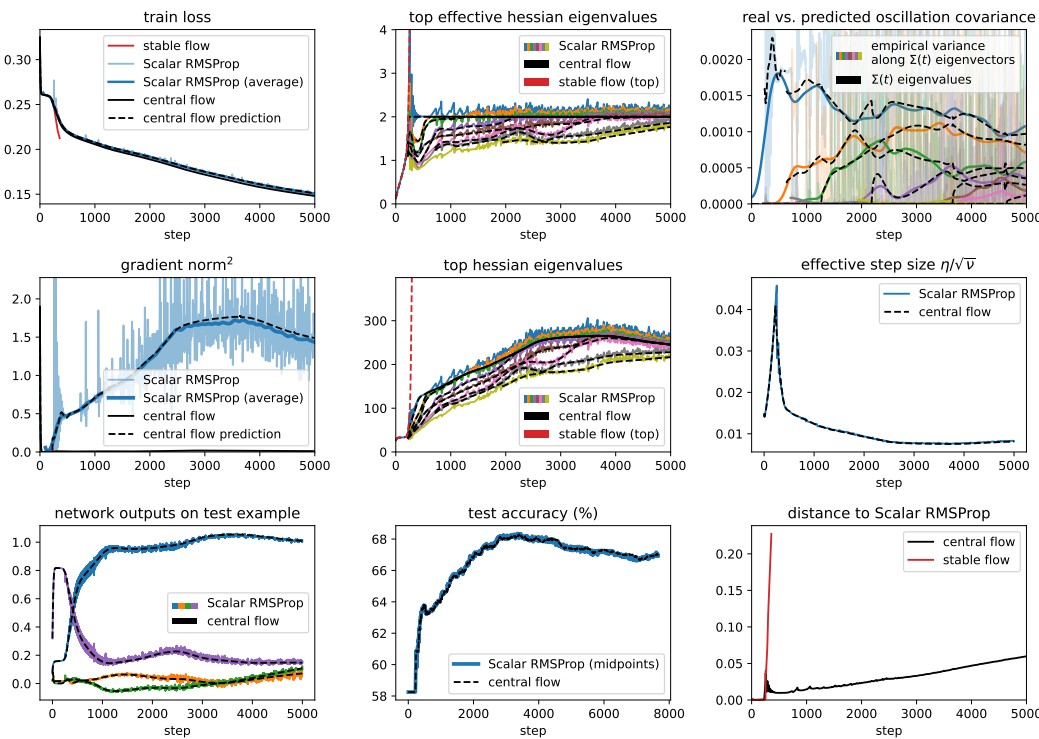

Figure 32.17: Scalar RMSProp central flow for a Mamba with MSE loss, $\eta$ =0.01, $\beta_2$ =0.99, and bias correction.

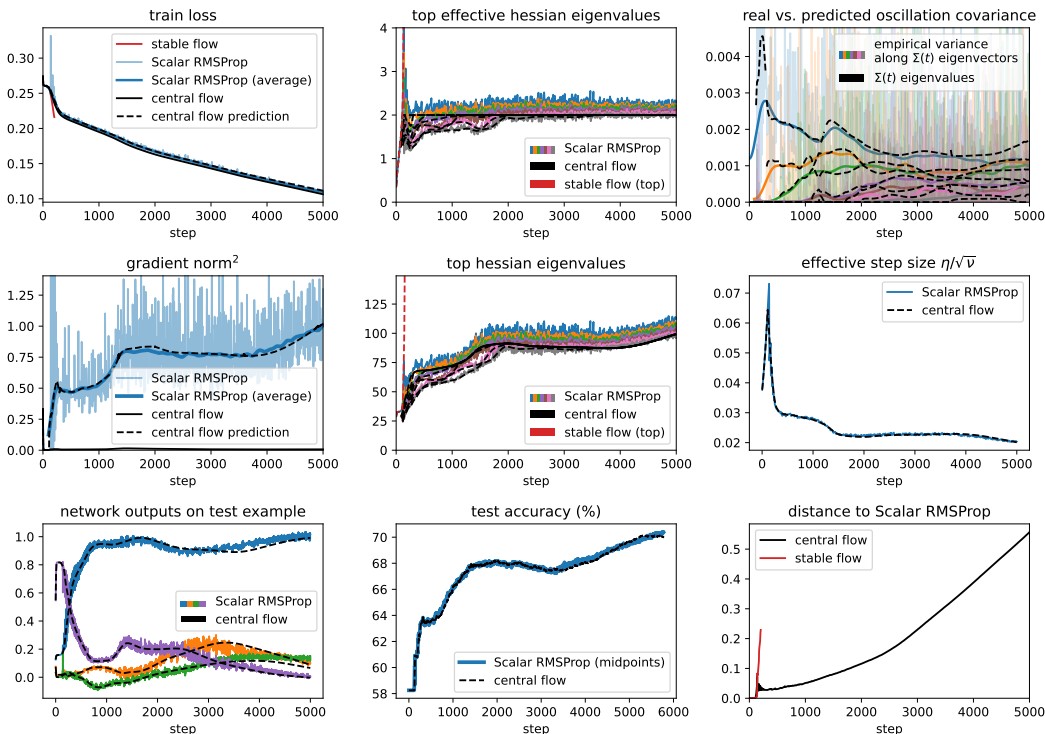

Figure 32.18: Scalar RMSProp central flow for a Mamba with MSE loss, $\eta =0.02$, $\beta_2 =0.99$, and bias correction.

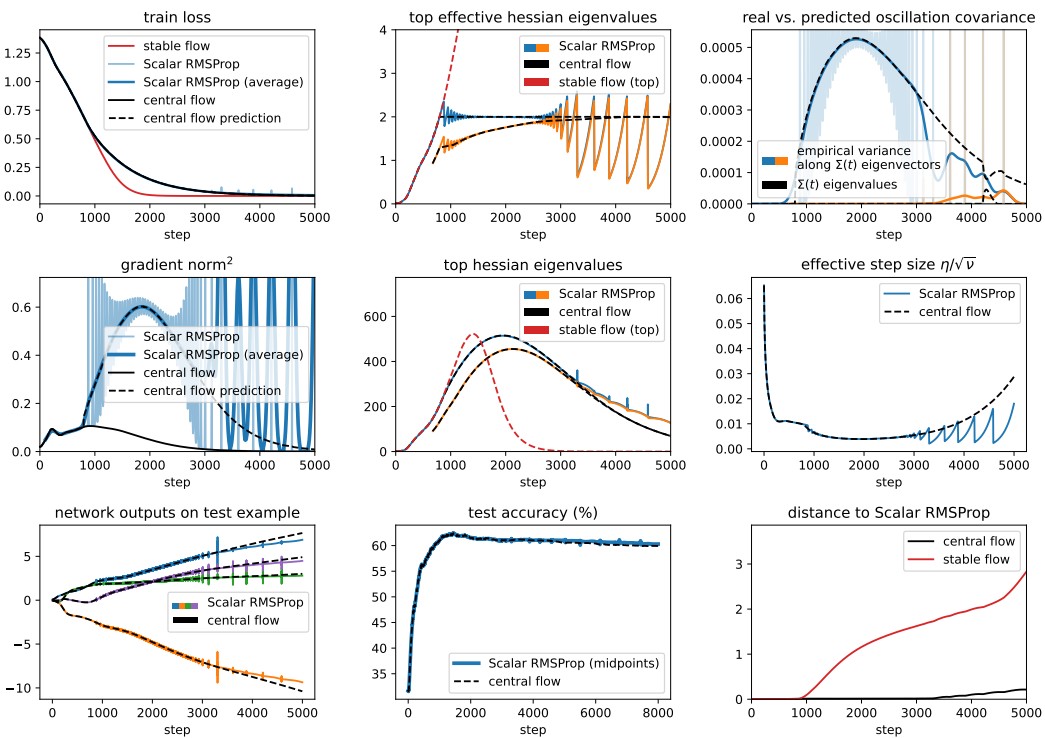

Figure 33.1: Scalar RMSProp central flow for a CNN with CE loss, $\eta = 0.003$, $\beta_2 = 0.99$, and bias correction.

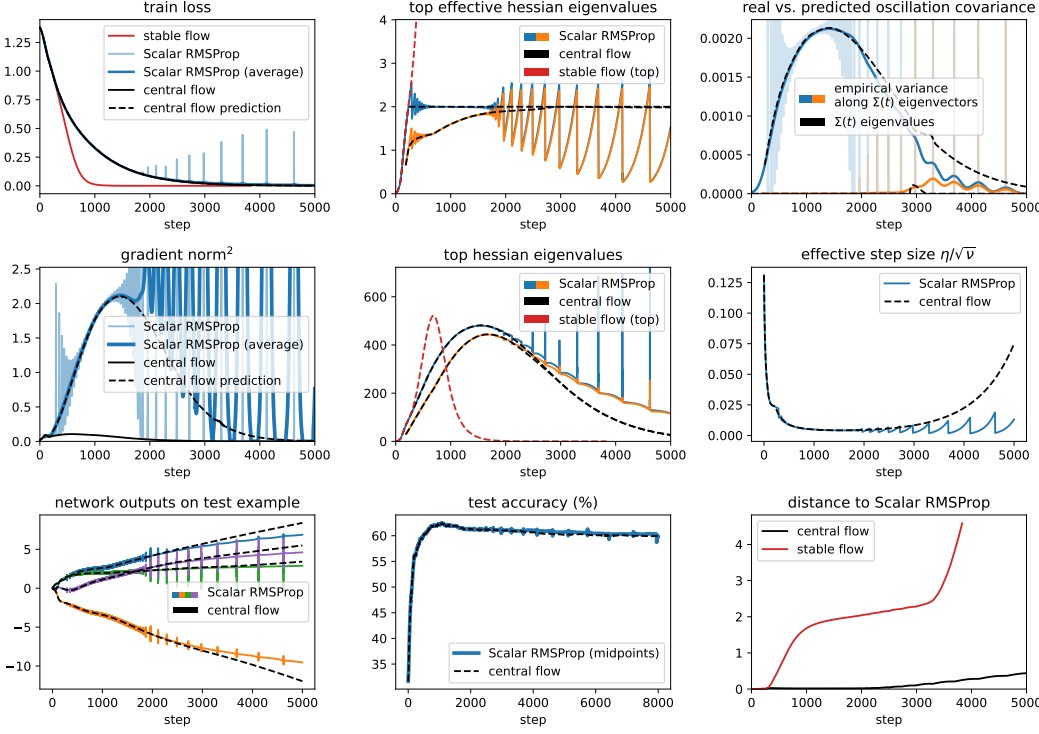

Figure 33.2: Scalar RMSProp central flow for a CNN with CE loss, $\eta = 0.006$, $\beta_2 = 0.99$, and bias correction.

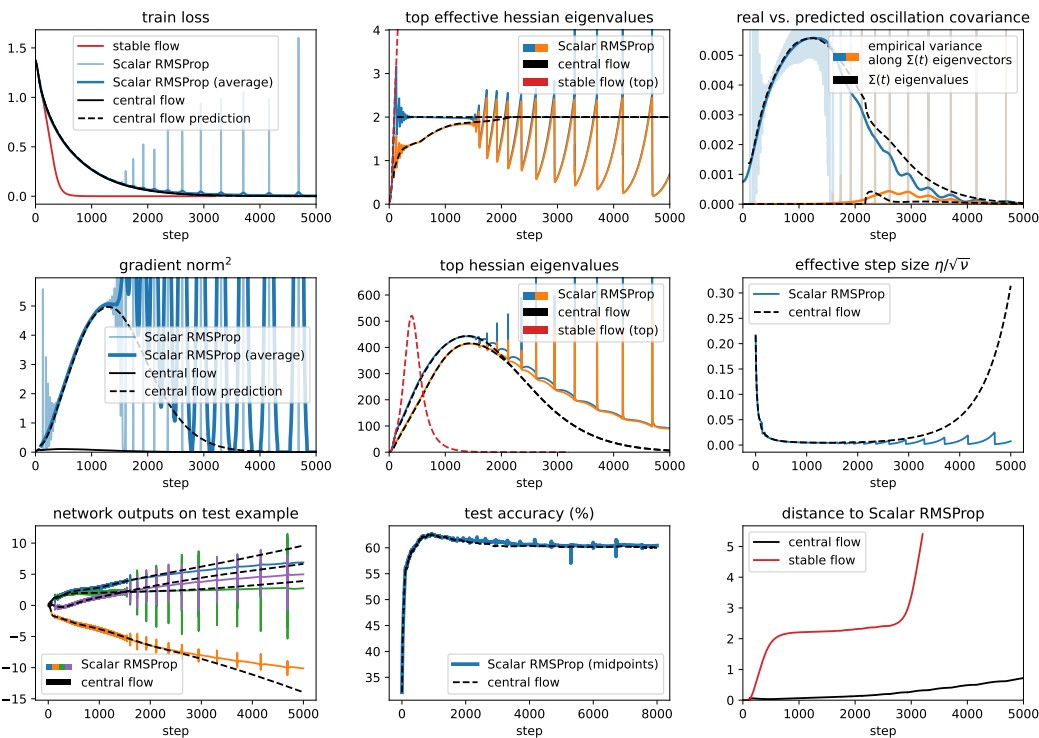

Figure 33.3: Scalar RMSProp central flow for a CNN with CE loss, $\eta$ =0.01, $\beta_2$ =0.99, and bias correction.

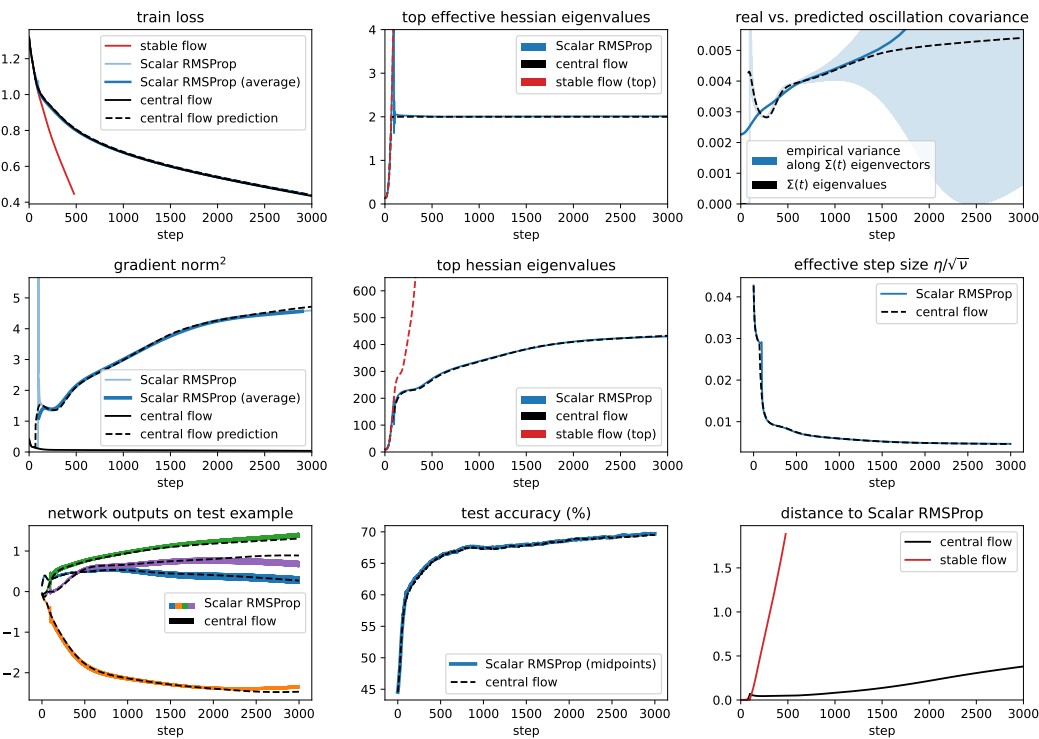

Figure 33.4: Scalar RMSProp central flow for a ResNet with CE loss, $\eta$ =0.01, $\beta_2$ =0.99, and bias correction.

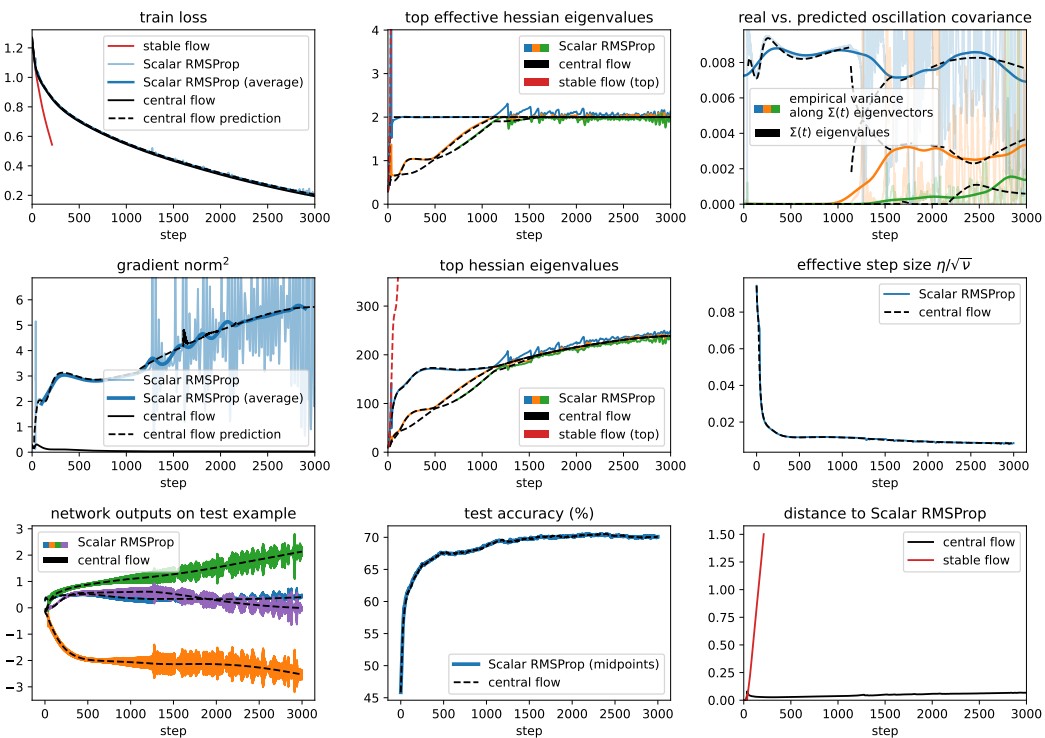

Figure 33.5: Scalar RMSProp central flow for a ResNet with CE loss, $\eta = 0.02$, $\beta_2 = 0.99$, and bias correction.

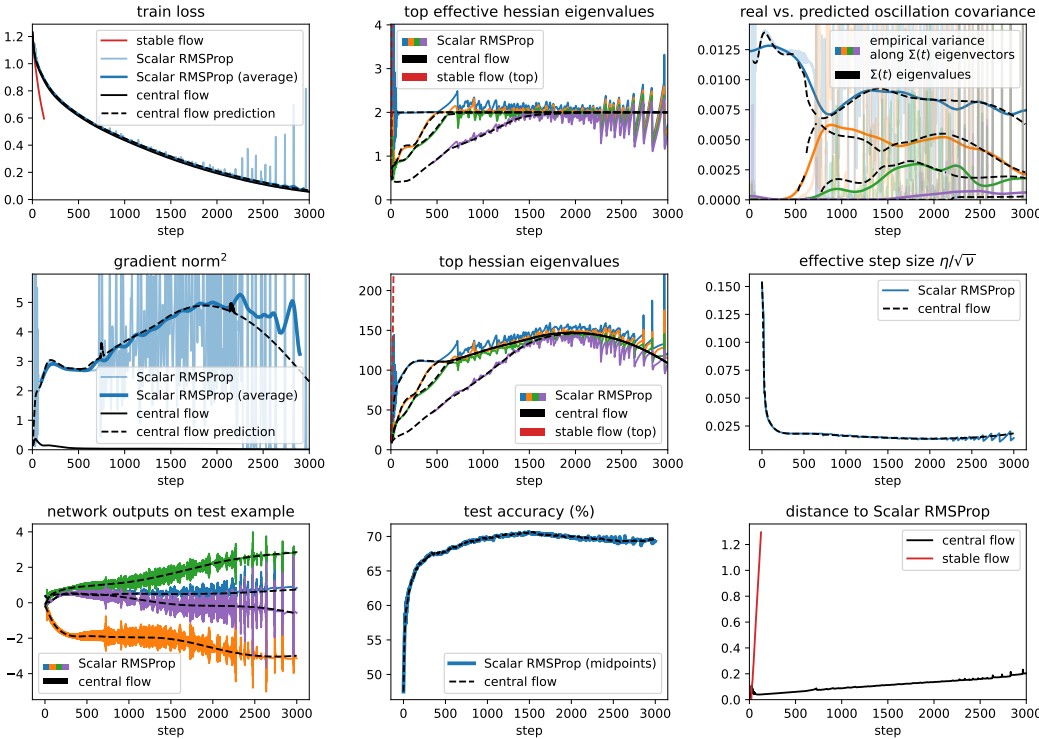

Figure 33.6: Scalar RMSProp central flow for a ResNet with CE loss, $\eta = 0.03$, $\beta_2 = 0.99$, and bias correction.

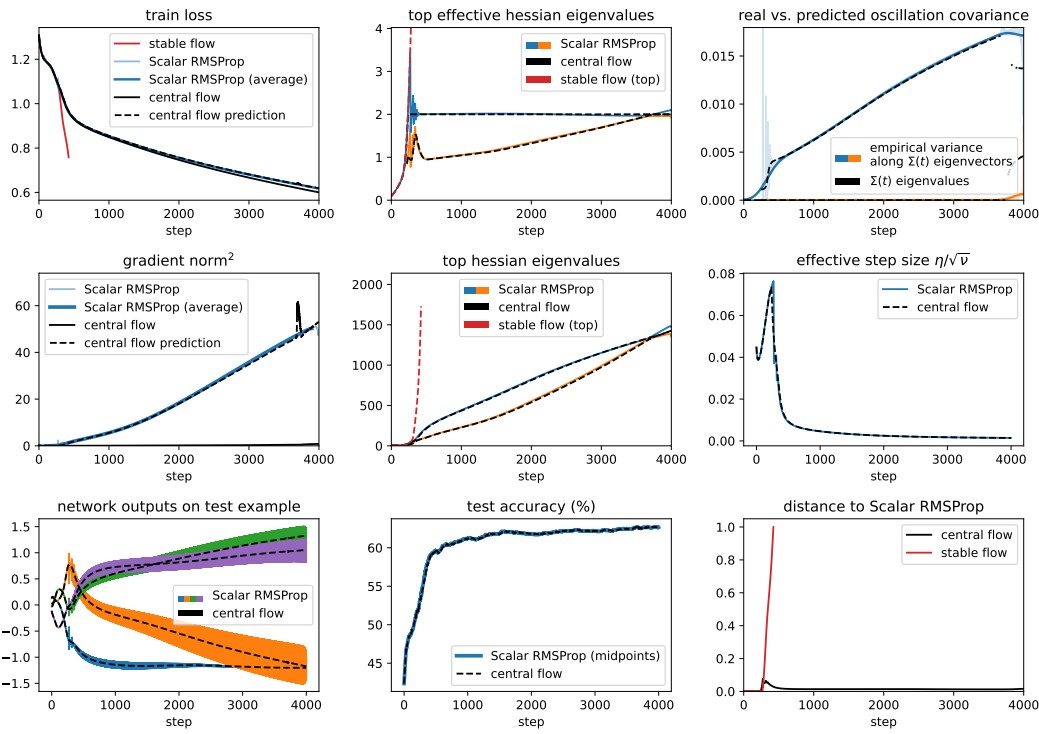

Figure 33.7: Scalar RMSProp central flow for a ViT with CE loss, $\eta = 0.01$, $\beta_2 = 0.99$, and bias correction.

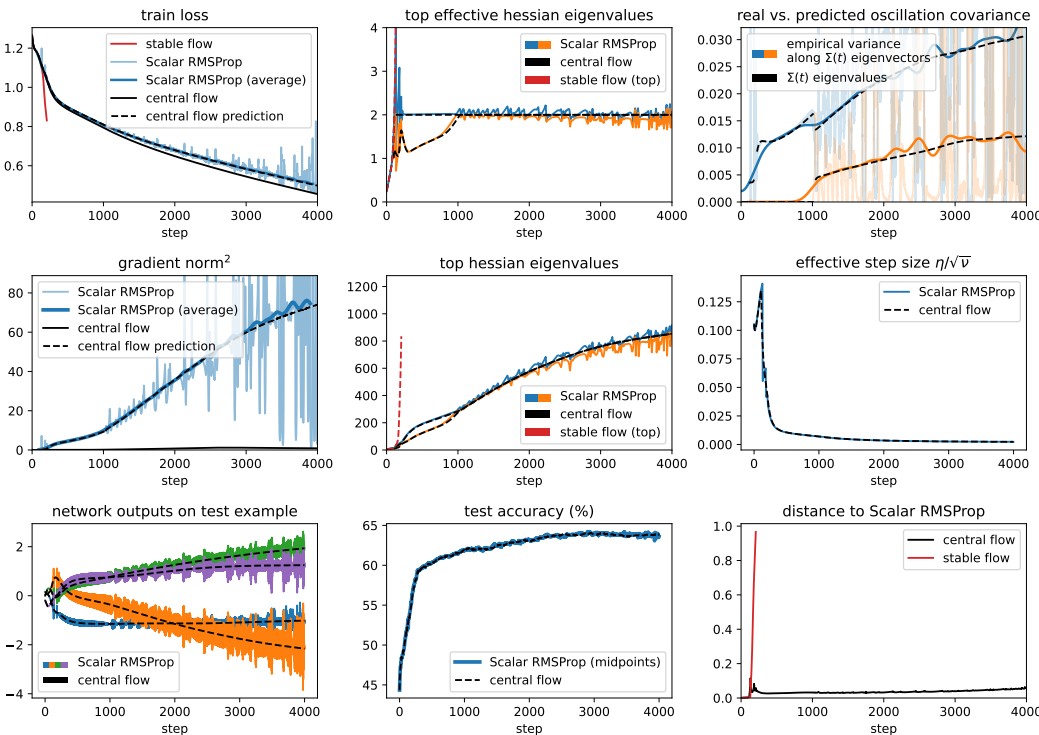

Figure 33.8: Scalar RMSProp central flow for a ViT with CE loss, $\eta = 0.02$, $\beta_2 = 0.99$, and bias correction.

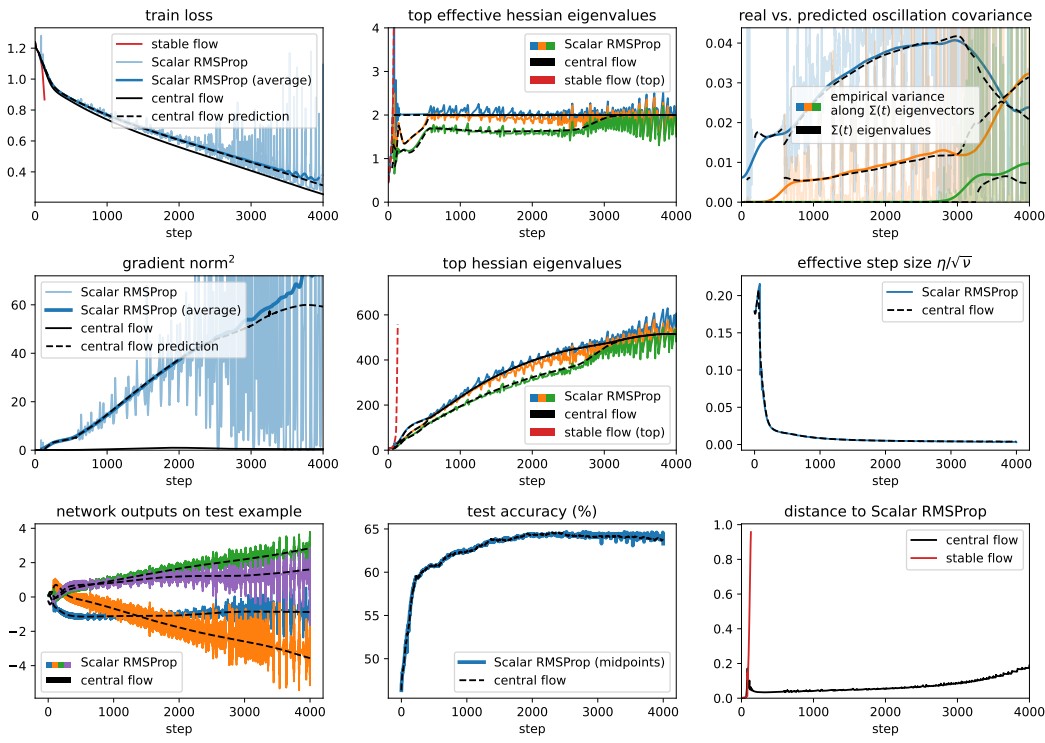

Figure 33.9: Scalar RMSProp central flow for a ViT with CE loss, $\eta =0.03$, $\beta_2 =0.99$, and bias correction.

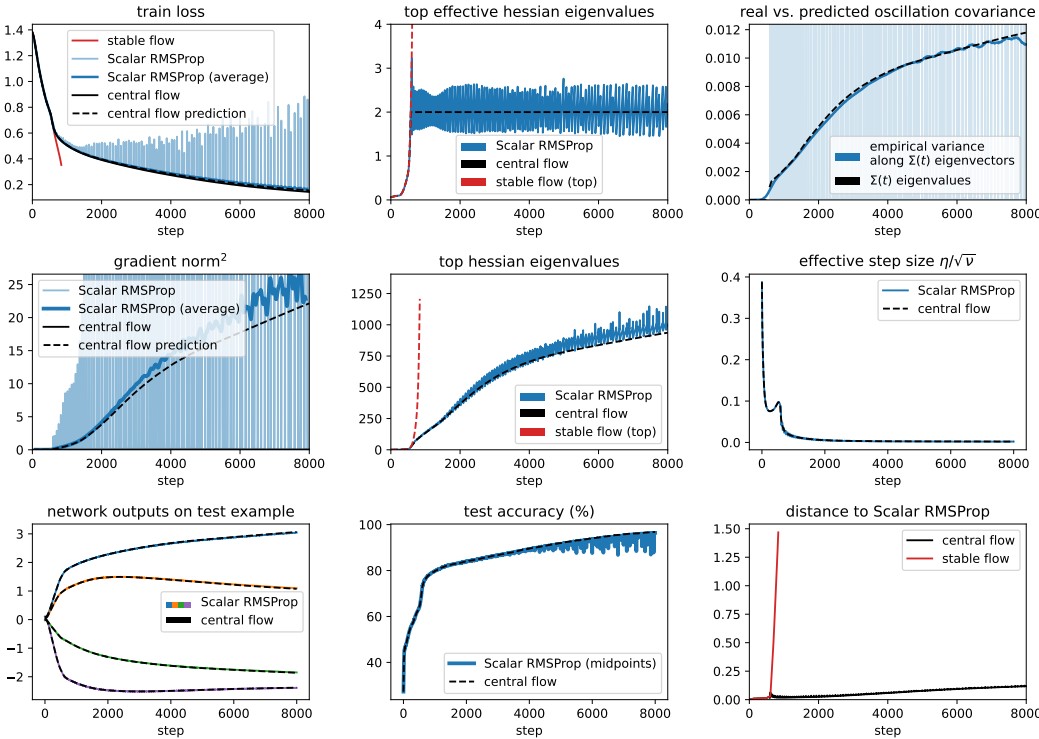

Figure 33.10: Scalar RMSProp central flow for a LSTM with CE loss, $\eta =0.01$, $\beta_2 =0.99$, and bias correction.

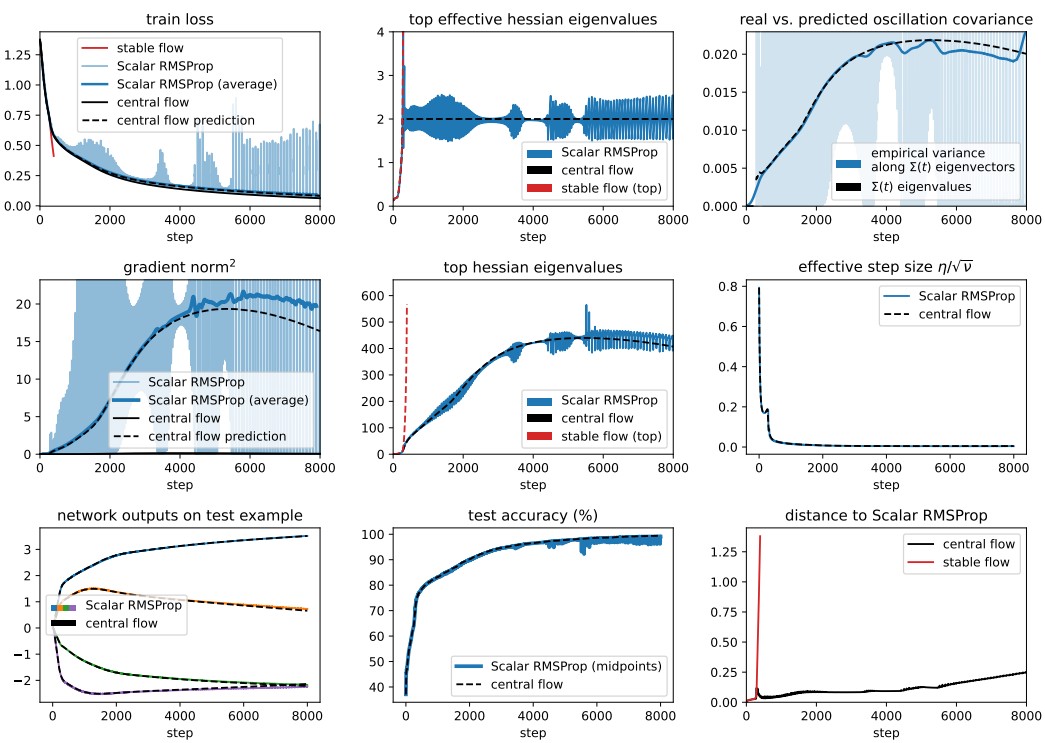

Figure 33.11: Scalar RMSProp central flow for a LSTM with CE loss, $\eta =0.02$, $\beta_2 =0.99$, and bias correction.

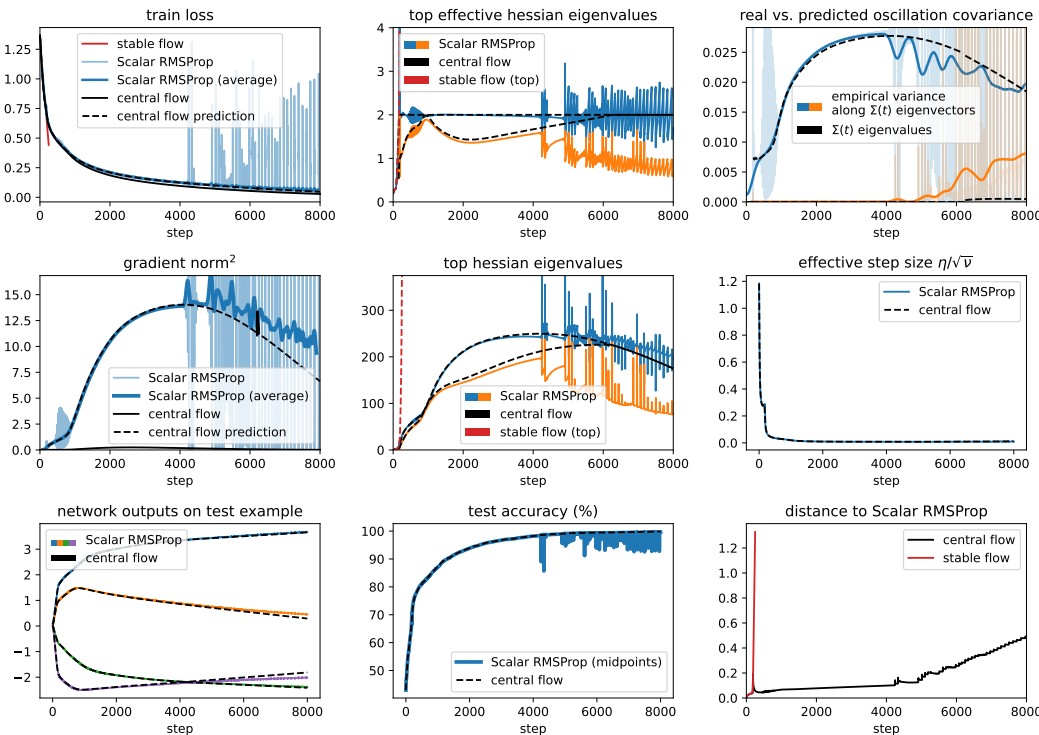

Figure 33.12: Scalar RMSProp central flow for a LSTM with CE loss, $\eta =0.03$, $\beta_2 =0.99$, and bias correction.

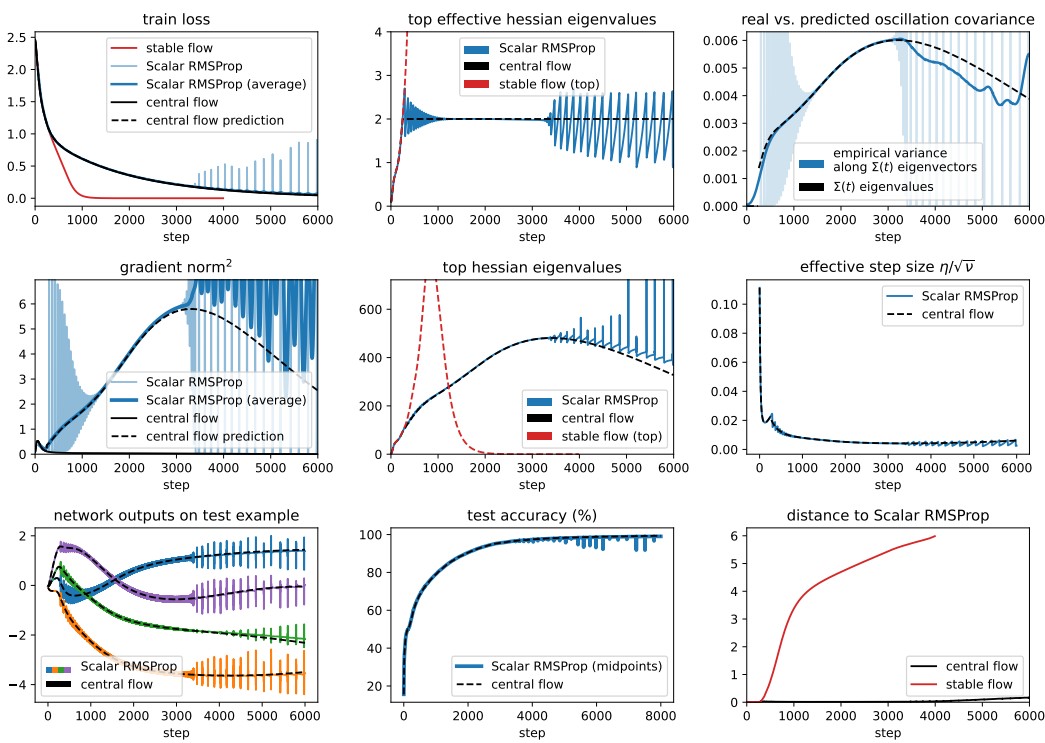

Figure 33.13: Scalar RMSProp central flow for a Transformer with CE loss, $\eta =0.01$, $\beta_2 =0.99$, and bias correction.

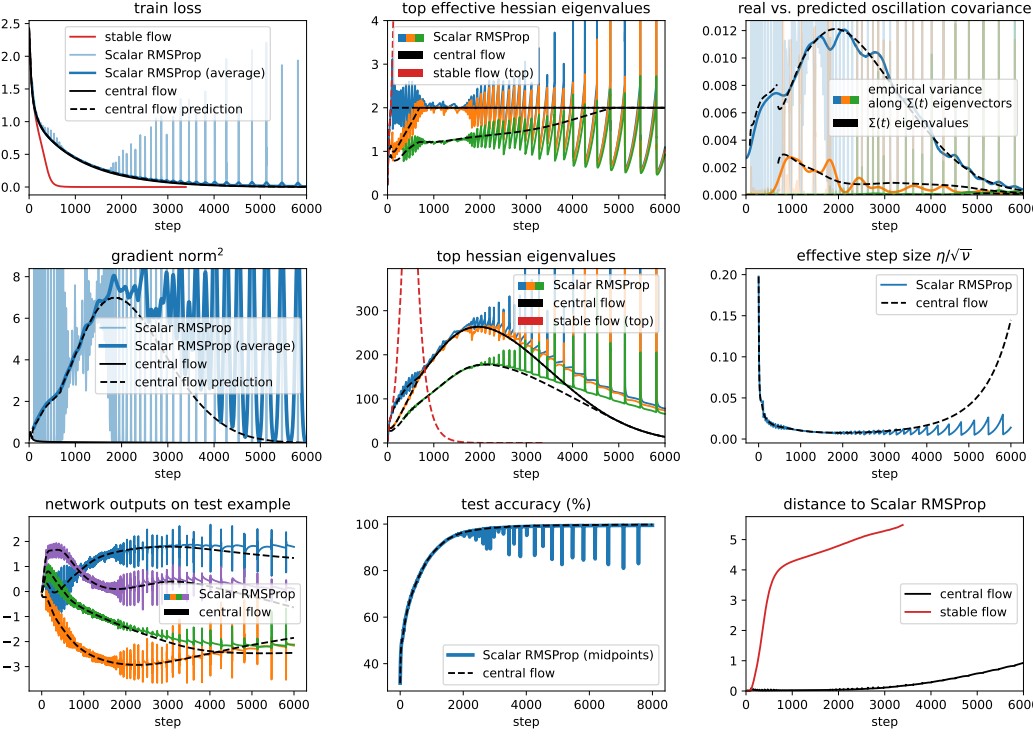

Figure 33.14: Scalar RMSProp central flow for a Transformer with CE loss, $\eta =0.02$, $\beta_2 =0.99$, and bias correction.

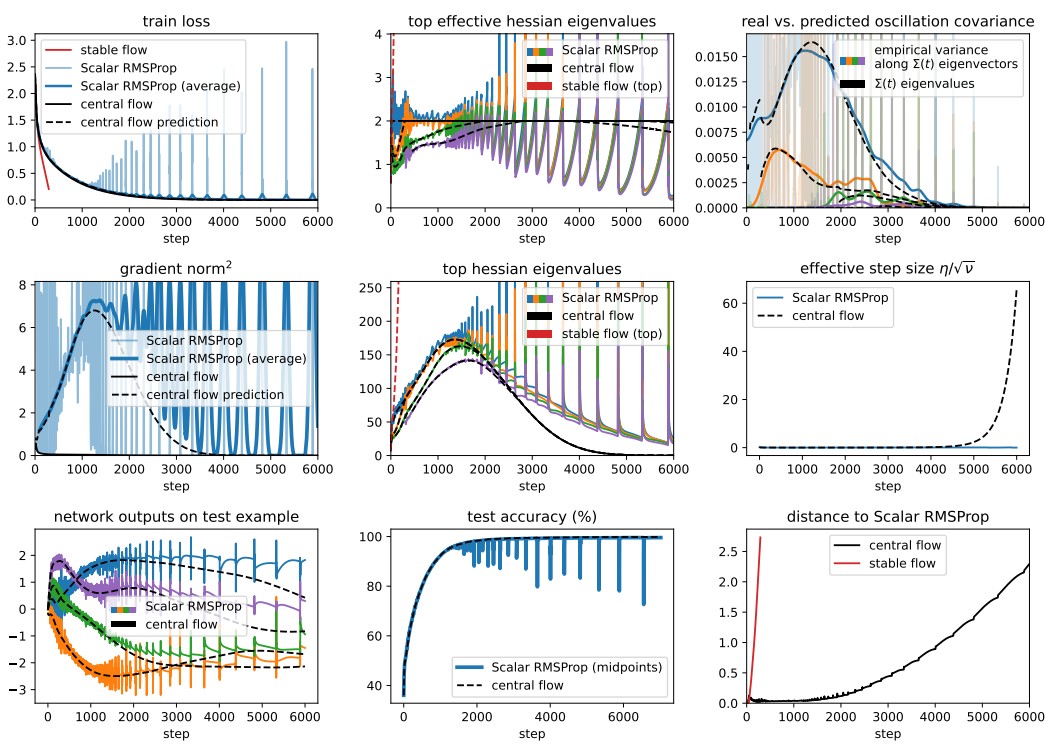

Figure 33.15: Scalar RMSProp central flow for a Transformer with CE loss, $\eta =0.03$, $\beta_2 =0.99$, and bias correction.

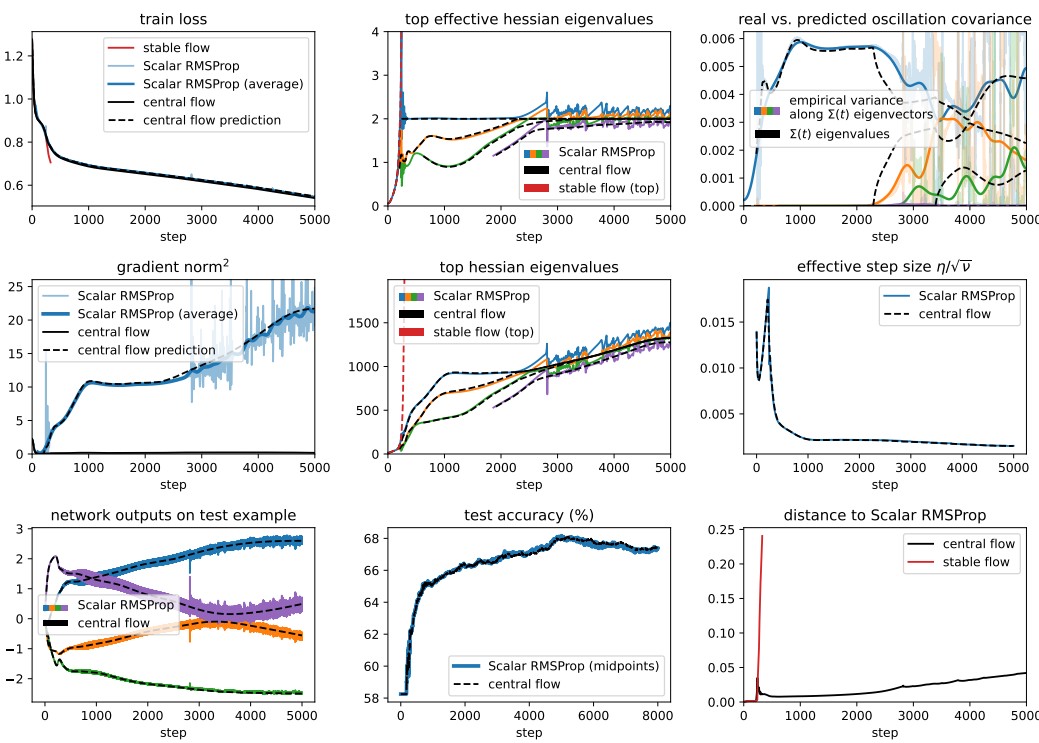

Figure 33.16: Scalar RMSProp central flow for a Mamba with CE loss, $\eta =0.007$, $\beta_2 =0.99$, and bias correction.

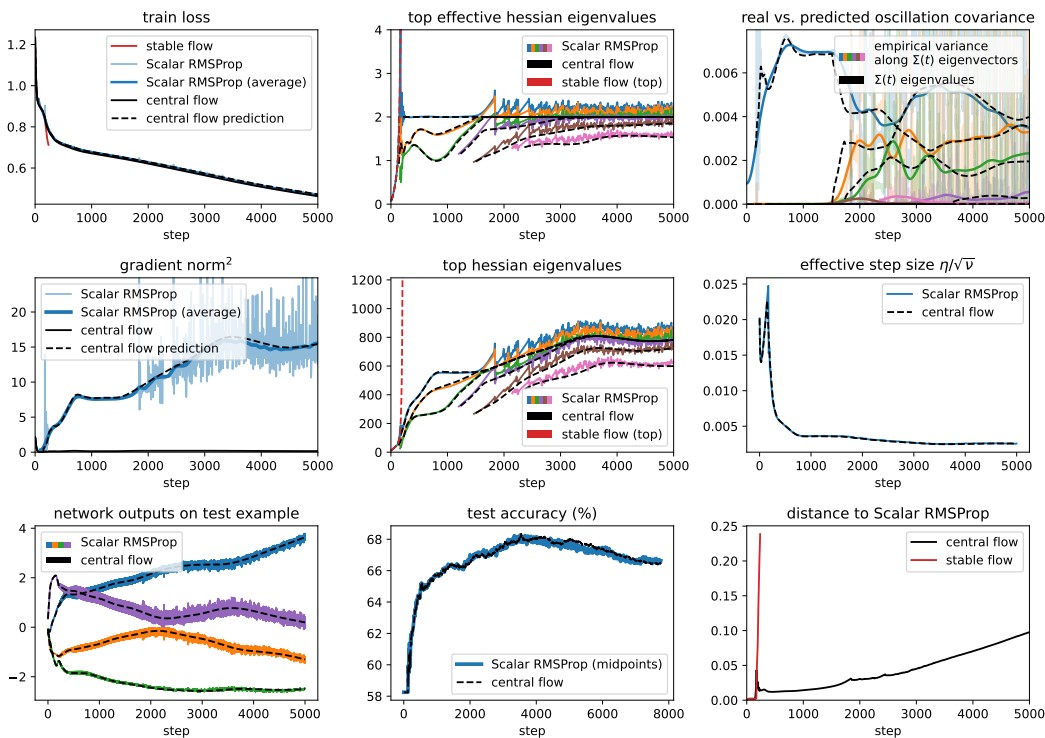

Figure 33.17: Scalar RMSProp central flow for a Mamba with CE loss, $\eta = 0.01$, $\beta_2 = 0.99$, and bias correction.

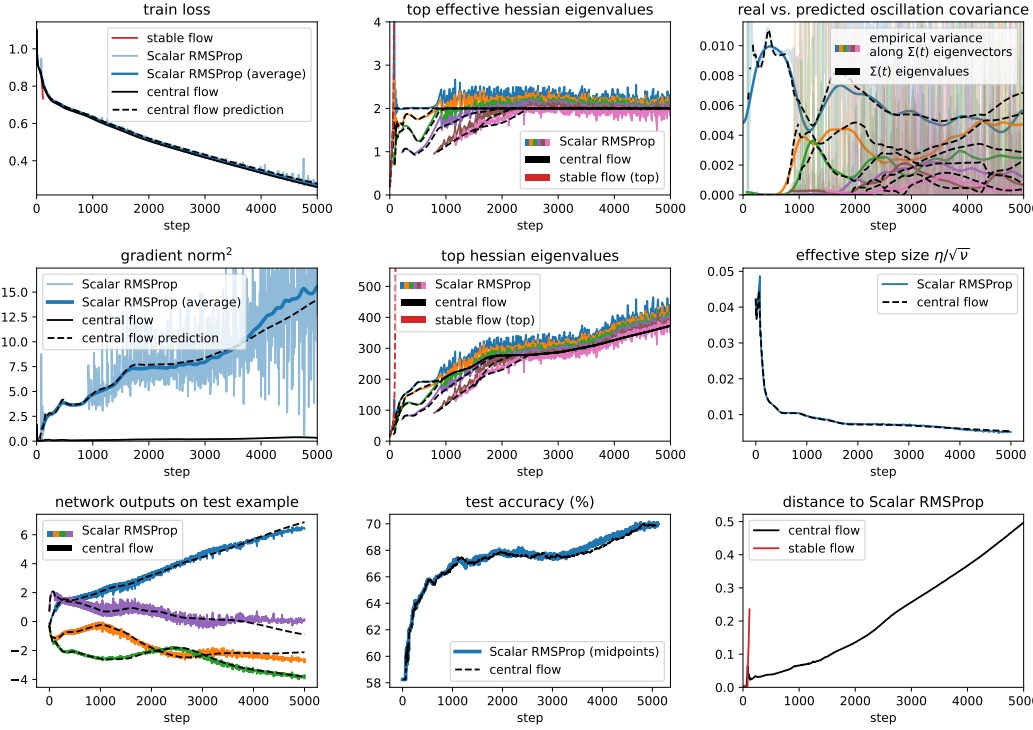

Figure 33.18: Scalar RMSProp central flow for a Mamba with CE loss, $\eta = 0.02$, $\beta_2 = 0.99$, and bias correction.

## C.3 RMSPROP

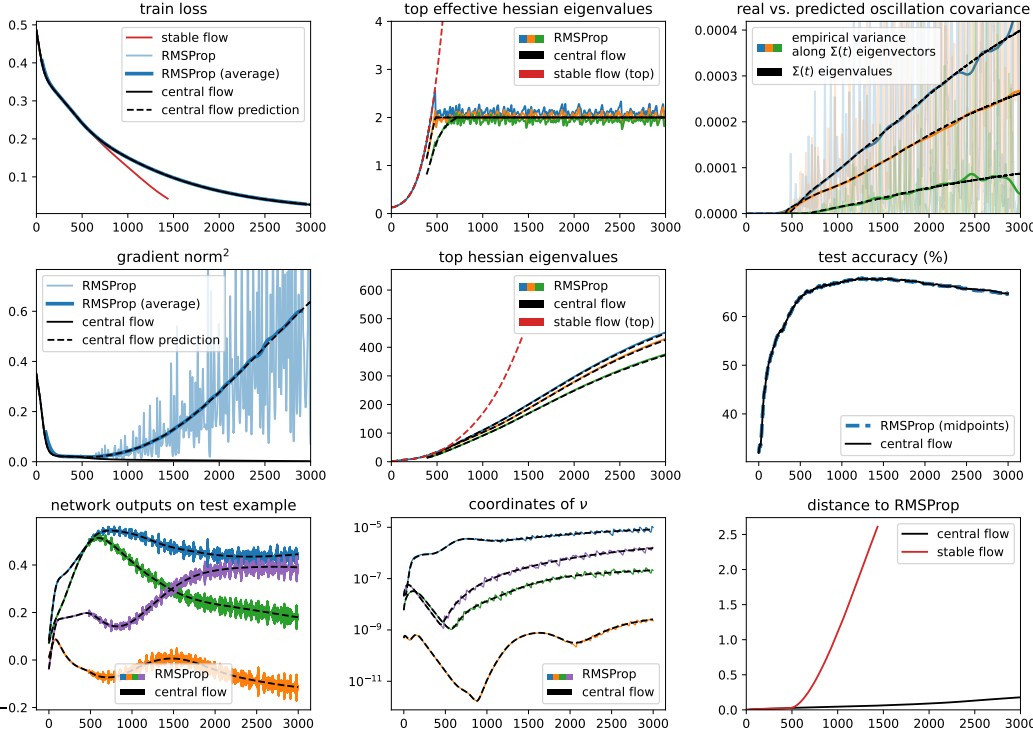

Figure 34.1: RMSProp central flow for a CNN with MSE loss, $\eta =$7e-06, $\beta_2 =$0.95, $\epsilon =$1e-08, and bias correction.

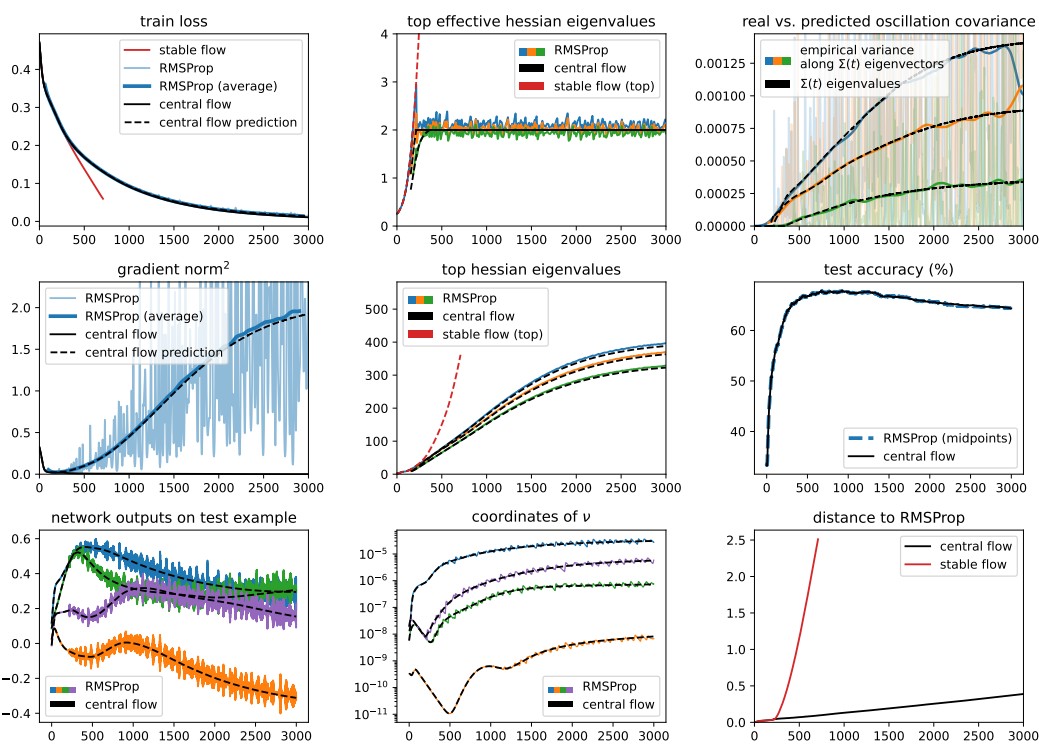

Figure 34.2: RMSProp central flow for a CNN with MSE loss, $\eta$ =1e-05, $\beta_2$ =0.95, $\epsilon$ =1e-08, and bias correction.

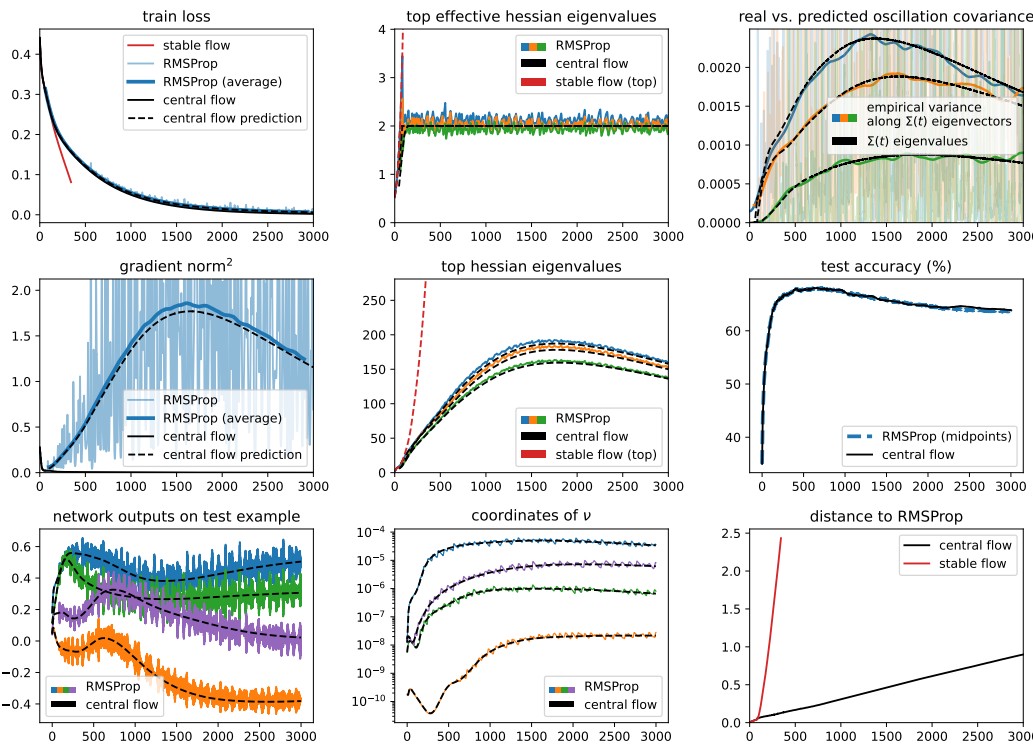

Figure 34.3: RMSProp central flow for a CNN with MSE loss, $\eta$ =2e-05, $\beta_2$ =0.95, $\epsilon$ =1e-08, and bias correction.

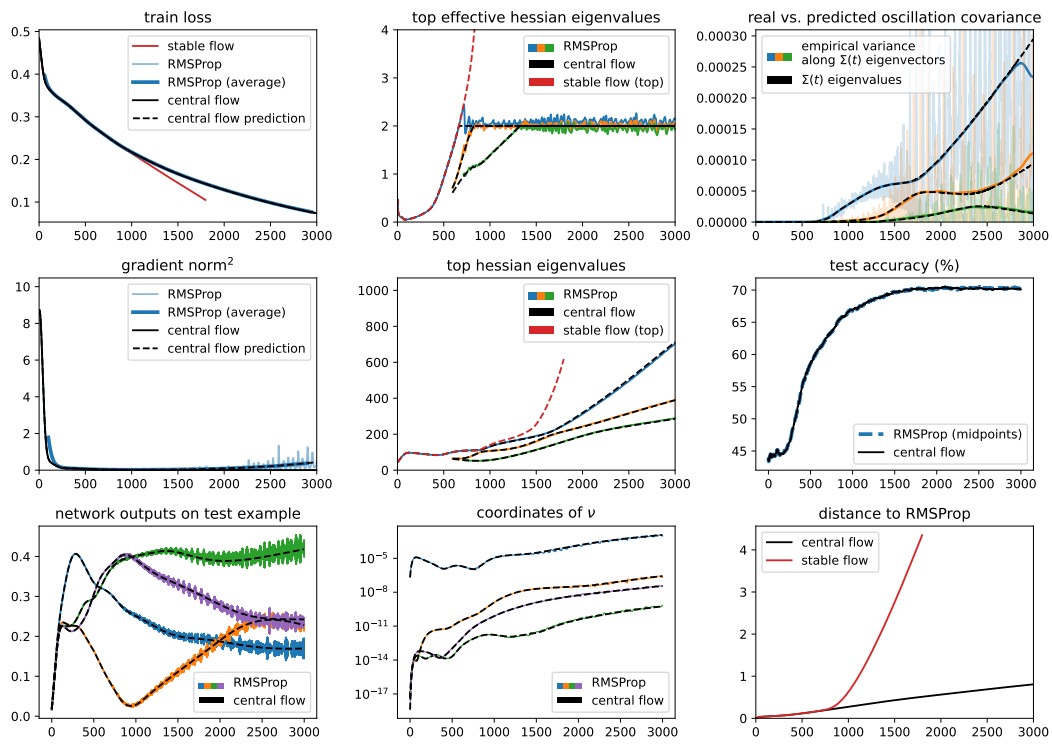

Figure 34.4: RMSProp central flow for a ResNet with MSE loss, $\eta =$1e-05, $\beta_2 =$0.95, $\epsilon =$1e-08, and bias correction.

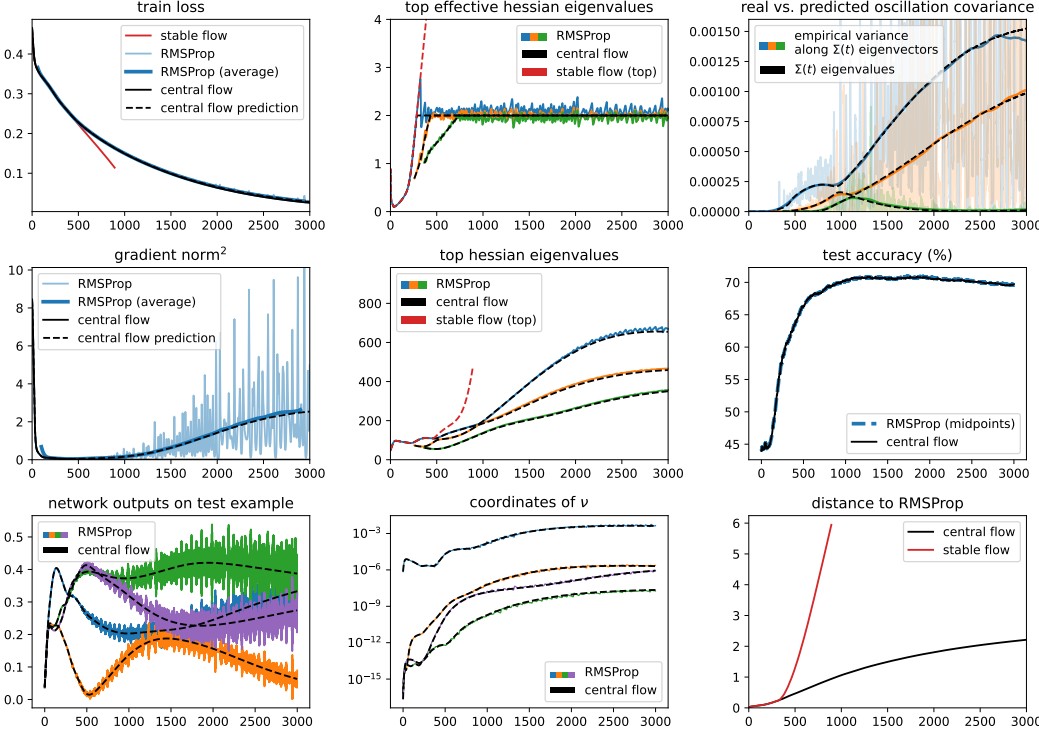

Figure 34.5: RMSProp central flow for a ResNet with MSE loss, $\eta =$2e-05, $\beta_2 =$0.95, $\epsilon =$1e-08, and bias correction.

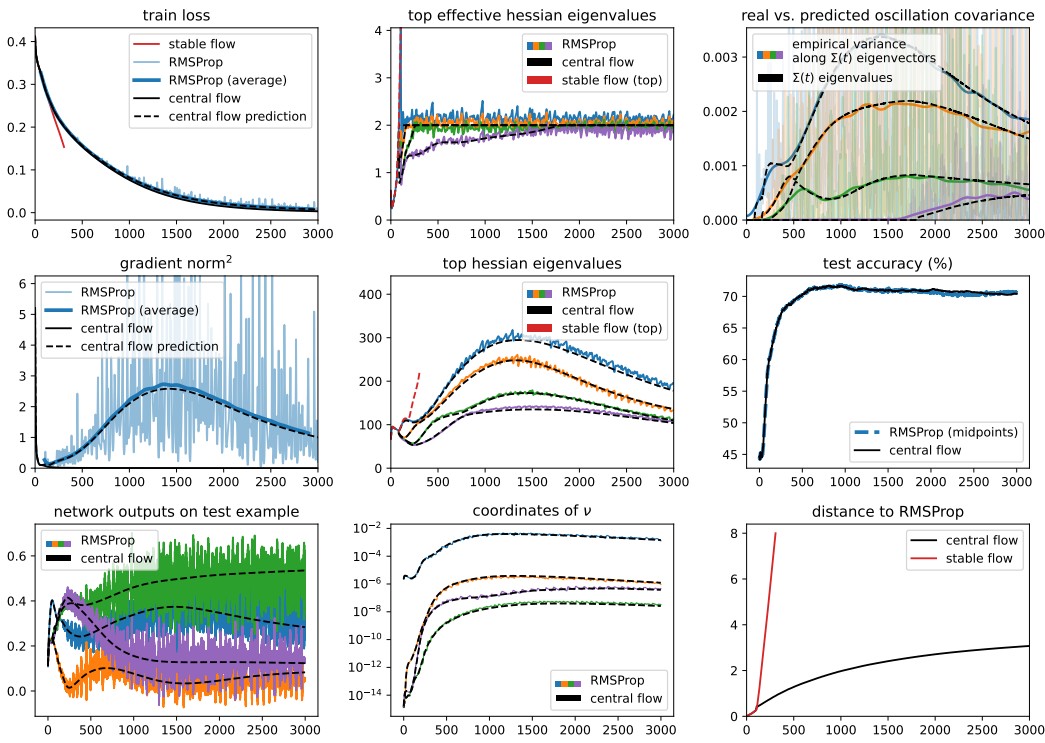

Figure 34.6: RMSProp central flow for a ResNet with MSE loss, $\eta$ =4e-05, $\beta_2$ =0.95, $\epsilon$ =1e-08, and bias correction.

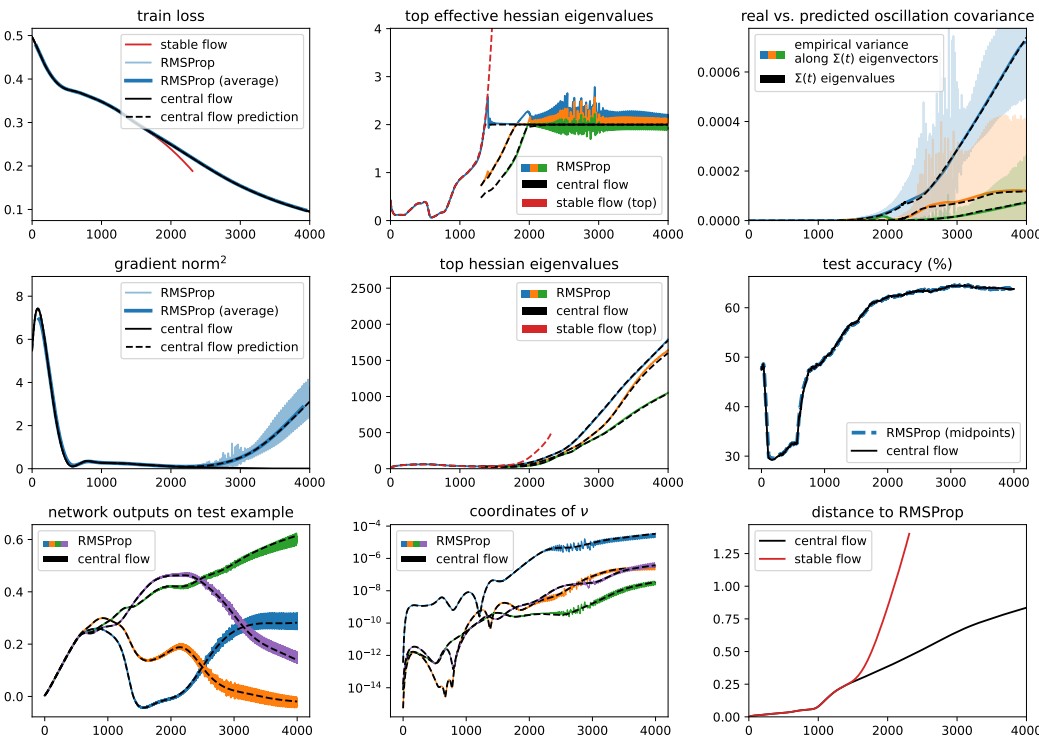

Figure 34.7: RMSProp central flow for a ViT with MSE loss, $\eta$ =5e-06, $\beta_2$ =0.95, $\epsilon$ =1e-08, and bias correction.

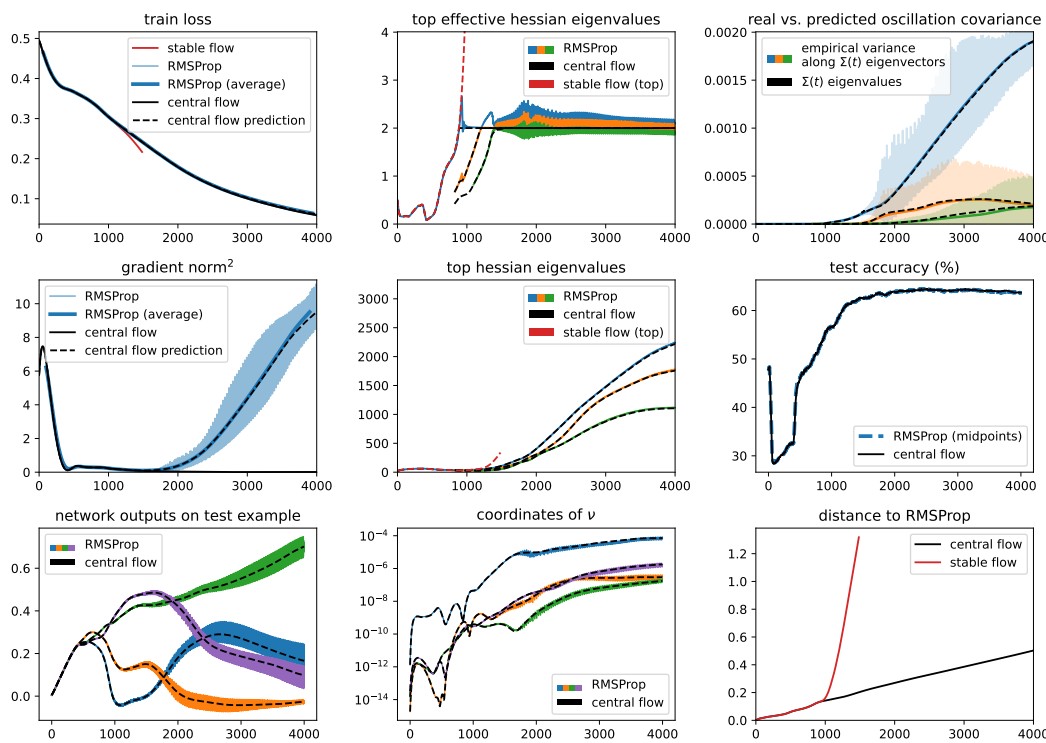

Figure 34.8: RMSProp central flow for a ViT with MSE loss, $\eta =$7e-06, $\beta_2 =$0.95, $\epsilon =$1e-08, and bias correction.

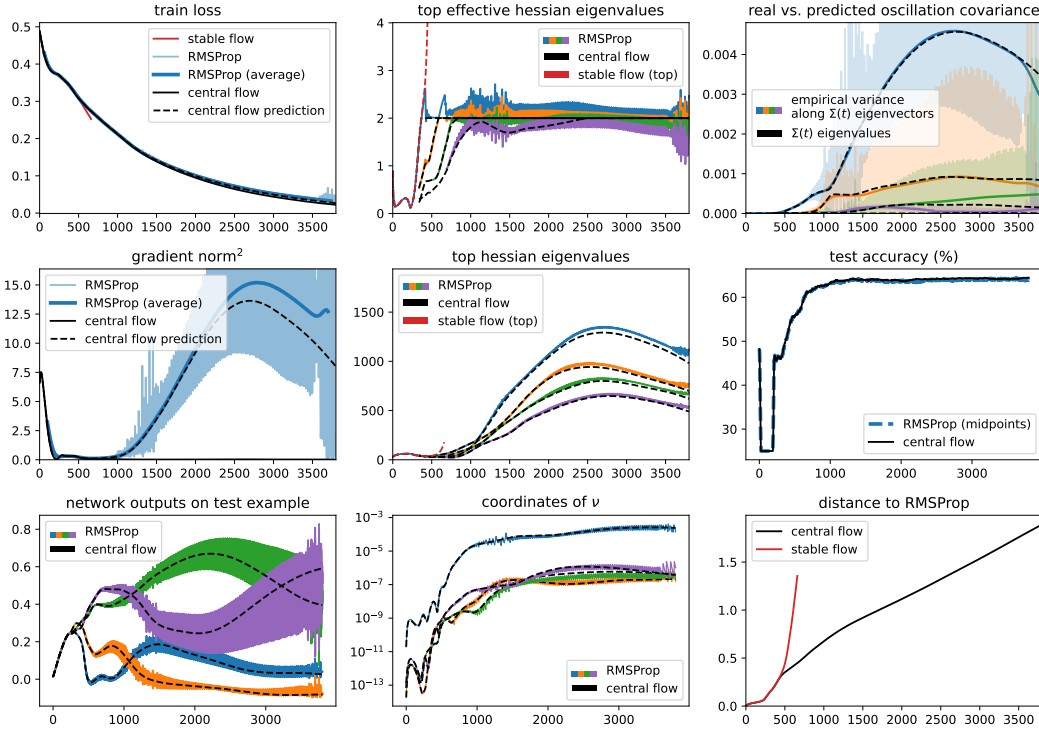

Figure 34.9: RMSProp central flow for a ViT with MSE loss, $\eta =$1e-05, $\beta_2 =$0.95, $\epsilon =$1e-08, and bias correction.

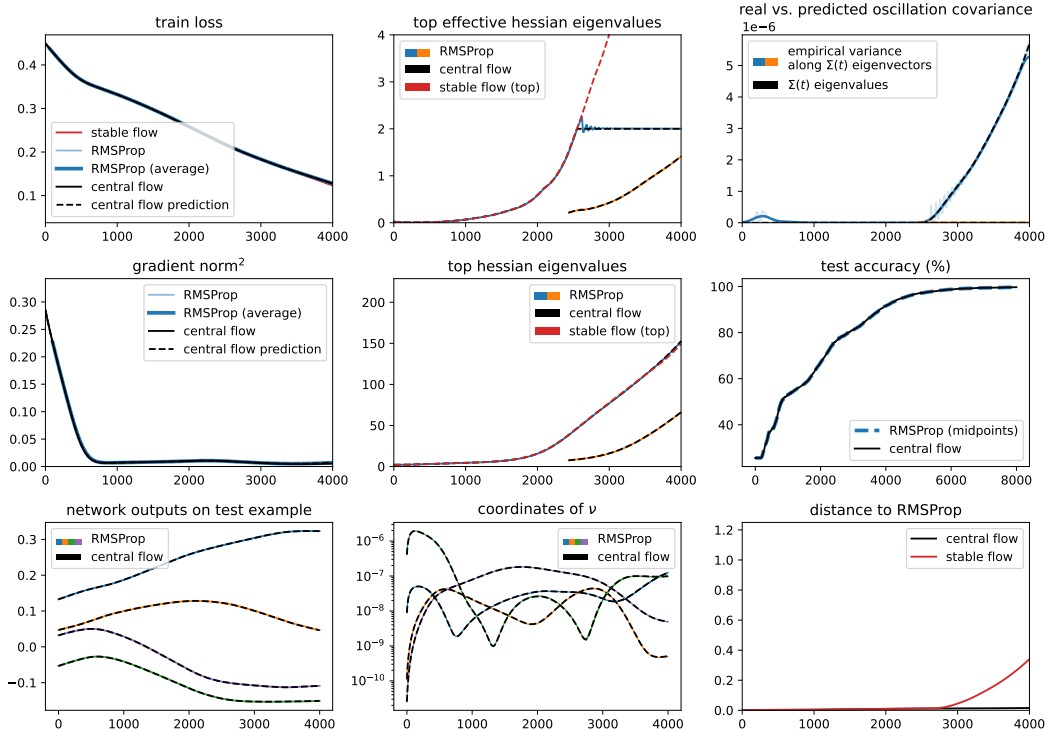

Figure 34.10: RMSProp central flow for a LSTM with MSE loss, $\eta =$1e-05, $\beta_2 =$0.95, $\epsilon =$1e-08, and bias correction.

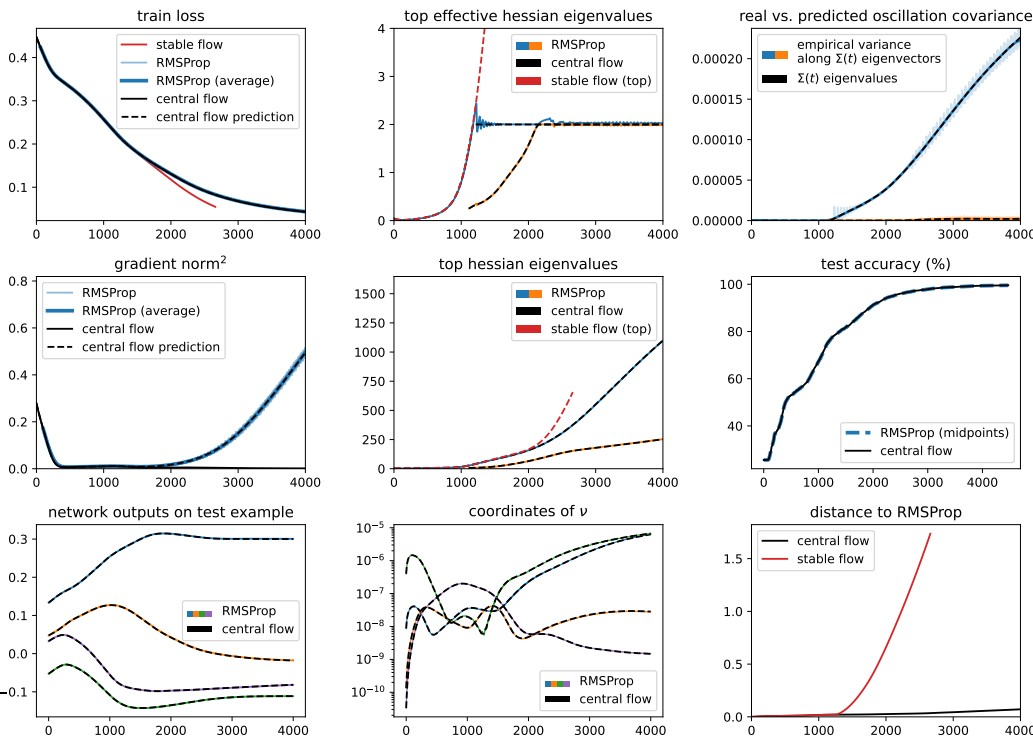

Figure 34.11: RMSProp central flow for a LSTM with MSE loss, $\eta =$2e-05, $\beta_2 =$0.95, $\epsilon =$1e-08, and bias correction.

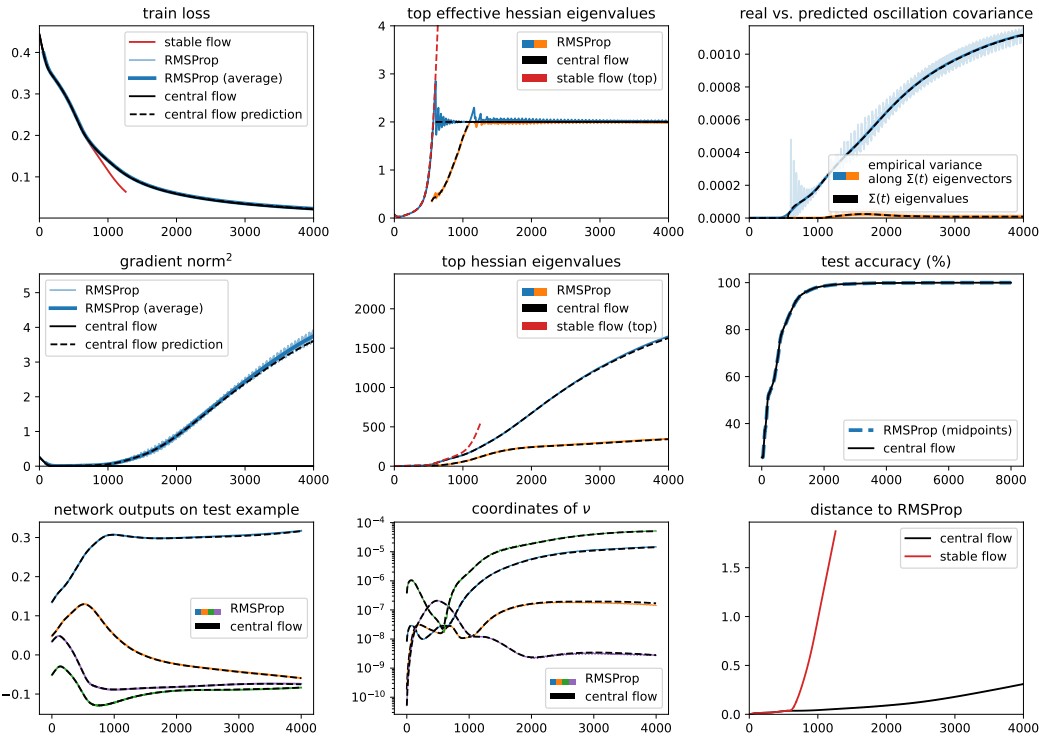

Figure 34.12: RMSProp central flow for a LSTM with MSE loss, $\eta$ =6e-05, $\beta_2$ =0.95, $\epsilon$ =1e-08, and bias correction.

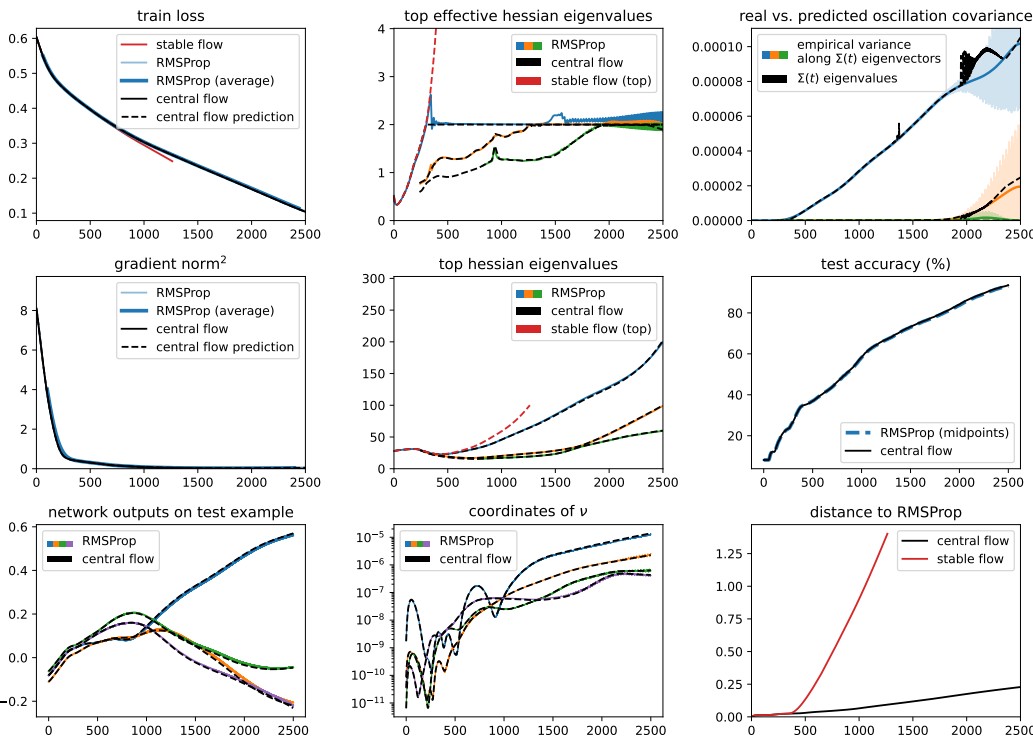

Figure 34.13: RMSProp central flow for a Transformer with MSE loss, $\eta$ =1e-05, $\beta_2$ =0.95, $\epsilon$ =1e-08, and bias correction.

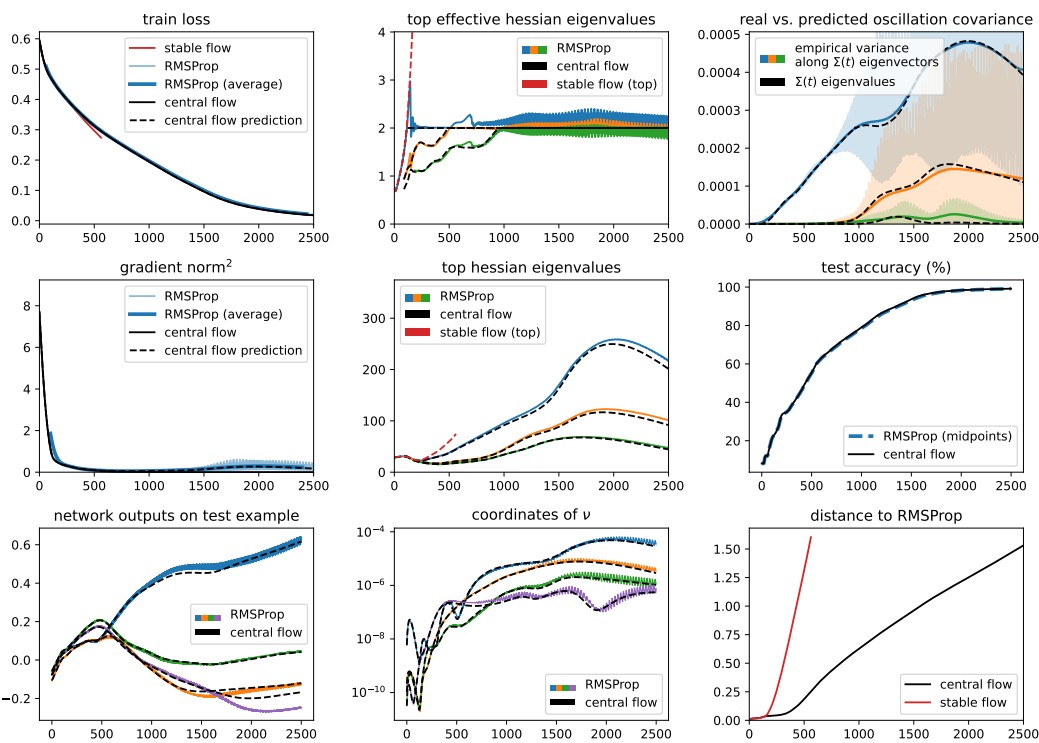

Figure 34.14: RMSProp central flow for a Transformer with MSE loss, $\eta$ =2e-05, $\beta_2$ =0.95, $\epsilon$ =1e-08, and bias correction.

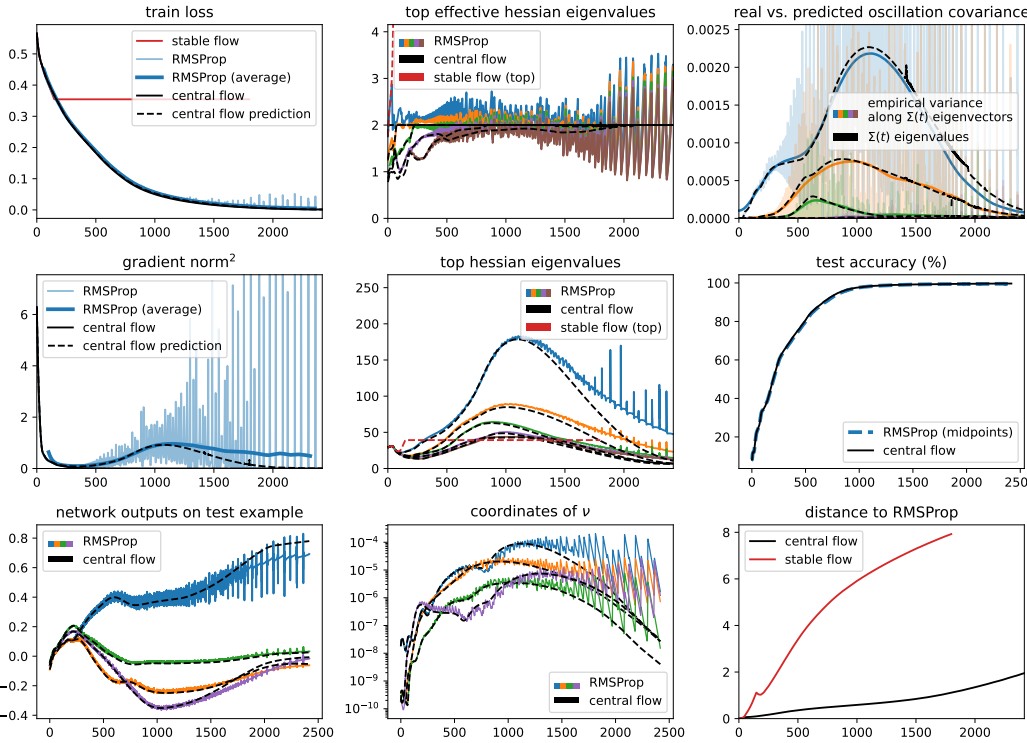

Figure 34.15: RMSProp central flow for a Transformer with MSE loss, $\eta$ =4e-05, $\beta_2$ =0.95, $\epsilon$ =1e-08, and bias correction.

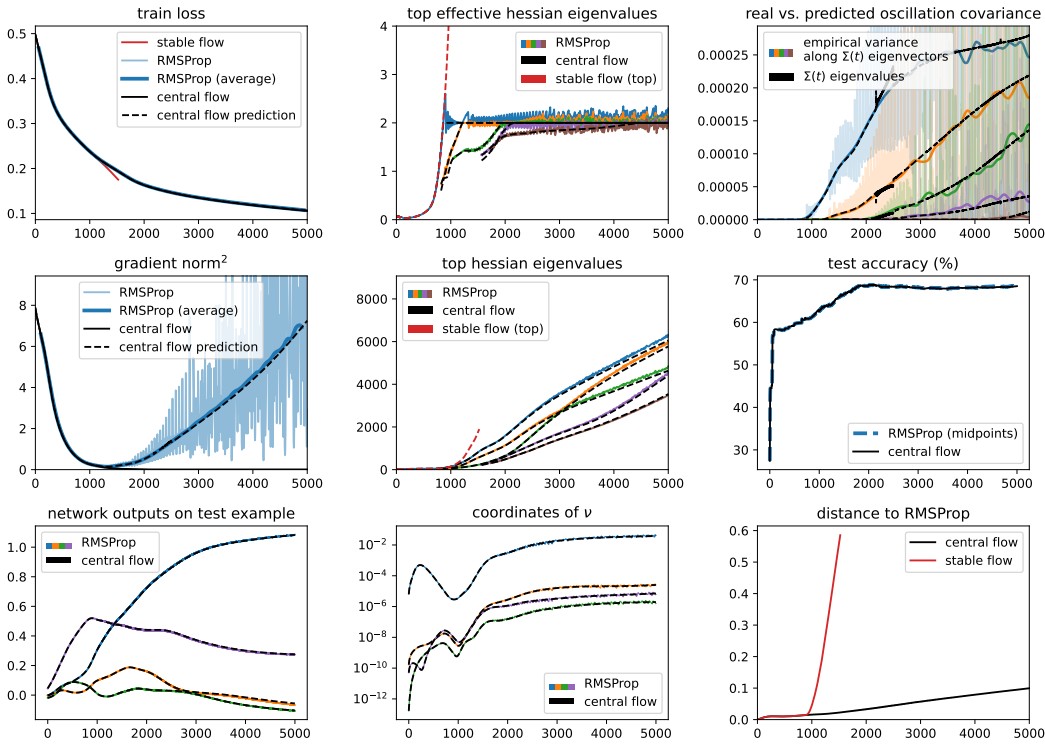

Figure 34.16: RMSProp central flow for a Mamba with MSE loss, $\eta$ =7e-06, $\beta_2$ =0.95, $\epsilon$ =1e-08, and bias correction.

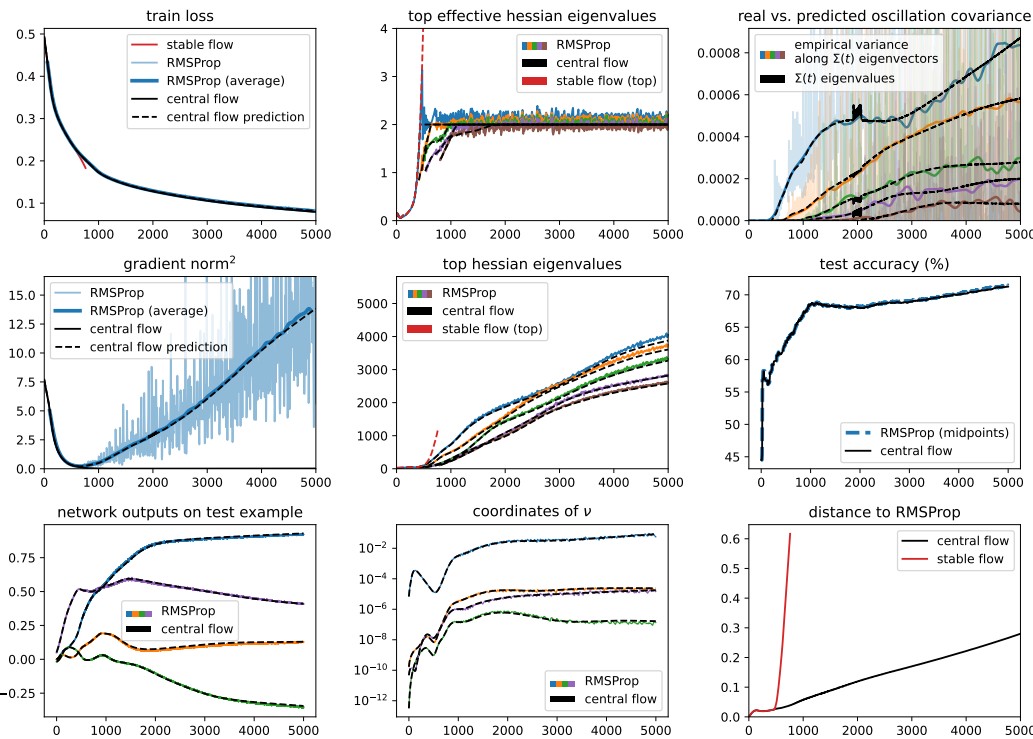

Figure 34.17: RMSProp central flow for a Mamba with MSE loss, $\eta$ =1e-05, $\beta_2$ =0.95, $\epsilon$ =1e-08, and bias correction.

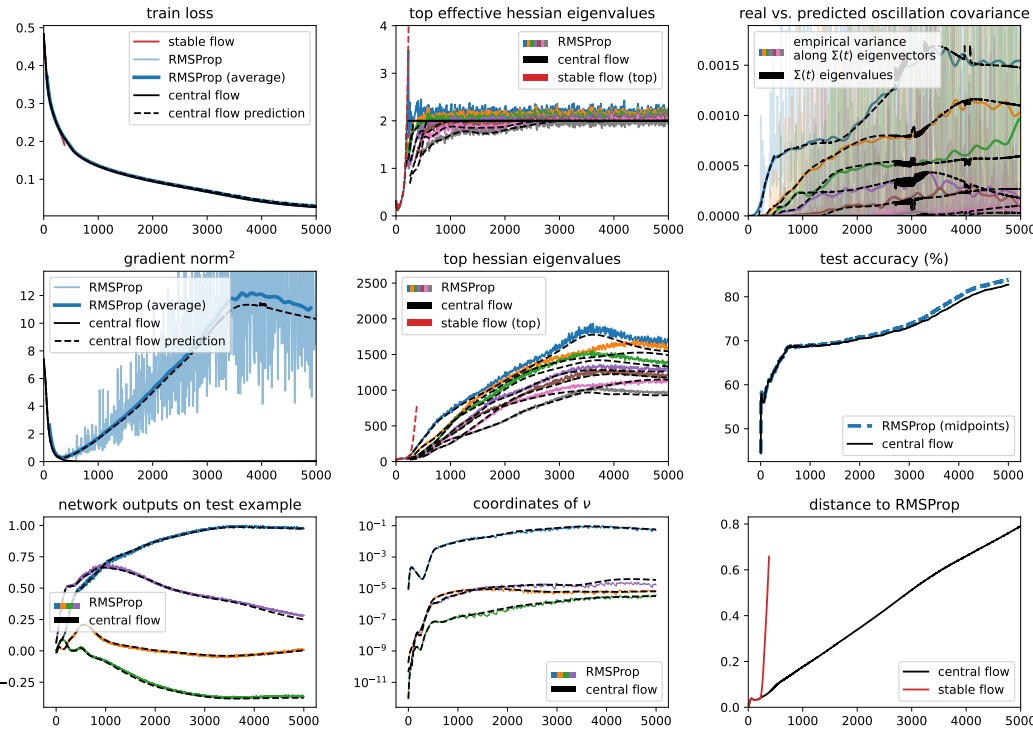

Figure 34.18: RMSProp central flow for a Mamba with MSE loss, $\eta$ =2e-05, $\beta_2$ =0.95, $\epsilon$ =1e-08, and bias correction.

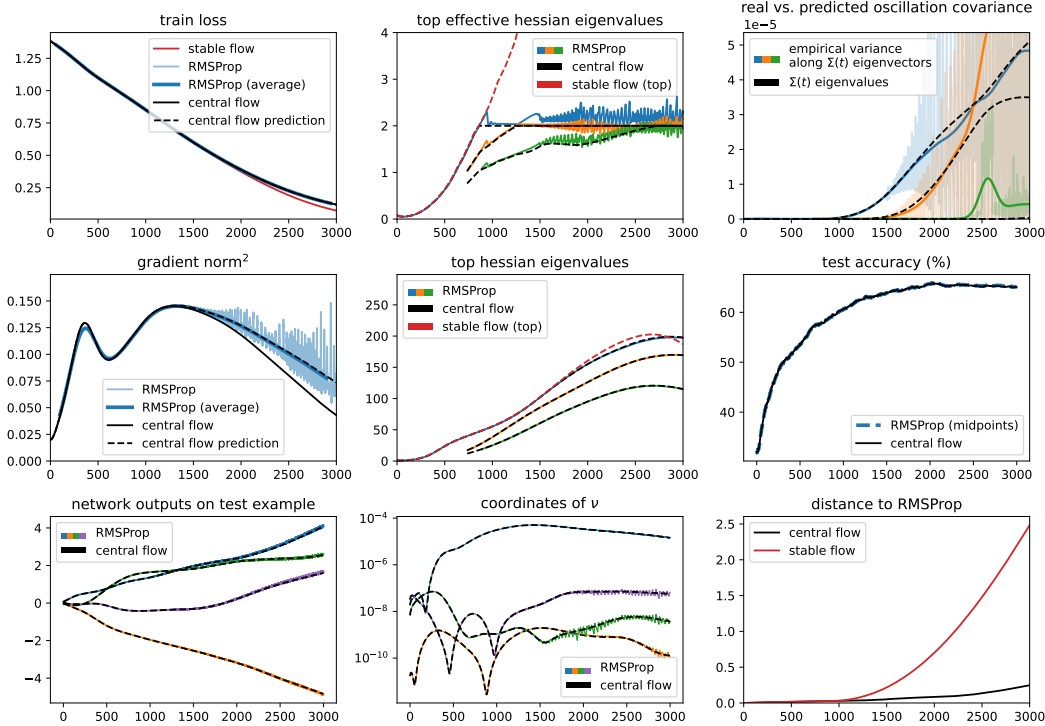

Figure 35.1: RMSProp central flow for a CNN with CE loss, $\eta =$7e-06, $\beta_2 =$0.95, $\epsilon =$1e-08, and bias correction.

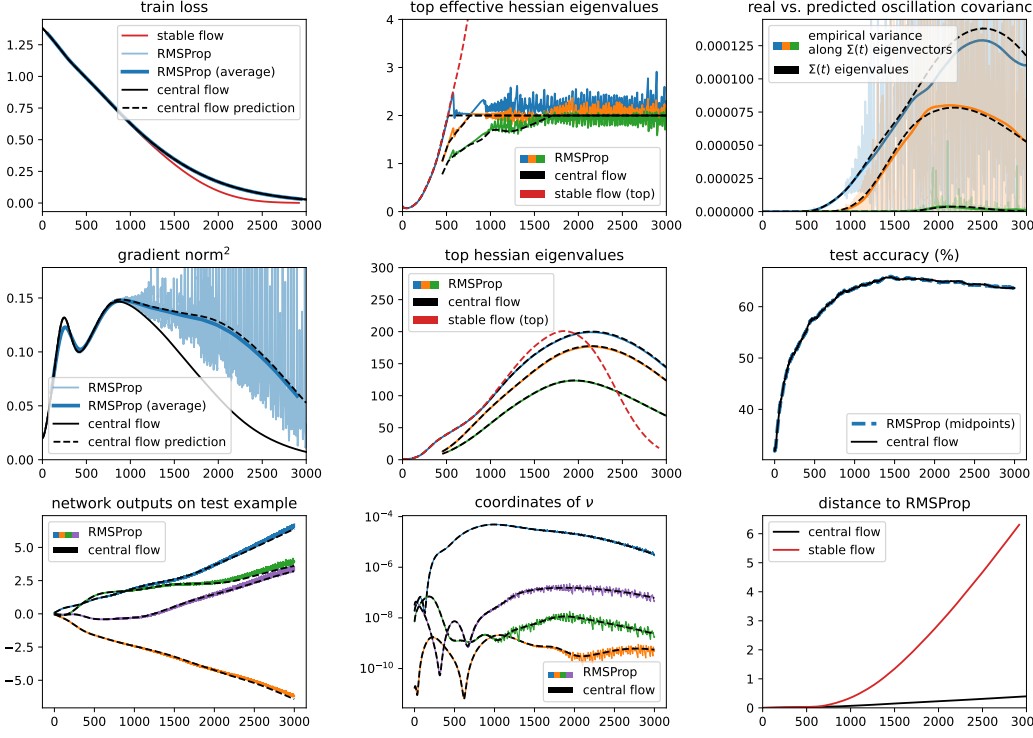

Figure 35.2: RMSProp central flow for a CNN with CE loss, $\eta =$1e-05, $\beta_2 =$0.95, $\epsilon =$1e-08, and bias correction.

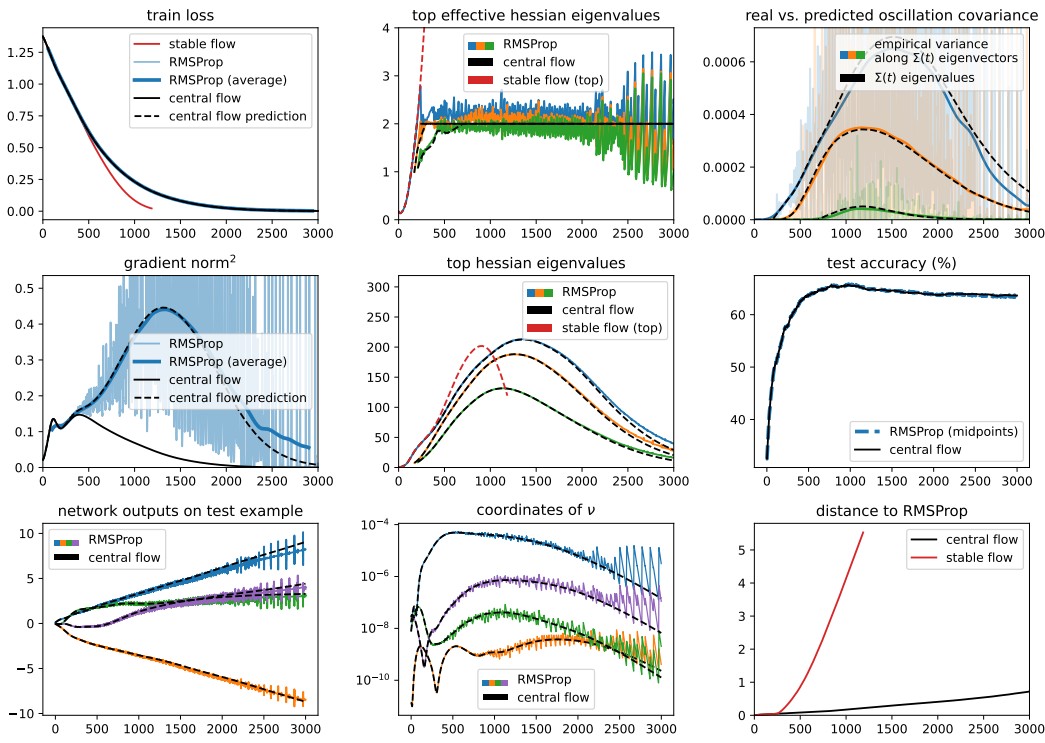

Figure 35.3: RMSProp central flow for a CNN with CE loss, $\eta =$2e-05, $\beta_2 =$0.95, $\epsilon =$1e-08, and bias correction.

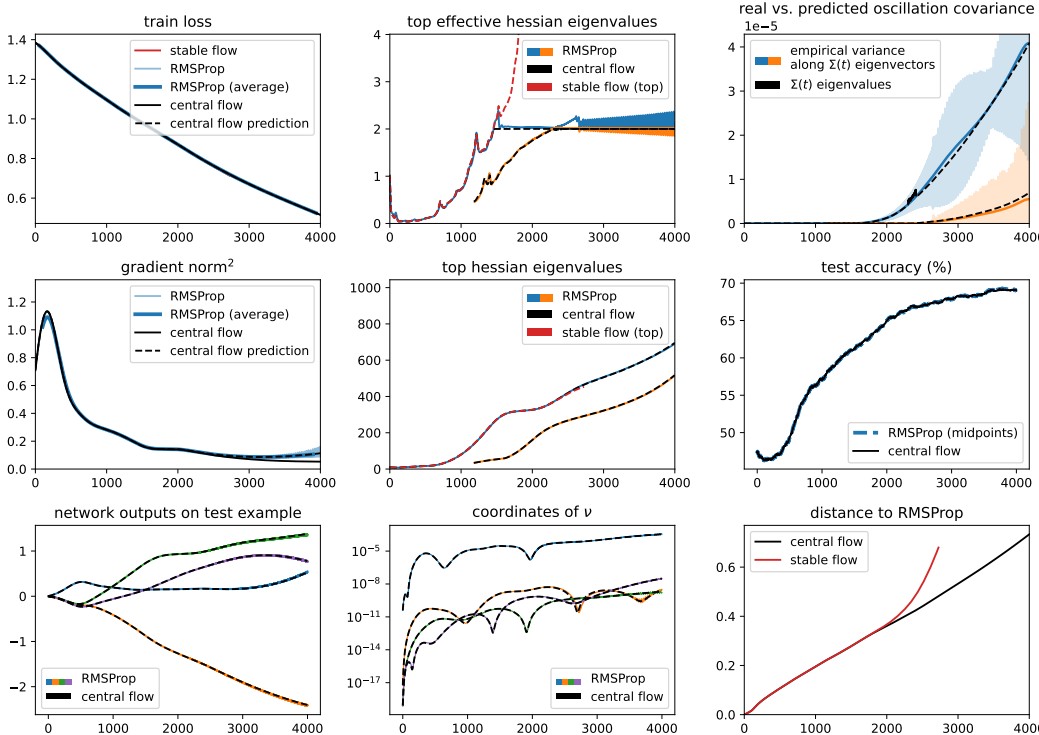

Figure 35.4: RMSProp central flow for a ResNet with CE loss, $\eta =$1e-05, $\beta_2 =$0.95, $\epsilon =$1e-08, and bias correction.

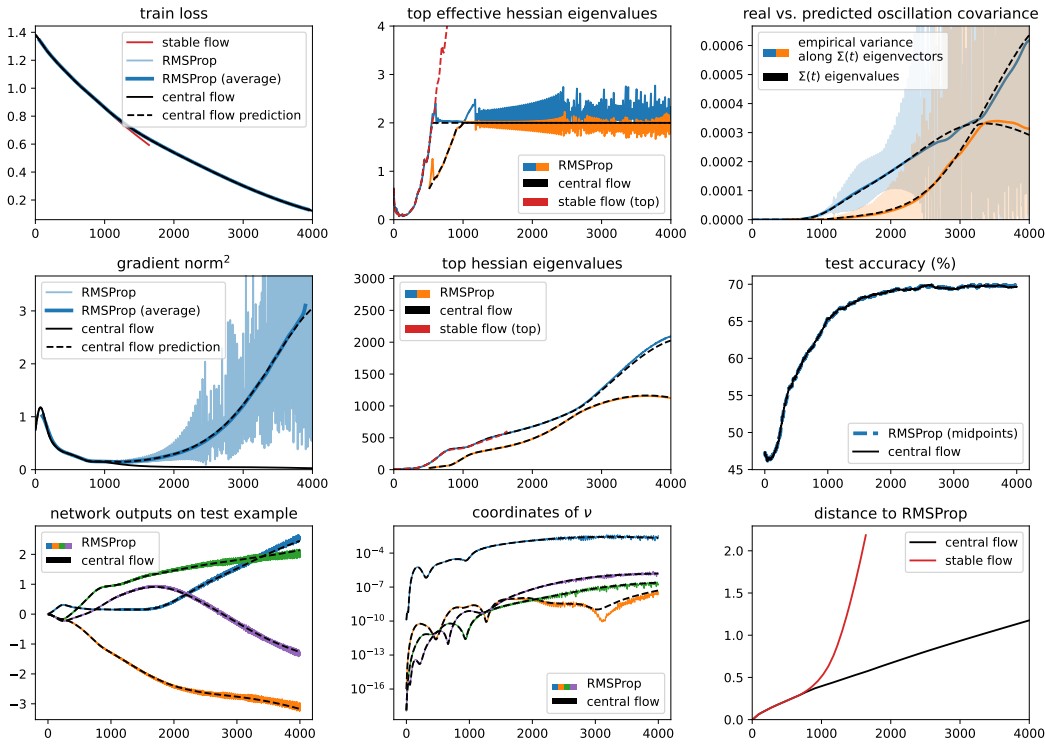

Figure 35.5: RMSProp central flow for a ResNet with CE loss, $\eta$ =2e-05, $\beta_2$ =0.95, $\epsilon$ =1e-08, and bias correction.

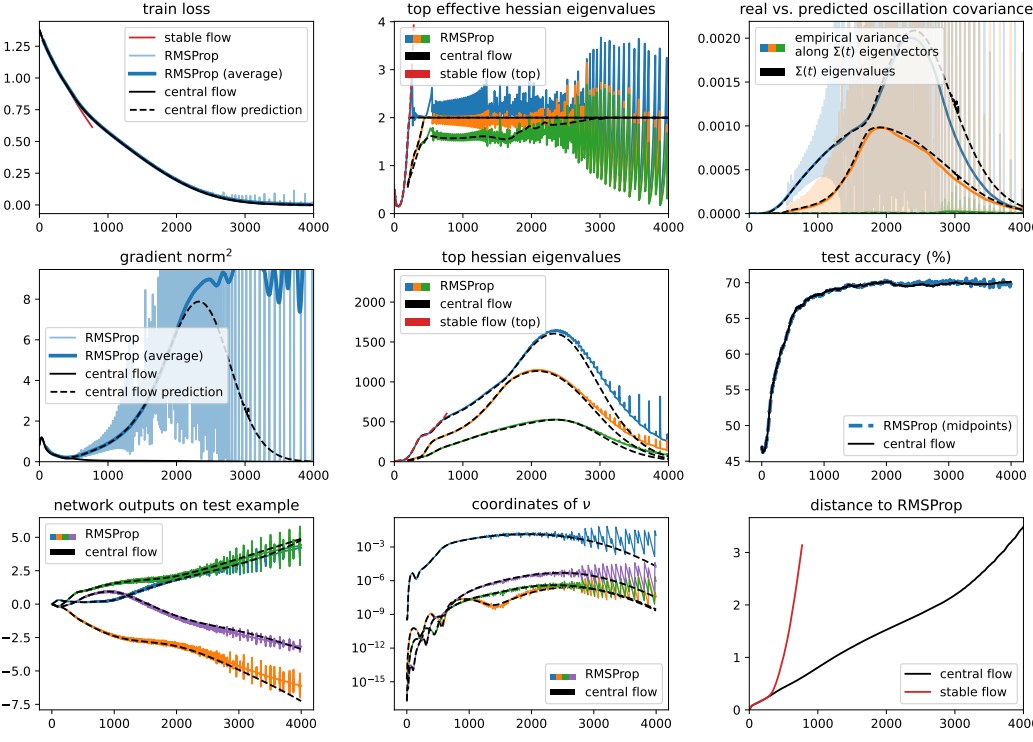

Figure 35.6: RMSProp central flow for a ResNet with CE loss, $\eta$ =4e-05, $\beta_2$ =0.95, $\epsilon$ =1e-08, and bias correction.

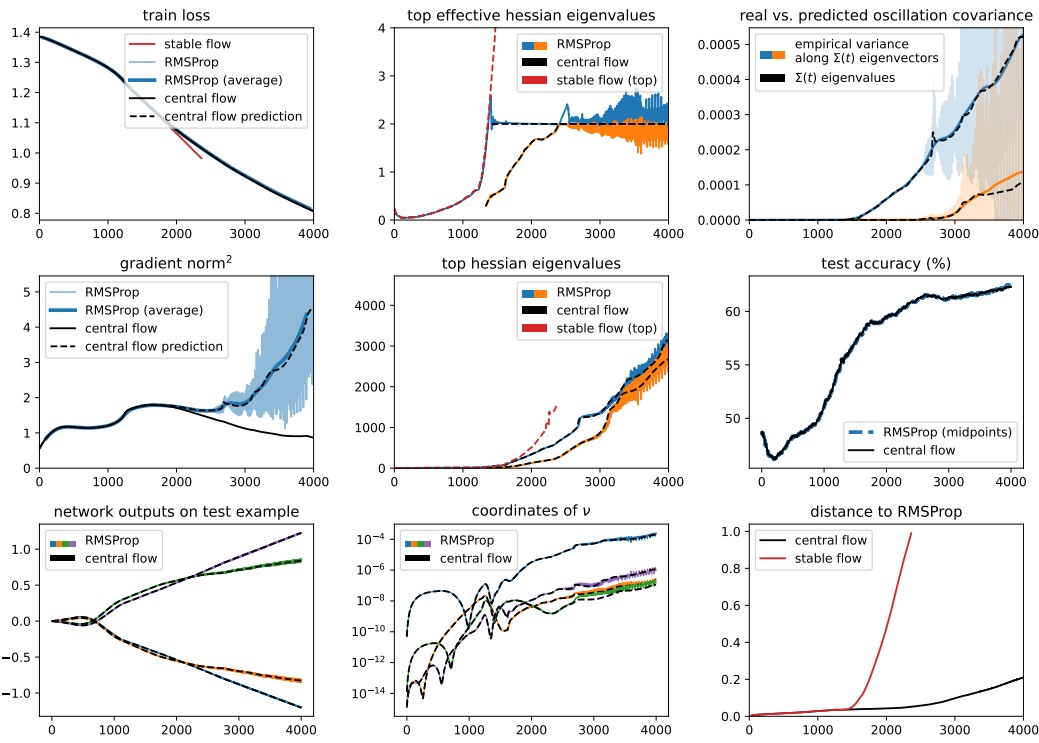

Figure 35.7: RMSProp central flow for a ViT with CE loss, $\eta =$5e-06, $\beta_2 =$0.95, $\epsilon =$1e-08, and bias correction.

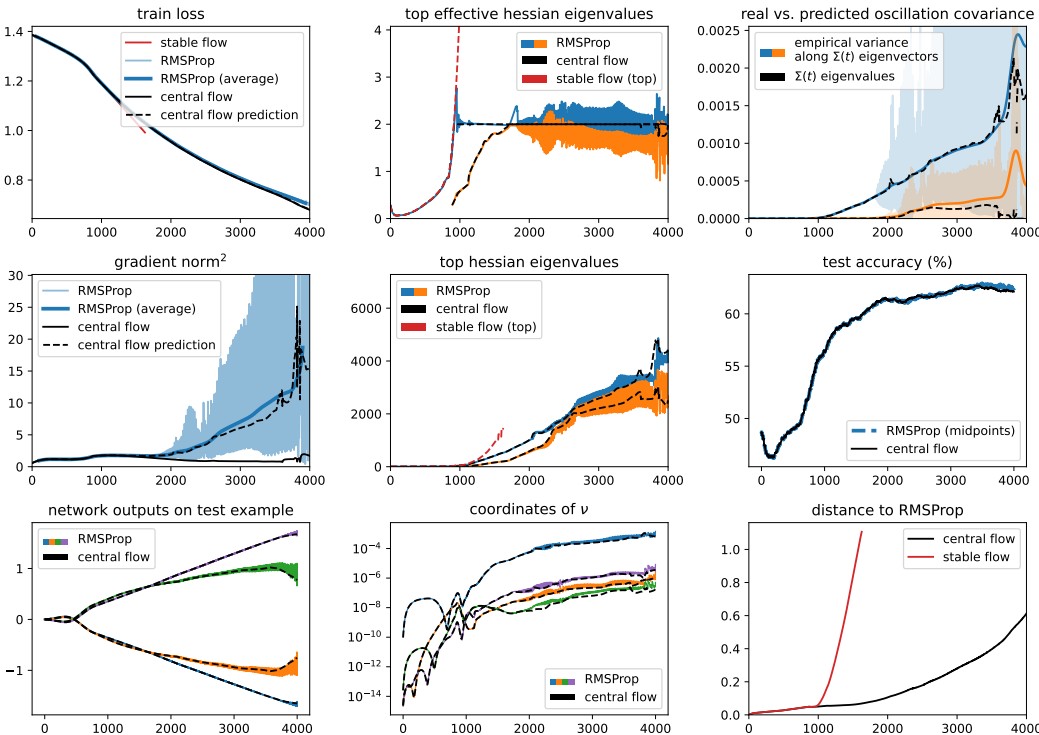

Figure 35.8: RMSProp central flow for a ViT with CE loss, $\eta =$7e-06, $\beta_2 =$0.95, $\epsilon =$1e-08, and bias correction.

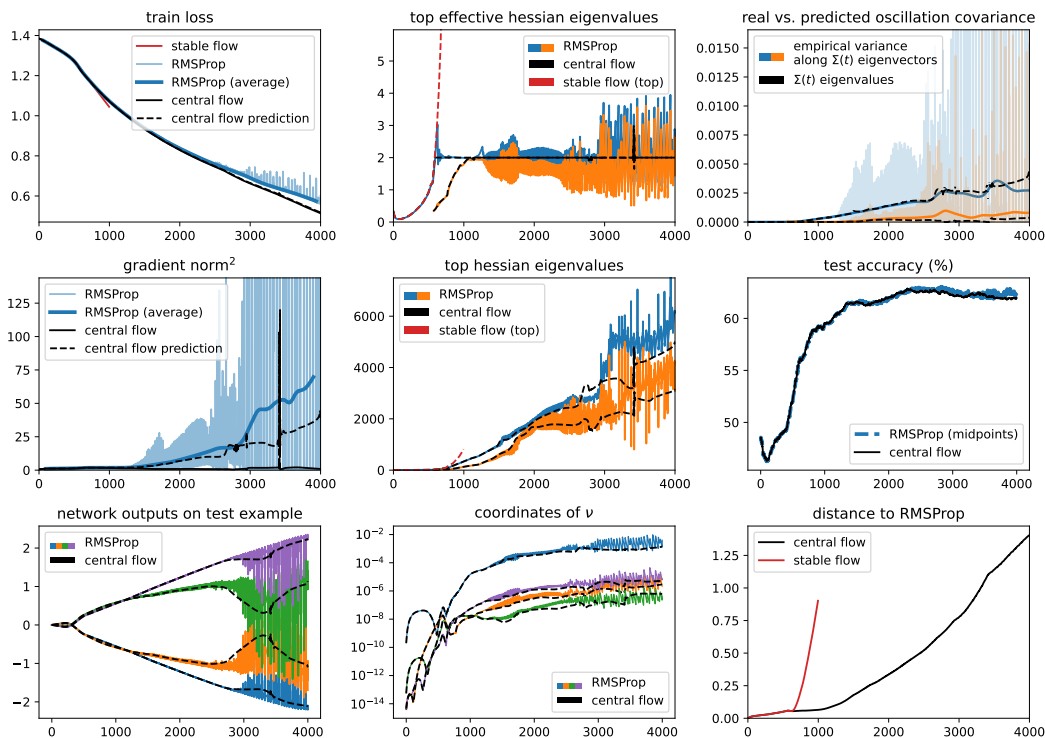

Figure 35.9: RMSProp central flow for a ViT with CE loss, $\eta$ =1e-05, $\beta_2$ =0.95, $\epsilon$ =1e-08, and bias correction.

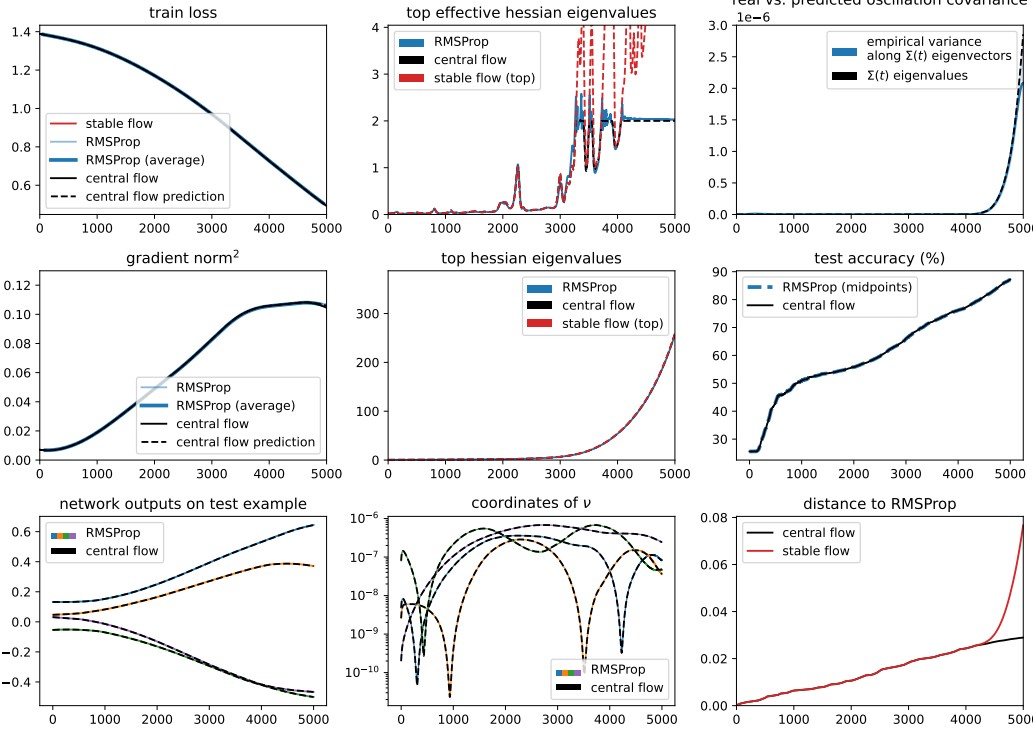

Figure 35.10: RMSProp central flow for a LSTM with CE loss, $\eta$ =1e-05, $\beta_2$ =0.95, $\epsilon$ =1e-08, and bias correction.

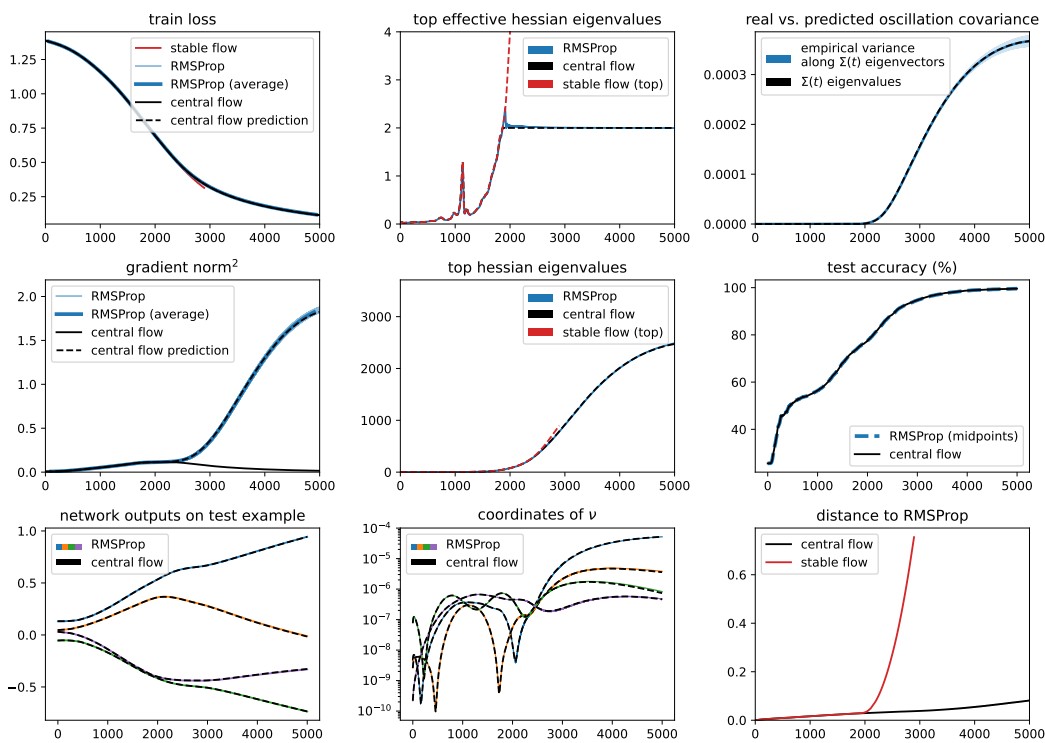

Figure 35.11: RMSProp central flow for a LSTM with CE loss, $\eta$ =2e-05, $\beta_2$ =0.95, $\epsilon$ =1e-08, and bias correction.

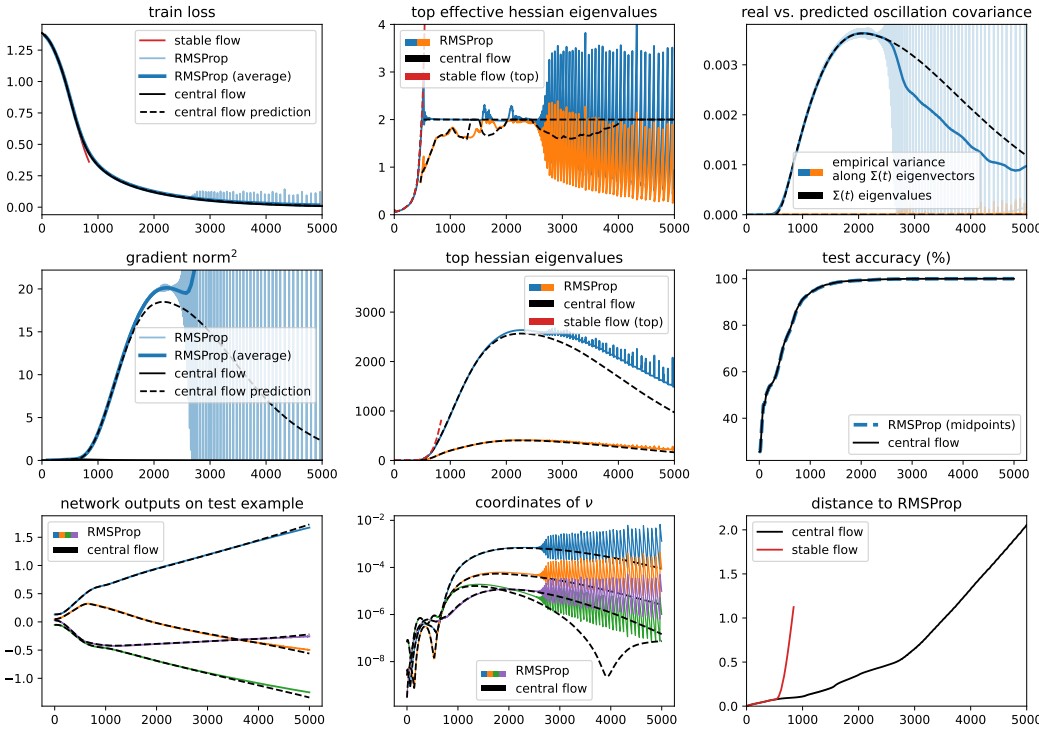

Figure 35.12: RMSProp central flow for a LSTM with CE loss, $\eta$ =6e-05, $\beta_2$ =0.95, $\epsilon$ =1e-08, and bias correction.

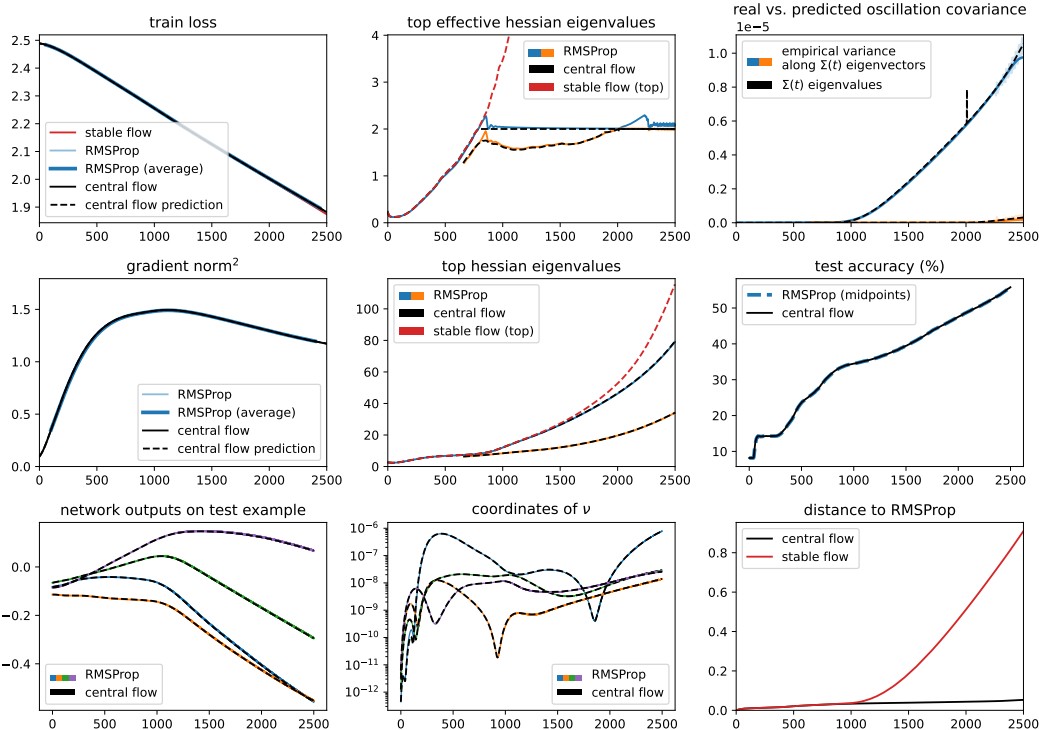

Figure 35.13: RMSProp central flow for a Transformer with CE loss, $\eta$ =1e-05, $\beta_2$ =0.95, $\epsilon$ =1e-08, and bias correction.

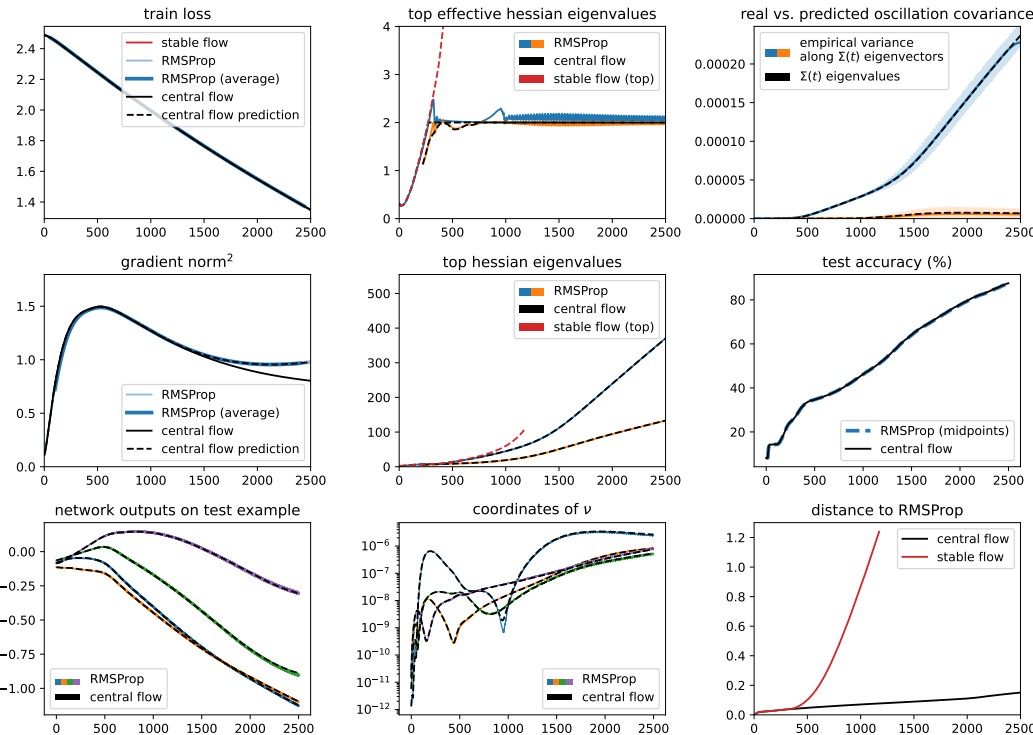

Figure 35.14: RMSProp central flow for a Transformer with CE loss, $\eta$ =2e-05, $\beta_2$ =0.95, $\epsilon$ =1e-08, and bias correction.

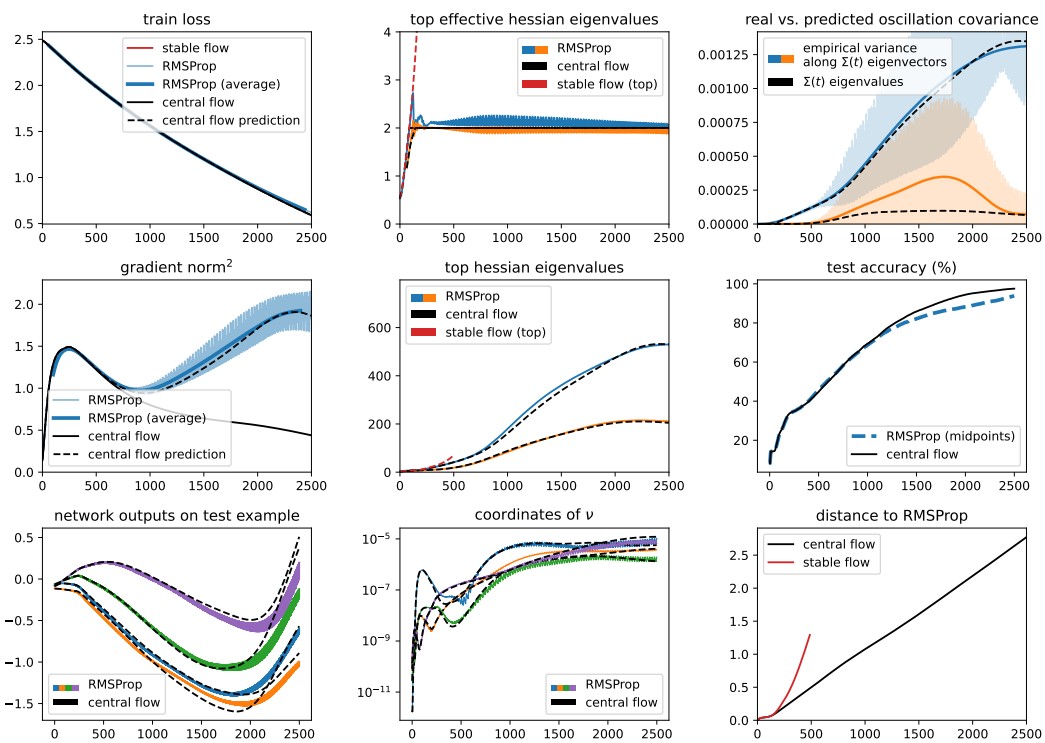

Figure 35.15: RMSProp central flow for a Transformer with CE loss, $\eta$ =4e-05, $\beta_2$ =0.95, $\epsilon$ =1e-08, and bias correction.

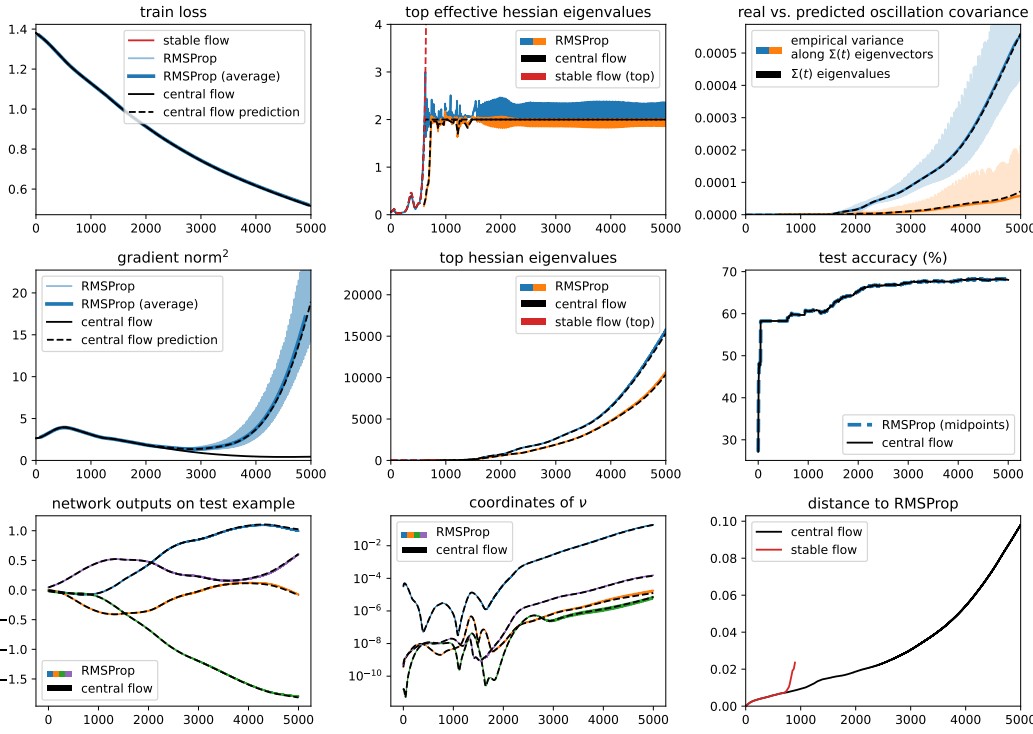

Figure 35.16: RMSProp central flow for a Mamba with CE loss, $\eta$ =7e-06, $\beta_2$ =0.95, $\epsilon$ =1e-08, and bias correction.

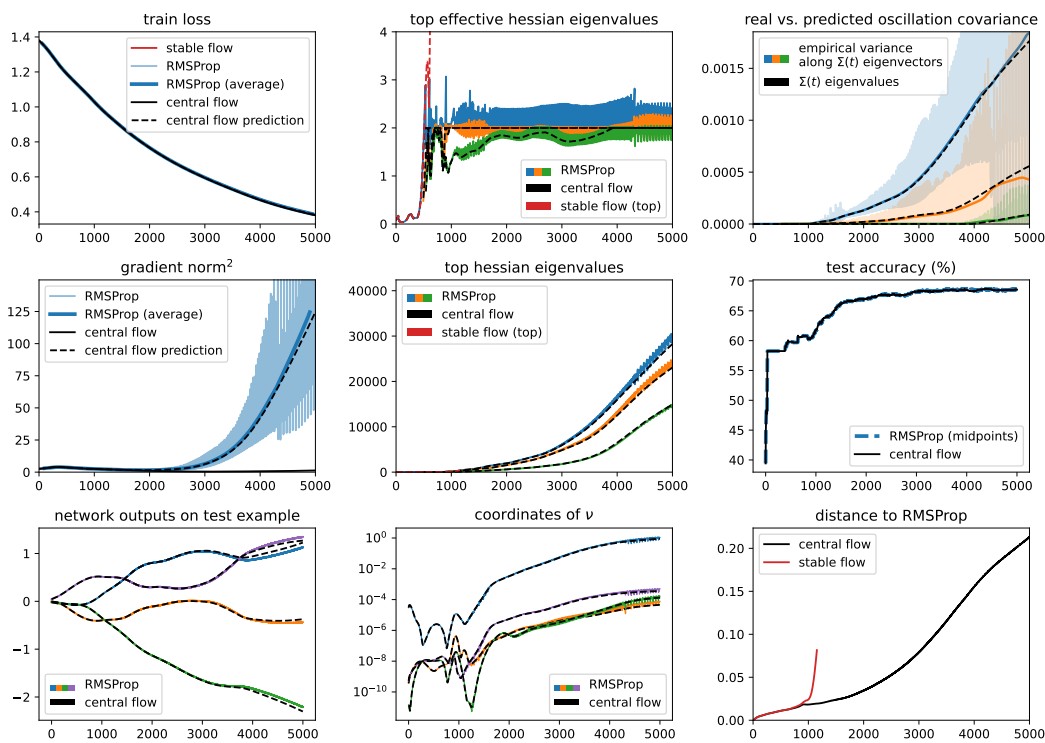

Figure 35.17: RMSProp central flow for a Mamba with CE loss, $\eta = $1e-05, $\beta_2 = $0.95, $\epsilon = $1e-08, and bias correction.

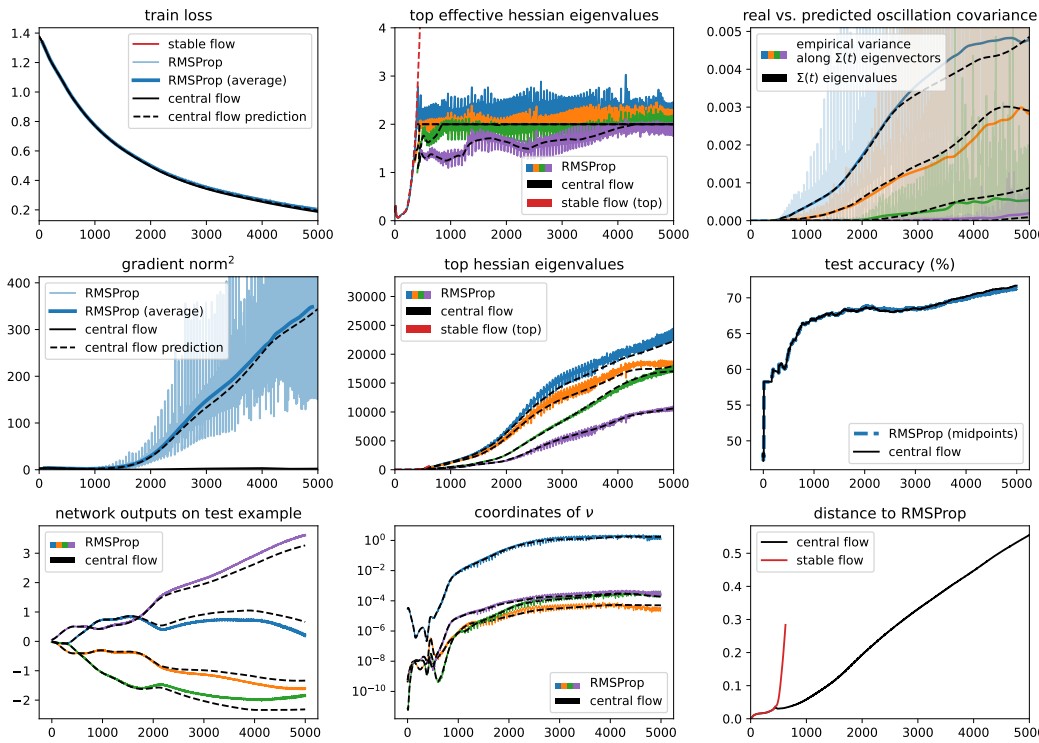

Figure 35.18: RMSProp central flow for a Mamba with CE loss, $\eta = $2e-05, $\beta_2 = $0.95, $\epsilon = $1e-08, and bias correction.

