# OpenReview forum: "Understanding Optimization in Deep Learning with Central Flows"
_ICLR.cc/2025/Conference — ICLR 2025 Poster_

### Official Review · Reviewer_VSTS · 2024-11-01

**Soundness:** 2
**Presentation:** 4
**Contribution:** 4
**Rating:** 8
**Confidence:** 3

**Summary:**

The paper uses mathematical reasoning to derive central flows of optimization functions. Central flows are functions which predict the averaged behavior of oscillating optimization functions when applied to neural networks.  These functions are empirically accurate, and reveal several interesting insights into the behavior of different optimization functions.

In gradient descent, central flows are used to describe the oscillatory effects of sharpness, and the sharpness-decreasing effects of oscillation, resulting in a steady-state (on average) of the sharpness of the points of the path taken by gradient descent.  These effects have been described previously, but not so generally.

In RMSProp, central flows are used to reason about what the paper calls effective sharpness.  They show that RMSProp, despite being a first-order algorithm, nonetheless adapt to the second-order curvature.  The path of the optimizer is driven to low-curvature spaces, where it can safely take large steps, improving its performance.

**Strengths:**

- The work is important.  Understanding the behaviors of optimization algorithms is very important, and this paper provides new insights through a novel approach.
- The paper is engaging.  It is well written and easy to read.
- Conclusions and key takeaways are clearly stated.
- The work is wide-ranging, and makes a number of important conclusions in a single paper.

**Weaknesses:**

The main weakness is that this is not really a conference paper.  Essential parts that justify the work (the mathematical derivations and experimental results that support the conclusions) are included in the 41-page appendix.  There is some point where the appendix is so essential, this cannot really be considered a 10-page paper.  The work is interesting, important, and well laid out, but the ten pages are not, on their own, an acceptable paper, as the appendices are so voluminous and essential that the reader would have insufficient reason to believe the claims of the authors without them. I cannot find anything in the reviewer guidelines to help me understand the expectations in this area, so I leave it to the AC for guidance and discussion.

If this is a problem for the conference, the most useful thing to promote to the core paper would be some of the experiments which demonstrate the fidelity of the predictions from the central flows to the true observed optimizations.  If it's not a problem, I believe this otherwise-excellent paper should be highlighted by the conference.

I have skimmed these appendices, but not closely checked the mathematics.

**Questions:**

Would the creation of central flows be easily created for other optimization algorithms, or do they depend upon specific characteristics of these two algorithms?  If they were second-order algorithms, would the derivation break down?

---

> ### Author Response · Authors · 2024-11-16
> **author response**
>
> We thank the reviewer for their insightful comments and questions. We address each point below and welcome further discussion.
>
> > The main weakness is that this is not really a conference paper. Essential parts that justify the work (the mathematical derivations and experimental results that support the conclusions) are included in the 41-page appendix.
>
> We certainly understand the reviewer’s point. Given the 10-page space constraint, we only had space to include the most important content in the main paper, while deferring most derivations and some supporting experiments to the appendix. We would be grateful for any suggestions regarding which content to highlight in the main paper, versus defer to the appendix.
>
> > The most useful thing to promote to the core paper would be some of the experiments which demonstrate the fidelity of the predictions from the central flows to the true observed optimizations
>
> Thank you for the suggestion!  Could you clarify whether you are referring to:
>
> - Plots which show a subset of the content in Figures 2,4,6 but replicated across more architectures and learning rates. For example, we could single out the real vs. predicted loss curves, or the distances in weight space between the flow and the discrete optimizer.
> - Plots with the supporting experiments from Section 5.1-5.2 (which are currently shown in figures 8-11 in the appendix)
> - Plots of the individual network outputs (i.e. the bottom row in all of the figures in the appendix), which vividly demonstrate how the central flow accurately approximates the discrete optimizer trajectory.
>
> We would also be grateful for any suggestions on which content to demote from the main paper in order to make way for these plots.  Perhaps the discussion of multiple unstable eigenvalues in section 3.2.2, or equation 10 on lines 366-370 could be moved to the appendix?
>
>
> > Would the creation of central flows be easily created for other optimization algorithms, or do they depend upon specific characteristics of these two algorithms? If they were second-order algorithms, would the derivation break down?
>
> Our analysis relies on the fact that for the optimizers studied in our paper, the effective sharpness equilibrates at a fixed value (i.e. $2/\eta$), which makes it possible to uniquely solve for the oscillation covariance. We believe that there are other optimizers which can be analyzed in the same fashion.  For example, Long et al. [1] empirically showed that SAM (Foret et al [2]) also obeys an analogous equilibrium rule, so we believe that our techniques could be straightforwardly used to derive a central flow for SAM.  We similarly suspect that our techniques could be used to understand other adaptive preconditioners beyond RMSProp, such as Shampoo [3].
>
> It is possible that for optimizers which do not admit a similar equilibrium rule, our analysis would not immediately carry over, as we would not be able to compute the oscillation covariance in the same fashion. That said, we still suspect that the underlying idea of modeling the optimizer as oscillating with some unknown covariance around its time-averaged trajectory may hold promise, even in these settings.
>
> Regarding second-order optimizers, there is no intrinsic difficulty that would render our analysis inapplicable for algorithms which leverage Hessian-vector products, but one would have to study each second-order optimizer on a case by case basis.
>
> ---
>
> [1] Phil Long and Peter Bartlett.  “Sharpness-Aware Minimization and the Edge of Stability.”  Journal of Machine Learning Research, 2024.
>
> [2] Pierre Foret et al.  “Sharpness-Aware Minimization for Efficiently Improving Generalization.”  ICLR, 2021.
>
> [3] Vineet Gupta et al. “Shampoo: Preconditioned Stochastic Tensor Optimization.”  ICML, 2018.

---

### Official Review · Reviewer_HuNC · 2024-11-02

**Soundness:** 4
**Presentation:** 4
**Contribution:** 4
**Rating:** 8
**Confidence:** 4

**Summary:**

The paper shows that an optimizer's implicit behavior comprising complex oscillatory dynamics (EOS) can be captured explicitly by ODE (termed as "central flow" by authors) modeling a time-averaged optimization trajectory. This a very interesting idea and probably the first explicit & numerically precise characterization that predicts a long-term optimization path well. Using central flow they analyze the local adaptivity of RMSProp and provide insight into an interesting phenomenon termed "acceleration via regularization" which attempts to reason the efficacy of adaptive optimizers in general.

**Strengths:**

1. This is a novel study to deeply understand an optimizer's implicit dynamics and attempt to explicitly characterize it. The authors exploit central flow as a tool to analyze adaptive optimizers (RMSProp) and consequently draw interesting observations regarding their efficacy. They provide a strong theoretically backed argument as to how the EOS regime leads to implicit step size selection and curvature reduction.
2. A new characterization of phenomena "acceleration via regularization" (observed using central flow) where they show how adaptive optimizers (RMSProp) navigate to a low generalization landscape where they can take larger steps. This claim is backed by a comprehensive set of experiments that showcase their claim precisely.
3. The development and intuition of central flow and how it tracks time-averaged optimization trajectory to account for precise implicit behavior is robust. While the derivation might have started from informal reasoning (as accounted in the paper) I find the subsequent logic intuitive and reasonable. Each claim is backed by theoretical analysis in the Appendix which also looks correct.
4. Detailed explanation of the experimental setup and good accounting of the limitations. They make it extremely clear what is the purpose of their work while also providing a comparison against other ODE methods like stable flow. Each claim is backed by a relevant ablation or experimental evaluation.
5. There are comprehensive notes for the reader to provide quick additional reference or context to a claim being made in the paper which is very useful.

**Weaknesses:**

Some comments
1. In the large-scale evaluation reported, I majorly see MSE loss being used. How does the ODE analysis deviate or negatively impacted by commonly used loss functions for classification tasks? While this might be part of future work I think addressing such limitations in the main body would improve the paper.
2. The analysis relies heavily on the largest eigenvalue estimation for ODE analysis. I think it might be helpful to account for how precise/reliable such estimations are when empirically implemented.
3. I might have missed this but I do not see any ablations on the behavior of $\sigma^2$, that models $\mathbb E[x_t^2]$ (equation 2). Is it because it has a fixed value due to the 3 desiderata described in line 249? In either case, I think it might be good to show its evolution across training to support its expected behavior.

Minor weakness
1. Figure 1: x-axis labels missing. While I can infer what is being plotted I think the figures should be precise on labels for faster understanding.
2. Figure 3, the legend is difficult to read might be possible to increase the font a bit.
3. Figure 4: The gray curves are mostly hard to see. I think it might be helpful to use a color palette that makes the curves more discernable.

**Questions:**

1. The ODE derivation is quite general. However, I wonder if it's possible to analyze non-convex objective functions with central flow and if they still track real optimization trajectories.
2. Can the authors comment on why they chose to analyze RMSProp only as opposed to using ADAM? What sort of issues did they face of trying with ADAM, if any?
3.  Central flow clearly shows how adaptive optimizers like RMSProp additionally account for the curvature regularization induced by oscillations. While it can be seen that they bias trajectory towards lower curvature, I wonder if central flow helps understand what happens in the presence of pathological curvatures (specifically low-curvature critical points such as saddle points). Does central flow demonstrate whether these points are circumvented when using adaptive optimizers?

---

> ### Author Response · Authors · 2024-11-16
> **author response**
>
> We thank the reviewer for their insightful comments and questions. We address each point below and welcome further discussion.
>
> > In the large-scale evaluation reported, I majorly see MSE loss being used. How does the ODE analysis deviate or negatively impacted by commonly used loss functions for classification tasks?
>
> First, we’d like to point out that while the figures in the main text use MSE loss, half of the figures in Appendix D used the more common cross-entropy loss, and we still observe good alignment between the central flow and the true optimization trajectory. For future revisions, we will include cross entropy experiments in the main text as well.
>
> As we briefly mentioned in line 1740, the central flow approximation is usually accurate over longer timescales when using MSE loss than when using cross-entropy loss.  The issue seems to be that for cross-entropy loss, the equilibrium point of the sharpness often differs slightly from 2/eta (think e.g. 2.05 / eta), whereas the central flow models the sharpness as being locked exactly at 2/eta.  This discrepancy causes error to gradually accumulate over long timescales.  Damian et al ‘22, in their Appendix F, used higher-order terms to accurately predict the corrected sharpness equilibrium point, and were therefore able to accurately model the trajectory even in these situations.   We think that a similar correction might be possible for our central flows, but we leave this to future work.
>
> In the revision, we will make sure to mention this point explicitly in the main paper.
>
> > The analysis relies heavily on the largest eigenvalue estimation for ODE analysis. I think it might be helpful to account for how precise/reliable such estimations are when empirically implemented.
>
> To compute the top Hessian eigenvalues, we used Scipy’s built-in implementation of the Lanczos iteration (`scipy.sparse.linalg.eigsh`); this library function only needs to access the Hessian using Hessian-vector products, which PyTorch can do.  Since this is a mature implementation, we have confidence that our eigenvalue computations were accurate. Indeed, whenever we have manually checked whether $Hv = \lambda v$, we have found this to be the case with high accuracy.
>
> > I might have missed this but I do not see any ablations on the behavior of $\sigma^2$, that models $E[x^2]$
>
> This is a great question.  We actually did verify our prediction for $\sigma^2(t)$, but it was not marked clearly; we apologize for this. In Figures 2, 4, and 6, the plot titled “Norm^2 of Oscillations” demonstrates that $tr(\Sigma(t))$, plotted in dashed black, closely matches the time average of the squared norm of oscillations $E[||\delta_t||^2]$, plotted in thick blue.  When there is exactly one unstable eigenvalue, the former quantity reduces to $\sigma^2(t)$, and the latter quantity reduces to $E[x_t^2]$.  Thus, this is the verification that you are requesting.  We will label this plot more clearly in the revision.
>
> > The ODE derivation is quite general. However, I wonder if it's possible to analyze non-convex objective functions with central flow and if they still track real optimization trajectories.
>
> We’re not sure that we completely understand the question, as our paper did empirically validate the central flow on non-convex neural network objective functions. Are you wondering whether it is possible to prove that the central flow tracks the real trajectory on a simple non-convex objective function?  Or whether the central flow still holds near a saddle point?  We are happy to discuss further.
>
> > Can the authors comment on why they chose to analyze RMSProp only as opposed to using ADAM? What sort of issues did they face of trying with ADAM, if any?
>
> The main reason why we did not cover Adam is that we still have not extended the central flows framework to momentum, and Adam is essentially RMSProp with momentum.  A key challenge is that momentum exhibits not only period-2 oscillations of the kind studied here, but also longer-range oscillations that arise from imaginary eigenvalues in the update’s Jacobian.  We do not yet know the best way to handle those longer-range oscillations. We are hopeful that a central flow can eventually be derived for both gradient descent with momentum and Adam.
>
> > I wonder if central flow helps understand what happens in the presence of pathological curvatures (specifically low-curvature critical points such as saddle points). Does central flow demonstrate whether these points are circumvented when using adaptive optimizers?
>
> We agree that one benefit of adaptive optimizers is that they can quickly escape from bad initializations near saddle points. However, this benefit can be observed even for the RMSProp stable flow, and may be somewhat orthogonal to the effects of the oscillations studied in our paper. We are not aware of any additional connections between the central flow and circumventing saddles, but this question certainly merits further investigation.

---

> > ### Author Response · Authors · 2024-11-16
> > **author response (continued)**
> >
> > > Minor weakness
> >
> > We thank the reviewer for catching these issues. We will fix them in a future revision.

---

> > > ### Comment · Reviewer_HuNC · 2024-11-25
> > >
> > > I thank the authors for their detailed response.
> > >
> > > > We’re not sure that we completely understand the question, as our paper did empirically validate the central flow on non-convex neural network objective functions. Or whether the central flow still holds near a saddle point?
> > >
> > > Yes, I am wondering whether central flow still holds near a saddle point. It would be interesting to visualize empirically the dynamics near such points. (Still, this is a minor point so I am satisfied with the authors' last responses).
> > >
> > > Thank you for answering my questions, they were quite helpful to my understanding. Your additional clarification of the points raised resolved all of my concerns (I commend the authors on validating their eigenvalue computations and other empirical issues mentioned). Thank you for your engagement. I had a great time reading and engaging in the review process for this work. I wish the authors all the best for future endeavors.

---

> > > > ### Author Response · Authors · 2024-11-28
> > > > **author response**
> > > >
> > > > > Yes, I am wondering whether central flow still holds near a saddle point.
> > > >
> > > > We do expect all the central flows to hold near saddle points – we are just saying that the central flow analysis likely isn’t necessary for explaining RMSProp’s ability to escape saddle points quickly.
> > > >
> > > > > Thank you for your engagement. I had a great time reading and engaging in the review process for this work.
> > > >
> > > > We would also like to thank you for your engagement with the review process!

---

### Official Review · Reviewer_X29f · 2024-11-03

**Soundness:** 3
**Presentation:** 2
**Contribution:** 2
**Rating:** 5
**Confidence:** 4

**Summary:**

The authors developed a methodology for creating optimizers in the context of deep learning. They empirically demonstrated that central flows govern long-term optimization trajectories, providing insight into the behavior of optimizers.

**Strengths:**

The work presents novel contributions. Additionally, it represents an advancement in the study of locally analyzed optimizers and suggests progress for other optimizers.

**Weaknesses:**

1. In the abstract, the introduction and related work the authors repeats the same phrase 'Optimization in deep learning remains poorly understood' (lines 11, 26 and 68).  Maybe the authors can phrases like: 'How optimization works in deep learning continues to be largely unclear' or 'The optimization process in deep learning is still not well understood.'
2. The contributions of the work are unclear in the introduction, which is somewhat difficult to follow in that sense. Maybe the authors can add a explicit numbered list with the contributions.
3. In general, it would have been valuable to provide a bit more discussion or formal mathematical argumentation on how some equations were derived (e.g., equations 1 or 7).
4. The proofs are generally well-written and clear, but since they are in the appendix, the flow of the paper's main narrative can sometimes be interrupted. Would be helpful to add some discussion of the results.

**Questions:**

What are the geometric implications of the curvature penalty?

---

> ### Author Response · Authors · 2024-11-16
> **author response**
>
> We thank the reviewer for their insightful comments and questions. We address each point below and welcome further discussion.
>
> > In the abstract, the introduction and related work the authors repeats the same phrase 'Optimization in deep learning remains poorly understood' (lines 11, 26 and 68). Maybe the authors can phrases like: 'How optimization works in deep learning continues to be largely unclear' or 'The optimization process in deep learning is still not well understood.'
>
> We thank the reviewer for pointing out that we repeat a phrase multiple times.  In future revisions, we will reword the phrase so as not to repeat it.
>
> > The contributions of the work are unclear in the introduction, which is somewhat difficult to follow in that sense. Maybe the authors can add a explicit numbered list with the contributions.
>
> We thank the reviewer for their helpful suggestion to include a numbered list of contributions.  The three main contributions of our paper are:
>    1. We develop a simple methodology for analyzing the time-averaged optimization trajectories of various optimizers. For each optimizer, the result of this analysis is a _central flow_: a differential equation which directly models this trajectory.
>    2. We empirically demonstrate that these central flows can predict long-term optimization trajectories of neural networks with a high degree of numerical accuracy across a variety of neural network settings. This represents an unprecedentedly high level of agreement between theory and experiment for an analysis of deep learning optimization.
>    3. By interpreting the central flows, we obtain new insights into each optimizer’s implicit behaviors. For example, our analysis reveals, for the first time, the precise sense in which RMSProp adapts its effective step sizes to the local loss landscape, and highlights a hidden “acceleration via regularization” mechanism that is key to RMSProp's success.
>
> > In general, it would have been valuable to provide a bit more discussion or formal mathematical argumentation on how some equations were derived (e.g., equations 1 or 7).
>
> We thank the reviewer for their feedback.  The proof for Equation 1 is given in Lemma 1 on page 22.  However, we apologize for not including a derivation for Equation 7, and will correct this in the revision. Since we model $w = \overline{w} + \delta$, where $\overline{w}$ represents the time-averaged iterate and $\delta$ represents the oscillations, a quadratic Taylor expansion of $L$ around $\overline{w}$ yields:
>
> $$L(w) = L(\overline{w} + \delta)
>             \approx L(\overline{w}) + \delta \cdot \nabla L(\overline{w}) + \tfrac{1}{2} \delta^T H(\overline{w}) \delta$$
>
> Taking the time average of both sides yields:
> $$E[L(w)] \approx L(\overline{w}) + E[\delta \cdot \nabla L(\overline{w})] + \tfrac{1}{2} E[\delta^T H(\overline{w}) \delta]$$
>
> The first-order term $E[\delta \cdot \nabla L(\overline{w})]$ is zero because $E[\delta] = 0$, i.e. the oscillations are mean zero. The second order term can be rewritten as $\tfrac{1}{2} \langle H(\overline{w}), \Sigma \rangle$ where $\Sigma = E[\delta \delta^T]$.  Thus we have:
>
> $$E[L(w)] \approx L(\overline{w}) + \tfrac{1}{2} \langle H(\overline{w}), \Sigma \rangle$$
>
> Finally, because $\Sigma$ is supported on the eigenvectors of $H(\overline{w})$ with eigenvalue $S(\overline{w})$, this simplifies to:
>
> $$ E[L(w)] \approx L(\overline{w}) + \tfrac{1}{2} S(\overline{w}) \textrm{tr}(\Sigma) $$
>
> > The proofs are generally well-written and clear, but since they are in the appendix, the flow of the paper's main narrative can sometimes be interrupted. Would be helpful to add some discussion of the results.
>
> The appendix mostly focuses on deriving the full central flows for each optimizer, including for multiple unstable directions. As these derivations are repetitive, we chose to only highlight the central flow for a single unstable direction in the main text, and defer the remaining derivations to the appendix. We used the rest of the space in the main text to shed light on these results, especially in the “interpretation” sections (3.3,4.3,5.2). Are there any specific results in the appendix that you think warrant additional discussion in the main text?
>
> > What are the geometric implications of the curvature penalty?
>
> We are not aware of any geometric implications of the curvature penalty, aside from steering the dynamics towards lower-curvature regions of the loss landscape. Interestingly, prior works have demonstrated that curvature penalties promote better generalization [Jastrzebski et al ‘21], though the underlying mechanism is not well-understood.
>
> Stanisław Jastrzębski et al.  "Catastrophic Fisher Explosion: Early Phase Fisher Matrix Impacts Generalization."  ICML ‘21.

---

> > ### Comment · Reviewer_X29f · 2024-11-25
> > **Response**
> >
> > Thank you for the reply! My concerns are addressed.

---

> > > ### Author Response · Authors · 2024-11-28
> > > **author response**
> > >
> > > That’s great to hear!   We greatly appreciate your engagement with the review process.  Since your concerns have been addressed, would you be open to reassessing the score?

---

### Official Review · Reviewer_mqGu · 2024-11-03

**Soundness:** 3
**Presentation:** 4
**Contribution:** 3
**Rating:** 6
**Confidence:** 3

**Summary:**

This work develops a central flow approach with which to study the training dynamics of neural networks. The authors use this approach to investigate the edge of stability and optimization via RMSProp. The authors argue that their analysis can shed new light on how RMSProp adapts to a local loss landscapes.

**Strengths:**

1. This work was well written and well structured, making it easy to follow along.

2. The authors analyzed a wide range of neural networks, including modern architectures such as ViT.

3. The authors developed their approach in settings where the interpretation was straightforward to make their point more clear.

4. The figures were well done and easy to read.

5. The authors convincingly show that their central flow approach accurately captures the general behavior of neural network model training.

6. The identification of a possible weakness of the pre-conditioning of RMSProp is interesting and suggestive of a way in which to use the authors' work to improve DNN optimization.

**Weaknesses:**

1. My main reservation of this work as it stands (which I hope the authors can easily clarify for me), is the claim that the central flow analysis of RMSProp-Norm demonstrates how the optimizer regularizes against sharpness and moves towards regions of parameter space with lower sharpness. While I understand the argument mathematically, and the analysis of the central flow (Fig. 5) illustrates this point, it does not seem consistent with the results of training ViT (Fig. 4). In particular, in Fig. 4 the Hessian eigenvalues increase with training and $2/S$ decreases (which more closely mirrors the ablation in Fig. 5). This seems to suggest that training is moving towards a region with greater sharpness. How do the authors reconcile this? Are the authors claiming that the network would be headed towards an even more sharp region if it was not for RMSProp-Norm, and that the optimizer is reducing sharpness, albeit not as completely as in the case of just analyzing the central flow?

2. I think this work would also benefit from showing the sharpness of the full RMSProp. Providing numerical evidence that use of RMSProp steers the network away from sharp areas of parameter would make this point more salient.

3. While the goal of understanding edge of stability was motivated, I think this work could also benefit from more discussion on how the central flow approach could be used for other training dynamics phenomena. In particular, as I understood this work, one of the keys to utilizing the method successfully was the ability to find $\sigma$. This was only possible since the training limited to some fixed value. How general do the authors expect this kind of solving for $\sigma$ to be?

4. The authors show that the distance between the central flow weights and the GD weights are small (Fig. 3). Is this because the distance is an average across many weights? Or is the difference between oscillations and no oscillations a matter of small differences in the weights?

5. One last point that was unclear to me is why $x_t u_t$ are not present in Eq. 2, since they are time varying and I would expect $d w(t) / dt$ to depend on their value.

MINOR POINTS:
1. PSD should be defined.

2. The authors should make a note somewhere that the footnotes are at the end of the main text. It was confusing not to see them at the bottom.

3. Figure 1 should be referenced in the text.

4. The figure in Sec. 4.3.1 is not referenced anywhere nor does it have a label.

**Questions:**

1. Why is there a mismatch between Fig. 4 and Fig. 5, in terms of sharpness and $2/S$?

2. How does the sharpness evolve (for a numerical example) when optimizing using full RMSProp?

3. Do the authors think this approach can be used when there may not be an obvious choice of $\sigma$?

4. Why are $x_t u_t$ not present in Eq. 2?

---

> ### Author Response · Authors · 2024-11-16
> **author response**
>
> We thank the reviewer for their insightful comments and questions. We address each point below and welcome further discussion.
>
> > Are the authors claiming that the network would be headed towards an even more sharp region if it was not for RMSProp-Norm?
>
> Yes, this is our precise claim.  When we say that the oscillations implicitly regularize sharpness, we mean that they _downwardly bias_ the rate of change in the sharpness.  So, if the unregularized flow $\tfrac{dw}{dt} = - \tfrac{2}{S(w)} \nabla L(w)$ would increase the sharpness, then the central flow, which additionally incorporates an implicit sharpness penalty,
>  1. might still increase the sharpness, but at a slower rate; or
>  2. might decrease the sharpness.
>
> In our view, there isn’t a meaningful difference between the two cases – both are implicit regularization.  You’ve pointed out that the two central flow trajectories shown in Figure 5 both depict the latter situation, where the implicit regularization is strong enough to push the sharpness down.  This was coincidental – we did not mean to suggest that the implicit regularization will always be strong enough to push the sharpness down.  We apologize for causing this confusion. In future revisions, we will be more clear about this.
>
> We emphasize that even in the case where the implicit regularization is too weak to overpower the growth in sharpness, it is still beneficial for the optimizer, in the sense that it causes the effective step sizes to decay less steeply than they would have otherwise.
>
> We are happy to discuss this topic further.  Please let us know if you have any remaining concerns.
>
> > I think this work would also benefit from showing the sharpness of the full RMSProp. Providing numerical evidence that use of RMSProp steers the network away from sharp areas of parameter would make this point more salient.
>
> Thank you for this suggestion.  In [this figure](https://imgur.com/a/VODSaxr), we plot both the effective sharpness and the sharpness, as a network is trained using RMSProp (blue lines).   We also run the central flow (black lines), and the stable flow (red lines).  Observe that after RMSProp hits EOS around step 300, the sharpness along the RMSProp & central flow trajectories is smaller than the sharpness along the stable flow.  This is a consequence of the curvature regularization that is implicit in RMSProp, and is rendered explicit by the central flow.
>
>
>
>
>
>
> That said, the sharpness is not an intrinsically significant quantity for full RMSProp.  Whereas RMSProp-Norm implicitly penalizes sharpness and then sets its effective step size to be 2 / sharpness, full RMSProp implicitly penalizes _preconditioned sharpness_ (using the current preconditioner), and then sets its effective step sizes by solving the convex program in equation 12.  It can be shown that RMSProp’s implicit curvature regularizer tends to make the _harmonic mean_ of its effective step sizes larger, which empirically tends to speed up training.
>
> We are happy to discuss this topic further.  Please let us know if you have any remaining concerns.
>
> > How general do the authors expect this kind of solving for σ to be?
>
> This is an excellent question. As you note, for all optimizers we studied in this paper, we were able to leverage that the effective sharpness equilibrates at 2/\eta to argue that there is a unique value that σ^2 can take.  We believe that there are other optimizers which can be analyzed in the same fashion.  For example, Long et al. [1] empirically showed that SAM (Foret et al [2]) also obeys an analogous equilibrium rule, so we believe that our techniques could be straightforwardly used to derive a central flow for SAM.  We similarly suspect that our techniques could be used to understand other adaptive preconditioners beyond RMSProp, such as Shampoo [3].
>
> However, it is possible that for optimizers which do not admit a similar equilibrium rule, our analysis would not immediately carry over, as we would not know how to set $\sigma^2$. That said, we suspect that the underlying idea of modeling the optimizer as oscillating with some unknown covariance around its time-averaged trajectory may still hold promise, even in these settings.
>
> [1] Phil Long and Peter Bartlett.  “Sharpness-Aware Minimization and the Edge of Stability.”  Journal of Machine Learning Research, 2024.
>
> [2] Pierre Foret et al.  “Sharpness-Aware Minimization for Efficiently Improving Generalization.”  ICLR, 2021.
>
> [3] Vineet Gupta et al. “Shampoo: Preconditioned Stochastic Tensor Optimization.”  ICML, 2018.
>
> > Minor points
>
> We thank the reviewer for catching these issues.  We will fix them in a future revision.

---

> ### Author Response · Authors · 2024-11-16
> **author response (continued)**
>
> > The authors show that the distance between the central flow weights and the GD weights are small (Fig. 3). Is this because the distance is an average across many weights? Or is the difference between oscillations and no oscillations a matter of small differences in the weights?
>
> In these figures, we are actually reporting the distance between the flows and the midpoint between successive iterates, i.e. the average across two consecutive weights.  We apologize that our draft was not clear about this, and we will clarify this point in future revisions. Here is a [revised figure](https://imgur.com/a/UpKyxPt) which plots both the distance to the midpoint (dashed black) and the distance to the “raw” iterate (dashed purple)
>
> As you can see, even the distances between the flow and the “raw” GD iterates (which includes the oscillations), remain small relative to the distance between GD and gradient flow. This is because while the oscillations remain bounded, the distance between GD and gradient flow grows over time and soon overtakes the size of the oscillations.
>
>
> > One last point that was unclear to me is why  xt ut are not present in Eq. 2, since they are time varying and I would expect dw(t)/dt to depend on their value.
>
> Intuitively, $x_t$ is flipping in sign at every step; indeed, this term is responsible for the period 2 oscillations in the GD trajectory.  As a result, $E[x_t]=0$, so this term does not affect the time-averaged trajectory, which is modeled by $w(t)$. The intuition is that the central flow averages out the oscillations, while retaining their lasting effect on the trajectory. The linear $x_t u_t$ term corresponds to “the oscillations”, whereas the quadratic term corresponds to “their lasting effect.”
>
> We are very happy to discuss any of these topics further.  Please let us know if you have more questions.

---

> > ### Comment · Reviewer_mqGu · 2024-11-16
> >
> > I thank the authors for their detailed response.
> >
> > The results in Fig. 4, and how they fit in with the larger picture, is more clear to me. Is it correct to say that, if there was not any sharpness regularization, then the optimization would follow the stable flow? So as long as the curves are below the stable flow, there is evidence of regularization (regardless of whether the sharpness decreases)?
> >
> > I now better understand the distance between GD and the central flow. Thank you for clarifying this.
> >
> > Thank you for plotting the results of the full RMSProp. It is indeed interesting that sharpness is not an intrinsically significant quantity. Showing this plot - to my mind - makes the authors' point (that the preconditioning could be done better) more salient.
> >
> > Lastly, thank you for the discussion on how estimating $\sigma^2$ impacts the generality of the method. It is nice to see that there is another case where there is an equilibrium that is reached. This broadens my view of the potential impact of this method. Including this discussion in the main text would be great.
> >
> > Having these questions clarified and seeing that 3 other reviewers viewed this work highly, I will increase my score.

---

> > > ### Author Response · Authors · 2024-11-18
> > > **author response**
> > >
> > > > Is it correct to say that, if there was not any sharpness regularization, then the optimization would follow the stable flow? So as long as the curves are below the stable flow, there is evidence of regularization (regardless of whether the sharpness decreases)?
> > >
> > > The answer to your first question is “yes, almost.”  The central flow differs from the stable flow in _two_ respects.  First, the oscillations affect the weights $w$ --- an effect which, as you note, is captured by the implicit sharpness penalty.  However, the oscillations also affect the EMA $\nu$; this has the effect of shrinking the central flow’s effective step size.  Because this is a potential confounder, the sharpness plot in Figure 4 is, by itself, insufficient evidence for our claim of implicit sharpness regularization and “acceleration via regularization.”
> > >
> > > To eliminate this potential confounder, our paper’s Figure 5 directly compared the central flow against an ablated flow which used the same effective step size as the central flow. However, as you pointed out, Figure 5 only depicted hyperparameters for which the central flow caused the sharpness to decrease.  Therefore, we’ve generated a new version of Figure 5 [here](https://imgur.com/a/zwZZ2Uu)  which additionally plots the central flow for $\eta = 0.01$ (in purple).  For this value of $\eta$, the implicit sharpness regularization is too weak to ‘overpower’ progressive sharpening, but still regularizes sharpness relative to the ablated central flow (in black).  We will add this figure to the revised paper.  Thank you for your feedback which has helped us improve our paper.

---

> > > > ### Comment · Reviewer_mqGu · 2024-11-19
> > > >
> > > > Ah ha!  Thank you for this additional explanation and the new figure. This is very helpful and I think makes the point even more clear now about how there is regularization against sharpness, even if it is not strong enough to cause a decrease is sharpness.
> > > >
> > > > Thank you for your engagement. I believe this paper has been strengthened.

---

> > > > > ### Author Response · Authors · 2024-11-28
> > > > > **author response**
> > > > >
> > > > > Thank you very much for participating in the review process, and for helping us improve our paper!

---

### Official Review · Reviewer_DEu8 · 2024-11-04

**Soundness:** 3
**Presentation:** 4
**Contribution:** 4
**Rating:** 8
**Confidence:** 3

**Summary:**

The paper investigates optimization methods in deep learning, focusing on the "edge of stability" regime. Since the gradient flow differential equation fails to capture both the oscillatory behaviors and the averaged trends of gradient descent in many neural network training scenarios, the authors introduce central flows, a new differential equation that approximates the "time-averaged" optimization dynamics up to high accuracy. The paper also provides valuable insights into adaptive optimizers, such as RMSProp, and introduces the concept of acceleration via regularization, where adaptive optimizers naturally steer trajectories toward low-curvature regions, enabling larger steps and improving optimization efficiency.

**Strengths:**

1. The article is clearly written and reader-friendly, with insightful discussions on the oscillatory behavior of optimization methods.

2. The time-averaging and Taylor expansion framework is intuitively sound and appears to be broadly applicable, capable of handling various adaptive optimizers, and could potentially be extended to more general optimization methods.

3. Extensive empirical results are presented, supporting the validity of the central flow.

Overall, I think this is a nice paper.

**Weaknesses:**

The central flow derivations are grounded in empirical observations, and several mathematical steps rely on informal reasoning. For example, it is not clear how to rigorously justify the time averaging step and quantify the high-order error in the Taylor expansion. Rigorous proofs or formal analysis would enhance the robustness of the claims.

**Questions:**

The authors present extensive numerical experiments that demonstrate the central flow as a strong approximation to average optimization dynamics. While the numerical results are impressive, could the authors discuss any potential limitations or scenarios where the central flow may not perform as effectively? Understanding the boundaries of the method’s applicability would provide helpful context for further investigation.

---

> ### Author Response · Authors · 2024-11-16
> **author response**
>
> We thank the reviewer for their insightful comments and questions. We address each point below and welcome further discussion.
>
> > The authors present extensive numerical experiments that demonstrate the central flow as a strong approximation to average optimization dynamics. While the numerical results are impressive, could the authors discuss any potential limitations or scenarios where the central flow may not perform as effectively? Understanding the boundaries of the method’s applicability would provide helpful context for further investigation.
>
> We agree that it is important to discuss the factors that affect the quality of the central flow approximation.  We included some preliminary discussion in Appendix D, lines 1739-1744, and we intended to link to this appendix on line 266 of the main paper (but we accidentally linked to Appendix B instead).  If the paper is accepted, our camera-ready revision will contain a discussion of this topic in the main paper.
>
> One important factor is the learning rate.  Empirically, we observe that the quality of the central flow approximation usually degrades as the learning rate grows increasingly large ([figure](https://imgur.com/a/K3xkkmb)), though we emphasize that even when the central flow is off in weight space, it often still successfully predicts derived quantities such as network outputs or the train loss. We hypothesize that the root factor here might be the magnitude of the fluctuations in sharpness around $2/\eta$, though further investigation is needed.
>
> Another factor is the loss criterion.  Empirically, the long-term accuracy of the central flow is often higher when using MSE loss than when using cross-entropy loss.  The issue seems to be that for cross-entropy loss, the equilibrium point of the sharpness often differs slightly from $2/\eta$ (think e.g. $2.05 / \eta$), whereas the central flow models the sharpness as being locked exactly at $2/\eta$.  This discrepancy causes errors to gradually accumulate over long timescales.  Damian et al ‘22, in their Appendix F, used higher-order terms to accurately predict the corrected sharpness equilibrium point, and were therefore able to accurately model the trajectory even in these situations.   We think that a similar correction might be possible for our central flows, but we leave this to future work.
>
> Finally, the regularity of the loss landscape also seems to affect the accuracy of the central flow.  When we consider a family of activation functions which interpolates between GeLU (smooth) and ReLU (not smooth), we find that the accuracy of the central flow decreases as the network’s activation function is interpolated from GeLU to ReLU (see this [figure](https://imgur.com/a/K3xkkmb)).
>
> We’ve also included both of the new figures in a new appendix (Appendix E) that can be found on the final page of the current PDF.

---

> > ### Comment · Reviewer_DEu8 · 2024-11-30
> >
> > Thanks for the detailed response. I have read the author's thoughtful response and other reviews. I think this is a strong paper and it is improved with the described revisions.

---

### Meta-Review · Area_Chair_oCm5 · 2024-12-10

**Metareview:**

The paper introduces a novel framework for understanding the long-term dynamics of optimization in deep learning, particularly in the "edge of stability" regime. By modeling the time-averaged behavior of GD and RMSProp, the paper reveals some new insights, including implicit regularization mechanisms that steer optimization trajectories toward low-curvature regions. The claims are supported by empirical results and detailed derivations.

### **Strengths**
- Introduces a new tool to analyze optimization trajectories
- Offers insights into RMSProp's implicit behavior, such as regularization against sharpness and "acceleration via regularization."

### **Weaknesses**
- A significant portion of the theoretical derivations and experimental results are relegated to the appendices, making the main paper feel incomplete.
 The framework relies on informal reasoning (e.g., time-averaging steps and Taylor expansions), and some equations (e.g., Eq. 7) are not fully derived in the main text.

The reviewers are overall positive and I therefore recommend acceptance but I strongly encourage the authors to better integrate key derivations into the main text.

**Additional Comments On Reviewer Discussion:**

The discussion properly addressed the important concerns of the reviewers. Only issues left are about the presentation which I think can be improved in a revised version.

---

### Decision · Program_Chairs · 2025-01-22

Accept (Poster)